# A Unified Framework for Variable Selection in Model-Based Clustering with Missing Not at Random

**Binh H. Ho**[*,1,2]**, Long Nguyen Chi**[*,3]**, TrungTin Nguyen**[*,†,4,5]**,**
**Binh T. Nguyen**[1,2]**, Van Ha Hoang**[1,2]**, Christopher Drovandi**[4,5]
[1]Faculty of Mathematics and Computer Science,
University of Science, Ho Chi Minh City, Vietnam.
[2]Vietnam National University Ho Chi Minh City, Vietnam.
[3]School of Information and Communications Technology,
Hanoi University of Science and Technology, Ha Noi, Vietnam.
[4]ARC Centre of Excellence for the Mathematical Analysis of Cellular Systems.
[5]School of Mathematical Sciences,
Queensland University of Technology, Brisbane City, Australia.

## Abstract

Model-based clustering integrated with variable selection is a powerful tool for uncovering latent structures within complex data. However, its effectiveness is often hindered by challenges such as identifying relevant variables that define heterogeneous subgroups and handling data that are missing not at random, a prevalent issue in fields like transcriptomics. While several notable methods have been proposed to address these problems, they typically tackle each issue in isolation, thereby limiting their flexibility and adaptability. This paper introduces a unified framework designed to address these challenges simultaneously. Our approach incorporates a data-driven penalty matrix into penalized clustering to enable more flexible variable selection, along with a mechanism that explicitly models the relationship between missingness and latent class membership. We demonstrate that, under certain regularity conditions, the proposed framework achieves both asymptotic consistency and selection consistency, even in the presence of missing data. This unified strategy significantly enhances the capability and efficiency of model-based clustering, advancing methodologies for identifying informative variables that define homogeneous subgroups in the presence of complex missing data patterns. The performance of the framework, including its computational efficiency, is evaluated through simulations and demonstrated using both synthetic and real-world transcriptomic datasets.

## 1 Introduction

**Model-based Clustering.** Model-based clustering formulates clustering as a probabilistic inference task, assuming data are generated from a finite mixture model, with each component corresponding to a cluster. This enables likelihood-based estimation and principled model selection. Gaussian mixture models (GMMs) [42, 43] are a classical instance, which can be estimated via the expectation-maximization (EM) algorithm [16], providing soft assignments and flexibility in modeling non-spherical and overlapping clusters. Bayesian mixture models extend this framework by treating parameters, and even the number of components, as random variables, thereby enabling the quantification of uncertainty. Techniques such as Markov chain Monte Carlo (MCMC) and variational inference are employed to sample the posterior distribution, although challenges such as

---

[*]Co-first author, [†]Corresponding author.

39th Conference on Neural Information Processing Systems (NeurIPS 2025).

label switching persist [64]. Bayesian nonparametric approaches, including reversible-jump MCMC and birth-death processes, impose prior constraints on model complexity and infer the number of clusters directly. Despite offering interpretability and robustness, especially in small-sample settings, Bayesian methods suffer from high computational costs, label switching, and convergence issues [29]. Therefore, this paper focuses on the maximum likelihood estimation (MLE) approach for dealing with variable selection and missing data in model-based clustering.

**Related Works on Variable Selection for Model-based Clustering.** Traditional variable selection methods like best-subset and stepwise regression rely on information criteria such as Akaike information criterion (AIC) [1] and Bayesian information criterion (BIC) [58], but become computationally infeasible as dimensionality increases. Penalized likelihood approaches, notably least absolute shrinkage and selection operator (LASSO) [70], address scalability by inducing sparsity; other earlier contributions include ridge regression [28] and the nonnegative garrote [9]. Efficient algorithms like least-angle regression [18] and coordinate descent enable application in high dimensions. To reduce dimensionality before modeling, screening methods such as sure independence screening [19] and its conditional extension [5] filter variables based on marginal or conditional correlations. In clustering, Law et al. [32] propose simultaneous variable selection and clustering using feature saliency. Andrews and McNicholas [3] introduce variable selection for clustering and classification, combining filter and wrapper strategies to discard noisy variables. Bayesian methods such as those by Tadesse et al. [67] and Kim et al. [30] extend variable relevance indicators to mixture models, though MCMC scalability remains a challenge. Raftery and Dean [54] develop a BIC-based stepwise method for identifying clustering-relevant variables, later optimized by Scrucca [59]. Finally, Maugis et al. [38] enrich this framework by incorporating redundancy modeling, enhancing selection accuracy in correlated settings at the cost of increased computational complexity. To overcome this computational complexity, Celeux et al. [13] propose a two-step strategy for selecting variables in mixture models. First, variables are ranked by optimizing a penalized likelihood function that shrinks both component means and precisions, following the approach of Zhou, Pan, and Shen [75]. This method is scalable to moderate dimensions and yields an ordered list where informative variables are expected to appear early. Second, roles are assigned in a single linear pass through the ranked list, replacing combinatorial search with a faster and interpretable procedure that maintains competitive clustering accuracy. However, there is no theoretical guarantee that this ranking recovers the true signal-redundant-uninformative partition.

**Handling Missing Data.** Missing data are commonly classified into three categories: missing completely at random (MCAR), missing at random (MAR), and missing not at random (MNAR). The last category, MNAR, is the most challenging, as the probability of missingness depends on unobserved values. Under the MAR assumption, the method of multiple imputation provides a principled framework by replacing each missing value with several plausible alternatives, then combining analyses across imputed datasets to account for uncertainty [57]. The multivariate imputation by chained equations algorithm [71] is a flexible implementation that fits a sequence of conditional models, accommodating mixed data types. In high-dimensional settings, estimating full joint or conditional models becomes unstable. To address this, regularized regression techniques, such as the LASSO and its Bayesian counterpart, have been integrated into imputation models to improve predictive performance by selecting and shrinking predictors [74]. Nonparametric and machine learning-based imputation methods have also been developed. The MissForest algorithm [63] uses random forests to iteratively impute missing values, capturing nonlinear relations. Deep generative models, such as the missing data importance-weighted autoencoder [36] and its federated variant [4], rely on latent variable representations to generate imputations, assuming data lie near a low-dimensional manifold. For MNAR data, model-based approaches face difficulties due to unidentifiability without external constraints. Two common strategies include selection models, which jointly specify the data and missingness mechanism, and pattern mixture models, which condition on observed missingness patterns. Both frameworks require strong assumptions or instruments to yield valid inferences. To address this challenge, Sportisse et al. [62] studied the identifiability of selection models under the MNAR assumption, particularly when missingness depends on unobserved values. They showed that identifiability can be achieved under structural assumptions, such as requiring each variable's missingness to be explained either by its value or by the latent class, but not both simultaneously. Their proposed joint modeling framework, termed MNARz, ensures that cluster-specific missingness reflects meaningful distributional differences. A key insight is that under MNARz, the missing data problem can be reformulated as MAR on an augmented data matrix, enabling tractable inference.

**Main Contribution.** Inspired by the work of Maugis et al. [39], who addressed variable selection under MAR without imputation, we generalize their framework to latent class MNARz. Under the MNARz mechanism, we extend the LASSO-based variable selection framework for model-based clustering proposed by [13], establishing identifiability and consistency guarantees in Section 4 for recovering the true signal-redundant-uninformative partition. Our method achieves substantial empirical improvements in Section 5 over existing approaches. This yields a unified, high-dimensional model-based clustering method that jointly addresses variable selection and MNAR inference in a principled manner.

**Paper Organization.** The remainder of the paper is structured as follows. Section 2 reviews variable selection in model-based clustering with MNAR data. Section 3 and Section 4 present our proposed framework and its theoretical guarantees, respectively. Simulation and real data results are reported in Section 5. We conclude with a summary, limitations, and future directions in Section 6. Proofs and additional details are provided in the supplementary material.

**Notation.** Throughout the paper, we use the shorthand $[N]$ to denote the index set $\{1, 2, \ldots, N\}$ for any positive integer $N \in \mathbb{N}$. We denote $\mathbb{R}$ as the set of real numbers. The notation $|\mathbb{S}|$ represents the cardinality of any set $\mathbb{S}$. For any vector $\boldsymbol{v} \in \mathbb{R}^D$, $\|\boldsymbol{v}\|_p$ denotes its $p$-norm. The operator $\odot$ indicates the element-wise (Hadamard) product between two matrices. We use $n \in [N]$ to index observations and $d \in [D]$ to index variables. The vector $\boldsymbol{y}_n \in \mathbb{R}^D$ denotes the full data vector for observation $n$, which may be further specified in more granular forms. The binary vector $\boldsymbol{c}_n = (c_{n1}, \ldots, c_{nD}) \in \{0, 1\}^D$ represents the missingness mask, where $c_{nd} = 1$ if $y_{nd}$ is missing. The latent variable $z_n \in [K]$ denotes the mixture component assignment for observation $n$, with indicator variable $z_{nk} = \mathbf{1}_{\{z_n = k\}}$. We use $\ell(\cdot)$ to denote the log-likelihood.

## 2 Background

We begin by introducing some preliminaries on model-based clustering with variable selection, penalized clustering, and MNARz formulation.

**Model-based Clustering and Parsimonious Mixture Form.** Model-based clustering adopts a probabilistic framework by assuming that the data matrix $\mathbf{Y} = (\boldsymbol{y}_1, \ldots, \boldsymbol{y}_N)^\top \in \mathbb{R}^{N \times D}$, with each $\boldsymbol{y}_n \in \mathbb{R}^D$, is independently drawn from a location-scale mixture distribution, a class known for its universal approximation capabilities and favorable convergence properties [25, 55, 14, 53, 52, 46, 47, 49, 60, 26, 27]. The density of an observation $\boldsymbol{y}_n$ under a $K$-component GMM with parameters $\boldsymbol{\alpha} = \{\boldsymbol{\pi}, \boldsymbol{\mu}, \boldsymbol{\Sigma}\}$ is given by: $f_{\text{GMM}}(\boldsymbol{y}_n \mid K, m, \boldsymbol{\alpha}) = \sum_{k=1}^K \pi_k \mathcal{N}(\boldsymbol{y}_n \mid \boldsymbol{\mu}_k, \boldsymbol{\Sigma}_k)$ where $\pi_k > 0$ and $\sum_{k=1}^K \pi_k = 1$. The term $\mathcal{N}(\boldsymbol{y}_n \mid \boldsymbol{\mu}_k, \boldsymbol{\Sigma}_k)$ denotes the multivariate normal density with mean vector $\boldsymbol{\mu}_k$ and covariance matrix $\boldsymbol{\Sigma}_k$, which is compactly denoted by $\boldsymbol{\Theta}_k = (\boldsymbol{\mu}_k, \boldsymbol{\Sigma}_k)$. The covariance matrix $\boldsymbol{\Sigma}_k$ encodes the structure determined by model form $m$, allowing for parsimonious modeling via spectral decompositions that control cluster volume, shape, and orientation [12]. These constraints are particularly valuable in high-dimensional settings, where $D \gg N$ and standard GMMs require estimating $\mathcal{O}(K^2 D)$ parameters, leading to overparameterization. Model selection criteria such as the BIC [58, 39, 22, 21, 48], slope heuristics [6, 51, 50], integrated classification likelihood (ICL) [7, 24], extended BIC (eBIC) [23, 45], dendrogram selection criterion [17, 69], and Sin-White information criterion (SWIC) [61, 72] can be employed to select both the number of components $K$ and the covariance structure $m$.

**Variable Selection as a Model Selection Problem in Model-based Clustering.** In high-dimensional settings, many variables may be irrelevant or redundant with respect to the underlying cluster structure, thereby degrading clustering performance and interpretability. To address this, [38] proposed the $\mathbb{SRUW}$ model, extending their earlier work [37], which assigns variables to one of four roles. Let $\mathbb{S}$ denote the set of relevant clustering variables. Its complement, $\mathbb{S}^c$, comprises the irrelevant variables and is partitioned into subsets $\mathbb{U}$ and $\mathbb{W}$. Variables in $\mathbb{U}$ are linearly explained by a subset $\mathbb{R} \subseteq \mathbb{S}$, while variables in $\mathbb{W}$ are assumed independent of all relevant variables. This framework facilitates variable-specific interpretation and avoids over-penalization from complex covariance structures, as typically encountered in constrained GMMs [12]. The joint density under the $\mathbb{SRUW}$ *model*, combining mixture components for clustering, regression, and independence, is defined as:

$$f_{\mathbb{SRUW}}(\boldsymbol{y}_n \mid K, m, r, l, \mathbb{V}, \boldsymbol{\Theta}) = f_{\text{clust}}(\boldsymbol{y}_n^{\mathbb{S}} \mid K, m, \boldsymbol{\alpha}) \, f_{\text{reg}}(\boldsymbol{y}_n^{\mathbb{U}} \mid r, \boldsymbol{a} + \boldsymbol{y}_n^{\mathbb{R}} \boldsymbol{\beta}, \boldsymbol{\Omega}) \, f_{\text{indep}}(\boldsymbol{y}_n^{\mathbb{W}} \mid l, \boldsymbol{\gamma}, \boldsymbol{\Gamma}). \tag{1}$$

Here, $\mathbb{V} = (\mathbb{S}, \mathbb{U}, \mathbb{R}, \mathbb{W})$ denotes the variable partition, and $\boldsymbol{\Theta}$ is the full parameter set. The components are defined as: $f_{\text{clust}} := f_{\text{GMM}}$, $f_{\text{reg}}(\boldsymbol{y}_n^{\mathbb{U}} \mid r, \boldsymbol{a} + \boldsymbol{y}_n^{\mathbb{R}}\boldsymbol{\beta}, \boldsymbol{\Omega}) := \mathcal{N}(\boldsymbol{y}_n^{\mathbb{U}} \mid \boldsymbol{a} + \boldsymbol{y}_n^{\mathbb{R}}\boldsymbol{\beta}, \boldsymbol{\Omega})$, with $\boldsymbol{a} \in \mathbb{R}^{1 \times |\mathbb{U}|}$, $\boldsymbol{\beta} \in \mathbb{R}^{|\mathbb{R}| \times |\mathbb{U}|}$, and $\boldsymbol{\Omega} \in \mathbb{R}^{|\mathbb{U}| \times |\mathbb{U}|}$ structured by $r$. The independent part is $f_{\text{indep}} := \mathcal{N}$, with variance structure $l$ and covariance $\boldsymbol{\Gamma}$. Model selection proceeds by maximizing a BIC-type criterion:

$$\text{crit}_{\text{BIC}}(K, m, r, l, \mathbb{V}) = \text{BIC}_{\text{clust}}(\boldsymbol{Y}^{\mathbb{S}} \mid K, m) + \text{BIC}_{\text{reg}}(\boldsymbol{Y}^{\mathbb{U}} \mid r, \boldsymbol{Y}^{\mathbb{R}}) + \text{BIC}_{\text{indep}}(\boldsymbol{Y}^{\mathbb{W}} \mid l), \quad (2)$$

over $(K, m, r, l, \mathbb{V})$, where each term scores the corresponding model component. Although this approach offers fine-grained variable treatment, it can be computationally demanding in high dimensions due to stepwise selection for clustering and regression.

**Penalized Log-Likelihood Methods for Simultaneous Clustering and Variable Selection.** We follow the framework of [11], which extends [75, 13], introducing cluster-specific penalties via group-wise weighting matrices $\boldsymbol{P}_k$ for adaptive regularization in Gaussian graphical mixture models. This method improves upon stepwise procedures [38, 37] by handling high-dimensional data more efficiently. The penalized log-likelihood is given by:

$$\sum_{n=1}^{N} \ln \Big[ \sum_{k=1}^{K} \pi_k \mathcal{N}(\bar{\boldsymbol{y}}_n \mid \boldsymbol{\mu}_k, \boldsymbol{\Sigma}_k) \Big] - \lambda \sum_{k=1}^{K} \|\boldsymbol{\mu}_k\|_1 - \rho \sum_{k=1}^{K} \sum_{d \neq d'} \big|(\boldsymbol{P}_k \odot \boldsymbol{\Sigma}_k^{-1})_{dd'}\big|. \quad (3)$$

Here, $\bar{\boldsymbol{y}}_n$ denotes centered and scaled observations, and $\boldsymbol{P}_k$ is typically chosen to be inversely proportional to initial partial correlations, enabling adaptive shrinkage akin to the adaptive lasso [77]. Given grids of regularization parameters $\mathcal{G}_\lambda$ and $\mathcal{G}_\rho$, the EM algorithm from [75] is used to estimate the parameters $\widehat{\boldsymbol{\alpha}}(\lambda, \rho) = \big(\widehat{\boldsymbol{\pi}}(\lambda, \rho), \widehat{\boldsymbol{\mu}}_1(\lambda, \rho), \dots, \widehat{\boldsymbol{\mu}}_K(\lambda, \rho), \widehat{\boldsymbol{\Sigma}}_1(\lambda, \rho), \dots, \widehat{\boldsymbol{\Sigma}}_K(\lambda, \rho)\big)$. For each variable $d \in [D]$, clustering scores $\mathcal{O}_K(d)$ are computed, and the relevant set $\mathbb{S}$ is formed from those that improve BIC, while $\mathbb{U}$ captures non-informative variables. The final model is selected by maximizing $\text{crit}_{\text{BIC}}\big(K, m, r, l, \mathbb{V}_{(K,m,r)}\big)$ in Equation (2), where $\mathbb{V}_{(K,m,r)} = \big(\mathbb{S}_{(K,m)}, \mathbb{R}_{(K,m,r)}, \mathbb{U}_{(K,m)}, \mathbb{W}_{(K,m)}\big)$.

## 3 Our Proposal: Variable Selection in Model-Based Clustering with MNAR

We propose a compact framework integrating adaptive variable selection with robust missing data handling in model-based clustering.

**Global GMM Representation of $\mathbb{SRUW}$ under MAR.** A key aspect is expressing the observed-data likelihood in a tractable form. Under the MAR assumption, the $\mathbb{SRUW}$ model admits a global GMM representation, extending the result of [39] for the $\mathbb{SR}$ model. Given the density in Equation (1) and Gaussian properties, the observed likelihood becomes

$$f(\boldsymbol{Y}^o \mid K, m, r, l, \mathbb{V}, \boldsymbol{\Theta}) = \prod_{n=1}^{N} \left( \sum_{k=1}^{K} \pi_k \, \mathcal{N}\left(\boldsymbol{y}_n^o \mid \tilde{\boldsymbol{\nu}}_{k,o}, \tilde{\boldsymbol{\Delta}}_{k,oo}\right) \right), \quad (4)$$

where $\tilde{\boldsymbol{\nu}}_{k,o} = \begin{pmatrix} \boldsymbol{\nu}_{k,o} \\ \boldsymbol{\gamma}_o \end{pmatrix}$, $\tilde{\boldsymbol{\Delta}}_{k,oo} = \begin{pmatrix} \boldsymbol{\Delta}_{k,oo} & \boldsymbol{0} \\ \boldsymbol{0} & \boldsymbol{\Gamma}_{oo} \end{pmatrix}$. Here, $\boldsymbol{\nu}_{k,o}, \boldsymbol{\gamma}_o$ and $\boldsymbol{\Delta}_{k,oo}, \boldsymbol{\Gamma}_{oo}$ correspond to the block means and covariances from the $\mathbb{SR}$ model. The derivation extends the technical proof from [39] and is detailed in the supplement. This representation offers: (i) a unified GMM encoding the variable roles in $\mathbb{SRUW}$ via observed-data parameters, (ii) compatibility with standard EM algorithms, enabling MAR-aware estimation and internal imputation that respects cluster structure, and (iii) theoretical guarantees of identifiability and consistency in selecting the true variable partition via Equation (2) (see Section 4).

**Incorporating MNARz Mechanism into $\mathbb{SRUW}$.** To explicitly address MNAR data, we integrate the MNARz mechanism from [62], where missingness depends on the latent class membership. Estimation proceeds via EM, while identifiability and the transformation of MNARz into MAR on augmented data (i.e., original data with the missingness indicator matrix $\boldsymbol{C}$) are retained (see Section 4). Let $\boldsymbol{y}_n = (\boldsymbol{y}_n^{\mathbb{S}}, \boldsymbol{y}_n^{\mathbb{U}}, \boldsymbol{y}_n^{\mathbb{W}})$ denote the full data for observation $n$, partitioned by role (Section 2), and let $\boldsymbol{c}_n$ be its binary missingness pattern. With $z_n \in [K]$ denoting the latent class, the complete-data density under $z_{nk} = 1$ is

$$f(\boldsymbol{y}_n, \boldsymbol{c}_n \mid z_{nk} = 1; \boldsymbol{\alpha}_k, \boldsymbol{\psi}_k) = f_{\text{clust}}(\boldsymbol{y}_n^{\mathbb{S}} \mid \boldsymbol{\alpha}_k) \, f_{\text{reg}}(\boldsymbol{y}_n^{\mathbb{U}} \mid \boldsymbol{y}_n^{\mathbb{R}}; \boldsymbol{\theta}_{\text{reg}})$$
$$\times f_{\text{indep}}(\boldsymbol{y}_n^{\mathbb{W}} \mid \boldsymbol{\theta}_{\text{indep}}) \, f_{\text{MNARz}}(\boldsymbol{c}_n \mid z_{nk} = 1; \boldsymbol{\psi}_k),$$

where $f_{\text{MNARz}}(\boldsymbol{c}_n \mid z_{nk} = 1; \boldsymbol{\psi}_k) = \prod_{d=1}^{D} \rho(\boldsymbol{\psi}_k)^{c_{nd}}(1 - \rho(\boldsymbol{\psi}_k))^{1-c_{nd}}$ and $\rho(\boldsymbol{\psi}_k)$ is the class-specific missingness probability. The model parameters are $\boldsymbol{\Theta} = (\pi_1, \ldots, \pi_K, \{\boldsymbol{\alpha}_k\}_{k=1}^{K}, \boldsymbol{\xi} = \{\boldsymbol{\theta}_{\text{reg}}, \boldsymbol{\theta}_{\text{indep}}\}, \{\boldsymbol{\psi}_k\}_{k=1}^{K})$. The observed-data log-likelihood is:

$$\ell(\boldsymbol{\Theta}; \boldsymbol{Y}, \boldsymbol{C}) = \sum_{n=1}^{N} \log \sum_{k=1}^{K} \pi_k f_k^o(\boldsymbol{y}_n^o; \boldsymbol{\alpha}_k, \boldsymbol{\xi}) f_c(\boldsymbol{c}_n; \boldsymbol{\psi}_k), \tag{5}$$

with the MNARz mechanism factored out due to independence. The complete-data log-likelihood is:

$$\ell_{\text{comp}}(\boldsymbol{\Theta}; \boldsymbol{Y}, \boldsymbol{C}) = \sum_{n=1}^{N} \sum_{k=1}^{K} z_{nk} \log\big(\pi_k f_k(\boldsymbol{y}_n; \boldsymbol{\alpha}_k, \boldsymbol{\xi}) f_c(\boldsymbol{c}_n; \boldsymbol{\psi}_k)\big). \tag{6}$$

To accommodate both MAR and MNAR patterns, we extend the model with a binary partition of variable indices: $\mathcal{D}_{\text{MAR}} \cup \mathcal{D}_{\text{MNAR}} = [D]$, $|\mathcal{D}_{\text{MAR}}| = D_M$, $|\mathcal{D}_{\text{MNAR}}| = D_{M'}$. Variables in $\mathcal{D}_{\text{MAR}}$ follow a MAR mechanism, and those in $\mathcal{D}_{\text{MNAR}}$ follow MNARz:

$$f_k(\boldsymbol{y}_n; \boldsymbol{\alpha}_k) = f_{\text{clust}}(\boldsymbol{y}_n^{\mathbb{S}} \mid \boldsymbol{\alpha}_k) f_{\text{reg}}(\boldsymbol{y}_n^{\mathbb{U}} \mid \boldsymbol{y}_n^{\mathbb{R}}; \boldsymbol{\theta}_{\text{reg}}) f_{\text{indep}}(\boldsymbol{y}_n^{\mathbb{W}} \mid \boldsymbol{\theta}_{\text{indep}}), \quad \boldsymbol{\xi} = (\boldsymbol{\theta}_{\text{reg}}, \boldsymbol{\theta}_{\text{indep}}),$$

$$f_c^{\text{MAR}}(\boldsymbol{c}_{n,M} \mid \boldsymbol{y}_n^o; \boldsymbol{\psi}_M) = \prod_{d \in \mathcal{D}_{\text{MAR}}} \rho_d(\boldsymbol{y}_n^o; \psi_{Md})^{c_{nd}} \left(1 - \rho_d(\boldsymbol{y}_n^o; \psi_{Md})\right)^{1-c_{nd}},$$

$$f_c^{\text{MNARz}}(\boldsymbol{c}_{n,N}; \boldsymbol{\psi}_k) = \prod_{d \in \mathcal{D}_{\text{MNAR}}} \rho(\boldsymbol{\psi}_k)^{c_{nd}} \left(1 - \rho(\boldsymbol{\psi}_k)\right)^{1-c_{nd}}.$$

The parameter blocks expand as: $\boldsymbol{\Theta} = \big(\pi_1, \ldots, \pi_K, \{\boldsymbol{\alpha}_k\}_{k=1}^{K}, \boldsymbol{\xi}, \boldsymbol{\psi}_M, \{\boldsymbol{\psi}_k\}_{k=1}^{K}\big)$. Let $\boldsymbol{y}_n = (\boldsymbol{y}_n^o, \boldsymbol{y}_n^m)$, and define $\boldsymbol{c}_{n,M} = (c_{nd})_{d \in \mathcal{D}_{\text{MAR}}}$, $\boldsymbol{c}_{n,M'} = (c_{nd})_{d \in \mathcal{D}_{\text{MNAR}}}$. Then, since the MNARz mask is independent of $\boldsymbol{y}_n$ given $k$, the observed-data likelihood becomes:

$$\ell(\boldsymbol{\Theta}; \boldsymbol{Y}, \boldsymbol{C}) = \sum_{n=1}^{N} \log \left[ \sum_{k=1}^{K} \pi_k \, f_c^{\text{MNARz}}(\boldsymbol{c}_{n,M'}; \boldsymbol{\psi}_k) f_{k,M}^o(\boldsymbol{y}_n^o; \boldsymbol{\alpha}_k, \boldsymbol{\psi}_M) \right]. \tag{7}$$

where $f_{k,M}^o(\boldsymbol{y}_n^o; \boldsymbol{\alpha}_k, \boldsymbol{\psi}_M) := \int f_k(\boldsymbol{y}_n; \boldsymbol{\alpha}_k) \, f_c^{\text{MAR}}(\boldsymbol{c}_{n,M} \mid \boldsymbol{y}_n^o; \boldsymbol{\psi}_M) \, d\boldsymbol{y}_n^m$. The MNARz term factors out of the integral, preserving the ignorability property for that block [62]. Only the MAR component requires integration or imputation.

**Adaptive Weighting for Penalty of Precision Matrix.** To enhance performance, we replace the inverse partial correlation scheme with a spectral-based computation of $\pi_{\mathbf{k}}$ in Equation (3). Starting from the initial precision matrix $\hat{\boldsymbol{\Psi}}_k^{(0)}$, we construct an unweighted, undirected graph $\mathcal{G}^{(k)} = (\mathcal{V}, \mathcal{E}^{(k)})$, where $\mathcal{V} = [D]$, and include edge $(i,j)$ in $\mathcal{E}^{(k)}$ if $\|\hat{\psi}_{k,ij}^{(0)}\| > \boldsymbol{\Gamma}_{adj}$. Let $\mathbf{A}^{(k)}$ and $\mathbf{D}^{(k)}$ denote the adjacency and degree matrices, respectively. The symmetrically normalized Laplacian is then defined as $\mathbf{L}_{sym}^{(k)} = \mathbf{I} - (\mathbf{D}^{(k)})^{-1/2}\mathbf{A}^{(k)}(\mathbf{D}^{(k)})^{-1/2}$, which offers a scale-invariant measure of connectivity. We aim to shrink $\boldsymbol{\Psi}_k$ toward a diagonal target (i.e., an empty graph), with $\mathbf{L}_{\text{target}} = \mathbf{0}$. The spectral distance between $\hat{\boldsymbol{\Psi}}_k^{(0)}$ and this target is $D_{\text{LS}}(\boldsymbol{\Psi}_k^{(0)}) = \|\text{spec}(\mathbf{L}_{\text{sym}}^k)\|_2$, and the corresponding adaptive weights are given by $\mathbf{P}_{k,ij} = \left(D_{\text{LS}}(\boldsymbol{\Psi}_k^{(0)}) + \epsilon\right)^{-1}$.

**Parameter Estimation.** Following [75], we adopt similar parameter estimation for variable ranking and focus here on estimating $\mathbb{SRUW}$ under the MNARz mechanism using the EM algorithm. Other estimation details are provided in the Supplement. Let each observation be partitioned as $\boldsymbol{y}_n = (\boldsymbol{y}_n^o, \boldsymbol{y}_n^m)$, and define $\boldsymbol{\xi} = (\boldsymbol{\theta}_{\text{reg}}, \boldsymbol{\theta}_{\text{indep}})$. Since $f_c$ is independent of $\boldsymbol{y}_n$, the observed-data log-likelihood becomes:

$$\ell(\boldsymbol{\Theta}; \boldsymbol{Y}^o, \boldsymbol{C}) = \sum_{n=1}^{N} \log \sum_{k=1}^{K} \pi_k f_c(\boldsymbol{c}_n; \boldsymbol{\psi}_k) f_k^o(\boldsymbol{y}_n^o; \boldsymbol{\alpha}_k, \boldsymbol{\xi}),$$

where $f_k^o(\boldsymbol{y}_n^o; \boldsymbol{\alpha}_k, \boldsymbol{\xi}) := \int f_{\text{clust}}(\boldsymbol{y}_n^S \mid \boldsymbol{\alpha}_k) f_{\text{reg}}(\boldsymbol{y}_n^U \mid \boldsymbol{y}_n^R; \boldsymbol{\theta}_{\text{reg}}) f_{\text{indep}}(\boldsymbol{y}_n^W \mid \boldsymbol{\theta}_{\text{indep}}) \, d\boldsymbol{y}_n^m$.

The E-step computes responsibilities:

$$t_{nk}^{(t)} = \frac{\pi_k^{(t-1)} f_k^o(\boldsymbol{y}_n^o; \boldsymbol{\alpha}_k^{(t-1)}, \boldsymbol{\xi}^{(t-1)}) f_c(\boldsymbol{c}_n; \boldsymbol{\psi}_k^{(t-1)})}{\sum_{l=1}^{K} \pi_l^{(t-1)} f_l^o(\boldsymbol{y}_n^o; \boldsymbol{\alpha}_l^{(t-1)}, \boldsymbol{\xi}^{(t-1)}) f_c(\boldsymbol{c}_n; \boldsymbol{\psi}_l^{(t-1)})}.$$

The $Q$-function to be maximized in the M-step is:

$$Q(\mathbf{\Theta}; \mathbf{\Theta}^{(t-1)}) = \sum_{n=1}^{N} \sum_{k=1}^{K} t_{nk}^{(t)} \left\{ \log \pi_k + g_y(\boldsymbol{\alpha}_k) + g_c(\boldsymbol{\psi}_k) \right\}.$$

$$g_y(\boldsymbol{\alpha}_k) = \mathbb{E}_{\mathbf{\Theta}^{(t-1)}} \left[ \log f_k(\boldsymbol{y}_n; \boldsymbol{\alpha}_k, \boldsymbol{\xi}_k) \mid \boldsymbol{y}_n^o, \boldsymbol{c}_n, z_{nk} = 1 \right],$$

$$g_c(\boldsymbol{\psi}_k) = \mathbb{E}_{\mathbf{\Theta}^{(t-1)}} \left[ \log f_c(\boldsymbol{c}_n \mid \boldsymbol{y}_n; \boldsymbol{\psi}_k) \mid \boldsymbol{y}_n^o, \boldsymbol{c}_n, z_{nk} = 1 \right].$$

In the M-step, we update:

$$\pi_k^{(t)} = \frac{1}{N} \sum_n t_{nk}^{(t)}, \quad \boldsymbol{\alpha}_k^{(t)} = \arg\max_{\boldsymbol{\alpha}_k} \sum_n t_{nk}^{(t)} g_y(\boldsymbol{\alpha}_k),$$

$$\boldsymbol{\psi}_k^{(t)} = \arg\max_{\boldsymbol{\psi}_k} \sum_n t_{nk}^{(t)} g_c(\boldsymbol{\psi}_k), \quad \boldsymbol{\xi}^{(t)} = \arg\max_{\boldsymbol{\xi}} \sum_{n,k} t_{nk}^{(t)} g_y(\boldsymbol{\xi}).$$

We now summarize the workflow; the penalty acts only in Stage A (ranking), while Stage B performs unpenalized SRUW model selection by a single pass over the ranked list.

---

**Algorithm 1** High level two-stage SRUW-MNARz procedure

---

**Input:** incomplete data $\boldsymbol{Y}$, mask $\boldsymbol{C}$, model grid $(K, m)$; regularization grids $\mathcal{G}_\lambda$ (and $\mathcal{G}_\rho$).
**Stage A:** Ranking (penalized GMM on a fast-imputed data)

1. Produce a fast single imputation $\tilde{\boldsymbol{Y}}$ *only to enable ranking*.

2. For each $(\lambda, \rho) \in \mathcal{G}_\lambda \times \mathcal{G}_\rho$, fit the penalized GMM to $\bar{\boldsymbol{Y}}$ where $\bar{\boldsymbol{Y}}$ denotes the centered/scaled $\tilde{\boldsymbol{Y}}$; record $\hat{\mu}_{kd}(\lambda, \rho)$.

3. Compute ranking score $\mathcal{O}_K(d)$ for each variable $d$ by counting along the path how often $\{\hat{\mu}_{kd}(\lambda, \rho)\}_k$ remain nonzero; sort variables by $\mathcal{O}_K(d)$ (see [13] for details).

**Stage B:** Role assignment (SRUW on *original* incomplete data)

1. Traverse ranked variables once (similar to [13]); at each step, fit unpenalized SRUW-MNARz on incomplete $\boldsymbol{Y}$ and decide the role (S/U/W) of the new variable using BIC criterion using Equation (2) as in [38]

2. Return $(\hat{\mathbb{S}}, \hat{\mathbb{R}}, \hat{\mathbb{U}}, \hat{\mathbb{W}})$, $\hat{K}, \hat{m}$, and parameter estimates.

**Output:** final role sets and estimates.

---

**Practical Initialization.** Like all mixture EM algorithms, ours is sensitive to initialization. We use a robust warm start: (i) fast single imputation $\tilde{\boldsymbol{Y}}$ (only for ranking); (ii) $K$-means++/hierarchical clustering/random start on $\tilde{\boldsymbol{Y}}$ for $(\pi_k, \boldsymbol{\mu}_k, \boldsymbol{\Sigma}_k)$; (iii) given initial partitions from step (ii) $\boldsymbol{\Psi}_k^{(0)}$ set to diagonal estimates using samples assigned to each cluster (cluster-aware estimates); (iv) $\boldsymbol{\psi}_k$ initialized from class-wise missing rates after a soft E-step. We then run a $\lambda$-path (as illustrated below) for Stage A, followed by unpenalized SRUW estimation for Stage B. Per-$K$ penalty grids are constructed by data-dependent upper bounds and a geometric path: For $\lambda$, with the hard partition $\boldsymbol{Z}^{(0)}$ from initialization,

$$\lambda_{\max} = \max_{k \in [K],\, d \in [D]} \left| (\boldsymbol{Z}^{(0)\top} \boldsymbol{X})_{kd} \right|, \qquad \lambda_\ell = \lambda_{\max} \left( \frac{\lambda_{\min}}{\lambda_{\max}} \right)^{\frac{\ell-1}{L-1}}, \ \lambda_{\min} = \xi \lambda_{\max}, \ \ell = [L].$$

This matches the KKT threshold at which the $\ell_1$ penalty zeros all $\boldsymbol{\mu}_k$ entries. For $\rho$, let $\boldsymbol{S}_k$ be the (hard-label) empirical covariance and note our Stage-A glasso objective uses the coefficient $(2\rho/n_k)$ in front of the weighted $\ell_1$ term. A diagonal-forcing threshold is

$$\rho_{\max} = \max_k \max_{i \neq j} \frac{n_k \left| (\boldsymbol{S}_k)_{ij} \right|}{(\boldsymbol{P}_k)_{ij}},$$

followed by the same $L$-point geometric path $\rho_\ell = \rho_{\max}(\rho_{\min}/\rho_{\max})^{(\ell-1)/(L-1)}$ with $\rho_{\min} = \xi \rho_{\max}$. We set $\mathrm{diag}(\boldsymbol{P}_k) = \boldsymbol{0}$ when we do not impose a penalty on diagonal entries.

# 4   Theoretical Properties

In this section, we present theoretical guarantees for our extended framework, including identifiability and selection consistency under the MNARz mechanism, and extend the results to the general $\mathbb{SRUW}$ model. We also show that these guarantees apply to the regularized approach of [13]. Given the known issues in finite mixture models, such as degeneracy and non-identifiability, we adopt standard assumptions, largely consistent with those in [38]. For any configuration $(\mathbb{S}, \mathbb{R}, \mathbb{U}, \mathbb{W})$, let $\boldsymbol{\theta}^*_{(\mathbb{S}, \mathbb{R}, \mathbb{U}, \mathbb{W})}$ denote the true parameter and $\hat{\boldsymbol{\theta}}_{(\mathbb{S}, \mathbb{R}, \mathbb{U}, \mathbb{W})}$ its maximum likelihood estimator.

**Theorem 1** (Informal:   Identifiability of the $\mathbb{SRUW}$ Model). *Let $(K, m, r, l, \mathbb{V})$ and $(K^*, m^*, r^*, l^*, \mathbb{V}^*)$ denote two models under the MNARz mechanism.   Let $\boldsymbol{\Theta}_{(K, m, r, l, \mathbb{V})} \subseteq \boldsymbol{\Upsilon}_{(K, m, r, l, \mathbb{V})}$ denote the parameter space such that each element $\boldsymbol{\theta} = (\boldsymbol{\alpha}, \boldsymbol{a}, \boldsymbol{\beta}, \boldsymbol{\Omega}, \boldsymbol{\gamma}, \boldsymbol{\Gamma})$ satisfies the following: (i) the component-specific parameters $(\boldsymbol{\mu}_k, \boldsymbol{\Sigma}_k)$ are distinct and satisfy, for all $s \subseteq \mathbb{S}$, there exist $1 \le k < k' \le K$ such that $\boldsymbol{\mu}_{k, \bar{s}|s} \ne \boldsymbol{\mu}_{k', \bar{s}|s}$   or   $\boldsymbol{\Sigma}_{k, \bar{s}|s} \ne \boldsymbol{\Sigma}_{k', \bar{s}|s}$   or   $\boldsymbol{\Sigma}_{k, \bar{s}\bar{s}|s} \ne \boldsymbol{\Sigma}_{k', \bar{s}\bar{s}|s}$, where $\bar{s}$ denotes the complement of $s$ in $\mathbb{S}$; (ii) if $\mathbb{U} \ne \emptyset$, then for every $j \in \mathbb{R}$, there exists $u \in \mathbb{U}$ such that $\boldsymbol{\beta}_{uj} \ne 0$, and each $u \in \mathbb{U}$ affects at least one $j \in \mathbb{R}$ with $\boldsymbol{\beta}_{uj} \ne 0$; and (iii) the parameters $\boldsymbol{\Gamma}$ and $\boldsymbol{\gamma}$ exactly respect the structural forms $r$ and $l$, respectively. If there exist $\boldsymbol{\theta} \in \boldsymbol{\Theta}_{(K, m, r, l, \mathbb{V})}$ and $\boldsymbol{\theta}^* \in \boldsymbol{\Theta}_{(K^*, m^*, r^*, l^*, \mathbb{V}^*)}$ such that $f(\cdot \mid K, m, r, l, \mathbb{V}, \boldsymbol{\theta}) = f(\cdot \mid K^*, m^*, r^*, l^*, \mathbb{V}^*, \boldsymbol{\theta}^*)$, then $(K, m, r, l, \mathbb{V}) = (K^*, m^*, r^*, l^*, \mathbb{V}^*)$ and $\boldsymbol{\theta} = \boldsymbol{\theta}^*$.*

**Theorem 2** (Informal: BIC consistency for the $\mathbb{SRUW}$ model). *Assume the MNARz mechanism, the regularity conditions specified in the Supplementary Material, and the existence of a unique tuple $(K_0, m_0, r_0, l_0, \mathbb{V}_0)$ such that the data-generating distribution satisfies $h = f\left(\cdot \mid \boldsymbol{\theta}^*_{(K_0, m_0, r_0, l_0, \mathbb{V}_0)}\right)$ for some true parameter $\boldsymbol{\theta}^*$ where $h$ is the density function of the sample $\boldsymbol{Y}$ and the model specification $(K_0, m_0, r_0, l_0, \mathbb{V}_0)$ is assumed to be known. Let $(K_0, m_0, r_0, l_0)$ be fixed. Then the variable selection procedure that selects the subsets $(\hat{\mathbb{S}}, \hat{\mathbb{R}}, \hat{\mathbb{U}}, \hat{\mathbb{W}})$ by maximizing the BIC criterion is consistent, in the sense that $\mathbb{P}\left((\hat{\mathbb{S}}, \hat{\mathbb{R}}, \hat{\mathbb{U}}, \hat{\mathbb{W}}) = (\mathbb{S}_0, \mathbb{R}_0, \mathbb{U}_0, \mathbb{W}_0)\right) \xrightarrow[N \to \infty]{} 1$.*

**Theorem 3** (Informal: Selection consistency of the two-stage procedure). *Under certain regularity conditions in Supplementary Material, the backward stepwise selection procedure in the $\mathbb{SRUW}$ model, guided by the variable ranking stage, consistently recovers the true relevant variable set $\mathbb{S}_0^*$ with high probability. That is, $\mathbb{P}(\hat{\mathbb{S}}^* = \mathbb{S}_0^*) \xrightarrow[N \to \infty]{} 1$.*

**Proof sketch.** The proof of Theorem 3 proceeds in three main steps: 1) We first establish the consistency of the penalized M-estimator used in the variable ranking stage, specifically the penalized GMMs. 2) This consistency implies that the variables are ranked in a manner that, with high probability, separates relevant variables from noise variables. 3) We then show that the subsequent BIC-based role determination step, when applied to this consistent ranking, correctly recovers the true variable roles. A key component of the proof involves establishing consistency in parameter estimation. In particular, we derive a Restricted Strong Convexity (RSC)-type condition for the negative GMM log-likelihood. More precisely, let $\mathcal{L}_N(\boldsymbol{\alpha})$ denote the empirical objective function based on sample size $N$, and let $\boldsymbol{\alpha}^\star$ be the true parameter. There exist constants $\gamma_1, \gamma_2 > 0$ such that for every perturbation $\boldsymbol{\Delta}$ in a restricted neighborhood of $\boldsymbol{\alpha}^\star$, $\mathcal{L}_N(\boldsymbol{\alpha}^\star + \boldsymbol{\Delta}) - \mathcal{L}_N(\boldsymbol{\alpha}^\star) - \langle \nabla \mathcal{L}_N(\boldsymbol{\alpha}^\star), \boldsymbol{\Delta} \rangle \ge \frac{\gamma_1}{2} \|\boldsymbol{\Delta}_{\boldsymbol{\mu}}\|_2^2 + \frac{\gamma_2}{2} \|\boldsymbol{\Delta}_{\boldsymbol{\Sigma}^{-1}}\|_F^2 - \tau_N(\boldsymbol{\Delta})$, where $\tau_N(\boldsymbol{\Delta})$ is a tolerance term that decays at rate $N^{-1/2}$. A complete and rigorous proof is provided in the Supplementary Material.

**Implications of the theorems.** In the framework of [13], the parameter $c$ in the BIC-based selection step (representing the number of consecutive non-positive BIC differences allowed before terminating inclusion into $\mathbb{S}$ or $\mathbb{W}$) governs the balance between false negatives (omitting true variables) and false positives (including irrelevant variables). Theorem 3 implies that for finite samples, choosing a moderately large $c$, such as $c \approx \log(D)$ (e.g., 3 to 5, as commonly practiced), provides robustness against spurious BIC fluctuations caused by noise variables. In particular, if the gap in the $\mathcal{O}_K(d)$ scores between the last true signal and the first noise variable is sufficiently pronounced, the exact value of $c$ is not overly critical for ensuring asymptotic consistency, as long as it adequately accounts for statistical noise in the BIC differences sequence. However, an excessively large $c$ could lead to the inclusion of noise variables, especially if the variable ranking is imperfect.

# 5 Experiments

Our primary objective in this section is to evaluate the clustering quality, imputation accuracy, as well as variable-role recovery under incomplete data scenarios. For the simulated dataset, we compare our method with its direct predecessors in [37, 13], denoting them as `Clustvarsel` and `Selvar` respectively, and design each of them to be a pre-imputation+clustering pipeline with the imputation method to be Random Forest [10] using missRanger package [40, 63]. For [37], we use the algorithm in [59] with forward direction and headlong process, with rationale being explained in [13].

We also benchmark with `VarSelLCM` illustrated in [35] since the model itself also recasts the variable selection problem into a model-selection one. For this case study, we adopt the same real high-dimensional dataset on transcriptome as in [37, 39, 13] and interpret the result in comparison with previous findings in [37, 39]. To measure imputation accuracy, we use WNRMSE - a weighted version of NRMSE where the weights are deduced from the proportion of missingness in each group, while clustering performance is measured by ARI. The variable selection will be assessed by the frequency the model chooses the correct set of relevant variables, which are controlled in the simulation.

**Simulated Dataset.** We generate two synthetic data sets with sample size $n = 2000$ for variable/model selection with missing values treatment benchmarking. The first data set mimics [39] where each observation $\boldsymbol{y}_n \in \mathbb{R}^7$ is drawn from a four-component Gaussian mixture on the clustering block $\mathbb{S} = \{\boldsymbol{y}_1, \boldsymbol{y}_2, \boldsymbol{y}_3\}$. The components are equiprobable and have means $(0, 0, 0)^{'}$, $(-6, 6, 0)^{'}$, $(0, 0, 6)^{'}$, $(-6, 6, 6)^{'}$ with a diagonal covariance matrix $\boldsymbol{\Sigma} = \text{diag}(6\sqrt{2}, 1, 2)\, \sigma_{\text{scale}}^2$. The redundant block $\mathbb{U} = \{\boldsymbol{y}_4, \boldsymbol{y}_5\}$ is generated by a linear regression $\mathbb{U} = (-1, 2)^{'} + \mathbf{X}_S[(0.5, 2)^{'}, (1, 0)^{'}] + \boldsymbol{\epsilon}$, where $\boldsymbol{\epsilon} \sim \mathcal{N}\big(\mathbf{0}, \mathbf{R}(\pi/\mathbf{6})\text{diag}(\mathbf{1}, \mathbf{3})\mathbf{R}(\pi/\mathbf{6})^{'}\big)$ and $\mathbf{R}(\boldsymbol{\theta})$ denotes the usual planar rotation. The remaining $W$ variables are iid $\mathcal{N}(0, 1)$ noise. In the second design, an observation $\boldsymbol{y}_n \in \mathbb{R}^{14}$. The clustering block now consists of $(\boldsymbol{y}_1, \boldsymbol{y}_2)$, drawn from an equiprobable four-component Gaussian mixture with means $(0, 0), (4, 0), (0, 2), (4, 2)$ and common covariance $0.5\,\mathbf{I_2}$. Conditional on $(\boldsymbol{y}_1, \boldsymbol{y}_2)$, the vector $\boldsymbol{y}_{3:14}$ follows a linear model $\boldsymbol{y}_{3:14} = \boldsymbol{\alpha} + (\boldsymbol{y}_1, \boldsymbol{y}_2)\boldsymbol{\beta} + \boldsymbol{\varepsilon}$, where $\boldsymbol{\alpha}$, the $2 \times 12$ coefficient matrix $\boldsymbol{\beta}$ and the diagonal covariance $\boldsymbol{\Omega}$ are varied across eight sub-scenarios: scenario 1 contains only pure noise, scenarios 2-7 introduce more regressors, and scenario 8 adds intercept shifts together with three additional pure-noise variables $\boldsymbol{y}_{12:14}$. In our setting, we consider $K \in \{2, 3, 4\}$ and every method except `VarSelLCM` is allowed to learn its own covariance structure. We vary the missing ratio $\{0.05, 0.1, 0.2, 0.3, 0.5\}$ under both MAR and MNAR patterns. For the second simulated dataset, we temporarily consider only scenario 8. Moreover, for `Selvar` and `SelvarMNARz`, we fix $c = 2$ and compute $\boldsymbol{P}_k$ with the spectral distance. Further details of the synthetic-data generation scheme are provided in the supplement. From Figure 1, we observe that `SelvarMNARz` delivers the highest ARI and the lowest WNRMSE across every missing-data level in both scenarios. Its performance declines only modestly as missingness increases, while the impute-then-cluster baselines deteriorate sharply, especially at 30% and 50% missingness under MNAR, pointing to the benefit of modeling the missing-data mechanism within the clustering algorithm. To statistically validate that our model exhibits a significantly slower performance decline under high missingness, we conducted one-sided Welch's t-tests (Bonferroni-corrected) at the 50% MNAR level. As shown in Table 1, the ARI of `SelvarMNARz` is significantly greater than all baseline models ($p < 0.001$) across 20 replications.

Table 1: Welch's t-test for ARI (MNAR, 50% missingness, $\alpha = 0.05$).

| Model | Mean ARI | Std | Comparison | p-value | Signif. |
|---|---|---|---|---|---|
| SelvarMNARz | 0.511 | 0.052 | – | – | NA |
| Clustvarsel | 0.363 | 0.088 | vs. SelvarMNARz | $< 0.001$ | $* * * *$ |
| Selvar | 0.348 | 0.108 | vs. SelvarMNARz | $< 0.001$ | $* * * *$ |
| VarSelLCM | 0.344 | 0.101 | vs. SelvarMNARz | $< 0.001$ | $* * * *$ |

While all methods show a monotonic performance drop as missingness increases, these results confirm that the superior performance of `SelvarMNARz` is not a random artifact but a statistically significant improvement, demonstrating its robustness in the challenging missing data scenarios. Figure 2 shows that `SelvarMNARz` consistently recovers the true cluster component and the correct set of clustering variables, regardless of the missing-data mechanism or rate. On Dataset 2, every method succeeds un-

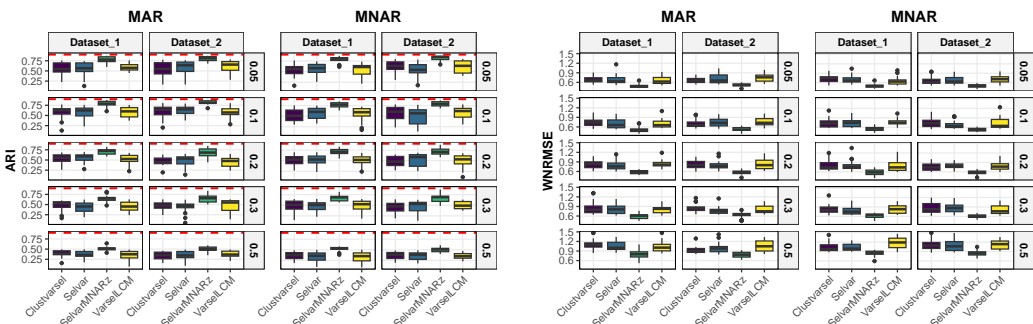

Figure 1: Comparison of four models under MAR and MNAR mechanisms over 20 replications; for ARI/WNRMSE, higher/lower boxplots indicate better performance.

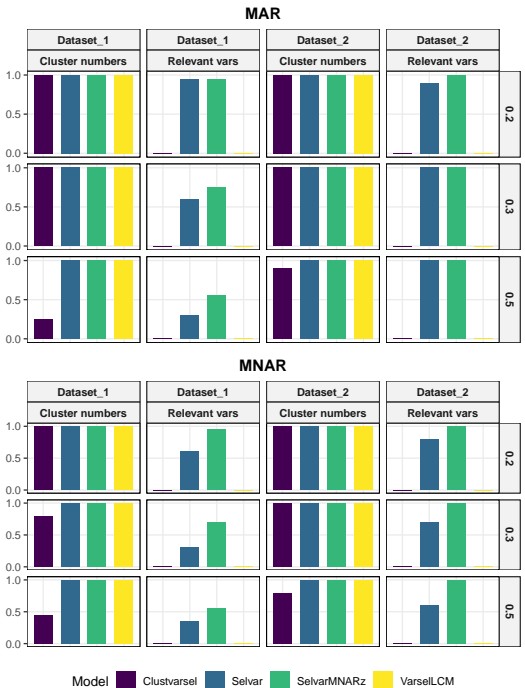

Figure 2: Proportions choosing correct relevant variables and cluster components over 20 replications.

der both MAR and MNAR, whereas on Dataset 1, the performance of `Clustvarsel` and `VarSelLCM` drops as missingness increases, leading to the omission of important variables. The contrast reflects the data-generating designs: Dataset 2 includes pure-noise variables $\mathbb{W}$ within the $\mathbb{SRUW}$ framework, whereas Dataset 1 aligns more closely with the assumptions behind `Clustvarsel` and `VarSelLCM`. Finally, workflows that rely on external imputation misidentify relevant variables once missingness exceeds 20% under either missing-data scenarios.

**Transcriptome Data.** We re-examine the 1,267-gene *Arabidopsis thaliana* transcriptome dataset previously analyzed using `SelvarClust` and `SelvarClustMV` in [37, 39] to illustrate clustering with variable selection. Full experimental details are provided in the Supplementary Material; here, we summarize the settings and key findings. We fitted `SelvarMNARz` for $K \in \{2, \dots, 20\}$ clusters with $c = 5$, spectral distance weights $\boldsymbol{P}_k$, and a $\pi_k LC$ mixture structure as in [39]. For conciseness, we refer to Project 1 as P1, Project 2 as P2, and so on. As shown in Figure 3, the proposed framework partitions the transcriptome into 18 clusters and reaffirms P1-P4 as core drivers, while globally assigning P5-P7 as being explained by these drivers. This finding is consistent with [39], which also identifies P2 as relevant for clustering, supporting the hypothesis that iron signaling responses are critical for defining co-expression groups when a broader gene set (including entries with missing

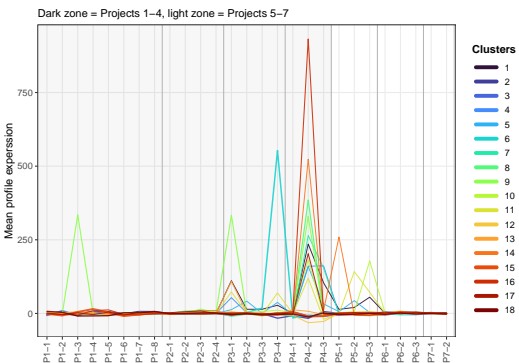

Figure 3: Mean expression profiles across 18 clusters. Light region indicates irrelevant P.

Table 2: Size of each mixture component and $R^2$ obtained by regressing the $\mathbb{U}$-block on the $\mathbb{S}$-block **within that cluster**. Values extremely close to 1 (marked by $^*$) occur because the $\mathbb{U}$-block is nearly constant in those groups.

| Cluster | # Genes | $R^2$ | Cluster | # Genes | $R^2$ | Cluster | # Genes | $R^2$ |
|---|---|---|---|---|---|---|---|---|
| 1 | 715 | 0.02 | 7 | 36 | 0.68 | 13 | 46 | 0.57 |
| 2 | 11 | $1.00^*$ | 8 | 13 | $1.00^*$ | 14 | 19 | $1.00^*$ |
| 3 | 6 | $1.00^*$ | 9 | 15 | $1.00^*$ | 15 | 6 | $1.00^*$ |
| 4 | 93 | 0.45 | 10 | 29 | 0.59 | 16 | 16 | $1.00^*$ |
| 5 | 124 | 0.12 | 11 | 71 | 0.43 | 17 | 24 | 0.90 |
| 6 | 9 | $1.00^*$ | 12 | 24 | 0.91 | 18 | 10 | $1.00^*$ |

data) is considered. Additionally, all three methods agree that P5 is not a primary clustering factor and is more likely a correlated effect explained by broader cellular processes. However, a key departure lies in the reclassification of P6 and P7, previously identified as informative clustering drivers [37, 39], into a redundant $\mathbb{U}$ block, explained by P1-P4. This suggests that, after accounting for MNAR effects, variation in P6-P7 is largely predictable from the core axes P1-P4, and P5-P7 all measure late-stage stress outputs. Further analysis of cluster sizes and the per-cluster $R^2$ values, shown in Table 2, validates the selected variables when regressing P5-P7 on P1-P4 within each cluster. Six biologically coherent clusters (6, 7, 8, 10, 12, 18) exhibit $R^2 > 0.60$, indicating that late-stress variation is well encoded by P1-P4. Cluster 1, although large, is transcriptionally flat across both $\mathbb{S}$ and $\mathbb{U}$; its low $R^2 = 0.02$ thus reflects a vanishing denominator rather than hidden structure. True residual structure (unexplained $\mathbb{U}$ signal) is concentrated in two groups: clusters 5 and 11, which are exactly where the 18th mixture component forms.

## 6 Conclusion, Limitations and Perspectives

We proposed a unified model-based clustering framework that jointly performs variable selection and handles MNAR data by combining adaptive penalization with explicit modeling of missingness-latent class dependencies. This enhances clustering flexibility and robustness in high-dimensional settings like transcriptomics. The method enjoys theoretical guarantees, including selection and asymptotic consistency, and shows strong empirical performance. However, it is currently limited to continuous data via Gaussian mixture models, making it less suitable for categorical or mixed-type variables. Extending the framework to categorical settings is a natural next step. For instance, Dean and Raftery [15] introduced variable selection for latent class models, later refined by Fop et al. [20] using a statistically grounded stepwise approach. Bontemps and Toussile [8] proposed a mixture of multivariate multinomial distributions with combinatorial variable selection and slope heuristics for penalty calibration. Building upon these, a future extension of our framework could incorporate discrete data distributions, adapt penalization accordingly, and model missingness mechanisms using latent class structures. Further directions include mixed-type data extensions and scalability improvements to accommodate larger, more heterogeneous datasets.

## Acknowledgments

TrungTin Nguyen and Christopher Drovandi acknowledge funding from the Australian Research Council Centre of Excellence for the Mathematical Analysis of Cellular Systems (CE230100001). Christopher Drovandi was also supported by an Australian Research Council Future Fellowship (FT210100260). This paper was also supported by AISIA Research Lab, Vietnam. The authors would like to express their special thanks to the authors of [37, 39], in particular Professor Cathy Maugis-Rabusseau and Dr. Marie-Laure Martin, for providing the Arabidopsis thaliana transcriptome dataset.

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

# Supplementary Materials for "A Unified Framework for Variable Selection in Model-Based Clustering with Missing Not at Random"

In this supplementary material, we first provide additional details on the main challenges as well as the technical and computational contributions in Section A, with the aim of enhancing the reader's understanding of our key theoretical and methodological developments. Next, we summarize the definitions and theoretical results for the $\mathbb{SR}$ and $\mathbb{SRUW}$ models in Section C. We then present the regularity assumptions and technical proofs for the main theoretical results in Sections C and D, respectively. Finally, we include additional experimental results and further discussion of related work in Sections F and G.

## Contents

# A   More Details on Main Challenges and Contributions

Model-based clustering with variable selection, particularly using the sophisticated SRUW framework [38], offers a powerful approach to understanding data structure by not only grouping data but also by assigning distinct functional roles to variables. However, extending this rich framework to handle missing data, especially MNAR, and to perform variable selection efficiently in high-dimensional settings presents substantial theoretical and practical challenges. We address some difficulties as follows:

**Integrating MNAR inside model-based clustering with variable selection:**   Unlike MAR or MCAR data, MNAR mechanisms are generally non-ignorable, meaning the missing data process must be explicitly modeled alongside the data distribution to avoid biased parameter estimates and incorrect clustering [33, 56]. Modeling this joint distribution significantly increases model complexity and the risk of misspecification. While various MNAR models exist (see [35, 62]), integration into complex variable selection clustering frameworks like SRUW, along with proving identifiability and estimator consistency, remains a frontier. Specifically, the MNARz mechanism (where missingness depends only on latent class membership) offers a tractable yet meaningful way to handle informative missingness, but its properties within a structured model like SRUW require careful elucidation.

**Variable selection with SRUW and missing data in high-dimensional regime:**   The SRUW model's strength lies in its detailed variable role specification. However, determining these roles from data is a combinatorial model selection problem. Original SRUW procedures often rely on stepwise algorithms that become computationally prohibitive as the number of variables $D$ increases. Introducing missing data further complicates parameter estimation within each candidate model. Penalized likelihood methods, common in high-dimensional regression (e.g., LASSO [70, 75, 11]), offer a promising avenue for simultaneous parameter estimation and variable selection. However, adapting these to the GMM likelihood and then to the structured SRUW model with an integrated MNARz mechanism requires establishing theoretical guarantees such as parameter estimation consistency and ranking/selection consistency for the chosen penalized objective. Some notable, nontrivial hurdles are appropriate curvature properties like RSC for non-convex log-likelihood loss functions in high dimensions or the cone conditions to ensure sparse recovery.

**Consistency of multi-stage procedures:**   Practical algorithms for variable selection tasks often involve multiple stages (e.g., initial ranking/variable screening, then detailed role assignment). Proving the consistency of such a multi-stage procedure requires advanced tools to handle.

Being aware of these difficulties, we attempt to make several contributions to address the aforementioned challenges, particularly focusing on providing a theoretical foundation for a two-stage variable selection procedure within the SRUW-MNARz framework:

1. Though the extension of the reinterpretation result from [62] is straightforward, this significantly simplifies the analysis and provides a way to handle identifiability of SRUW under MNAR missingness, which potentially facilitates algorithmic design.

2. Building on the MAR reinterpretation and existing results for SRUW data model identifiability and MNARz mechanism identifiability, we establish conditions under which the full SRUW-MNARz model parameters (including SRUW structure, GMM parameters, regression coefficients, and MNARz probabilities) are identifiable from the observed data. This is important since identifiability is a prerequisite for consistent parameter estimation.

3. To underpin the variable ranking stage, which uses a LASSO-like penalized GMM, we provide a rigorous analysis of the penalized GMM estimator based on [66, 44, 34]: First, we establish a non-asymptotic sup-norm bound on the gradient of the GMM negative log-likelihood at the true parameters, which is crucial for selecting the appropriate magnitude for regularization parameters. Subsequently, we formally state the RSC condition needed for the GMM loss function, and the error vector of the penalized GMM estimator lies in a specific cone, which is essential for sparse recovery. These settings provide the necessary

high-dimensional statistical guarantees for the penalized GMM used in the ranking stage, ensuring that its estimated sparse means are reliable.

4. Leveraging the parameter consistency of the penalized GMM, we prove that the variable ranking score $\mathcal{O}_K(j)$ can consistently distinguish between truly relevant variables (for clustering means) and irrelevant variables. This involves establishing precise "signal strength" conditions; thereby, validating the first step of the two-stage variable selection procedure in [13].

5. For the main theorem on selection consistency, the process combines ranking consistency with an analysis of the BIC-based stepwise decisions for assigning roles. We show that under appropriate conditions on local BIC decision error rates and the choice of the stopping parameter $c$, the entire two-stage procedure consistently recovers the true variable partition. This provides the first (to our knowledge) formal consistency proof for such a two-stage variable selection in model-based clustering (SRUW) that incorporates an initial LASSO-like ranking, especially in the context of MNAR pattern. It bridges the theoretical gap between the computationally intensive and a more practical high-dimensional strategy.

6. While full optimality of $c$ is not derived, we provide a condition for choosing $c$ to achieve a target false positive rate in relevant variable selection, given the single-decision error probability. This moves beyond purely heuristic choices for $c$ in [13], paving the way for more adaptive or theoretically grounded choices of $c$ in practice. This offers a more principled approach to selecting this single hyperparameter in the algorithm, enhancing the reliability of the selection procedure.

## B   Preliminary of $\mathbb{SR}$ and $\mathbb{SRUW}$ Models

This part summarizes definitions and theoretical results of the $\mathbb{SR}$ and $\mathbb{SRUW}$ models in [37, 38]. We start with the following two definitions:

**Definition 1** ($\mathbb{SR}$ Model for Complete Data). *Let $\boldsymbol{y}_n \in \mathbb{R}^D$ be the $n$-th observation, for $n = 1, \ldots, N$. A model $\mathcal{M}_{\mathbb{SR}} = (K, m, \mathbb{S}, \mathbb{R})$ is defined by:*

1. *$K$: The number of mixture components (clusters).*

2. *$m$: The covariance structure of the GMM for variables in $\mathbb{S}$.*

3. *$\mathbb{S} \subseteq \{1, \ldots, D\}$: The non-empty set of relevant clustering variables.*

4. *$\mathbb{R} \subseteq \mathbb{S}$: The subset of relevant variables in $\mathbb{S}$ used to explain the irrelevant variables $\mathbb{S}^c = \{1, \ldots, D\} \setminus \mathbb{S}$ via linear regression.*

*The parameters of the model are $\boldsymbol{\theta} = (\boldsymbol{\alpha}, \boldsymbol{a}, \boldsymbol{\beta}, \boldsymbol{\Omega})$, where $\boldsymbol{\alpha}$ are the GMM parameters for variables in $\mathbb{S}$. $\boldsymbol{a}$ is the intercept vector, $\boldsymbol{\beta}$ is the matrix of regression coefficients for explaining $\boldsymbol{y}^{\mathbb{S}^c}$ from $\boldsymbol{y}^{\mathbb{R}}$, and $\boldsymbol{\Omega}$ is the covariance matrix of the regression component. The complete data likelihood for one observation $\boldsymbol{y}_n$ is:*

$$f(\boldsymbol{y}_n; K, m, \mathbb{S}, \mathbb{R}, \boldsymbol{\theta}) = f_{clust}(\boldsymbol{y}_n^{\mathbb{S}}; K, m, \boldsymbol{\alpha}) f_{reg}(\boldsymbol{y}_n^{\mathbb{S}^c}; \boldsymbol{a} + \boldsymbol{y}_n^{\mathbb{R}} \boldsymbol{\beta}, \boldsymbol{\Omega})$$

*where $\boldsymbol{y}_n^{\mathbb{S}}$ denotes the sub-vector of $\boldsymbol{y}_n$ corresponding to variables in $\mathbb{S}$, and similarly for $\boldsymbol{y}_n^{\mathbb{S}^c}$ and $\boldsymbol{y}_n^{\mathbb{R}}$. $f_{clust}$ is a K-component GMM density, and $f_{reg}$ is a multivariate Gaussian density.*

*Importantly, $\mathbb{SR}$ model can be equivalently written as a single K-component GMM for the full data vector $\boldsymbol{y}_n \in \mathbb{R}^D$ [37]:*

$$f(\boldsymbol{y}_n; K, m, \mathbb{S}, \mathbb{R}, \boldsymbol{\theta}) = \sum_{k=1}^{K} \pi_k \Phi(\boldsymbol{y}_n; \boldsymbol{\nu}_k, \boldsymbol{\Delta}_k) \tag{8}$$

*where $\Phi(\cdot; \boldsymbol{\nu}, \boldsymbol{\Delta})$ is the multivariate Gaussian density with mean $\boldsymbol{\nu}$ and covariance $\boldsymbol{\Delta}$. The parameters $(\boldsymbol{\nu}_k, \boldsymbol{\Delta}_k)$ are constructed from the original $\mathbb{SR}$ parameters $(\boldsymbol{\alpha}_k, \boldsymbol{a}, \boldsymbol{\beta}, \boldsymbol{\Omega})$ as follows: Let $\boldsymbol{\Lambda}$ be a $|\mathbb{S}| \times |\mathbb{S}^c|$ matrix derived from $\boldsymbol{\beta}$ (which is $|\mathbb{R}| \times |\mathbb{S}^c|$). For a variable $j \in \mathbb{S}$ and $l \in \mathbb{S}^c$, $\Lambda_{jl} = \beta_{j'l}$ if the j-th variable of $\mathbb{S}$ is the j'-th variable of $\mathbb{R}$, and $\Lambda_{jl} = 0$ if the j-th variable of $\mathbb{S}$ is in $\mathbb{S} \setminus \mathbb{R}$. Then, for each component k:*

- *The mean vector $\boldsymbol{\nu}_k \in \mathbb{R}^D$ has elements:*

$$\nu_{kj} = \begin{cases} \mu_{kj} & \text{if variable } j \in \mathbb{S} \\ (\boldsymbol{a} + \boldsymbol{\mu}_k^{\mathbb{S}}\boldsymbol{\Lambda})_j & \text{if variable } j \in \mathbb{S}^c \end{cases}$$

*where $\boldsymbol{\mu}_k^{\mathbb{S}}$ is the mean of the $k$-th component for variables in $\mathbb{S}$.*

- *The covariance matrix $\boldsymbol{\Delta}_k \in \mathbb{R}^{D \times D}$ has blocks:*

$$\Delta_{k,jl} = \begin{cases} \Sigma_{k,jl} & \text{if } j \in \mathbb{S}, l \in \mathbb{S} \\ (\boldsymbol{\Sigma}_k\boldsymbol{\Lambda})_{jl} & \text{if } j \in \mathbb{S}, l \in \mathbb{S}^c \\ (\boldsymbol{\Lambda}^T\boldsymbol{\Sigma}_k)_{jl} & \text{if } j \in \mathbb{S}^c, l \in \mathbb{S} \\ (\boldsymbol{\Omega} + \boldsymbol{\Lambda}^T\boldsymbol{\Sigma}_k\boldsymbol{\Lambda})_{jl} & \text{if } j \in \mathbb{S}^c, l \in \mathbb{S}^c \end{cases}$$

**Definition 2** ($\mathbb{SRUW}$ Model for Complete Data). *Let $\boldsymbol{y}_n \in \mathbb{R}^D$ be the $n$-th observation. An $\mathbb{SRUW}$ model $\mathcal{M}_{\mathbb{SRUW}} = (K, m, r, l, \mathbb{V})$ is defined by:*

1. *$K$: The number of mixture components.*

2. *$m$: The covariance structure of the GMM for variables in $\mathbb{S}$.*

3. *$r$: The form of the covariance matrix $\boldsymbol{\Omega}$ for the regression of $\mathbb{U}$ on $\mathbb{R}$.*

4. *$l$: The form of the covariance matrix $\boldsymbol{\Gamma}$ for the independent variables $\mathbb{W}$.*

5. *$\mathbb{V} = (\mathbb{S}, \mathbb{R}, \mathbb{U}, \mathbb{W})$: A partition of the $D$ variables, where*

   - *$\mathbb{S}$: Non-empty set of relevant clustering variables.*
   - *$\mathbb{R} \subseteq \mathbb{S}$: Subset of $\mathbb{S}$ regressing variables in $\mathbb{U}$. ($\mathbb{R} = \emptyset$ if $\mathbb{U} = \emptyset$).*
   - *$\mathbb{U}$: Set of irrelevant variables explained by $\mathbb{R}$.*
   - *$\mathbb{W}$: Set of irrelevant variables independent of all variables in $\mathbb{S}$.*
   - *$\mathbb{S} \cup \mathbb{U} \cup \mathbb{W} = \{1, \dots, D\}$ and these sets are disjoint.*

*The parameters are $\boldsymbol{\theta} = (\boldsymbol{\alpha}, \boldsymbol{a}, \boldsymbol{\beta}, \boldsymbol{\Omega}, \boldsymbol{\gamma}, \boldsymbol{\Gamma})$, where $\boldsymbol{\alpha}$ is the GMM parameters. The complete data density for one observation $\boldsymbol{y}_n$ is:*

$$f(\boldsymbol{y}_n; K, m, r, l, \mathbb{V}, \boldsymbol{\theta}) = f_{clust}(\boldsymbol{y}_n^{\mathbb{S}}; K, m, \boldsymbol{\alpha}) f_{reg}(\boldsymbol{y}_n^{\mathbb{U}}; \boldsymbol{a} + \boldsymbol{y}_n^{\mathbb{R}}\boldsymbol{\beta}, \boldsymbol{\Omega}) f_{indep}(\boldsymbol{y}_n^{\mathbb{W}}; \boldsymbol{\gamma}, \boldsymbol{\Gamma})$$

*This can be written as a $K$-component GMM for the full data vector $\boldsymbol{y}_n \in \mathbb{R}^D$:*

$$f(\boldsymbol{y}_n; K, m, r, l, \mathbb{V}, \boldsymbol{\theta}) = \sum_{k=1}^{K} \pi_k \Phi(\boldsymbol{y}_n; \tilde{\boldsymbol{\nu}}_k, \tilde{\boldsymbol{\Delta}}_k) \tag{9}$$

*where $\tilde{\boldsymbol{\nu}}_k$ and $\tilde{\boldsymbol{\Delta}}_k$ are constructed from $\boldsymbol{\theta}$. Let $(\boldsymbol{\nu}_k^{(\mathbb{S},\mathbb{U})}, \boldsymbol{\Delta}_k^{(\mathbb{S},\mathbb{U})})$ be the mean and covariance for the $(\mathbb{S}, \mathbb{U})$ part derived as in the $\mathbb{SR}$ model (where $\mathbb{U}$ plays the role of $\mathbb{S}_{\mathbb{SR}}^c$). Then:*

$$\tilde{\boldsymbol{\nu}}_k = \left( (\boldsymbol{\nu}_k^{(\mathbb{S},\mathbb{U})})^T, \boldsymbol{\gamma}^T \right)^T$$

$$\tilde{\boldsymbol{\Delta}}_k = \begin{pmatrix} \boldsymbol{\Delta}_k^{(\mathbb{S},\mathbb{U})} & \boldsymbol{0} \\ \boldsymbol{0} & \boldsymbol{\Gamma} \end{pmatrix}$$

*due to the independence of $\mathbb{W}$ from $\mathbb{S}$ and $\mathbb{U}$ (given the cluster $k$, which is implicit in the construction of $\boldsymbol{\nu}_k^{(\mathbb{S},\mathbb{U})}$ using $\boldsymbol{\mu}_k^{\mathbb{S}}$).*

Similar to the global GMM representation of $\mathbb{SR}$ model under MAR, we can obtain such a global form of $\mathbb{SRUW}$ model under the same missingness pattern, which is illustrated by the following claim.

**Proposition 1.** *Under MAR, $\mathbb{SRUW}$ has a natural global GMM representation with appropriate parameters.*

*Proof of Proposition 1.* Let $\boldsymbol{y}_n = (\boldsymbol{y}_n^{\mathbb{S}}, \boldsymbol{y}_n^{\mathbb{U}}, \boldsymbol{y}_n^{\mathbb{W}})$ be the complete data vector for observation $n \in \{1, \ldots, N\}$, partitioned according to the $\mathbb{SRUW}$ model structure $\mathbb{V} = (\mathbb{S}, \mathbb{R}, \mathbb{U}, \mathbb{W})$. The parameters are $\boldsymbol{\theta} = (\boldsymbol{\alpha}, \boldsymbol{a}, \boldsymbol{\beta}, \boldsymbol{\Omega}, \boldsymbol{\gamma}, \boldsymbol{\Gamma})$, where $\boldsymbol{\alpha} = (\pi_1, \ldots, \pi_k, \boldsymbol{\mu}_1, \ldots, \boldsymbol{\mu}_K, \boldsymbol{\Sigma}_1, \ldots, \boldsymbol{\Sigma}_K)$.

The complete-data density for a single observation $\boldsymbol{y}_n$, given its membership to cluster $k$ (denoted by $z_{ik} = 1$), is:

$$f(\boldsymbol{y}_n \mid z_{ik} = 1, \mathbb{V}, \boldsymbol{\theta}) = f_{\text{clust}}(\boldsymbol{y}_n^{\mathbb{S}}; \boldsymbol{\mu}_k, \boldsymbol{\Sigma}_k) \cdot f_{\text{reg}}(\boldsymbol{y}_n^{\mathbb{U}}; \boldsymbol{a} + \boldsymbol{y}_n^{\mathbb{R}}\boldsymbol{\beta}, \boldsymbol{\Omega}) \cdot f_{\text{indep}}(\boldsymbol{y}_n^{\mathbb{W}}; \boldsymbol{\gamma}, \boldsymbol{\Gamma})$$

The marginal complete-data density for $\boldsymbol{y}_n$, by summing over latent cluster memberships is:

$$f(\boldsymbol{y}_n; \mathbb{V}, \boldsymbol{\theta}) = \sum_{k=1}^{K} \pi_k \left[ f_{\text{clust}}(\boldsymbol{y}_n^{\mathbb{S}}; \boldsymbol{\mu}_k, \boldsymbol{\Sigma}_k) \cdot f_{\text{reg}}(\boldsymbol{y}_n^{\mathbb{U}}; \boldsymbol{a} + \boldsymbol{y}_n^{\mathbb{R}}\boldsymbol{\beta}, \boldsymbol{\Omega}) \cdot f_{\text{indep}}(\boldsymbol{y}_n^{\mathbb{W}}; \boldsymbol{\gamma}, \boldsymbol{\Gamma}) \right] \qquad (10)$$

Due to the independence of $\boldsymbol{y}_n^{\mathbb{W}}$ from $(\boldsymbol{y}_n^{\mathbb{S}}, \boldsymbol{y}_n^{\mathbb{U}})$ given the cluster $k$ and its parameters $\boldsymbol{\gamma}, \boldsymbol{\Gamma}$ are not $k$-specific, we can rewrite the term inside the sum as follows: suppose that $f_k(\boldsymbol{y}_n^{\mathbb{S}}, \boldsymbol{y}_n^{\mathbb{U}}) = \Phi(\boldsymbol{y}_n^{\mathbb{S}}; \boldsymbol{\mu}_k, \boldsymbol{\Sigma}_k) \cdot \Phi(\boldsymbol{y}_n^{\mathbb{U}}; \boldsymbol{a} + \boldsymbol{y}_n^{\mathbb{R}}\boldsymbol{\beta}, \boldsymbol{\Omega})$. This is the density of $(\boldsymbol{y}_n^{\mathbb{S}}, \boldsymbol{y}_n^{\mathbb{U}})$ given cluster $k$. Then, Equation (10) becomes:

$$f(\boldsymbol{y}_n; \mathbb{V}, \boldsymbol{\theta}) = \left( \sum_{k=1}^{K} \pi_k f_k(\boldsymbol{y}_n^{\mathbb{S}}, \boldsymbol{y}_n^{\mathbb{U}}) \right) \cdot f_{\text{indep}}(\boldsymbol{y}_n^{\mathbb{W}}; \boldsymbol{\gamma}, \boldsymbol{\Gamma})$$

The term $\Phi(\boldsymbol{y}_n^{\mathbb{W}}; \boldsymbol{\gamma}, \boldsymbol{\Gamma})$ factors out of the sum over $k$ because $\boldsymbol{\gamma}$ and $\boldsymbol{\Gamma}$ are not indexed by $k$.

Let $\boldsymbol{y}_n = (\boldsymbol{y}_n^o, \boldsymbol{y}_n^m)$ be the partition of $\boldsymbol{y}_n$ into observed and missing parts for observation $n$. The missing parts are $\boldsymbol{y}_n^m = (\boldsymbol{y}_n^{\mathbb{S} \cap \mathbb{M}_n}, \boldsymbol{y}_n^{\mathbb{U} \cap \mathbb{M}_n}, \boldsymbol{y}_n^{\mathbb{W} \cap \mathbb{M}_n})$, where $\mathbb{M}_n$ denotes the set of missing variable indices for observation $n$. The observed-data density for observation $n$ is obtained by integrating $f(\boldsymbol{y}_n; \mathbb{V}, \boldsymbol{\theta})$ over $\boldsymbol{y}_n^m$:

$$f(\boldsymbol{y}_n^o; \mathbb{V}, \boldsymbol{\theta}) = \int f(\boldsymbol{y}_n^o, \boldsymbol{y}_n^m; \mathbb{V}, \boldsymbol{\theta}) \, d\boldsymbol{y}_n^m$$

$$= \int \left[ \left( \sum_{k=1}^{K} \pi_k f_k(\boldsymbol{y}_n^{\mathbb{S} \cap \mathbb{O}_n}, \boldsymbol{y}_n^{\mathbb{S} \cap \mathbb{M}_n}, \boldsymbol{y}_n^{\mathbb{U} \cap \mathbb{O}_n}, \boldsymbol{y}_n^{\mathbb{U} \cap \mathbb{M}_n}) \right) \right.$$
$$\left. \times f_{\text{indep}}(\boldsymbol{y}_n^{\mathbb{W} \cap \mathbb{O}_n}, \boldsymbol{y}_n^{\mathbb{W} \cap \mathbb{M}_n}; \boldsymbol{\gamma}, \boldsymbol{\Gamma}) \right] d\boldsymbol{y}_n^{\mathbb{S} \cap \mathbb{M}_n} \, d\boldsymbol{y}_n^{\mathbb{U} \cap \mathbb{M}_n} \, d\boldsymbol{y}_n^{\mathbb{W} \cap \mathbb{M}_n}$$

where $\mathbb{O}_n$ denotes the set of observed variable indices for observation $n$. We can swap the sum and the integral:

$$f(\boldsymbol{y}_n^o; \mathbb{V}, \boldsymbol{\theta}) = \sum_{k=1}^{K} \pi_k \int \left[ f_k(\boldsymbol{y}_n^{\mathbb{S} \cap \mathbb{O}_n}, \boldsymbol{y}_n^{\mathbb{S} \cap \mathbb{M}_n}, \boldsymbol{y}_n^{\mathbb{U} \cap \mathbb{O}_n}, \boldsymbol{y}_n^{\mathbb{U} \cap \mathbb{M}_n}) \right.$$
$$\left. \times f_{\text{indep}}(\boldsymbol{y}_n^{\mathbb{W} \cap \mathbb{O}_n}, \boldsymbol{y}_n^{\mathbb{W} \cap \mathbb{M}_n}; \boldsymbol{\gamma}, \boldsymbol{\Gamma}) \right] d\boldsymbol{y}_n^{\mathbb{S} \cap \mathbb{M}_n} \, d\boldsymbol{y}_n^{\mathbb{U} \cap \mathbb{M}_n} \, d\boldsymbol{y}_n^{\mathbb{W} \cap \mathbb{M}_n}$$

Since $f_k(\cdot)$ only involves variables in $\mathbb{S}$ and $\mathbb{U}$, and $f_{\text{clust}}(\cdot; \boldsymbol{\gamma}, \boldsymbol{\Gamma})$ only involves variables in $\mathbb{W}$, and these sets are disjoint, the integration can be separated:

$$f(\boldsymbol{y}_n^o; \mathbb{V}, \boldsymbol{\theta}) = \sum_{k=1}^{K} \pi_k \left[ \int f_k(\boldsymbol{y}_n^{\mathbb{S} \cap \mathbb{O}_n}, \boldsymbol{y}_n^{\mathbb{S} \cap \mathbb{M}_n}, \boldsymbol{y}_n^{\mathbb{U} \cap \mathbb{O}_n}, \boldsymbol{y}_n^{\mathbb{U} \cap \mathbb{M}_n}) \, d\boldsymbol{y}_n^{\mathbb{S} \cap \mathbb{M}_n} \, d\boldsymbol{y}_n^{\mathbb{U} \cap \mathbb{M}_n} \right]$$
$$\times \left[ \int f_{\text{indep}}(\boldsymbol{y}_n^{\mathbb{W} \cap \mathbb{O}_n}, \boldsymbol{y}_n^{\mathbb{W} \cap \mathbb{M}_n}; \boldsymbol{\gamma}, \boldsymbol{\Gamma}) \, d\boldsymbol{y}_n^{\mathbb{W} \cap \mathbb{M}_n} \right]$$

Let $f_k(\boldsymbol{y}_n^{(\mathbb{S}, \mathbb{U}) \cap \mathbb{O}_n}) = \int f_k(\boldsymbol{y}_n^{\mathbb{S} \cap \mathbb{O}_n}, \boldsymbol{y}_n^{\mathbb{S} \cap \mathbb{M}_n}, \boldsymbol{y}_n^{\mathbb{U} \cap \mathbb{O}_n}, \boldsymbol{y}_n^{\mathbb{U} \cap \mathbb{M}_n}) \, d\boldsymbol{y}_n^{\mathbb{S} \cap \mathbb{M}_n} \, d\boldsymbol{y}_n^{\mathbb{U} \cap \mathbb{M}_n}$. This is the marginal density of the observed parts of $(\mathbb{S}, \mathbb{U})$ variables for component $k$. It is Gaussian, $\Phi(\boldsymbol{y}_n^{(\mathbb{S}, \mathbb{U}) \cap \mathbb{O}_n}; \boldsymbol{\nu}_{k,o}^{(\mathbb{S}, \mathbb{U})(n)}, \boldsymbol{\Delta}_{k,oo}^{(\mathbb{S}, \mathbb{U})(n)})$, where these parameters are derived from the complete-data $\mathbb{SR}$ parameters for the $(\mathbb{S}, \mathbb{U})$ part. Let $\Phi(\boldsymbol{y}_n^{\mathbb{W} \cap \mathbb{O}_n}; \boldsymbol{\gamma}_o^{(n)}, \boldsymbol{\Gamma}_{oo}^{(n)}) = \int \Phi(\boldsymbol{y}_n^{\mathbb{W} \cap \mathbb{O}_n}, \boldsymbol{y}_n^{\mathbb{W} \cap \mathbb{M}_n}; \boldsymbol{\gamma}, \boldsymbol{\Gamma}) \, d\boldsymbol{y}_n^{\mathbb{W} \cap \mathbb{M}_n}$. This is the marginal density of the observed parts of $\mathbb{W}$ variables. Then,

$$f(\boldsymbol{y}_n^o; \mathbb{V}, \boldsymbol{\theta}) = \left( \sum_{k=1}^{K} \pi_k f_{\text{clust}}(\boldsymbol{y}_n^{(\mathbb{S}, \mathbb{U}) \cap \mathbb{O}_n}; \boldsymbol{\nu}_{k,o}^{(\mathbb{S}, \mathbb{U})(n)}, \boldsymbol{\Delta}_{k,oo}^{(\mathbb{S}, \mathbb{U})(n)}) \right) \cdot f_{\text{indep}}(\boldsymbol{y}_n^{\mathbb{W} \cap \mathbb{O}_n}; \boldsymbol{\gamma}_o^{(n)}, \boldsymbol{\Gamma}_{oo}^{(n)}) \quad (11)$$

From Equation (11), we can get

$$\prod_{n=1}^{N} \left( \sum_{k=1}^{K} \pi_k \, f_{\text{clust}} \left( \boldsymbol{y}_n^o ; \tilde{\boldsymbol{\nu}}_{k,o}^{(n)}, \tilde{\boldsymbol{\Delta}}_{k,oo}^{(n)} \right) \right)$$

by allowing $f_{\text{indep}}(\boldsymbol{y}_n^{\mathbb{W} \cap \mathbb{O}_n}; \boldsymbol{\gamma}_o^{(n)}, \boldsymbol{\Gamma}_{oo}^{(n)})$ to be "absorbed" into the sum over $k$. So, we can write:

$$f(\boldsymbol{y}_n^o; \mathbb{V}, \boldsymbol{\theta}) = \sum_{k=1}^{K} \left( \pi_k f_{\text{clust}}(\boldsymbol{y}_n^{(\mathbb{S},\mathbb{U}) \cap \mathbb{O}_n}; \boldsymbol{\nu}_{k,o}^{(\mathbb{S},\mathbb{U})(n)}, \boldsymbol{\Delta}_{k,oo}^{(\mathbb{S},\mathbb{U})(n)}) \cdot f_{\text{indep}}(\boldsymbol{y}_n^{\mathbb{W} \cap \mathbb{O}_n}; \boldsymbol{\gamma}_o^{(n)}, \boldsymbol{\Gamma}_{oo}^{(n)}) \right)$$

Since the variables in $(\mathbb{S}, \mathbb{U})$ and $\mathbb{W}$ are conditionally independent given $k$, the product of their marginal observed densities is the marginal observed density of their union. Let $\boldsymbol{y}_n^o = (\boldsymbol{y}_n^{(\mathbb{S},\mathbb{U}) \cap \mathbb{O}_n}, \boldsymbol{y}_n^{\mathbb{W} \cap \mathbb{O}_n})$. Let $\tilde{\boldsymbol{\nu}}_{k,o}^{(n)} = \left( (\boldsymbol{\nu}_{k,o}^{(\mathbb{S},\mathbb{U})(n)})^T, (\boldsymbol{\gamma}_o^{(n)})^T \right)^T$. Let $\tilde{\boldsymbol{\Delta}}_{k,oo}^{(n)} = \begin{pmatrix} \boldsymbol{\Delta}_{k,oo}^{(\mathbb{S},\mathbb{U})(n)} & \mathbf{0} \\ \mathbf{0} & \boldsymbol{\Gamma}_{oo}^{(n)} \end{pmatrix}$.

Then, $f_{\text{clust}}(\boldsymbol{y}_n^{(\mathbb{S},\mathbb{U}) \cap \mathbb{O}_n}; \boldsymbol{\nu}_{k,o}^{(\mathbb{S},\mathbb{U})(n)}, \boldsymbol{\Delta}_{k,oo}^{(\mathbb{S},\mathbb{U})(n)}) \cdot f_{\text{indep}}(\boldsymbol{y}_n^{\mathbb{W} \cap \mathbb{O}_n}; \boldsymbol{\gamma}_o^{(n)}, \boldsymbol{\Gamma}_{oo}^{(n)}) = f_{\text{clust}}(\boldsymbol{y}_n^o; \tilde{\boldsymbol{\nu}}_{k,o}^{(n)}, \tilde{\boldsymbol{\Delta}}_{k,oo}^{(n)})$.

Therefore, the observed-data density for a single observation $n$ is:

$$f(\boldsymbol{y}_n^o; \mathbb{V}, \boldsymbol{\theta}) = \sum_{k=1}^{K} \pi_k f_{\text{clust}}(\boldsymbol{y}_n^o; \tilde{\boldsymbol{\nu}}_{k,o}^{(n)}, \tilde{\boldsymbol{\Delta}}_{k,oo}^{(n)}) \tag{12}$$

And for $N$ i.i.d. observations, the total observed-data likelihood is:

$$\prod_{n=1}^{N} f(\boldsymbol{y}_n^o; K, m, r, l, \mathbb{V}, \boldsymbol{\theta}) = \prod_{n=1}^{N} \left( \sum_{k=1}^{K} \pi_k f_{\text{clust}}(\boldsymbol{y}_n^o; \tilde{\boldsymbol{\nu}}_{k,o}^{(n)}, \tilde{\boldsymbol{\Delta}}_{k,oo}^{(n)}) \right) \tag{13}$$

$\square$

When the parameters meet some distinct properties, $\mathbb{SRUW}$ under complete-data can achieve identifiability, as shown in Theorem 1.

**Fact 1** (Theorem 1 in [38]). *Let $\boldsymbol{\Theta}_{(K,m,r,l,\mathbb{V})}$ be a subset of the parameter set $\Upsilon_{(K,m,r,l,\mathbb{V})}$ such that elements $\boldsymbol{\theta} = (\boldsymbol{\alpha}, \boldsymbol{a}, \boldsymbol{\beta}, \boldsymbol{\Omega}, \boldsymbol{\gamma}, \boldsymbol{\Gamma})$*

- *contain distinct couples $(\mu_k, \Sigma_k)$ fulfilling $\forall s \subseteq \mathbb{S}, \exists (k, k'), 1 \leq k < k' \leq K$;*

$$\mu_{k,\bar{s}|s} \neq \mu_{k',\bar{s}|s} \text{ or } \Sigma_{k,\bar{s}|s} \neq \Sigma_{k',\bar{s}|s} \text{ or } \Sigma_{k,\bar{s}\bar{s}|s} \neq \Sigma_{k',\bar{s}\bar{s}|s} \tag{14}$$

  *where $\bar{s}$ denotes the complement in $\mathbb{S}$ of any nonempty subset $s$ of $\mathbb{S}$*

- *if $\mathbb{U} \neq \emptyset$,*

  - *for all variables $j$ of $\mathbb{R}$, there exists a variable $u$ of $\mathbb{U}$ such that the restriction $\boldsymbol{\beta}_{uj}$ of the regression coefficient matrix $\boldsymbol{\beta}$ associated to $j$ and $u$ is not equal to zero.*
  - *for all variables $u$ of $\mathbb{U}$, there exists a variable $j$ of $\mathbb{R}$ such that $\boldsymbol{\beta}_{uj} \neq 0$.*

- *parameters $\boldsymbol{\Omega}$ and $\tau$ exactly respect the forms $r$ and $l$ respectively. They are both diagonal matrices with at least two different eigenvalues if $r = [LB]$ and $l = [LB]$ and $\boldsymbol{\Omega}$ has at least a non-zero entry outside the main diagonal if $r = [LC]$.*

*Let $(K, m, r, l, \mathbb{V})$ and $(K^*, m^*, r^*, l^*, \mathbb{V}^*)$ be two models. If there exist $\boldsymbol{\theta} \in \boldsymbol{\Theta}_{(K,m,r,l,\mathbb{V})}$ and $\boldsymbol{\theta}^* \in \boldsymbol{\Theta}_{(K^*,m^*,r^*,l^*,\mathbb{V}^*)}$ such that*

$$f(\cdot | K, m, r, l, \mathbb{V}, \boldsymbol{\theta}) = f(\cdot | K^*, m^*, r^*, l^*, \mathbb{V}^*, \boldsymbol{\theta}^*)$$

*then $(K, m, r, l, \mathbb{V}) = (K^*, m^*, r^*, l^*, \mathbb{V}^*)$ and $\boldsymbol{\theta} = \boldsymbol{\theta}^*$ (up to a permutation of mixture components).*

## C    Regular Assumptions for Main Theoretical Results

For a fixed tuple of $(K_0, m_0, r_0, l_0, \mathbb{V}_0)$, we can write $f(\boldsymbol{y}; \boldsymbol{\theta})$ instead of $f(\boldsymbol{y}; K_0, m_0, \mathbb{S}_0, \mathbb{R}_0, \boldsymbol{\theta})$ or $f(\boldsymbol{y}; \mathbb{V}, \boldsymbol{\theta})$ for short notation.

**Assumption 1** ($\mathbb{SR}$). *There exists a unique $(K_0, m_0, \mathbb{S}_0, \mathbb{R}_0)$ such that*

$$h = f\big(\cdot; \boldsymbol{\theta}^*_{(K_0, m_0, \mathbb{S}_0, \mathbb{R}_0)}\big)$$

*for some parameter value $\boldsymbol{\theta}^*$, and the pair $(K_0, m_0)$ is assumed known. (In the following, we omit explicit notation of the dependence on $(K_0, m_0)$ for brevity.)*

**Assumption 2.** *The vectors $\boldsymbol{\theta}^*_{(\mathbb{S},\mathbb{R})}$ and $\hat{\boldsymbol{\theta}}_{(\mathbb{S},\mathbb{R})}$ belong to a compact subset*

$$\boldsymbol{\Theta}'_{(\mathbb{S},\mathbb{R})} \subset \boldsymbol{\Theta}_{(\mathbb{S},\mathbb{R})}$$

*defined by*

$$\boldsymbol{\Theta}'_{(\mathbb{S},\mathbb{R})} = \mathcal{P}_{K_0-1} \times \mathcal{B}(\eta, |\mathbb{S}|)^{K_0} \times \mathcal{D}^{K_0}_{|\mathbb{S}|} \times \mathcal{B}(\rho, |\mathbb{S}^c|) \times \mathcal{B}(\rho, |\mathbb{R}|, |\mathbb{S}^c|) \times \mathcal{D}_{|\mathbb{S}^c|}$$

*where the components are defined as follows:*

- $\mathcal{P}_{K-1} = \{(\pi_1, \ldots, \pi_K) \in [0,1]^K : \sum_{k=1}^K \pi_k = 1\}$ *(set of proportions);*

- $\mathcal{B}(\eta, r) = \{x \in \mathbb{R}^r : \|x\| \leq \eta\}$ *with* $\|x\| = \sqrt{\sum_{i=1}^r x_i^2}$;

- $\mathcal{B}(\rho, q, r) = \{A \in \mathcal{M}_{q \times r}(\mathbb{R}) : \|\!|A|\!\| \leq \rho\}$, *with norm*

$$\|\!|A|\!\| = \sup_{y \in \mathbb{R}^r, \|y\|=1} \|Ay\|;$$

- $\mathcal{D}_r$ *is the set of $r \times r$ positive definite matrices with eigenvalues in $[s_m, s_M]$ (with $0 < s_m < s_M$)*

**Assumption 3.** *The optimal parameter $\boldsymbol{\theta}^*_{(\mathbb{S}_0, \mathbb{R}_0)}$ is an interior point of $\boldsymbol{\Theta}'_{(\mathbb{S}_0, \mathbb{R}_0)}$.*

The assumptions can be extended to the $\mathbb{SRUW}$ framework. We define the optimal parameter as $\boldsymbol{\theta}^*_{\mathbb{V}}$, and the maximum likelihood estimator as $\hat{\boldsymbol{\theta}}_{\mathbb{V}}$. In this modified framework, the assumptions become:

**Assumption 4** ($\mathbb{SRUW}$). *There exists a unique $(K_0, m_0, r_0, l_0, \mathbb{V}_0)$ such that*

$$h = f\big(\cdot; \boldsymbol{\theta}^\star_{(K_0, m_0, r_0, l_0, \mathbb{V}_0)}\big)$$

*for some $\boldsymbol{\theta}^\star$, and the model $(K_0, m_0, r_0, l_0, \mathbb{V}_0)$ is assumed known.*

**Assumption 5.** *Vectors $\boldsymbol{\theta}^*_{\mathbb{V}}$ and $\hat{\boldsymbol{\theta}}_{\mathbb{V}}$ belong to the compact subspace $\boldsymbol{\Theta}'_{\mathbb{V}}$ of the compact set $\boldsymbol{\Theta}_{\mathbb{V}}$, defined by:*

$$\boldsymbol{\Theta}'_{\mathbb{V}} = \mathcal{P}_{K_0-1} \times \mathcal{B}(\eta, |\mathbb{S}|)^{K_0} \times \mathcal{D}^{K_0}_{|\mathbb{S}|} \times \mathcal{B}(\rho, |\mathbb{U}|) \times \mathcal{B}(\rho, |\mathbb{R}|, |\mathbb{U}|) \times \mathcal{D}_{|\mathbb{U}|} \times \mathcal{B}(\eta, |\mathbb{W}|) \times \mathcal{D}_{|\mathbb{W}|}.$$

**Assumption 6.** *The optimal parameter $\boldsymbol{\theta}^*_{\mathbb{V}}$ is an interior point of $\boldsymbol{\Theta}'_{\mathbb{V}}$.*

## D    Proof of Main Results

Throughout the proofs, we denote the full parameter vector of the $K$-component Gaussian Mixture Model (GMM) by $\boldsymbol{\alpha} = (\boldsymbol{\pi}, \{\boldsymbol{\mu}_k\}_{k=1}^K, \{\boldsymbol{\Psi}_k\}_{k=1}^K)$, where $\boldsymbol{\pi}$ are the mixing proportions, $\boldsymbol{\mu}_k \in \mathbb{R}^D$ are the component means for $D$ variables, and $\boldsymbol{\Psi}_k$ are the component precision matrices (inverse of covariance matrices $\boldsymbol{\Sigma}_k$). The true parameter vector is denoted by $\boldsymbol{\alpha}^*$. The negative single-observation log-likelihood is $\ell_1(\boldsymbol{y}_n; \boldsymbol{\alpha}) = -\ln f_{\text{clust}}(\boldsymbol{y}_n; \boldsymbol{\alpha})$, and its sample average is $\ell_N(\boldsymbol{\alpha})$. The score function components are $S_j(\boldsymbol{y}; \boldsymbol{\alpha}^*) = -\frac{\partial \ell_1(\boldsymbol{y}; \boldsymbol{\alpha})}{\partial \alpha_j}|_{\boldsymbol{\alpha}=\boldsymbol{\alpha}^*}$. The Fisher Information Matrix (FIM) is $\boldsymbol{I}(\boldsymbol{\alpha}^*)$, and the empirical Hessian is $\boldsymbol{H}_N(\boldsymbol{\alpha})$. The set of true non-zero penalized parameters in $\boldsymbol{\alpha}^*$ is $\mathbb{S}_0$, with cardinality $s_0$. The error vector is $\boldsymbol{\Delta} = \hat{\boldsymbol{\alpha}} - \boldsymbol{\alpha}^*$. For ranking, $\mathcal{O}_K(j)$ is the ranking score for variable $j$ based on a grid of regularization parameters $\mathcal{G}_\lambda$. Standard notations like $\mathbb{E}[\cdot]$ for expectation, $\|\cdot\|_1, \|\cdot\|_2, \|\cdot\|_\infty$ for $L_1, L_2, L_\infty$ norms. Other abbreviations include KKT (Karush-Kuhn-Tucker), RSC (Restricted Strong Convexity), and SRUW (for the variable role framework).

### D.1 Proof of Theorem 1

**Proof Sketch.** The proof involves two main steps. First, we establish that equality of observed-data likelihoods under MAR implies equality of the parameters of an equivalent full-data GMM representation for the $\mathbb{SRUW}$ model. Second, we apply the identifiability arguments for the complete-data $\mathbb{SRUW}$ model similar to [38], which itself relies on the identifiability of the $\mathbb{SR}$ model [37]

*Proof.* Let $\mathcal{M}_1 = (K, m, r, l, \mathbb{V}, \boldsymbol{\theta})$ and $\mathcal{M}_2 = (K^*, m^*, r^*, l^*, \mathbb{V}^*, \boldsymbol{\theta}^*)$ be two $\mathbb{SRUW}$ models with $\mathcal{M}_1, \mathcal{M}_2 \in \boldsymbol{\Theta}_{(K,m,r,l,\mathbb{V})}$. The complete-data density for an observation $\boldsymbol{y}_n$ under $\mathcal{M}_1$ and Equation (12) is given by:

$$f(\boldsymbol{y}_n^o; K, m, r, l, \mathbb{V}, \boldsymbol{\theta}, \boldsymbol{M}_n) = \sum_{k=1}^{K} p_k f_{\text{clust}}(\boldsymbol{y}_n^o; \tilde{\boldsymbol{\nu}}_{k,o}^{(n)}, \tilde{\boldsymbol{\Delta}}_{k,oo}^{(n)}),$$

Analogously, we can derive the same expression for $\mathcal{M}_2$. Consider the specific missingness pattern $\boldsymbol{M}_0$ where no data are missing for observation $n$. For this pattern, $\boldsymbol{y}_n^o = \boldsymbol{y}_n$, $\tilde{\boldsymbol{\nu}}_{k,o}^{(n)} = \tilde{\boldsymbol{\nu}}_k$, and $\tilde{\boldsymbol{\Delta}}_{k,oo}^{(n)} = \tilde{\boldsymbol{\Delta}}_k$. Under the premise of the theorem, if the equality of observed-data likelihoods for pattern $\boldsymbol{M}_0$, we get:

$$\sum_{k=1}^{K} \pi_k f_{\text{clust}}(\boldsymbol{y}_n; \tilde{\boldsymbol{\nu}}_k, \tilde{\boldsymbol{\Delta}}_k) = \sum_{k'=1}^{K^*} p_{k'}^* f_{\text{clust}}(\boldsymbol{y}_n; \tilde{\boldsymbol{\nu}}_{k'}^*, \tilde{\boldsymbol{\Delta}}_{k'}^*), \quad \forall \boldsymbol{y}_n \in \mathbb{R}^D.$$

From theorem 1 in [38], since $\mathbb{SRUW}$ are identifiable under complete-data scenario this implies $K = K^*$, and up to a permutation of component labels:

$$\pi_k = \pi_k^*, \quad \tilde{\boldsymbol{\nu}}_k = \tilde{\boldsymbol{\nu}}_k^*, \quad \text{and} \quad \tilde{\boldsymbol{\Delta}}_k = \tilde{\boldsymbol{\Delta}}_k^* \quad \text{for all } k = 1, \ldots, K. \tag{15}$$

For any $\boldsymbol{y} \in \mathbb{R}^D$:

$$f_{\text{reg}}(\boldsymbol{y}^{\mathbb{U}}; r, \boldsymbol{a} + \boldsymbol{y}^{\mathbb{R}}\boldsymbol{\beta}, \boldsymbol{\Omega}) f_{\text{indep}}(\boldsymbol{y}^{\mathbb{W}}; l, \boldsymbol{\gamma}, \boldsymbol{\Gamma}) = f_{\text{clust}}(\boldsymbol{y}^{\mathbb{U} \cup \mathbb{W}}; \check{\boldsymbol{a}} + \boldsymbol{y}^{\mathbb{R}}\check{\boldsymbol{\beta}}, \check{\boldsymbol{\Omega}})$$

where $\mathbb{S}^c = \mathbb{U} \cup \mathbb{W}$, $\check{\boldsymbol{a}} = ((\boldsymbol{a})^T, (\boldsymbol{\gamma})^T)^T$, $\check{\boldsymbol{\beta}} = \begin{pmatrix} \boldsymbol{\beta} \\ \boldsymbol{0} \end{pmatrix}$ (with $|\boldsymbol{0}| = |\mathbb{W}| \times |\mathbb{R}|$), and $\check{\boldsymbol{\Omega}} = $ blockdiag$(\boldsymbol{\Omega}, \boldsymbol{\Gamma})$. Here, blockdiag$(\boldsymbol{\Omega}, \boldsymbol{\Gamma})$ is the block diagonal matrix created by aligning the input matrices $\boldsymbol{\Omega}, \boldsymbol{\Gamma}$. The $\mathbb{SRUW}$ density can be expressed as an equivalent $\mathbb{SR}$ model density:

$$f(\boldsymbol{y}; K, m, r, l, \mathbb{V}, \boldsymbol{\theta}) = f_{\text{clust}}(\boldsymbol{y}^{\mathbb{S}}; K, m, \boldsymbol{\alpha}) f_{\text{clust}}(\boldsymbol{y}^{\mathbb{S}^c}; \check{\boldsymbol{a}} + \boldsymbol{y}^{\mathbb{R}}\check{\boldsymbol{\beta}}, \check{\boldsymbol{\Omega}})$$
$$= f_{\mathbb{SR}\text{-eq}}(\boldsymbol{y}; K, m, \mathbb{S}, \mathbb{R}, \boldsymbol{\theta}_{\mathbb{SR}\text{-eq}})$$

where $\boldsymbol{\theta}_{\mathbb{SR}\text{-eq}} = (\boldsymbol{\alpha}, \check{\boldsymbol{a}}, \check{\boldsymbol{\beta}}, \check{\boldsymbol{\Omega}})$. From Section D.1, the equivalent $\mathbb{SR}$ densities for the two models, where the second model is denoted with $^*$) must be equal:

$$f_{\mathbb{SR}\text{-eq}}(\boldsymbol{y}; K, m, \mathbb{S}, \mathbb{R}, \boldsymbol{\theta}_{\mathbb{SR}\text{-eq}}) = f_{\mathbb{SR}\text{-eq}}(\boldsymbol{y}; K^*, m^*, \mathbb{S}^*, \mathbb{R}^*, \boldsymbol{\theta}_{\mathbb{SR}\text{-eq}}^*)$$

By the identifiability of the $\mathbb{SR}$ model [37] and three conditions in Theorem 1 , we deduce: $K = K^*$, $m = m^*, \mathbb{S} = \mathbb{S}^*, \mathbb{R} = \mathbb{R}^*, \boldsymbol{\alpha} = \boldsymbol{\alpha}^*, \check{\boldsymbol{a}} = \check{\boldsymbol{a}}^*, \check{\boldsymbol{\beta}} = \check{\boldsymbol{\beta}}^*$, and $\check{\boldsymbol{\Omega}} = \check{\boldsymbol{\Omega}}^*$.

Since $\mathbb{S} = \mathbb{S}^*$ and $\mathbb{S}^c = \mathbb{S}^{*c}$, supposed that, for contradiction, that $\mathbb{U}^* \cap \mathbb{W} \neq \emptyset$. Let $j \in \mathbb{U}^* \cap \mathbb{W}$. Since $j \in \mathbb{W}$, for all $q \in \mathbb{U}^*$ the rows of $(\check{\boldsymbol{\beta}})_{qj}$ are $\boldsymbol{0}^T$. Additionally, since $j \in \mathbb{U}^*$, there exists some $q \in \mathbb{R}* = \mathbb{R}$ such that $(\check{\boldsymbol{\beta}}^*)_{qj} \neq \boldsymbol{0}^T$. However, we have established $\check{\boldsymbol{\beta}} = \check{\boldsymbol{\beta}}^*$. This is a contradiction: $(\check{\boldsymbol{\beta}})_{qj}$ must be simultaneously $\boldsymbol{0}^T$ and non-zero for all $q \in sR$. Therefore, $\mathbb{U}^* \cap \mathbb{W} = \emptyset$. By symmetry, $\mathbb{U} \cap \mathbb{W}^* = \emptyset$. Given $\mathbb{U} \cup \mathbb{W} = \mathbb{U}^* \cup \mathbb{W}^*$, the disjoint conditions imply $\mathbb{U} \subseteq \mathbb{U}^*$ and $\mathbb{U}^* \subseteq \mathbb{U}$, hence $\mathbb{U} = \mathbb{U}^*$. Consequently, $\mathbb{W} = \mathbb{W}^*$. This establishes $\mathbb{V} = \mathbb{V}^*$. The parameter equalities $\boldsymbol{a} = \boldsymbol{a}^*, \boldsymbol{\beta} = \boldsymbol{\beta}^*, \boldsymbol{\gamma} = \boldsymbol{\gamma}^*, \boldsymbol{\Omega} = \boldsymbol{\Omega}^*, \boldsymbol{\Gamma} = \boldsymbol{\Gamma}^*$ and $r = r^*, l = l^*$ then follow directly from comparing the components of $\check{\boldsymbol{a}} = \check{\boldsymbol{a}}^*, \check{\boldsymbol{\beta}} = \check{\boldsymbol{\beta}}^*, \check{\boldsymbol{\Omega}} = \check{\boldsymbol{\Omega}}^*$, condition 3 in Theorem 1. Thus, the SRUW parameters under MAR are identifiable.

Now, we extend this to the full model which includes the MNARz mechanism parameters $\boldsymbol{\psi} = \{\psi_k\}_{k=1}^K$. The full set of parameters for the MNARz-SRUW model is $(\boldsymbol{\theta}, \boldsymbol{\psi})$. Suppose that the two models $\mathcal{M}_A$ and $\mathcal{M}_B$ produce the same observed-data likelihood for all observed data $(\boldsymbol{y}_n^o, \boldsymbol{c}_n)$:

$$L_{\text{MNARz-SRUW}}(\boldsymbol{y}_n^o, \boldsymbol{c}_n; \boldsymbol{\theta}_A^{\text{data}}, \boldsymbol{\psi}_A^{\text{MNARz}}) = L_{\text{MNARz-SRUW}}(\boldsymbol{y}_n^o, \boldsymbol{c}_n; \boldsymbol{\theta}_B^{\text{data}}, \boldsymbol{\psi}_B^{\text{MNARz}}) \quad \forall (\boldsymbol{y}_n^o, \boldsymbol{c}_n),$$

By Theorem 5, the equality of observed-data likelihoods under MNARz-SRUW:

$$L_{\text{MNARz-SRUW}}(\boldsymbol{y}_n^o, \boldsymbol{c}_n; \boldsymbol{\theta}_A^{\text{data}}, \boldsymbol{\psi}_A^{\text{MNARz}}) = L_{\text{MNARz-SRUW}}(\boldsymbol{y}_n^o, \boldsymbol{c}_n; \boldsymbol{\theta}_B^{\text{data}}, \boldsymbol{\psi}_B^{\text{MNARz}})$$

for all $(\boldsymbol{y}_n^o, \boldsymbol{c}_n)$ implies the equality of the likelihoods for the augmented observed data $\tilde{\boldsymbol{y}}_n^o = (\boldsymbol{y}_n^o, \boldsymbol{c}_n)$ under the MAR interpretation where $f(\boldsymbol{c}_n|z_{nk} = 1; \psi_k)$ is treated as part of the component density:

$$\sum_{k=1}^{K_A} (\pi_A)_k f_k(\boldsymbol{y}_n^o|z_{nk} = 1; (\theta_A^{\text{SRUW}})_k) f(\boldsymbol{c}_n|z_{nk} = 1; (\psi_A)_k)$$

$$= \sum_{k'=1}^{K_B} (\pi_B)_{k'} f_{k'}(\boldsymbol{y}_n^o|z_{nk'} = 1; (\theta_B^{\text{SRUW}})_{k'}) f(\boldsymbol{c}_n|z_{nk'} = 1; (\psi_B)_{k'}) \tag{16}$$

for all $(\boldsymbol{y}_n^o, \boldsymbol{c}_n)$. Let $g_k(\boldsymbol{y}_n^o, \boldsymbol{c}_n; (\theta_A)_k, (\psi_A)_k) = f_k(\boldsymbol{y}_n^o|z_{nk} = 1; (\theta_A^{\text{SRUW}})_k) f(\boldsymbol{c}_n|z_{nk} = 1; (\psi_A)_k)$. This means two finite mixture models on the augmented data $(\boldsymbol{y}_n^o, \boldsymbol{c}_n)$ are identical. If $\sum_{k=1}^{K_A} (\pi_A)_k g_k(\cdot) = \sum_{k'=1}^{K_B} (\pi_B)_{k'} g_{k'}'(\cdot)$, and the family of component densities $\{g_k\}$ (and $\{g_{k'}'\}$) is identifiable and satisfies certain linear independence conditions ([68], [2]), then $K_A = K_B = K$, and after a permutation of labels, $(\pi_A)_k = (\pi_B)_k$ and $g_k(\cdot) = g_k'(\cdot)$ for all $k$. Then, we need to ensure the family of component densities $h_k(\boldsymbol{y}_n^o, \boldsymbol{c}_n; \theta_k, \psi_k) = f_k(\boldsymbol{y}_n^o|z_{nk} = 1; \theta_k) f(\boldsymbol{c}_n|z_{nk} = 1; \psi_k)$ meets such conditions. Hence, we further assume:

1. The parameter sets $\boldsymbol{\theta}^{\text{SRUW}}$ and $\boldsymbol{\psi}^{\text{MNARz}}$ are functionally independent

2. The support of $\boldsymbol{y}_n$ and $\boldsymbol{c}_n$ is rich enough such that equalities holding for all $(\boldsymbol{y}_n, \boldsymbol{c}_n)$ imply functional equalities. This implies there is *sufficient variability* of $\boldsymbol{y}_n$ and $\boldsymbol{c}_n$.

These assumptions hold because $\psi$ parametrizes only $f(\boldsymbol{c}_n|\boldsymbol{z})$ and $\theta$ only $f(\boldsymbol{y}_n^o|\boldsymbol{z})$; hence there is no functional overlap. Applying those assumptions to Equation (16), we conclude that $K_A = K_B = K$, and up to a permutation of labels:

$$(\pi_A)_k = (\pi_B)_k \quad \text{for all } k, \tag{17}$$

and

$$f_k(\boldsymbol{y}_n^o|z_{nk} = 1; (\theta_A^{\text{SRUW}})_k) f(\boldsymbol{c}_n|z_{nk} = 1; (\psi_A)_k)$$
$$= \tag{18}$$
$$f_k(\boldsymbol{y}_n^o|z_{nk} = 1; (\theta_B^{\text{SRUW}})_k) f(\boldsymbol{c}_n|z_{nk} = 1; (\psi_B)_k)$$

for all $(\boldsymbol{y}_n^o, \boldsymbol{c}_n)$ and for each $k = 1, \ldots, K$.

Now we need to deduce that $(\theta_A^{\text{SRUW}})_k = (\theta_B^{\text{SRUW}})_k$ and $(\psi_A)_k = (\psi_B)_k$. Equation (18) can be written as:

$$A_1(\boldsymbol{y}_n^o) B_1(\boldsymbol{c}_n) = A_2(\boldsymbol{y}_n^o) B_2(\boldsymbol{c}_n)$$

where $A_1(\boldsymbol{y}_n^o) = f_k(\boldsymbol{y}_n^o|z_{nk} = 1; (\theta_A^{\text{SRUW}})_k)$, $B_1(\boldsymbol{c}_n) = f(\boldsymbol{c}_n|z_{nk} = 1; (\psi_A)_k)$. By defined assumptions, this equality holding for all $\boldsymbol{y}_n^o, \boldsymbol{c}_n$ implies a relationship between these functions. Since $\boldsymbol{y}_n^o$ and $\boldsymbol{c}_n$ can vary independently [1] and the parameters $(\theta_A)_k, (\psi_A)_k$ are functionally independent, if $\int A_1 B_1 d\boldsymbol{y}^o d\boldsymbol{c} = \int A_2 B_2 d\boldsymbol{y}^o d\boldsymbol{c} = 1$, and $A_1 B_1 = A_2 B_2$ pointwise, and assuming $A_1, B_1, A_2, B_2$ are non-zero almost everywhere on their support: Pick a $\boldsymbol{c}_n^0$ such that $f(\boldsymbol{c}_n^0|k; (\psi_A)_k) \neq 0$ and $f(\boldsymbol{c}_n^0|k; (\psi_B)_k) \neq 0$. Then for this fixed $\boldsymbol{c}_n^0$:

$$f_k(\boldsymbol{y}_n^o|z_{nk} = 1; (\theta_A^{\text{SRUW}})_k) \cdot C_A = f_k(\boldsymbol{y}_n^o|z_{nk} = 1; (\theta_B^{\text{SRUW}})_k) \cdot C_B$$

where $C_A = f(\boldsymbol{c}_n^0|k; (\psi_A)_k)$ and $C_B = f(\boldsymbol{c}_n^0|k; (\psi_B)_k)$. Since both $f_k(\boldsymbol{y}_n^o|\ldots)$ are conditional densities, this implies $C_A = C_B$ and thus

$$f_k(\boldsymbol{y}_n^o|z_{nk} = 1; (\theta_A^{\text{SRUW}})_k) = f_k(\boldsymbol{y}_n^o|z_{nk} = 1; (\theta_B^{\text{SRUW}})_k) \quad \text{for all } \boldsymbol{y}_n^o.$$

By identifiability of SRUW under MAR, this implies that the structural SRUW parameters $(m_A, r_A, l_A, \mathbb{V}_A)$ must equal $(m_B, r_B, l_B, \mathbb{V}_B)$ and the parameters $(\theta_A^{\text{SRUW}})_k = (\theta_B^{\text{SRUW}})_k$.

---

[1]Since, for each component $k$, the joint component density factorizes as $f_k(\boldsymbol{y}_n^o|z_{nk} = 1; \theta_k) f(\boldsymbol{c}_n|z_{nk} = 1; \psi_k)$ and both factors are positive on sets of positive measure, pick $c_0$ in that common positive-measure set; integrating $A_1(\boldsymbol{y}_n^o) B_1(c_0) = A_2(\boldsymbol{y}_n^o) B_2(c_0)$ over $\boldsymbol{y}_n^o$ gives $B_1(c_0) = B_2(c_0)$, hence $A_1 = A_2$, and then $B_1 = B_2$.

Since $C_A = C_B$ means $f(\boldsymbol{c}_n^0|k;(\psi_A)_k) = f(\boldsymbol{c}_n^0|k;(\psi_B)_k)$, and this must hold for all $\boldsymbol{c}_n^0$ (by choosing different fixed $\boldsymbol{c}_n^0$ or by varying $\boldsymbol{y}_n^o$ first), this implies:

$$f(\boldsymbol{c}_n|k;(\psi_A)_k) = f(\boldsymbol{c}_n|k;(\psi_B)_k) \quad \text{for all } \boldsymbol{c}_n.$$

By idenitifiability of MNARz, this implies $(\psi_A)_k = (\psi_B)_k$.

Thus, we have $K_A = K_B = K$, and up to a common permutation of labels for $k = 1, \ldots, K$:

$$(\pi_A)_k = (\pi_B)_k$$
$$(m_A, r_A, l_A, \mathbb{V}_A) = (m_B, r_B, l_B, \mathbb{V}_B)$$
$$(\theta_A^{\text{SRUW}})_k = (\theta_B^{\text{SRUW}})_k$$
$$(\psi_A)_k = (\psi_B)_k$$

This implies the full set of model specifications and parameters for $\mathcal{M}_A$ and $\mathcal{M}_B$ is identical. Therefore, the SRUW model under the specified MNARz mechanism is identifiable. $\qquad\square$

### D.2 Proof of Theorem 2

**Proof Sketch.**

Suppose that the observations are from a parametric model $\mathcal{P}_{\boldsymbol{\Theta}} = \{f(\cdot; \boldsymbol{\theta}) : \boldsymbol{\theta} \in \boldsymbol{\Theta}\}$, and assume that the distributions in $\mathcal{P}_{\boldsymbol{\Theta}}$ are dominated by a common $\sigma$-finite measure $\upsilon$ with respect to which they have probability density functions $f(\cdot; \boldsymbol{\theta})$.

Now, we need to prove the consistency of the sample KL, which will lead to the consistency of the BIC criterion in both the $\mathbb{SR}$ and $\mathbb{SRUW}$ models under the MAR mechanism. First, we prove the consistency of the sample KL in the $\mathbb{SR}$ model:

**Proposition 2.** *Under Assumption 1 and 2, for all $(\mathbb{S}, \mathbb{R})$,*

$$\frac{1}{N} \sum_{n=1}^{N} \ln \left( \frac{h(\boldsymbol{y}_n^o)}{f(\boldsymbol{y}_n^o; \hat{\boldsymbol{\theta}}_{(\mathbb{S}, \mathbb{R})})} \right) \xrightarrow[n \to \infty]{P} D_{\text{KL}} \Big[ h, f(\cdot; \boldsymbol{\theta}_{(\mathbb{S}, \mathbb{R})}^*) \Big].$$

*Proof.* Let $\mathbb{O} \subset \{1, \ldots, D\}$ denote the indices of the observed components.

Recall that the full covariance matrix for the component $k$, $\boldsymbol{\Delta}_k$, is built as:

$$\Delta_{k,jl} = \begin{cases} \Sigma_{k,jl}, & j, l \in \mathbb{S}, \\ (\boldsymbol{\Sigma}_k \boldsymbol{\Lambda})_{jl}, & j \in \mathbb{S}, l \in \mathbb{S}^c, \\ (\boldsymbol{\Lambda}^T \boldsymbol{\Sigma}_k)_{jl}, & j \in \mathbb{S}^c, l \in \mathbb{S}, \\ (\boldsymbol{\Omega} + \boldsymbol{\Lambda}^T \boldsymbol{\Sigma}_k \boldsymbol{\Lambda})_{jl}, & j, l \in \mathbb{S}^c. \end{cases}$$

For indices $j, l \in \mathbb{S}$: As before,
$$\Delta_{k,jl} = \Sigma_{k,jl}.$$

Thus, eigenvalues of $\boldsymbol{\Delta}_{k,\mathbb{SS}}$ are in $[s_m, s_M]$.

For $j \in \mathbb{S}, l \in \mathbb{S}^c$:
$$\Delta_{k,jl} = (\boldsymbol{\Sigma}_k \boldsymbol{\Lambda})_{jl}.$$

The norm of the product satisfies

$$\|\boldsymbol{\Sigma}_k \boldsymbol{\Lambda}\| \le \|\boldsymbol{\Sigma}_k\| \cdot \|\boldsymbol{\Lambda}\|.$$

Since $\boldsymbol{\Sigma}_k \in \mathcal{D}_{|\mathbb{S}|}$,
$$\|\boldsymbol{\Sigma}_k\| = \lambda_{\max}(\boldsymbol{\Sigma}_k) \le s_M.$$

Given $\boldsymbol{\beta} \in \mathcal{B}(\rho, |\mathbb{R}|, |\mathbb{S}^c|)$,
$$\|\boldsymbol{\Lambda}\| \le \rho.$$

Thus,
$$\|\boldsymbol{\Sigma}_k \boldsymbol{\Lambda}\| \le s_M \rho.$$

This bounds the norm of the $(\mathbb{S} \times \mathbb{S}^c)$ block of $\boldsymbol{\Delta}_k$.

For $j, l \in \mathbb{S}^c$:
$$\Delta_{k,jl} = (\boldsymbol{\Omega} + \boldsymbol{\Lambda}^T \boldsymbol{\Sigma}_k \boldsymbol{\Lambda})_{jl}.$$

We can bound the eigenvalues of this sum using the spectral norm:
$$\|\boldsymbol{\Omega} + \boldsymbol{\Lambda}^T \boldsymbol{\Sigma}_k \boldsymbol{\Lambda}\| \leq \|\boldsymbol{\Omega}\| + \|\boldsymbol{\Lambda}^T \boldsymbol{\Sigma}_k \boldsymbol{\Lambda}\|.$$

Since $\boldsymbol{\Omega} \in \mathcal{D}_{|\mathbb{S}^c|}$,
$$\|\boldsymbol{\Omega}\| = \lambda_{\max}(\boldsymbol{\Omega}) \leq s_M.$$

Next,
$$\|\boldsymbol{\Lambda}^T \boldsymbol{\Sigma}_k \boldsymbol{\Lambda}\| \leq \|\boldsymbol{\Lambda}^T\| \cdot \|\boldsymbol{\Sigma}_k\| \cdot \|\boldsymbol{\Lambda}\| = \|\boldsymbol{\Lambda}\|^2 \cdot \|\boldsymbol{\Sigma}_k\| \leq \rho^2 s_M.$$

Thus,
$$\|\boldsymbol{\Omega} + \boldsymbol{\Lambda}^T \boldsymbol{\Sigma}_k \boldsymbol{\Lambda}\| \leq s_M + \rho^2 s_M = s_M(1 + \rho^2).$$

Given the above bounds, there exist refined constants $\tilde{s}_m > 0$ and
$$\tilde{s}_M = \max\{s_M(1 + \rho^2),\ s_M\rho,\ s_M\} = s_M(1 + \rho^2)$$

such that
$$\tilde{s}_m I \preceq \boldsymbol{\Delta}_k \preceq \tilde{s}_M I.$$

Note that $\tilde{s}_M$ will depend on the sizes of the blocks, and it is finite being dependent on $s_M$, $\rho$, and the structure of $\boldsymbol{\Omega}$, $\boldsymbol{\beta}$, and $\boldsymbol{\Sigma}_k$.

Since $\boldsymbol{\Delta}_{k,oo}$ is a principal submatrix of $\boldsymbol{\Delta}_k$ and $\boldsymbol{\Delta}_k$ is a symmetric matrix, by the interlacing theorem:

$$\tilde{s}_m \leq \lambda_{\min}(\boldsymbol{\Delta}_{k,oo}) \leq \lambda_{\max}(\boldsymbol{\Delta}_{k,oo}) \leq \tilde{s}_M.$$

Thus,
$$\tilde{s}_m\, I_{|\mathbb{O}|} \preceq \boldsymbol{\Delta}_{k,oo} \preceq \tilde{s}_M\, I_{|\mathbb{O}|},$$

and
$$(\boldsymbol{\Delta}_{k,oo})^{-1} \preceq \frac{1}{\tilde{s}_m}\, I_{|\mathbb{O}|}.$$

We are now moving to the main proof. By hypothesis, the true observed-data density is given by

$$h(\boldsymbol{y}^o) = \sum_{k=1}^K \pi_k\, \Phi\Big(\boldsymbol{y}^o; \boldsymbol{\nu}_{k,o}^*, \Delta_{k,oo}^*\Big)$$

and if we can show that
$$\mathbb{E}_{\boldsymbol{y}^o}\left[|\ln h(\boldsymbol{y}^o)|\right] < \infty.$$

then by the LLN,

$$\frac{1}{N} \sum_{n=1}^N \ln\big(h(\boldsymbol{y}_n^o)\big) \xrightarrow[n\to\infty]{P} \mathbb{E}_{\boldsymbol{y}^o}\big[\ln h(\boldsymbol{y}^o)\big].$$

Moreover, the observed-data likelihood under the model is

$$f(\boldsymbol{y}^o; \boldsymbol{\theta}) = \sum_{k=1}^K \pi_k\, \Phi\Big(\boldsymbol{y}^o; \boldsymbol{\nu}_{k,o}, \boldsymbol{\Delta}_{k,oo}\Big).$$

and we later show in Proposition 3 that

$$\frac{1}{N} \sum_{n=1}^N \ln\big(f(\boldsymbol{y}_n^o; \hat{\boldsymbol{\theta}}_{(\mathbb{S},\mathbb{R})})\big) \xrightarrow[n\to\infty]{P} \mathbb{E}_{\boldsymbol{y}^o}\big[\ln f(\boldsymbol{y}^o; \boldsymbol{\theta}_{(\mathbb{S},\mathbb{R})}^*)\big].$$

To do so, we must verify that the class of functions

$$\mathcal{F}_{(\mathbb{S},\mathbb{R})} = \Big\{\boldsymbol{y}^o \mapsto \ln f(\boldsymbol{y}^o; \boldsymbol{\theta}) : \boldsymbol{\theta} \in \Theta'_{(\mathbb{S},\mathbb{R})}\Big\}$$

satisfies the conditions for uniform convergence under Assumption 2. In particular, by Assumption 2, the parameter space $\boldsymbol{\Theta}'_{(\mathbb{S},\mathbb{R})}$ is compact, and for every $\boldsymbol{y}^o \in \mathbb{R}^{|\mathbb{O}|}$ the mapping

$$\boldsymbol{\theta} \mapsto \ln f(\boldsymbol{y}^o; \boldsymbol{\theta})$$

is continuous. Next, we verify that there is an $h$-integrable envelope function $F \in \mathcal{F}_{(\mathbb{S},\mathbb{R})}$. Recalling that for $\boldsymbol{y}^o \in \mathbb{R}^{|\mathbb{O}|}$, the Gaussian component density is

$$\Phi\Big(\boldsymbol{y}^o; \boldsymbol{\nu}_{k,o}, \boldsymbol{\Delta}_{k,oo}\Big) = (2\pi)^{-\frac{|\mathbb{O}|}{2}} |\boldsymbol{\Delta}_{k,oo}|^{-\frac{1}{2}} \exp\Big( -\tfrac{1}{2}(\boldsymbol{y}^o - \boldsymbol{\nu}_{k,o})^T \boldsymbol{\Delta}_{k,oo}^{-1} (\boldsymbol{y}^o - \boldsymbol{\nu}_{k,o}) \Big).$$

We can derive the bounds for this as follows.

For the upper bound, since $\lambda_{\min}(\boldsymbol{\Delta}_{k,oo}) \geq \tilde{s}_m$,

$$|\boldsymbol{\Delta}_{k,oo}|^{-\frac{1}{2}} \leq (\tilde{s}_m)^{-\frac{|\mathbb{O}|}{2}}.$$

With $\exp(-\cdot) \leq 1$ and $\boldsymbol{\Delta}_{k,oo}$ is positive definite

$$\Phi\Big(\boldsymbol{y}^o; \boldsymbol{\nu}_{k,o}, \boldsymbol{\Delta}_{k,oo}\Big) \leq (2\pi)^{-\frac{|\mathbb{O}|}{2}} (\tilde{s}_m)^{-\frac{|\mathbb{O}|}{2}}.$$

Summing over $k$ and using $\sum_{k=1}^{K} \pi_k = 1$,

$$f(\boldsymbol{y}^o; \boldsymbol{\theta}) \leq (2\pi\tilde{s}_m)^{-\frac{|\mathbb{O}|}{2}}.$$

Taking logarithms:

$$\ln f(\boldsymbol{y}^o; \boldsymbol{\theta}) \leq -\frac{|\mathbb{O}|}{2} \ln(2\pi\tilde{s}_m).$$

For the lower bound, for each $k$,

$$\ln \Phi\Big(\boldsymbol{y}^o; \boldsymbol{\nu}_{k,o}, \boldsymbol{\Delta}_{k,oo}\Big) = -\frac{|\mathbb{O}|}{2} \ln(2\pi) - \frac{1}{2} \ln|\boldsymbol{\Delta}_{k,oo}| - \frac{1}{2}(\boldsymbol{y}^o - \boldsymbol{\nu}_{k,o})^T \boldsymbol{\Delta}_{k,oo}^{-1}(\boldsymbol{y}^o - \boldsymbol{\nu}_{k,o}).$$

Since $\lambda_{\max}(\boldsymbol{\Delta}_{k,oo}) \leq \tilde{s}_M$,

$$\ln|\boldsymbol{\Delta}_{k,oo}| \leq |\mathbb{O}| \ln(\tilde{s}_M),$$

and $\boldsymbol{\Delta}_{k,oo}^{-1} \preceq \frac{1}{\tilde{s}_m} I$,

$$(\boldsymbol{y}^o - \boldsymbol{\nu}_{k,o})^T \boldsymbol{\Delta}_{k,oo}^{-1}(\boldsymbol{y}^o - \boldsymbol{\nu}_{k,o}) \leq \frac{\|\boldsymbol{y}^o - \boldsymbol{\nu}_{k,o}\|^2}{\tilde{s}_m}.$$

Using $\|\boldsymbol{y}^o - \boldsymbol{\nu}_{k,o}\|^2 \leq 2(\|\boldsymbol{y}^o\|^2 + \|\boldsymbol{\nu}_{k,o}\|^2)$,

$$(\boldsymbol{y}^o - \boldsymbol{\nu}_{k,o})^T \boldsymbol{\Delta}_{k,oo}^{-1}(\boldsymbol{y}^o - \boldsymbol{\nu}_{k,o}) \leq \frac{2(\|\boldsymbol{y}^o\|^2 + \|\boldsymbol{\nu}_{k,o}\|^2)}{\tilde{s}_m}.$$

Given the construction of the mean vector $\boldsymbol{\nu}_k$ for each component $k$:

$$\nu_{kj} = \begin{cases} \mu_{kj}, & \text{if } j \in \mathbb{S}, \\ (a + \boldsymbol{\mu}_k \Lambda)_j, & \text{if } j \in \mathbb{S}^c, \end{cases}$$

and knowing that $a$ belongs to the closed ball $\mathcal{B}(\rho, 1, |\mathbb{S}^c|)$-a set of $1 \times |\mathbb{S}^c|$ matrices with norm bounded by $\rho$-we seek to derive a uniform bound for $\boldsymbol{\nu}_k$. For indices $j \in \mathbb{S}$, $\nu_{kj} = \mu_{kj}$, and since the parameter space $\boldsymbol{\Theta}'_V$ is compact, the cluster means $\mu_{kj}$ are bounded by some constant $\eta > 0$. For indices $j \in \mathbb{S}^c$, we have $\nu_{kj} = (a + \boldsymbol{\mu}_k \Lambda)_j = a_j + (\boldsymbol{\mu}_k \Lambda)_j$. Using the elementary inequality $(u + v)^2 \leq 2(u^2 + v^2)$, it follows that

$$\|\nu_{kj}\|^2 \leq 2\Big(\|a_j\|^2 + \|(\boldsymbol{\mu}_k \Lambda)_j\|^2\Big).$$

Given that $a$ lies in $\mathcal{B}(\rho, 1, |\mathbb{S}^c|)$, $\|a_j\|^2 \leq \rho^2$ for each $j \in \mathbb{S}^c$. In addition, the term $(\boldsymbol{\mu}_k \Lambda)_j$ can be bounded by $\|\boldsymbol{\mu}_k\|_2 \|\Lambda_{\cdot j}\|_2$, where $\Lambda_{\cdot j}$ is the $j$-th column of $\Lambda$. With $\boldsymbol{\mu}_k$ belonging to a compact set $\mathcal{B}(\eta, |\mathbb{S}|)$, $\|\boldsymbol{\mu}_k\|^2 \leq \eta^2$, and because $\Lambda$ is derived from bounded parameters (including

$\boldsymbol{\beta}$ with norm being bounded by $\rho^2$), each column $\Lambda_{.j}$ is bounded in norm by $\rho^2$. Consequently, $|(\boldsymbol{\mu}_k \Lambda)_j| \le \rho^2 + \eta^2 \rho^2 = \rho^2(1+\eta^2)$ for each $j \in \mathbb{S}^c$. Combining these results, for any $j \in \mathbb{S}^c$,

$$\|\nu_{kj}\|^2 \le 2\rho^2(1+\eta^2).$$

Thus, all entries of $\boldsymbol{\nu}_k$, irrespective of whether they correspond to indices in $\mathbb{S}$ or $\mathbb{S}^c$, are bounded by a constant that depends on $\eta$, $\rho$. Hence,

$$
\begin{aligned}
\|\boldsymbol{\nu}_k\|^2 &= \sum_j \|\boldsymbol{\nu}_{k,j}\|^2 \\
&\le \sum_j 2\rho^2(1+\eta^2) \\
&= 2D\rho^2(1+\eta^2)
\end{aligned}
$$

uniformly for all $k$.

Consider again the Gaussian density for the observed variables

$$\ln \Phi\Big(\boldsymbol{y}^o; \boldsymbol{\nu}_{k,o}, \boldsymbol{\Delta}_{k,oo}\Big) = -\frac{|\mathbb{O}|}{2}\ln(2\pi) - \frac{1}{2}\ln|\boldsymbol{\Delta}_{k,oo}| - \frac{1}{2}(\boldsymbol{y}^o - \boldsymbol{\nu}_{k,o})^T \boldsymbol{\Delta}_{k,oo}^{-1}(\boldsymbol{y}^o - \boldsymbol{\nu}_{k,o}).$$

To control the quadratic term in the exponent, we note that

$$\|\boldsymbol{y}^o - \boldsymbol{\nu}_{k,o}\|^2 \le 2\Big(\|\boldsymbol{y}^o\|^2 + \|\boldsymbol{\nu}_{k,o}\|^2\Big).$$

Given that $\|\boldsymbol{\nu}_k\| = \sum_{j \in \mathbb{O}}\|\nu_{kj}\|^2 \le 2|\mathbb{O}|\rho^2(1+\eta^2)$ is uniformly bounded, it follows that

$$\|\boldsymbol{y}^o - \boldsymbol{\nu}_{k,o}\|^2 \le 2\Big(\|\boldsymbol{y}^o\|^2 + 2|\mathbb{O}|\rho^2(1+\eta^2)\Big).$$

This bound ensures that the quadratic form $(\boldsymbol{y}^o - \boldsymbol{\nu}_{k,o})^T \boldsymbol{\Delta}_{k,oo}^{-1}(\boldsymbol{y}^o - \boldsymbol{\nu}_{k,o})$ remains finite. The uniform boundedness of $\boldsymbol{\nu}_k$ thus allows us to assert the existence of an integrable envelope function $F(\boldsymbol{y}^o)$ that uniformly bounds $|\ln f(\boldsymbol{y}^o; \boldsymbol{\theta})|$ for all $\boldsymbol{\theta}$ within the compact parameter space $\boldsymbol{\Theta}_V'$.

Thus,

$$\ln \Phi\Big(\boldsymbol{y}^o; \boldsymbol{\nu}_{k,o}, \boldsymbol{\Delta}_{k,oo}\Big) \ge -\frac{|\mathbb{O}|}{2}\ln(2\pi) - \frac{|\mathbb{O}|}{2}\ln(\tilde{s}_M) - \frac{1}{2}\cdot\frac{2(\|\boldsymbol{y}^o\|^2 + 2|\mathbb{O}|\rho^2(1+\eta^2))}{\tilde{s}_m}.$$

Simplifying,

$$\ln \Phi\Big(\boldsymbol{y}^o; \boldsymbol{\nu}_{k,o}, \boldsymbol{\Delta}_{k,oo}\Big) \ge -\frac{|\mathbb{O}|}{2}\ln(2\pi\tilde{s}_M) - \frac{\|\boldsymbol{y}^o\|^2 + 2|\mathbb{O}|\rho^2(1+\eta^2)}{\tilde{s}_m}.$$

Using Jensen's inequality over the mixture:

$$\ln f(\boldsymbol{y}^o; \boldsymbol{\theta}) \ge \sum_{k=1}^{K} \pi_k \left( -\frac{|\mathbb{O}|}{2}\ln(2\pi\tilde{s}_M) - \frac{\|\boldsymbol{y}^o\|^2 + 2|\mathbb{O}|\rho^2(1+\eta^2)}{\tilde{s}_m} \right).$$

Since $\sum_{k=1}^{K} \pi_k = 1$,

$$\ln f(\boldsymbol{y}^o; \boldsymbol{\theta}) \ge -\frac{|\mathbb{O}|}{2}\ln(2\pi\tilde{s}_M) - \frac{\|\boldsymbol{y}^o\|^2 + 2|\mathbb{O}|\rho^2(1+\eta^2)}{\tilde{s}_m}.$$

Combining the refined upper and lower bounds:

$$-\frac{|\mathbb{O}|}{2}\ln(2\pi\tilde{s}_M) - \frac{\|\boldsymbol{y}^o\|^2 + 2|\mathbb{O}|\rho^2(1+\eta^2)}{\tilde{s}_m} \le \ln f(\boldsymbol{y}^o; \boldsymbol{\theta}_{(\mathbb{S},\mathbb{R})}) \le -\frac{|\mathbb{O}|}{2}\ln(2\pi\tilde{s}_m).$$

As a final note, these bounds also rely on the eigenvalue constraints on $\boldsymbol{\Sigma}_k$, $\boldsymbol{\Omega}$, and the norm constraint on $\boldsymbol{\beta}$.

Now we prove that the envelope function $F$, which is related to the upper and lower bounds above, is $h$-integrable. The true density $h(\boldsymbol{y})$ corresponds to a Gaussian mixture model $f(\boldsymbol{y}; \boldsymbol{\theta}^*_{(\mathbb{S}_0, \mathbb{R}_0)})$ with parameters in a compact set. The observed-data density for $\boldsymbol{y}^o$, obtained by marginalizing over the missing components, is given by

$$h(\boldsymbol{y}^o) = \sum_{k=1}^K \pi_k \, \Phi\Big(\boldsymbol{y}^o; \boldsymbol{\nu}^*_{k,o}, \Delta^*_{k,oo}\Big).$$

To verify that the envelope function $F$ is $h$-integrable, we need to show

$$\int \|\boldsymbol{y}^o\|^2 h(\boldsymbol{y}^o) \, d\boldsymbol{y}^o < \infty.$$

We proceed by examining the second moment of the observed components. First, observe that

$$\int \|\boldsymbol{y}^o\|^2 h(\boldsymbol{y}^o) \, d\boldsymbol{y}^o = \sum_{k=1}^K \pi_k \int \|\boldsymbol{y}^o\|^2 \, \Phi\Big(\boldsymbol{y}^o; \boldsymbol{\nu}^*_{k,o}, \Delta^*_{k,oo}\Big) d\boldsymbol{y}^o.$$

$$\leq 2\|\boldsymbol{\nu}^*_{k,o}\|^2 + 2\operatorname{tr}(\Delta^*_{k,oo}).$$

by using Lemma. By the compactness of the parameter space $\boldsymbol{\Theta}_{\mathbb{V}}$ and since $\sum_{k=1}^K \pi_k = 1$ and the bounds derived earlier, we have

$$\int \|\boldsymbol{y}^o\|^2 h(\boldsymbol{y}^o) \, d\boldsymbol{y}^o \leq |\mathbb{O}|\rho^2(1 + \eta^2) + 2s_M|\mathbb{S}_{0,o}| < \infty$$

Therefore, $F$ is h-integrable, i.e.,

$$\int F(\boldsymbol{y}^o) h(\boldsymbol{y}^o) \, d\boldsymbol{y}^o < \infty,$$

since $\int \|\boldsymbol{y}^o\|^2 h(\boldsymbol{y}^o) \, d\boldsymbol{y}^o < \infty$.

Finally, because $|\ln(h(\boldsymbol{y}^o))| \leq F(\boldsymbol{y}^o)$, it implies

$$\mathbb{E}[|\ln(h(\boldsymbol{y}^o))|] = \int |\ln(h(\boldsymbol{y}^o))| h(\boldsymbol{y}^o) \, d\boldsymbol{y}^o \leq \int F(\boldsymbol{y}^o) h(\boldsymbol{y}^o) \, d\boldsymbol{y}^o < \infty.$$

Hence, the envelope function $F$ is $h$-integrable and $\mathbb{E}[|\ln(h(\boldsymbol{y}^o))|] < \infty$. Thus, we can apply the law of large numbers to obtain the consistency of the sample KL divergence. $\qquad\square$

**Proposition 3.** *Assume that*

1. *$(\boldsymbol{y}^o_1, \ldots, \boldsymbol{y}^o_n)$ are i.i.d. observed vectors with unknown density $h$.*

2. *$\boldsymbol{\Theta}$ is a compact metric space.*

3. *$\boldsymbol{\theta} \in \boldsymbol{\Theta} \mapsto \ln[f(\boldsymbol{y}^o; \boldsymbol{\theta})]$ is continuous for every $\boldsymbol{y}^o \in \mathbb{R}^{|\mathbb{O}|}$.*

4. *$F$ is an envelope function of $\mathcal{F} := \{\ln[f(\cdot; \boldsymbol{\theta})]; \boldsymbol{\theta} \in \boldsymbol{\Theta}\}$ which is $h$-integrable.*

5. *$\boldsymbol{\theta}^* = \operatorname{argmax}_{\boldsymbol{\theta} \in \boldsymbol{\Theta}} D_{\mathrm{KL}}[h, f(\cdot; \boldsymbol{\theta})]$*

6. *$\hat{\boldsymbol{\theta}} = \operatorname{argmax}_{\boldsymbol{\theta} \in \boldsymbol{\Theta}} \sum_{n=1}^N \ln f(\boldsymbol{y}_n; \boldsymbol{\theta}).$*

*Then, as $n \to \infty$,*

$$\frac{1}{N} \sum_{n=1}^N \ln f(\boldsymbol{y}^o_n; \hat{\boldsymbol{\theta}}_{(\mathbb{S}, \mathbb{R})}) \xrightarrow[n \to \infty]{P} \mathbb{E}_{\boldsymbol{y}^o}\Big[\ln f(\boldsymbol{y}^o; \boldsymbol{\theta}^*_{(\mathbb{S}, \mathbb{R})})\Big].$$

*Proof.* We consider the following inequality:

$$\left| \mathbb{E}\Big[\ln f(\boldsymbol{y}^o; \boldsymbol{\theta}^*_{(\mathbb{S}, \mathbb{R})})\Big] - \frac{1}{N} \sum_{n=1}^N \ln f(\boldsymbol{y}^o_n; \hat{\boldsymbol{\theta}}_{(\mathbb{S}, \mathbb{R})}) \right|$$

$$\leq \left| \mathbb{E}\left[\ln f(\boldsymbol{y}^o; \boldsymbol{\theta}^*_{(\mathbb{S},\mathbb{R})})\right] - \mathbb{E}\left[\ln f(\boldsymbol{y}^o; \hat{\boldsymbol{\theta}}_{(\mathbb{S},\mathbb{R})})\right]\right| + \sup_{\boldsymbol{\theta} \in \boldsymbol{\Theta}_{(\mathbb{S},\mathbb{R})}} \left| \mathbb{E}\left[\ln f(\boldsymbol{y}^o; \boldsymbol{\theta})\right] - \frac{1}{N}\sum_{n=1}^{N}\ln f(\boldsymbol{y}^o_n; \boldsymbol{\theta})\right|.$$

By the definition of $\boldsymbol{\theta}^*_{(\mathbb{S},\mathbb{R})}$, we have

$$\mathbb{E}\left[\ln f(\boldsymbol{y}^o; \boldsymbol{\theta}^*_{(\mathbb{S},\mathbb{R})})\right] - \mathbb{E}\left[\ln f(\boldsymbol{y}^o; \hat{\boldsymbol{\theta}}_{(\mathbb{S},\mathbb{R})})\right] \geq 0.$$

Note that,

$$\mathbb{E}\left[\ln f(\boldsymbol{y}^o; \boldsymbol{\theta}^*_{(\mathbb{S},\mathbb{R})})\right] - \mathbb{E}\left[\ln f(\boldsymbol{y}^o; \hat{\boldsymbol{\theta}}_{(\mathbb{S},\mathbb{R})})\right] = \underbrace{\mathbb{E}\left[\ln f(\boldsymbol{y}^o; \boldsymbol{\theta}^*_{(\mathbb{S},\mathbb{R})})\right] - \frac{1}{N}\sum_{n=1}^{N}\ln f(\boldsymbol{y}^o; \boldsymbol{\theta}^*_{(\mathbb{S},\mathbb{R})})}_{I}$$

$$+ \underbrace{\frac{1}{N}\sum_{n=1}^{N}\ln f(\boldsymbol{y}^o; \boldsymbol{\theta}^*_{(\mathbb{S},\mathbb{R})}) - \frac{1}{N}\sum_{n=1}^{N}\ln f(\boldsymbol{y}^o; \hat{\boldsymbol{\theta}}_{(\mathbb{S},\mathbb{R})})}_{II}$$

$$- \underbrace{\frac{1}{N}\sum_{n=1}^{N}\ln f(\boldsymbol{y}^o; \hat{\boldsymbol{\theta}}_{(\mathbb{S},\mathbb{R})}) + \mathbb{E}\left[\ln f(\boldsymbol{y}^o; \hat{\boldsymbol{\theta}}_{(\mathbb{S},\mathbb{R})})\right]}_{III}$$

For term $II$, sine $\hat{\boldsymbol{\theta}}$ maximizes the empirical log-likelihood,

$$\frac{1}{N}\sum_{n=1}^{N}\ln(f(X_n|\hat{\boldsymbol{\theta}}_{(\mathbb{S},\mathbb{R})}) \geq \frac{1}{N}\sum_{n=1}^{N}\ln(f(X_n|\boldsymbol{\theta}^*_{(\mathbb{S},\mathbb{R})}),$$

which implies that $II \leq 0$. Therefore,

$$\mathbb{E}\left[\ln f(\boldsymbol{y}^o; \boldsymbol{\theta}^*_{(\mathbb{S},\mathbb{R})})\right] - \mathbb{E}\left[\ln f(\boldsymbol{y}^o; \hat{\boldsymbol{\theta}}_{(\mathbb{S},\mathbb{R})})\right] \leq |I| + |III|.$$

Moreover, since both $|I|$ and $|III|$ are bounded by the uniform deviation:

$$|I| \leq \sup_{\boldsymbol{\theta} \in \boldsymbol{\Theta}_{(\mathbb{S},\mathbb{R})}} \left| \mathbb{E}\left[\ln f(\boldsymbol{y}^o; \boldsymbol{\theta})\right] - \frac{1}{N}\sum_{n=1}^{N}\ln f(\boldsymbol{y}^o_n; \boldsymbol{\theta})\right|$$

and

$$|II| \leq \sup_{\boldsymbol{\theta} \in \boldsymbol{\Theta}_{(\mathbb{S},\mathbb{R})}} \left| \mathbb{E}\left[\ln f(\boldsymbol{y}^o; \boldsymbol{\theta})\right] - \frac{1}{N}\sum_{n=1}^{N}\ln f(\boldsymbol{y}^o_n; \boldsymbol{\theta})\right|$$

Hence,

$$\mathbb{E}\left[\ln f(\boldsymbol{y}^o; \boldsymbol{\theta}^*_{(\mathbb{S},\mathbb{R})})\right] - \mathbb{E}\left[\ln f(\boldsymbol{y}^o; \hat{\boldsymbol{\theta}}_{(\mathbb{S},\mathbb{R})})\right] \leq |I| + |III|$$

$$\leq 2\sup_{\boldsymbol{\theta} \in \boldsymbol{\Theta}_{(\mathbb{S},\mathbb{R})}} \left| \mathbb{E}\left[\ln f(\boldsymbol{y}^o; \boldsymbol{\theta})\right] - \frac{1}{N}\sum_{n=1}^{N}\ln f(\boldsymbol{y}^o_n; \boldsymbol{\theta})\right|.$$

Hence, the left-hand side is bounded by three times the uniform deviation:

$$\left| \mathbb{E}\left[\ln f(\boldsymbol{y}^o; \boldsymbol{\theta}^*_{(\mathbb{S},\mathbb{R})})\right] - \frac{1}{N}\sum_{n=1}^{N}\ln f(\boldsymbol{y}^o_n; \hat{\boldsymbol{\theta}}_{(\mathbb{S},\mathbb{R})})\right| \leq 3\sup_{\boldsymbol{\theta} \in \boldsymbol{\Theta}_{(\mathbb{S},\mathbb{R})}} \left| \mathbb{E}\left[\ln f(\boldsymbol{y}^o; \boldsymbol{\theta})\right] - \frac{1}{N}\sum_{n=1}^{N}\ln f(\boldsymbol{y}^o_n; \boldsymbol{\theta})\right|.$$

By the argument in the Proposition 2-namely, using the compactness of $\boldsymbol{\Theta}_{(\mathbb{S},\mathbb{R})}$, the continuity of $\boldsymbol{\theta} \mapsto \ln f(\cdot; \boldsymbol{\theta})$, and the existence of an $h$-integrable envelope $F$-the class $\mathcal{F}_{(\mathbb{S},\mathbb{R})}$ is P-Glivenko-Cantelli by applying Example 19.8 in van der Vaart (1998) to conclude on the finiteness of bracketing numbers of $\mathcal{F}$ under the assumptions. In particular,

$$\sup_{\boldsymbol{\theta} \in \boldsymbol{\Theta}_{(\mathbb{S},\mathbb{R})}} \left| \mathbb{E}\left[\ln f(\boldsymbol{y}^o; \boldsymbol{\theta})\right] - \frac{1}{N}\sum_{n=1}^{N}\ln f(\boldsymbol{y}^o_n; \boldsymbol{\theta})\right| \xrightarrow[n \to \infty]{P} 0.$$

Therefore,

$$\frac{1}{N} \sum_{n=1}^{N} \ln f(\boldsymbol{y}_n^o; \hat{\boldsymbol{\theta}}_{(\mathbb{S},\mathbb{R})}) \xrightarrow[n\to\infty]{P} \mathbb{E}\Big[\ln f(\boldsymbol{y}^o; \boldsymbol{\theta}_{(\mathbb{S},\mathbb{R})}^*)\Big].$$

which concludes the proof of Proposition 2. □

Now, we prove the consistency of the sample KL in the $\mathbb{SRUW}$ model:

**Proposition 4.** *Under Assumption 4 and 5, for all $\mathbb{V}$,*

$$\frac{1}{N} \sum_{n=1}^{N} \ln\left(\frac{h(\boldsymbol{y}_n^o)}{f(\boldsymbol{y}_n^o; \hat{\boldsymbol{\theta}}_{\mathbb{V}})}\right) \xrightarrow[n\to\infty]{P} D_{\mathrm{KL}}\Big[h, f(\cdot; \boldsymbol{\theta}_{\mathbb{V}}^*)\Big].$$

*Proof.* We carry the proof in the same manner as in the $\mathbb{SR}$ model. Let $\mathbb{V} = (\mathbb{S}, \mathbb{R}, \mathbb{U}, \mathbb{W})$ and $\mathbb{O} \subset \{1, \ldots, D\}$ denotes the observed variables. We still want to apply the Proposition 3 with the new family

$$\mathcal{F}_{(\mathbb{V})} := \{\ln[f(\cdot|\boldsymbol{\theta})]; \boldsymbol{\theta} \in \boldsymbol{\Theta}_{\mathbb{V}}'\}$$

similarly to the proof of Proposition 2 to achieve

$$\frac{1}{N} \sum_{n=1}^{N} \ln\big(f(\boldsymbol{y}_n^o; \hat{\boldsymbol{\theta}}_{\mathbb{V}})\big) \xrightarrow[n\to\infty]{P} \mathbb{E}_{\boldsymbol{y}^o}\big[\ln f(\boldsymbol{y}^o; \boldsymbol{\theta}_{\mathbb{V}}^*)\big].$$

To do so, we must initially verify that the class of functions

$$\mathcal{F}_{(\mathbb{V})} = \Big\{\boldsymbol{y}^o \mapsto \ln f(\boldsymbol{y}^o; \boldsymbol{\theta}) : \boldsymbol{\theta} \in \boldsymbol{\Theta}_{\mathbb{V}}'\Big\}$$

satisfies the conditions for uniform convergence under Assumption 5. In particular, by Assumption 5, the parameter space $\boldsymbol{\Theta}_{\mathbb{V}}'$ is compact, and for every $\boldsymbol{y}^o \in \mathbb{R}^{|\mathbb{O}|}$ the mapping

$$\boldsymbol{\theta} \mapsto \ln f(\boldsymbol{y}^o; \boldsymbol{\theta})$$

is continuous. Next, we verify that there is an $h$-integrable envelope function $F \in \mathcal{F}_{(\mathbb{V})}$. Recalling that for $\boldsymbol{y}^o \in \mathbb{R}^{|\mathbb{O}|}$ and a given component $k$, the current Gaussian component density is

$$\Phi\Big(\boldsymbol{y}^o; \tilde{\boldsymbol{\nu}}_{k,o}, \tilde{\boldsymbol{\Delta}}_{k,oo}\Big) = (2\pi)^{-\frac{|\mathbb{O}|}{2}} |\tilde{\boldsymbol{\Delta}}_{k,oo}|^{-\frac{1}{2}} \exp\Big(-\tfrac{1}{2}(\boldsymbol{y}^o - \tilde{\boldsymbol{\nu}}_{k,o})^T \tilde{\boldsymbol{\Delta}}_{k,oo}^{-1}(\boldsymbol{y}^o - \tilde{\boldsymbol{\nu}}_{k,o})\Big).$$

We will bound the density function as usual. From Proposition 2, we know that there exists $\tilde{s}_m > 0$ and $\tilde{s}_M' = \max\{s_M(1 + \rho^2), s_M\rho, s_M\} = s_M(1 + \rho^2)$ such that

$$\tilde{s}_m I \preceq \boldsymbol{\Delta}_k \preceq \tilde{s}_M I.$$

Thus,

$$\tilde{s}_m I_{|\mathbb{O}|} \preceq \boldsymbol{\Delta}_{k,oo} \preceq \tilde{s}_M I_{|\mathbb{O}|},$$

and

$$(\boldsymbol{\Delta}_{k,oo})^{-1} \preceq \frac{1}{\tilde{s}_m} I_{|\mathbb{O}|}.$$

Given that $\boldsymbol{\Gamma} \in \mathcal{D}_{|\mathbb{W}|}$ and is independent of any relevant variables, we have that the principal sub-matrix $\boldsymbol{\Gamma}_{oo}$ is bounded by $s_M$

$$\lambda_{\max}(\boldsymbol{\Gamma}_{oo}) \leq s_M.$$

Since $\tilde{\boldsymbol{\Delta}}_{k,oo}$ is block-diagonal, its eigenvalues are the union of the eigenvalues of $\boldsymbol{\Delta}_{k,oo}$ and $\boldsymbol{\Gamma}_{oo}$. Hence,

$$\lambda_{\max}(\tilde{\boldsymbol{\Delta}}_{k,oo}) \leq \max\{s_M(1 + \rho^2), s_M\} = s_M(1 + \rho^2).$$

We define the upper bound by

$$\tilde{s}_M := s_M(1 + \rho^2).$$

The lower bound of the structured block, together with the lower bound on the independent block, implies that the covariance $\tilde{\boldsymbol{\Delta}}_{k,oo}$ satisfies

$$\tilde{s}_m I \preceq \tilde{\boldsymbol{\Delta}}_{k,oo} \preceq \tilde{s}_M I.$$

Because $\tilde{\mathbf{\Delta}}_{k,oo}^{-1} \preceq \frac{1}{\tilde{s}_m} I$, it follows that

$$(\boldsymbol{y}^o - \tilde{\boldsymbol{\nu}}_{k,o})^T \tilde{\mathbf{\Delta}}_{k,oo}^{-1} (\boldsymbol{y}^o - \tilde{\boldsymbol{\nu}}_{k,o}) \le \frac{\|\boldsymbol{y}^o - \tilde{\boldsymbol{\nu}}_{k,o}\|^2}{\tilde{s}_m}.$$

Using the elementary inequality

$$\|\boldsymbol{y}^o - \tilde{\boldsymbol{\nu}}_{k,o}\|^2 \le 2\Big(\|\boldsymbol{y}^o\|^2 + \|\tilde{\boldsymbol{\nu}}_{k,o}\|^2\Big),$$

we obtain

$$(\boldsymbol{y}^o - \tilde{\boldsymbol{\nu}}_{k,o})^T \tilde{\mathbf{\Delta}}_{k,oo}^{-1} (\boldsymbol{y}^o - \tilde{\boldsymbol{\nu}}_{k,o}) \le \frac{2\Big(\|\boldsymbol{y}^o\|^2 + \|\tilde{\boldsymbol{\nu}}_{k,o}\|^2\Big)}{\tilde{s}_m}.$$

Now we process the bound for $\tilde{\boldsymbol{\nu}}_{k,o}$. Recalling that if we denote $D_{\mathbb{SU}}$ denote the number of coordinates in the structured part (i.e. $\mathbb{S} \cup \mathbb{U}$); then

$$\|\boldsymbol{\nu}_{k,o}\|^2 = \sum_{j \in o} \|\nu_{kj}\|^2 \le D_{\mathbb{SU},o} \, \rho^2(1 + \eta^2)$$

is uniformly bounded and $\|\gamma_o\|^2 \le \eta^2$ since $\gamma$ belongs to the compact set $\mathcal{B}(\eta, |\mathbb{W}|)$, it follows that

$$\|\tilde{\boldsymbol{\nu}}_{k,o}\|^2 \le D_{\mathbb{SU},o} \rho^2 (1 + \eta)^2 + \eta^2,$$

We deduce that

$$\ln \Phi\Big(\boldsymbol{y}^o; \tilde{\boldsymbol{\nu}}_{k,o}, \tilde{\mathbf{\Delta}}_{k,oo}\Big) \ge -\frac{|\mathbb{O}|}{2} \ln\big(2\pi \, \tilde{s}_M\big) - \frac{\|\boldsymbol{y}^o\|^2 + D_{\mathbb{SU},o}\rho^2(1 + \eta^2) + \eta^2}{\tilde{s}_m}.$$

Using Jensen's inequality over the mixture and $\sum_{k=1}^K \pi_k = 1$, we have

$$\ln f(\boldsymbol{y}^o; \boldsymbol{\theta}) \ge \sum_{k=1}^K \pi_k \left( -\frac{|\mathbb{O}|}{2} \ln(2\pi \tilde{s}_M) - \frac{\|\boldsymbol{y}^o\|^2 + D_{\mathbb{SU},o}\rho^2(1 + \eta^2) + \eta^2}{\tilde{s}_m} \right).$$

$$\ge -\frac{|\mathbb{O}|}{2} \ln(2\pi \tilde{s}_M) - \frac{\|\boldsymbol{y}^o\|^2 + D_{\mathbb{SU},o}\rho^2(1 + \eta^2) + \eta^2}{\tilde{s}_m}$$

Therefore, each family's member in $\mathcal{F}_{(\mathbb{V})}$ is bounded by

$$-\frac{|\mathbb{O}|}{2} \ln(2\pi \tilde{s}_M) - \frac{\|\boldsymbol{y}^o\|^2 + D_{\mathbb{SU},o}\rho^2(1 + \eta^2) + \eta^2}{\tilde{s}_m} \le \ln f(\boldsymbol{y}^o; \boldsymbol{\theta}_{(\mathbb{V})}) \le -\frac{|\mathbb{O}|}{2} \ln(2\pi \tilde{s}_m).$$

Now we prove that $F$ is $h$-integrable. The true observed-data density under the $\mathbb{SRUW}$ model is given by

$$h(\boldsymbol{y}^o) = \sum_{k=1}^K \pi_k \, \Phi\Big(\boldsymbol{y}^o; \tilde{\boldsymbol{\nu}}_{k,o}^*, \tilde{\Delta}_{k,oo}^*\Big),$$

where the "$*$" indicates that the parameters are the true ones and the density is that of a Gaussian mixture with parameters in a compact set. In our derivation we have shown that, for any $\boldsymbol{\theta}$ in the family $\mathcal{F}_{(\mathbb{V})}$, the log-density satisfies

$$-\frac{|\mathbb{O}|}{2} \ln(2\pi \tilde{s}_M) - \frac{\|\boldsymbol{y}^o\|^2 + D_{\mathbb{SU},o} \, \rho^2(1 + \eta^2) + \eta^2}{\tilde{s}_m} \le \ln f(\boldsymbol{y}^o; \boldsymbol{\theta}) \le -\frac{|\mathbb{O}|}{2} \ln(2\pi \tilde{s}_m).$$

In other words, every function $\ln f(\boldsymbol{y}^o; \boldsymbol{\theta})$ is bounded in absolute value by

$$F(\boldsymbol{y}^o) = \frac{|\mathbb{O}|}{2} \ln\big(2\pi \tilde{s}_M\big) + \frac{\|\boldsymbol{y}^o\|^2 + D_{\mathbb{SU},o} \, \rho^2(1 + \eta^2) + \eta^2}{\tilde{s}_m}.$$

Hence, for all $\boldsymbol{\theta}$ in the compact parameter space,

$$\Big|\ln f(\boldsymbol{y}^o; \boldsymbol{\theta})\Big| \le F(\boldsymbol{y}^o).$$

To verify that $F$ is $h$-integrable, we must show that

$$\int F(\boldsymbol{y}^o)\, h(\boldsymbol{y}^o)\, d\boldsymbol{y}^o < \infty.$$

Since the envelope function $F$ is of the form

$$F(\boldsymbol{y}^o) = C_0 + \frac{1}{\tilde{s}_m}\|\boldsymbol{y}^o\|^2,$$

with

$$C_0 = \frac{|\mathbb{O}|}{2}\ln\big(2\pi\tilde{s}_M\big) + \frac{D_{\mathbb{SU},o}\,\rho^2(1+\eta^2)+\eta^2}{\tilde{s}_m},$$

it is clear that

$$\int F(\boldsymbol{y}^o)\, h(\boldsymbol{y}^o)\, d\boldsymbol{y}^o \le C_0 + \frac{1}{\tilde{s}_m}\int \|\boldsymbol{y}^o\|^2\, h(\boldsymbol{y}^o)\, d\boldsymbol{y}^o.$$

Because $h(\boldsymbol{y}^o)$ is a Gaussian mixture with parameters in a compact set, standard properties of Gaussian mixtures guarantee that the second moment is finite, that is,

$$\int \|\boldsymbol{y}^o\|^2\, h(\boldsymbol{y}^o)\, d\boldsymbol{y}^o < \infty.$$

Thus,

$$\int F(\boldsymbol{y}^o)\, h(\boldsymbol{y}^o)\, d\boldsymbol{y}^o < \infty.$$

Finally, since for every $\boldsymbol{y}^o$ we have

$$\big|\ln h(\boldsymbol{y}^o)\big| \le F(\boldsymbol{y}^o),$$

It follows that

$$\mathbb{E}\Big[\big|\ln h(\boldsymbol{y}^o)\big|\Big] = \int |\ln h(\boldsymbol{y}^o)|\, h(\boldsymbol{y}^o)\, d\boldsymbol{y}^o \le \int F(\boldsymbol{y}^o)\, h(\boldsymbol{y}^o)\, d\boldsymbol{y}^o < \infty.$$

Therefore, the envelope function $F$ is $h$-integrable and $\mathbb{E}[|\ln(h(\boldsymbol{y}^o))|] < \infty$, and consequently, we can apply the law of large numbers and uniform convergence to conclude the proof. $\qquad\square$

*Proof of Theorem 2.* Define the BIC score of a variable partition $\mathbb{V} = (\mathbb{S}, \mathbb{R}, \mathbb{U}, \mathbb{W})$ by

$$\mathrm{BIC}(\mathbb{V}) = 2\ell_N\big(\hat{\boldsymbol{\theta}}_{\mathbb{V}}\big) - \Xi_{(\mathbb{V})}\log N,$$

where $\Xi_{(\mathbb{V})}$ is the number of free parameters. Let $\mathbb{V}_0$ be the true partition and set $\Delta\mathrm{BIC}(\mathbb{V}) = \mathrm{BIC}(\mathbb{V}_0) - \mathrm{BIC}(\mathbb{V})$.

Let $\mathbb{V}_1 = \big\{\mathbb{V} \ne \mathbb{V}_0 : D_{\mathrm{KL}}[h, f(\cdot; \boldsymbol{\theta}_{\mathbb{V}}^\star)] > 0\big\}$. The Identifiability Theorem 1 implies every $\mathbb{V} \ne \mathbb{V}_0$ belongs to $\mathbb{V}_1$. For $\mathbb{V} \in \mathbb{V}_1$,

$$\Delta\mathrm{BIC}(\mathbb{V}) = 2N\left[\frac{1}{N}\sum_{n=1}^N \ln\left(\frac{f(\boldsymbol{y}_n^o; \hat{\boldsymbol{\theta}}_{\mathbb{V}_0})}{h(\boldsymbol{y}_n^o)}\right) - \frac{1}{N}\sum_{n=1}^N \ln\left(\frac{f(\boldsymbol{y}_n^o; \hat{\boldsymbol{\theta}}_{\mathbb{V}})}{h(\boldsymbol{y}_n^o)}\right)\right] + \big[\Xi_{(\mathbb{V})} - \Xi_{(\mathbb{V}_0)}\big]\log N \tag{19}$$

To prove the theorem, we will prove that:

$$\forall\, \mathbb{V} \in \mathbb{V}_1, \mathbb{P}[\Delta\mathrm{BIC}(\mathbb{V}) < 0] \underset{N\to\infty}{\to} 0$$

Denoting $\mathbb{M}_N(\mathbb{V}) = \frac{1}{N}\sum_{n=1}^N \ln\left(\frac{f(\boldsymbol{y}_n^o; \hat{\boldsymbol{\theta}}_{\mathbb{V}})}{h(\boldsymbol{y}_n^o)}\right)$, $M(\mathbb{V}) = -D_{\mathrm{KL}}[h, f(\cdot; \boldsymbol{\theta}_{\mathbb{V}}^*)]$, from Equation (19), we get:

$$\mathbb{P}(\Delta\mathrm{BIC}(\mathbb{V}) < 0)$$
$$= \mathbb{P}(2N(\mathbb{M}_N(\mathbb{V}_0) - \mathbb{M}_N(\mathbb{V})) + \big[\Xi_{(\mathbb{V})} - \Xi_{(\mathbb{V}_0)}\big]\log N < 0)$$
$$= \mathbb{P}(\mathbb{M}_N(\mathbb{V}_0) - M(\mathbb{V}_0) + M(\mathbb{V}_0) - M(\mathbb{V}) + M(\mathbb{V}) - \mathbb{M}_N(\mathbb{V}) + \frac{\big[\Xi_{(\mathbb{V})} - \Xi_{(\mathbb{V}_0)}\big]\log N}{2N} < 0)$$
$$\le \mathbb{P}(\mathbb{M}_N(\mathbb{V}_0) - M(\mathbb{V}_0) > \epsilon) + \mathbb{P}(M(\mathbb{V}) - \mathbb{M}_N(\mathbb{V}) > \epsilon) + \mathbb{P}(M(\mathbb{V}_0) - M(\mathbb{V}) + \frac{\big[\Xi_{(\mathbb{V})} - \Xi_{(\mathbb{V}_0)}\big]\log N}{2N} < 2\epsilon)$$

From Proposition 4, we know that

$$\frac{1}{N}\sum_{n=1}^{N}\ln\left(\frac{h(\boldsymbol{y}_n^o)}{f(\boldsymbol{y}_n^o;\hat{\boldsymbol{\theta}}_{\mathbb{V}})}\right) \xrightarrow[n\to\infty]{P} D_{\mathrm{KL}}\Big[h, f(\cdot;\boldsymbol{\theta}_{\mathbb{V}}^*)\Big].$$

This leads to $\mathbb{M}_N(\mathbb{V}) \xrightarrow[n\to\infty]{P} M(\mathbb{V})$. Similar to the proof for the $\mathbb{SRUW}$ model in Maugis [38], we prove the theorem. $\qquad\square$

### D.3 Proof of Theorem 3

Throughout this proof, we analyze the estimator defined by minimizing the *negative log-likelihood* penalized. Accordingly, although we denote by

$$\ell(\boldsymbol{y};\boldsymbol{\alpha}) = \ln f(\boldsymbol{y};\boldsymbol{\alpha})$$

the usual log-likelihood of a single observation, we consistently work with its negative $-\ell$ as the optimization objective. To streamline notation, we therefore use the term "score" to mean the gradient of the *negative* log-likelihood:

$$S_j(\boldsymbol{y}_n;\boldsymbol{\alpha}) := -\frac{\partial\ell(\boldsymbol{y}_n;\boldsymbol{\alpha})}{\partial\alpha_j}.$$

This convention differs by a minus sign from the standard statistical score, but it ensures that all gradients, Hessians, and Fisher information matrices below are taken with respect to the minimized objective $-\ell_N$.

We start with presenting some useful lemmas before proving our main theorem. These lemmas establish fundamental properties of the penalized GMM estimators and the variable ranking procedure.

**Lemma 1** (Score Function Components)**.** *Let $f_{clust}(\boldsymbol{y}_n;\boldsymbol{\alpha}) = \sum_{m=1}^{K}\pi_m\Phi(\boldsymbol{y}_n;\boldsymbol{\mu}_m,\boldsymbol{\Sigma}_m)$ be the p.d.f of a $K$-component GMM for an observation $\boldsymbol{y}_n$, where $\boldsymbol{\alpha} = (\boldsymbol{\pi},\{\boldsymbol{\mu}_k\}_{k=1}^{K},\{\boldsymbol{\Sigma}_k\}_{k=1}^{K})$ represents the GMM parameters. Let $\boldsymbol{\Psi}_k = \boldsymbol{\Sigma}_k^{-1}$ be the precision matrix for component $k$. The log-likelihood for observation $\boldsymbol{y}_n$ is $\ell(\boldsymbol{y}_n;\boldsymbol{\alpha}) = \ln f_{clust}(\boldsymbol{y}_n;\boldsymbol{\alpha})$. The responsibility for component $k$ given observation $\boldsymbol{y}_n$ and parameters $\boldsymbol{\alpha}$ is $t_k(\boldsymbol{y}_n;\boldsymbol{\alpha}) = \frac{\pi_k\Phi(\boldsymbol{y}_n;\boldsymbol{\mu}_k,\boldsymbol{\Sigma}_k)}{f_{clust}(\boldsymbol{y}_n;\boldsymbol{\alpha})}$. The score components, $S_j(\boldsymbol{y}_n;\boldsymbol{\alpha}^*)$, where $\alpha_j$ is a parameter in $\boldsymbol{\alpha}$ and $\boldsymbol{\alpha}^*$ are the true parameter values, are given as follows:*

1. ***Mixing proportions** $\pi_k$: When parameterizing $\pi_K = 1 - \sum_{j=1}^{K-1}\pi_j$, the score component for $\pi_k$, $k \in \{1,\ldots,K-1\}$, is:*

$$S_{\pi_k}(\boldsymbol{y}_n;\boldsymbol{\alpha}^*) = \frac{t_k(\boldsymbol{y}_n;\boldsymbol{\alpha}^*)}{\pi_K^*} - \frac{t_k(\boldsymbol{y}_n;\boldsymbol{\alpha}^*)}{\pi_k^*}$$

2. ***Mean parameters** $(\boldsymbol{\mu}_k)_d$: Let $(\boldsymbol{\mu}_k)_d$ be the $d$-th component of the mean vector $\boldsymbol{\mu}_k$. The score component is:*

$$S_{(\boldsymbol{\mu}_k)_d}(\boldsymbol{y}_n;\boldsymbol{\alpha}^*) = -t_k(\boldsymbol{y}_n;\boldsymbol{\alpha}^*)(\boldsymbol{\Psi}_k^*(\boldsymbol{y}_n - \boldsymbol{\mu}_k^*))_d$$

3. ***Precision matrix parameters** $(\boldsymbol{\Psi}_k)_{rs}$: Let $(\boldsymbol{\Psi}_k)_{rs}$ be the element $(r,s)$ of the symmetric precision matrix $\boldsymbol{\Psi}_k$. The score component is:*

$$S_{(\boldsymbol{\Psi}_k)_{rs}}(\boldsymbol{y}_n;\boldsymbol{\alpha}^*) = -t_k(\boldsymbol{y}_n;\boldsymbol{\alpha}^*)\frac{(2-\delta_{rs})}{2}\left((\boldsymbol{\Sigma}_k^*)_{rs} - (\boldsymbol{y}_n-\boldsymbol{\mu}_k^*)_r(\boldsymbol{y}_n-\boldsymbol{\mu}_k^*)_s\right)$$

*where $\delta_{rs}$ is the Kronecker delta.*

*Proof of Lemma 1.* Let $\Phi_m(\boldsymbol{y}_n;\boldsymbol{\alpha}) = \Phi(\boldsymbol{y}_n;\boldsymbol{\mu}_m,\boldsymbol{\Sigma}_m)$. The log-likelihood for a single observation $\boldsymbol{y}_n$ is $\ell(\boldsymbol{y}_n;\boldsymbol{\alpha}) = \ln\left(\sum_{m=1}^{K}\pi_m\Phi_m(\boldsymbol{y}_n;\boldsymbol{\alpha})\right)$. The score component for a generic parameter $\alpha_j$ is $S_j(\boldsymbol{y}_n;\boldsymbol{\alpha}^*) = -\frac{\partial\ell(\boldsymbol{y}_n;\boldsymbol{\alpha})}{\partial\alpha_j}|_{\boldsymbol{\alpha}=\boldsymbol{\alpha}^*}.$

**Score components for mixing proportions** $\pi_k$**.** We parameterize $\pi_K = 1 - \sum_{j=1}^{K-1} \pi_j$. For $k \in \{1, \ldots, K-1\}$:

$$\frac{\partial \ell(\boldsymbol{y}_n; \boldsymbol{\alpha})}{\partial \pi_k} = \frac{1}{f_{\text{clust}}(\boldsymbol{y}_n; \boldsymbol{\alpha})} \frac{\partial}{\partial \pi_k} \left( \sum_{m=1}^{K-1} \pi_m \Phi_m(\boldsymbol{y}_n; \boldsymbol{\alpha}) + \left( 1 - \sum_{j=1}^{K-1} \pi_j \right) \Phi_K(\boldsymbol{y}_n; \boldsymbol{\alpha}) \right)$$

$$= \frac{\Phi_k(\boldsymbol{y}_n; \boldsymbol{\alpha}) - \Phi_K(\boldsymbol{y}_n; \boldsymbol{\alpha})}{f_{\text{clust}}(\boldsymbol{y}_n; \boldsymbol{\alpha})}$$

Since $t_m(\boldsymbol{y}_n; \boldsymbol{\alpha}) = \frac{\pi_m \Phi_m(\boldsymbol{y}_n; \boldsymbol{\alpha})}{f_{\text{clust}}(\boldsymbol{y}_n; \boldsymbol{\alpha})}$, we have $\frac{\Phi_m(\boldsymbol{y}_n; \boldsymbol{\alpha})}{f_{\text{clust}}(\boldsymbol{y}_n; \boldsymbol{\alpha})} = \frac{t_m(\boldsymbol{y}_n; \boldsymbol{\alpha})}{\pi_m}$. Thus,

$$\frac{\partial \ell(\boldsymbol{y}_n; \boldsymbol{\alpha})}{\partial \pi_k} = \frac{t_k(\boldsymbol{y}_n; \boldsymbol{\alpha})}{\pi_k} - \frac{t_k(\boldsymbol{y}_n; \boldsymbol{\alpha})}{\pi_K}$$

The score component at $\boldsymbol{\alpha}^*$ is:

$$S_{\pi_k}(\boldsymbol{y}_n; \boldsymbol{\alpha}^*) = - \left( \frac{t_k(\boldsymbol{y}_n; \boldsymbol{\alpha}^*)}{\pi_k^*} - \frac{t_k(\boldsymbol{y}_n; \boldsymbol{\alpha}^*)}{\pi_K^*} \right) = \frac{t_k(\boldsymbol{y}_n; \boldsymbol{\alpha}^*)}{\pi_K^*} - \frac{t_k(\boldsymbol{y}_n; \boldsymbol{\alpha}^*)}{\pi_k^*}.$$

**Score components for mean parameters** $(\boldsymbol{\mu}_k)_d$**.** Let $(\boldsymbol{\mu}_k)_d$ be the $d$-th component of $\boldsymbol{\mu}_k$.

$$\frac{\partial \ell(\boldsymbol{y}_n; \boldsymbol{\alpha})}{\partial (\boldsymbol{\mu}_k)_d} = \frac{1}{f_{\text{clust}}(\boldsymbol{y}_n; \boldsymbol{\alpha})} \frac{\partial}{\partial (\boldsymbol{\mu}_k)_d} \left( \sum_{m=1}^{K} \pi_m \Phi_m(\boldsymbol{y}_n; \boldsymbol{\alpha}) \right)$$

$$= \frac{\pi_k}{f_{\text{clust}}(\boldsymbol{y}_n; \boldsymbol{\alpha})} \frac{\partial \Phi_k(\boldsymbol{y}_n; \boldsymbol{\alpha})}{\partial (\boldsymbol{\mu}_k)_d}$$

Since $\frac{\partial \Phi_k}{\partial (\boldsymbol{\mu}_k)_d} = \Phi_k \frac{\partial \ln \Phi_k}{\partial (\boldsymbol{\mu}_k)_d}$, and for $\Phi_k$, $\ln \Phi_k(\boldsymbol{y}_n; \boldsymbol{\alpha}) = C_k - \frac{1}{2}(\boldsymbol{y}_n - \boldsymbol{\mu}_k)^\top \boldsymbol{\Psi}_k (\boldsymbol{y}_n - \boldsymbol{\mu}_k)$, we have:

$$\frac{\partial \ln \Phi_k(\boldsymbol{y}_n; \boldsymbol{\alpha})}{\partial (\boldsymbol{\mu}_k)_d} = (\boldsymbol{\Psi}_k (\boldsymbol{y}_n - \boldsymbol{\mu}_k))_d$$

Substituting this back:

$$\frac{\partial \ell(\boldsymbol{y}_n; \boldsymbol{\alpha})}{\partial (\boldsymbol{\mu}_k)_d} = \frac{\pi_k \Phi_k(\boldsymbol{y}_n; \boldsymbol{\alpha})}{f_{\text{clust}}(\boldsymbol{y}_n; \boldsymbol{\alpha})} (\boldsymbol{\Psi}_k (\boldsymbol{y}_n - \boldsymbol{\mu}_k))_d = t_k(\boldsymbol{y}_n; \boldsymbol{\alpha})(\boldsymbol{\Psi}_k (\boldsymbol{y}_n - \boldsymbol{\mu}_k))_d$$

The score component at $\boldsymbol{\alpha}^*$ is:

$$S_{(\boldsymbol{\mu}_k)_d}(\boldsymbol{y}_n; \boldsymbol{\alpha}^*) = -t_k(\boldsymbol{y}_n; \boldsymbol{\alpha}^*)(\boldsymbol{\Psi}_k^* (\boldsymbol{y}_n - \boldsymbol{\mu}_k^*))_d.$$

**Score components for precision matrix parameters** $(\boldsymbol{\Psi}_k)_{rs}$**.** Let $(\boldsymbol{\Psi}_k)_{rs}$ be an element of the symmetric precision matrix $\boldsymbol{\Psi}_k$.

$$\frac{\partial \ell(\boldsymbol{y}_n; \boldsymbol{\alpha})}{\partial (\boldsymbol{\Psi}_k)_{rs}} = \frac{\pi_k}{f_{\text{clust}}(\boldsymbol{y}_n; \boldsymbol{\alpha})} \frac{\partial \Phi_k(\boldsymbol{y}_n; \boldsymbol{\alpha})}{\partial (\boldsymbol{\Psi}_k)_{rs}} = t_k(\boldsymbol{y}_n; \boldsymbol{\alpha}) \frac{1}{\pi_k} \pi_k \frac{\partial \ln \Phi_k(\boldsymbol{y}_n; \boldsymbol{\alpha})}{\partial (\boldsymbol{\Psi}_k)_{rs}}$$

The log-density of a single Gaussian component is $\ln \Phi_k(\boldsymbol{y}_n; \boldsymbol{\alpha}) = C_k' + \frac{1}{2} \ln \det(\boldsymbol{\Psi}_k) - \frac{1}{2}(\boldsymbol{y}_n - \boldsymbol{\mu}_k)^\top \boldsymbol{\Psi}_k (\boldsymbol{y}_n - \boldsymbol{\mu}_k)$. For a symmetric matrix $\boldsymbol{\Psi}_k$, the derivative with respect to an element $(\boldsymbol{\Psi}_k)_{rs}$ (using the "symmetric" derivative convention where $\frac{\partial}{\partial X_{rs}}$ means varying $X_{rs}$ and $X_{sr}$ simultaneously if $r \neq s$) is:

$$\frac{\partial \ln \det(\boldsymbol{\Psi}_k)}{\partial (\boldsymbol{\Psi}_k)_{rs}} = (2 - \delta_{rs})(\boldsymbol{\Psi}_k^{-1})_{rs} = (2 - \delta_{rs})(\boldsymbol{\Sigma}_k)_{rs}$$

$$\frac{\partial (\boldsymbol{y}_n - \boldsymbol{\mu}_k)^\top \boldsymbol{\Psi}_k (\boldsymbol{y}_n - \boldsymbol{\mu}_k)}{\partial (\boldsymbol{\Psi}_k)_{rs}} = (2 - \delta_{rs})(\boldsymbol{y}_n - \boldsymbol{\mu}_k)_r (\boldsymbol{y}_n - \boldsymbol{\mu}_k)_s$$

Therefore,

$$\frac{\partial \ln \Phi_k(\boldsymbol{y}_n; \boldsymbol{\alpha})}{\partial (\boldsymbol{\Psi}_k)_{rs}} = \frac{1}{2}(2 - \delta_{rs})(\boldsymbol{\Sigma}_k)_{rs} - \frac{1}{2}(2 - \delta_{rs})(\boldsymbol{y}_n - \boldsymbol{\mu}_k)_r (\boldsymbol{y}_n - \boldsymbol{\mu}_k)_s$$

So,

$$\frac{\partial \ell(\boldsymbol{y}_n; \boldsymbol{\alpha})}{\partial (\boldsymbol{\Psi}_k)_{rs}} = t_k(\boldsymbol{y}_n; \boldsymbol{\alpha}) \frac{(2 - \delta_{rs})}{2} \left( (\boldsymbol{\Sigma}_k)_{rs} - (\boldsymbol{y}_n - \boldsymbol{\mu}_k)_r (\boldsymbol{y}_n - \boldsymbol{\mu}_k)_s \right)$$

The score component at $\boldsymbol{\alpha}^*$ is:

$$S_{(\boldsymbol{\Psi}_k)_{rs}}(\boldsymbol{y}_n; \boldsymbol{\alpha}^*) = -t_k(\boldsymbol{y}_n; \boldsymbol{\alpha}^*) \frac{(2 - \delta_{rs})}{2} \left( (\boldsymbol{\Sigma}_k^*)_{rs} - (\boldsymbol{y}_n - \boldsymbol{\mu}_k^*)_r (\boldsymbol{y}_n - \boldsymbol{\mu}_k^*)_s \right) .$$

$\square$

For the two lemmas below, we will use the following assumptions:

**Assumption 7** (Identifiability & Smoothness). *The GMM density $f_{clust}(\boldsymbol{y}; \boldsymbol{\alpha})$ is identifiable. $\ell_1(\boldsymbol{y}; \boldsymbol{\alpha})$. is three times continuously differentiable w.r.t. $\boldsymbol{\alpha}$ in an open ball $\mathbb{B}(\boldsymbol{\alpha}^*, r_0)$ around $\boldsymbol{\alpha}^*$.*

**Assumption 8** (Compact Parameter Space & True Parameter Properties). *$\boldsymbol{\alpha}^*$ is an interior point of a compact set $\boldsymbol{\Theta}'_{\mathbb{V}} \subset \mathbb{B}(\boldsymbol{\alpha}^*, r_0)$. This implies:*

- *Mixing proportions: $\pi_k^* \geq \pi_{\min} > 0$ for all $k = 1, \ldots, K$, for some constant $\pi_{\min} \in (0, 1/K]$.*

- *Means: $\|\boldsymbol{\mu}_k^*\|_2 \leq \eta < \infty$ for all $k$.*

- *Covariance and Precision Matrices: The covariance matrices $\boldsymbol{\Sigma}_k^*$ have eigenvalues $\lambda(\boldsymbol{\Sigma}_k^*)$ such that $0 < s_m \leq \lambda(\boldsymbol{\Sigma}_k^*) \leq s_M < \infty$. Consequently, for the precision matrices $\boldsymbol{\Psi}_k^* = (\boldsymbol{\Sigma}_k^*)^{-1}$, their eigenvalues $\lambda(\boldsymbol{\Psi}_k^*)$ satisfy $0 < 1/s_M \leq \lambda(\boldsymbol{\Psi}_k^*) \leq 1/s_m < \infty$. We define $\sigma_{\min}^2 = s_m$, $\sigma_{\max}^2 = s_M$. And for precision matrices, $\theta_{\min} = 1/s_M$, $\theta_{\max} = 1/s_m$.*

*All $\boldsymbol{\alpha} \in \boldsymbol{\Theta}'_{\mathbb{V}}$ satisfy these bounds. Let $L_3$ be an upper bound on the norm of the third derivative tensor of $\ell_1(\boldsymbol{y}; \boldsymbol{\alpha})$ for $\boldsymbol{\alpha} \in \boldsymbol{\Theta}'_{\mathbb{V}}$, such that $\mathbb{E}[L_3(\boldsymbol{y})] < \infty$.*

**Assumption 9** (Data Distribution). *Observations $\boldsymbol{y}_1, \ldots, \boldsymbol{y}_N$ are i.i.d. from $f_{clust}(\boldsymbol{y}; \boldsymbol{\alpha}^*)$. For each $\boldsymbol{y}_n$, there exists a latent class variable $\mathrm{z}_n \in \{1, \ldots, K\}$ with $P(\mathrm{z}_n = k) = \pi_k^*$, such that $\boldsymbol{y}_n | \mathrm{z}_n = k \sim \mathcal{N}(\boldsymbol{\mu}_k^*, \boldsymbol{\Sigma}_k^*)$.*

**Assumption 10** (Bounded Posteriors). *For $\boldsymbol{\alpha} \in \boldsymbol{\Theta}'_{\mathbb{V}}$, the posterior probabilities $t_k(\boldsymbol{y}; \boldsymbol{\alpha}) = \frac{\pi_k \phi(\boldsymbol{y} | \boldsymbol{\mu}_k, \boldsymbol{\Sigma}_k)}{\sum_{j=1}^K \pi_j \phi(\boldsymbol{y} | \boldsymbol{\mu}_j, \boldsymbol{\Sigma}_j)}$ satisfy $0 < t_k(\boldsymbol{y}; \boldsymbol{\alpha}) \leq 1$.*

The empirical Hessian is $\boldsymbol{H}_N(\boldsymbol{\alpha}) = \nabla^2 \ell_N(\boldsymbol{\alpha})$. The Fisher Information Matrix is $\boldsymbol{I}(\boldsymbol{\alpha}^*) = \mathbb{E}_{\boldsymbol{\alpha}^*}[\nabla^2 \ell(\boldsymbol{y}; \boldsymbol{\alpha}^*)]$. Moreover, we have the following definition of the true support of penalized parameters:

**Definition 3** (True Support of Penalized Parameters $\mathbb{S}_0$). *Let $\boldsymbol{\alpha}^* = (\boldsymbol{\pi}^*, \{\boldsymbol{\mu}_k^*\}_{k=1}^K, \{\boldsymbol{\Psi}_k^*\}_{k=1}^K)$ be the true GMM parameter vector. Consider the penalty*

$$P(\boldsymbol{\alpha}) = \lambda \sum_{k=1}^K \|\boldsymbol{\mu}_k\|_1 + \rho \sum_{k=1}^K \|\boldsymbol{\Psi}_k\|_1.$$

*We define the true support set $\mathbb{S}_0$ as the collection of indices corresponding to nonzero parameters in $\boldsymbol{\alpha}^*$ that are subject to penalization:*

$$\mathbb{S}_0 = \mathbb{S}_{\boldsymbol{\mu}}^* \cup \mathbb{S}_{\boldsymbol{\Psi}}^*, \qquad s_0 = |\mathbb{S}_0| = s_{\boldsymbol{\mu}}^* + s_{\boldsymbol{\Psi}}^*.$$

*Here:*

1. *Support of means*

$$\mathbb{S}_{\boldsymbol{\mu}}^* = \left\{ (k, d) : 1 \leq k \leq K, \ 1 \leq d \leq p, \ (\boldsymbol{\mu}_k^*)_d \neq 0 \right\}, \quad s_{\boldsymbol{\mu}}^* = |\mathbb{S}_{\boldsymbol{\mu}}^*|.$$

2. *Support of precision matrices*

$$\mathbb{S}_{\boldsymbol{\Psi}}^* = \left\{ (k, r, s) : 1 \leq k \leq K, \ 1 \leq r, s \leq p, \ (\boldsymbol{\Psi}_k^*)_{rs} \neq 0 \right\}, \quad s_{\boldsymbol{\Psi}}^* = |\mathbb{S}_{\boldsymbol{\Psi}}^*|.$$

**Assumption 11** (Restricted Eigenvalue Condition). *For any parameter increment vector $\boldsymbol{\Delta}$ indexed compatibly with $\boldsymbol{\alpha}$, we $\boldsymbol{\Delta}_{\mathbb{S}_0}$ for its restriction to indices in $\mathbb{S}_0$ and $\boldsymbol{\Delta}_{(\mathbb{S}_0)^c}$ for its complement. For $c_0 \geq 1$, define the cone*

$$\mathcal{C}(c_0, \mathbb{S}_0) = \big\{ \boldsymbol{\Delta} : \|\boldsymbol{\Delta}_{(\mathbb{S}_0)^c}\|_1 \leq c_0 \|\boldsymbol{\Delta}_{\mathbb{S}_0}\|_1 \big\}.$$

*Assume the FIM $\boldsymbol{I}(\boldsymbol{\alpha}^*)$ is positive definite, then there exists $\kappa_I > 0$ such that, for all $\boldsymbol{\Delta} \in \mathcal{C}(c_0, \mathbb{S}_0)$,*

$$\boldsymbol{\Delta}^\top \boldsymbol{I}(\boldsymbol{\alpha}^*)\boldsymbol{\Delta} \geq \kappa_I \|\boldsymbol{\Delta}\|_2^2.$$

**Remark 1.** *The constant $c_0$ is chosen to match the cone where the estimation error $\widehat{\boldsymbol{\alpha}} - \boldsymbol{\alpha}^*$ lies. A sufficient choice is $c_0 \geq \frac{A_0+1}{A_0-1}$ when the regularization level satisfies $\lambda \geq A_0 \|\nabla \ell_N(\boldsymbol{\alpha}^*)\|_\infty$ with $A_0 > 1$.*

**Assumption 12** (Uniform Hessian Concentration). *et $d_\alpha = (K-1) + KD + KD(D+1)/2$ denote the number of free parameters in $\boldsymbol{\alpha}$. There exist constants $C_H > 0$, $c_1^H, c_2^H > 0$ and a radius $\delta_R > 0$ such that, for $N \gtrsim s_0 \ln d_\alpha$, with probability at least $1 - c_1^H d_\alpha^{-c_2^H}$,*

$$\sup_{\substack{\tilde{\boldsymbol{\alpha}} \in \mathbb{B}(\boldsymbol{\alpha}^*, \delta_R) \cap \boldsymbol{\Theta}_{\mathbb{V}}' \\ \boldsymbol{\Delta} \in \mathcal{C}(c_0, \mathbb{S}_0),\, \|\boldsymbol{\Delta}\|_2 = 1}} \left| \boldsymbol{\Delta}^\top \big(\boldsymbol{H}_N(\tilde{\boldsymbol{\alpha}}) - \boldsymbol{I}(\tilde{\boldsymbol{\alpha}})\big)\boldsymbol{\Delta} \right| \ \leq \ C_H \sqrt{\frac{s_0 \ln d_\alpha}{N}}.$$

*The constant $C_H$ depends on the bounds in Assumption 8 and on moment bounds for derivatives of $\ell$. The radius can be taken as $\delta_R \asymp \sqrt{s_0 \ln d_\alpha / N}$.*

**Lemma 2** (Gradient Bound). *Let $\boldsymbol{\alpha}^* = (\boldsymbol{\pi}^*, \boldsymbol{\mu}_1^*, \ldots, \boldsymbol{\mu}_K^*, \boldsymbol{\Psi}_1^*, \ldots, \boldsymbol{\Psi}_K^*)$ be the true parameter vector of a $K$-component GMM with $D$-dimensional components. Suppose Assumptions 7-10 hold. Let $d_\alpha$ be defined in Assumption 12. Then there exist absolute constants $C_1, C_2 > 0$ and $C_g > 0$ such that, for any $N \geq 1$,*

$$\mathbb{P}\left( \|\nabla \ell_N(\boldsymbol{\alpha}^*)\|_\infty \leq C_g \sqrt{\frac{\ln d_\alpha}{N}} \right) \ \geq \ 1 - C_1 \, d_\alpha^{-C_2},$$

*where $\ell_N(\boldsymbol{\alpha}) = \frac{1}{N} \sum_{n=1}^N \ell(\boldsymbol{y}_n; \boldsymbol{\alpha})$ is the empirical negative log-likelihood (recall $\ell(\boldsymbol{y}; \boldsymbol{\alpha}) = -\ln f_{clust}(\boldsymbol{y}; \boldsymbol{\alpha})$). An explicit choice is $C_g = \sqrt{2(C_2+1)/c_0}\,\nu_{\max}$, where $c_0$ is an absolute constant in Bernstein's inequality (e.g., $c_0 = 1/2$). This bound holds provided $N \geq C_B \ln d_\alpha$, with*

$$C_B \ = \ \frac{2(C_2+1)\,\nu_{\max}^2}{c_0 \big( \min_{j:\, b_j \neq 0} (\mathbb{E}[S_{jn}^2]/b_j) \big)^2},$$

*$\nu_{\max}^2 = \max_j \mathbb{E}[S_{jn}^2]$, and $b_j$ is the sub-exponential scale of $S_{jn}$ ($b_j = 0$ for sub-Gaussian components).*

*Proof of Lemma 2.* By Assumptions 7 and 9 (regularity and correct specification),

$$\mathbb{E}_{\boldsymbol{\alpha}^*}[S_j(\boldsymbol{y}_n; \boldsymbol{\alpha}^*)] = 0 \quad \text{for all } j = 1, \ldots, d_\alpha$$

Let $\nu_j^2 = \mathbb{E}[S_j(\boldsymbol{y}_n; \boldsymbol{\alpha}^*)^2]$ be the variance of the score evaluated at $\alpha_j^*$. We bound variances and tail parameters for each parameter block.

**Mixing proportions $S_{\pi_k}(\boldsymbol{y}_n; \boldsymbol{\alpha}^*)$.** By Lemma 1,

$$S_{\pi_k}(\boldsymbol{y}_n; \boldsymbol{\alpha}^*) = \frac{t_K(\boldsymbol{y}_n; \boldsymbol{\alpha}^*)}{\pi_K^*} - \frac{t_k(\boldsymbol{y}_n; \boldsymbol{\alpha}^*)}{\pi_k^*}.$$

By Assumption 10, $0 < t_m(\boldsymbol{y}_n; \boldsymbol{\alpha}^*) \leq 1$, and by Assumption 8, $\pi_m^* \geq \pi_{\min} > 0$; hence $|S_{\pi_k}(\boldsymbol{y}_n; \boldsymbol{\alpha}^*)| \leq 2/\pi_{\min}$. Thus $S_{\pi_k}$ is bounded sub-Gaussian with $\nu_{\pi_k}^2 = \mathbb{E}[S_{\pi_k}^2] \leq (2/\pi_{\min})^2$ and $b_{\pi_k} = 0$.

**Mean parameters $S_{(\boldsymbol{\mu}_k)_d}(\boldsymbol{y}_n; \boldsymbol{\alpha}^*)$.** From Lemma 1,

$$S_{(\boldsymbol{\mu}_k)_d}(\boldsymbol{y}_n; \boldsymbol{\alpha}^*) = -\, t_k(\boldsymbol{y}_n; \boldsymbol{\alpha}^*) \big(\boldsymbol{\Psi}_k^*(\boldsymbol{y}_n - \boldsymbol{\mu}_k^*)\big)_d.$$

Let $U_{nkd} := \left(\boldsymbol{\Psi}_k^*(\boldsymbol{y}_n - \boldsymbol{\mu}_k^*)\right)_d$. Conditionally on $z_n = \ell$, we have $\boldsymbol{y}_n = \boldsymbol{\mu}_\ell^* + \varepsilon_\ell$ with $\varepsilon_\ell \sim \mathcal{N}(\mathbf{0}, \boldsymbol{\Sigma}_\ell^*)$. Then

$$\mathrm{Var}\left(U_{nkd} \mid z_n = \ell\right) = e_d^\top \boldsymbol{\Psi}_k^* \boldsymbol{\Sigma}_\ell^* \boldsymbol{\Psi}_k^* e_d \;\leq\; \|\boldsymbol{\Psi}_k^*\|_2^2 \, \|\boldsymbol{\Sigma}_\ell^*\|_2 \;\leq\; \theta_{\max}^2 \, \sigma_{\max}^2,$$

and

$$\left|\mathbb{E}[U_{nkd} \mid z_n = \ell]\right| = \left|e_d^\top \boldsymbol{\Psi}_k^*(\boldsymbol{\mu}_\ell^* - \boldsymbol{\mu}_k^*)\right| \;\leq\; \|\boldsymbol{\Psi}_k^*\|_2 \, \|\boldsymbol{\mu}_\ell^* - \boldsymbol{\mu}_k^*\|_2 \;\leq\; \theta_{\max}\,(2\eta).$$

Hence $\mathbb{E}[U_{nkd}^2] \leq \theta_{\max}^2(\sigma_{\max}^2 + 4\eta^2)$. Since $|t_k| \leq 1$,

$$\nu_{(\boldsymbol{\mu}_k)_d}^2 = \mathbb{E}\left[S_{(\boldsymbol{\mu}_k)_d}^2\right] \;\leq\; \mathbb{E}[U_{nkd}^2] \;\leq\; \theta_{\max}^2(\sigma_{\max}^2 + 4\eta^2).$$

Moreover, $U_{nkd}$ is (conditionally) Gaussian and hence sub-Gaussian; with the bounded multiplier $t_k$, $S_{(\boldsymbol{\mu}_k)_d}$ is sub-Gaussian, so $b_{(\boldsymbol{\mu}_k)_d} = 0$.

**Precision entries $S_{(\boldsymbol{\Psi}_k)_{rs}}(\boldsymbol{y}_n; \boldsymbol{\alpha}^*)$.** From Lemma 1,

$$S_{(\boldsymbol{\Psi}_k)_{rs}}(\boldsymbol{y}_n; \boldsymbol{\alpha}^*) = -\, t_k(\boldsymbol{y}_n; \boldsymbol{\alpha}^*)\, \frac{(2 - \delta_{rs})}{2} \left((\boldsymbol{\Sigma}_k^*)_{rs} - (\boldsymbol{y}_n - \boldsymbol{\mu}_k^*)_r (\boldsymbol{y}_n - \boldsymbol{\mu}_k^*)_s\right).$$

Let $V_{nkrs} := (\boldsymbol{\Sigma}_k^*)_{rs} - (\boldsymbol{y}_n - \boldsymbol{\mu}_k^*)_r (\boldsymbol{y}_n - \boldsymbol{\mu}_k^*)_s$. Each $(\boldsymbol{y}_n - \boldsymbol{\mu}_k^*)_r$ is sub-Gaussian with parameter $\lesssim \sigma_{\max} + \eta$, so the product $(\boldsymbol{y}_n - \boldsymbol{\mu}_k^*)_r (\boldsymbol{y}_n - \boldsymbol{\mu}_k^*)_s$ is sub-exponential; hence $V_{nkrs}$ is sub-exponential. Since $|t_k| \leq 1$, $S_{(\boldsymbol{\Psi}_k)_{rs}}$ is sub-exponential. Thus there exist constants $C_{\Psi,\nu}, C_{\Psi,b} > 0$ such that

$$\nu_{(\boldsymbol{\Psi}_k)_{rs}}^2 = \mathbb{E}\left[S_{(\boldsymbol{\Psi}_k)_{rs}}^2\right] \;\leq\; C_{\Psi,\nu}\,(\sigma_{\max}^2 + \eta^2)^2, \qquad b_{(\boldsymbol{\Psi}_k)_{rs}} \;\leq\; C_{\Psi,b}\,(\sigma_{\max}^2 + \eta^2).$$

**Union bound.** Let $\nu_{\max}^2 = \max_j \mathbb{E}[S_{jn}^2]$ and $b_{\max} = \max_j\{b_j : b_j \neq 0\}$; by the above bounds these depend only on $(\pi_{\min}, \eta, s_m, s_M)$. By Bernstein's inequality, for each coordinate $j$ and any $t > 0$,

$$\mathbb{P}\left(\left|\frac{1}{N}\sum_{n=1}^N S_{jn}\right| \geq t\right) \;\leq\; 2\exp\left(-c_0 N \min\left(\frac{t^2}{\nu_j^2}, \frac{t}{b_j}\right)\right),$$

with the convention that if $b_j = 0$ then $\min(\cdot) = t^2/\nu_j^2$. Set $t_N = C_g \sqrt{\frac{\ln d_\alpha}{N}}$ and assume $N \geq C_B \ln d_\alpha$ so that $t_N \leq \min_{j:b_j \neq 0} \nu_j^2/b_j$. Then for all $j$,

$$\mathbb{P}\left(\left|\frac{1}{N}\sum_{n=1}^N S_{jn}\right| \geq t_N\right) \;\leq\; 2\exp\left(-\frac{c_0 N t_N^2}{2\nu_j^2}\right) \;\leq\; 2\exp\left(-\frac{c_0 N t_N^2}{2\nu_{\max}^2}\right) = 2\, d_\alpha^{-\frac{c_0 C_g^2}{2\nu_{\max}^2}}.$$

A union bound over $j = 1, \ldots, d_\alpha$ yields

$$\mathbb{P}\left(\|\nabla \ell_N(\boldsymbol{\alpha}^*)\|_\infty \geq t_N\right) \;\leq\; 2\, d_\alpha^{1 - \frac{c_0 C_g^2}{2\nu_{\max}^2}}.$$

Choosing $C_1 = 2$ and $C_g$ so that $\frac{c_0 C_g^2}{2\nu_{\max}^2} = C_2 + 1$ gives the claim, i.e., $C_g = \sqrt{2(C_2 + 1)/c_0}\,\nu_{\max}$. Finally, the condition $N \geq C_B \ln d_\alpha$ is guaranteed by taking

$$C_B \;=\; \frac{C_g^2}{\left(\min_{j:\, b_j \neq 0} \nu_j^2/b_j\right)^2} \;=\; \frac{2(C_2 + 1)\,\nu_{\max}^2}{c_0\left(\min_{j:\, b_j \neq 0}(\mathbb{E}[S_{jn}^2]/b_j)\right)^2}.$$

$\square$

**Lemma 3** (Parameter Consistency for Penalized GMM Estimator). *Let $\widehat{\boldsymbol{\alpha}}$ be any local minimizer of the penalized negative average log-likelihood*

$$Q(\boldsymbol{\alpha}) = \ell_N(\boldsymbol{\alpha}) + P(\boldsymbol{\alpha}),$$

*where*

$$\ell_N(\boldsymbol{\alpha}) = -\frac{1}{N}\sum_{n=1}^N \ln f_{clust}(\boldsymbol{y}_n; \boldsymbol{\alpha}), \qquad P(\boldsymbol{\alpha}) = \lambda \sum_{k=1}^K \|\boldsymbol{\mu}_k\|_1 + \rho \sum_{k=1}^K \|\boldsymbol{\Psi}_k\|_1.$$

Let $\boldsymbol{\alpha}^*$ be the true GMM parameter vector, and $\mathbb{S}_0$ be the true support of the penalized parameters in $\boldsymbol{\alpha}^*$, with sparsity $s_0 = |\mathbb{S}_0|$. Let $d_\alpha$ be the total number of free parameters in $\boldsymbol{\alpha}$ as defined in Assumptions 12.

Suppose Assumptions 7-12 hold and the gradient bound in Lemma 2 is satisfied. Choose the regularization parameters $\lambda = \rho = \lambda_{chosen}$, where

$$\lambda_{chosen} = A_0 C_g \sqrt{\frac{\ln d_\alpha}{N}}$$

for a constant $A_0 > 1$ (e.g. $A_0 = 3$). Then, provided $N \gtrsim s_0 \ln d_\alpha$, with probability at least $1 - C_1^{grad} d_\alpha^{-C_2^{grad}} - c_1^H d_\alpha^{-c_2^H}$:

1. (*$L_2$-norm consistency*):

$$\|\widehat{\boldsymbol{\alpha}} - \boldsymbol{\alpha}^*\|_2 \leq \frac{2(A_0 + 1)C_g}{\kappa_I}\sqrt{\frac{s_0 \ln d_\alpha}{N}}.$$

2. (*$L_1$-norm consistency*):

$$\|\widehat{\boldsymbol{\alpha}} - \boldsymbol{\alpha}^*\|_1 \leq \frac{4A_0(A_0 + 1)C_g}{(A_0 - 1)\kappa_I} s_0 \sqrt{\frac{\ln d_\alpha}{N}}.$$

Here $C_g$ is the gradient bound constant from Lemma 2, and $\kappa_I$ is the restricted eigenvalue constant from Assumption 11. The factor $A_0$ is a user-chosen constant controlling the regularization strength.

*Proof of Lemma 3.* Let $\widehat{\boldsymbol{\alpha}}$ be a minimizer of the penalized negative log-likelihood $Q(\boldsymbol{\alpha}) = \ell_N(\boldsymbol{\alpha}) + P(\boldsymbol{\alpha})$, where $P(\boldsymbol{\alpha}) = \lambda \sum_k \|\boldsymbol{\mu}_k\|_1 + \rho \sum_k \|\boldsymbol{\Psi}_k\|_1$ and $\ell_N(\boldsymbol{\alpha}) = \frac{1}{N} \sum_{n=1}^N \ell(\boldsymbol{y}_n; \boldsymbol{\alpha})$ with $\ell(\boldsymbol{y}; \boldsymbol{\alpha}) = -\ln f_{\text{clust}}(\boldsymbol{y}; \boldsymbol{\alpha})$. Since $\widehat{\boldsymbol{\alpha}}$ minimizes $Q$, we have

$$\ell_N(\widehat{\boldsymbol{\alpha}}) - \ell_N(\boldsymbol{\alpha}^*) \leq P(\boldsymbol{\alpha}^*) - P(\widehat{\boldsymbol{\alpha}}). \tag{20}$$

Let $\boldsymbol{\Delta} = \widehat{\boldsymbol{\alpha}} - \boldsymbol{\alpha}^*$ and assume $\widehat{\boldsymbol{\alpha}} \in \boldsymbol{\Theta}'_\mathbb{V}$ so that $\boldsymbol{\alpha}^* + \boldsymbol{\Delta} \in \boldsymbol{\Theta}'_\mathbb{V}$. A second-order Taylor expansion gives

$$\ell_N(\widehat{\boldsymbol{\alpha}}) - \ell_N(\boldsymbol{\alpha}^*) = \langle \nabla \ell_N(\boldsymbol{\alpha}^*), \boldsymbol{\Delta} \rangle + \frac{1}{2} \boldsymbol{\Delta}^\top \boldsymbol{H}_N(\boldsymbol{\alpha}^* + t_0 \boldsymbol{\Delta}) \boldsymbol{\Delta}$$

for some $t_0 \in (0, 1)$. Write $\tilde{\boldsymbol{\alpha}} = \boldsymbol{\alpha}^* + t_0 \boldsymbol{\Delta}$. We lower bound $\frac{1}{2} \boldsymbol{\Delta}^\top \boldsymbol{H}_N(\tilde{\boldsymbol{\alpha}}) \boldsymbol{\Delta}$ by decomposing

$$\boldsymbol{\Delta}^\top \boldsymbol{H}_N(\tilde{\boldsymbol{\alpha}}) \boldsymbol{\Delta} = \boldsymbol{\Delta}^\top \boldsymbol{I}(\boldsymbol{\alpha}^*) \boldsymbol{\Delta} + \boldsymbol{\Delta}^\top \big( \boldsymbol{I}(\tilde{\boldsymbol{\alpha}}) - \boldsymbol{I}(\boldsymbol{\alpha}^*) \big) \boldsymbol{\Delta} + \boldsymbol{\Delta}^\top \big( \boldsymbol{H}_N(\tilde{\boldsymbol{\alpha}}) - \boldsymbol{I}(\tilde{\boldsymbol{\alpha}}) \big) \boldsymbol{\Delta}.$$

*From this point, all bounds are stated for arbitrary $\boldsymbol{\Delta}$ in the cone $\mathcal{C}(c_0, \mathbb{S}_0)$; the fact that the actual error $\widehat{\boldsymbol{\alpha}} - \boldsymbol{\alpha}^*$ lies in $\mathcal{C}(c_0, \mathbb{S}_0)$ will be established later by Lemma 4.*

By Assumption 11, for $\boldsymbol{\Delta} \in \mathcal{C}(c_0, \mathbb{S}_0)$,

$$\boldsymbol{\Delta}^\top \boldsymbol{I}(\boldsymbol{\alpha}^*) \boldsymbol{\Delta} \geq \kappa_I \|\boldsymbol{\Delta}\|_2^2.$$

By Assumption 7 and compactness (Assumption 8), $\boldsymbol{\alpha} \mapsto \boldsymbol{I}(\boldsymbol{\alpha}) = \mathbb{E}[\nabla^2 \ell(\boldsymbol{y}; \boldsymbol{\alpha})]$ is Lipschitz on $\mathbb{B}(\boldsymbol{\alpha}^*, r_0) \cap \boldsymbol{\Theta}'_\mathbb{V}$ in spectral norm: there exists $L_I > 0$ such that

$$\|\boldsymbol{I}(\tilde{\boldsymbol{\alpha}}) - \boldsymbol{I}(\boldsymbol{\alpha}^*)\|_2 \leq L_I \|\tilde{\boldsymbol{\alpha}} - \boldsymbol{\alpha}^*\|_2 = L_I t_0 \|\boldsymbol{\Delta}\|_2 \leq L_I \|\boldsymbol{\Delta}\|_2,$$

and therefore

$$\big| \boldsymbol{\Delta}^\top (\boldsymbol{I}(\tilde{\boldsymbol{\alpha}}) - \boldsymbol{I}(\boldsymbol{\alpha}^*)) \boldsymbol{\Delta} \big| \leq \|\boldsymbol{I}(\tilde{\boldsymbol{\alpha}}) - \boldsymbol{I}(\boldsymbol{\alpha}^*)\|_2 \|\boldsymbol{\Delta}\|_2^2 \leq L_I \|\boldsymbol{\Delta}\|_2^3 \leq L_I \delta_R \|\boldsymbol{\Delta}\|_2^2,$$

provided $\|\boldsymbol{\Delta}\|_2 \leq \delta_R$. By Assumption 12, if $\|\boldsymbol{\Delta}\|_2 \leq \delta_R$ and $\boldsymbol{\Delta} \in \mathcal{C}(c_0, \mathbb{S}_0)$, then with probability at least $1 - c_1^H d_\alpha^{-c_2^H}$,

$$\big| \boldsymbol{\Delta}^\top (\boldsymbol{H}_N(\tilde{\boldsymbol{\alpha}}) - \boldsymbol{I}(\tilde{\boldsymbol{\alpha}})) \boldsymbol{\Delta} \big| \leq C_H \sqrt{\frac{s_0 \ln d_\alpha}{N}} \|\boldsymbol{\Delta}\|_2^2.$$

Combining the three representations, for $\boldsymbol{\Delta} \in \mathcal{C}(c_0, \mathbb{S}_0)$ with $\|\boldsymbol{\Delta}\|_2 \leq \delta_R$,

$$\frac{1}{2}\boldsymbol{\Delta}^\top \boldsymbol{H}_N(\tilde{\boldsymbol{\alpha}})\boldsymbol{\Delta} \geq \frac{1}{2}\left(\kappa_I - L_I\|\boldsymbol{\Delta}\|_2 - C_H\sqrt{\frac{s_0 \ln d_\alpha}{N}}\right)\|\boldsymbol{\Delta}\|_2^2.$$

Choose $\delta_R$ and $N$ so that $L_I \delta_R \leq \kappa_I/4$ and $C_H\sqrt{\frac{s_0 \ln d_\alpha}{N}} \leq \kappa_I/4$ (e.g., $N \gtrsim (C_H^2/\kappa_I^2)s_0 \ln d_\alpha$). Then, with the same probability,

$$\frac{1}{2}\boldsymbol{\Delta}^\top \boldsymbol{H}_N(\tilde{\boldsymbol{\alpha}})\boldsymbol{\Delta} \geq \frac{\kappa_I}{4}\|\boldsymbol{\Delta}\|_2^2.$$

Hence we obtain the *restricted strong convexity* (RSC) inequality on the cone:

$$\ell_N(\boldsymbol{\alpha}^* + \boldsymbol{\Delta}) - \ell_N(\boldsymbol{\alpha}^*) - \langle\nabla\ell_N(\boldsymbol{\alpha}^*), \boldsymbol{\Delta}\rangle \geq \frac{\kappa_L}{2}\|\boldsymbol{\Delta}\|_2^2, \qquad \text{for all } \boldsymbol{\Delta} \in \mathcal{C}(c_0, \mathbb{S}_0), \ \|\boldsymbol{\Delta}\|_2 \leq \delta_R, \tag{21}$$

where $\kappa_L = \kappa_I/2 > 0$.

**Stochastic term.** By Lemma 2, if $N \geq C_B \ln d_\alpha$, then with probability at least $1 - C_1^{\text{grad}} d_\alpha^{-C_2^{\text{grad}}}$,

$$\|\nabla\ell_N(\boldsymbol{\alpha}^*)\|_\infty \leq C_g\sqrt{\frac{\ln d_\alpha}{N}}.$$

By Hölder's inequality $\left(\langle x, y\rangle <= \|x\|_\infty\|y\|_1\right)$,

$$|\langle\nabla\ell_N(\boldsymbol{\alpha}^*), \boldsymbol{\Delta}\rangle| \leq \|\nabla\ell_N(\boldsymbol{\alpha}^*)\|_\infty\|\boldsymbol{\Delta}\|_1 \leq C_g\sqrt{\frac{\ln d_\alpha}{N}}\|\boldsymbol{\Delta}\|_1. \tag{22}$$

**Penalty difference.** Let $\boldsymbol{\Delta}_{\boldsymbol{\mu}_k} = \widehat{\boldsymbol{\mu}}_k - \boldsymbol{\mu}_k^*$ and $\boldsymbol{\Delta}_{\boldsymbol{\Psi}_k} = \widehat{\boldsymbol{\Psi}}_k - \boldsymbol{\Psi}_k^*$, and denote by $\mathbb{S}_{\boldsymbol{\mu}}^*$ and $\mathbb{S}_{\boldsymbol{\Psi}}^*$ the true supports (Definition 3). Using the $L_1$-triangle inequality on supports: for any vector $\boldsymbol{a}, \boldsymbol{b}$ and support $\mathbb{S}$ of $\boldsymbol{a}$: $\|\boldsymbol{a}\|_1 - \|\boldsymbol{b}\|_1 \leq \|\boldsymbol{a}_{\mathbb{S}} - \boldsymbol{b}_{\mathbb{S}}\|_1 - \|\boldsymbol{a}_{(\mathbb{S})^c} - \boldsymbol{b}_{(\mathbb{S})^c}\|_1 + 2\|\boldsymbol{a}_{(\mathbb{S})^c}\|_1$. Since $\boldsymbol{a}_{(\mathbb{S})^c} = \boldsymbol{0}$:

$$\|\boldsymbol{\mu}_k^*\|_1 - \|\widehat{\boldsymbol{\mu}}_k\|_1 \leq \|\boldsymbol{\Delta}_{\boldsymbol{\mu}_k, \mathbb{S}_{\boldsymbol{\mu}}^*}\|_1 - \|\boldsymbol{\Delta}_{\boldsymbol{\mu}_k, (\mathbb{S}_{\boldsymbol{\mu}}^*)^c}\|_1, \qquad \|\boldsymbol{\Psi}_k^*\|_1 - \|\widehat{\boldsymbol{\Psi}}_k\|_1 \leq \|\boldsymbol{\Delta}_{\boldsymbol{\Psi}_k, \mathbb{S}_{\boldsymbol{\Psi}}^*}\|_1 - \|\boldsymbol{\Delta}_{\boldsymbol{\Psi}_k, (\mathbb{S}_{\boldsymbol{\Psi}}^*)^c}\|_1.$$

Summing over $k$ and writing $\mathbb{S}_0 = \mathbb{S}_{\boldsymbol{\mu}}^* \cup \mathbb{S}_{\boldsymbol{\Psi}}^*$,

$$P(\boldsymbol{\alpha}^*) - P(\widehat{\boldsymbol{\alpha}}) \leq \lambda\big(\|\boldsymbol{\Delta}_{\boldsymbol{\mu}, \mathbb{S}_0}\|_1 - \|\boldsymbol{\Delta}_{\boldsymbol{\mu}, (\mathbb{S}_0)^c}\|_1\big) + \rho\big(\|\boldsymbol{\Delta}_{\boldsymbol{\Psi}, \mathbb{S}_0}\|_1 - \|\boldsymbol{\Delta}_{\boldsymbol{\Psi}, (\mathbb{S}_0)^c}\|_1\big). \tag{23}$$

**Combining.** With probability at least $1 - C_1^{\text{grad}} d_\alpha^{-C_2^{\text{grad}}} - c_1^H d_\alpha^{-c_2^H}$, combining Equation (21), Equation (22), and Equation (23) with Equation (20) yields, for all $\boldsymbol{\Delta} \in \mathcal{C}(c_0, \mathbb{S}_0)$ with $\|\boldsymbol{\Delta}\|_2 \leq \delta_R$,

$$\frac{\kappa_L}{2}\|\boldsymbol{\Delta}\|_2^2 \leq C_g\sqrt{\frac{\ln d_\alpha}{N}}\|\boldsymbol{\Delta}\|_1 + \lambda\big(\|\boldsymbol{\Delta}_{\boldsymbol{\mu}, \mathbb{S}_0}\|_1 - \|\boldsymbol{\Delta}_{\boldsymbol{\mu}, (\mathbb{S}_0)^c}\|_1\big) + \rho\big(\|\boldsymbol{\Delta}_{\boldsymbol{\Psi}, \mathbb{S}_0}\|_1 - \|\boldsymbol{\Delta}_{\boldsymbol{\Psi}, (\mathbb{S}_0)^c}\|_1\big). \tag{24}$$

Invoking the inequality Equation (29) from Lemma 4 (the cone lemma, proved via KKT and the gradient bound, thus independent of RSC),

$$\frac{\kappa_L}{2}\|\boldsymbol{\Delta}\|_2^2 \leq \left(1 + \frac{1}{A_0}\right)\sum_{j \in \mathbb{S}_0}\lambda_j|\Delta_j| - \left(1 - \frac{1}{A_0}\right)\sum_{j \in (\mathbb{S}_0)^c}\lambda_j|\Delta_j|.$$

Since the second term is non-positive (as $A_0 > 1$ and $\lambda_j|\Delta_j| \geq 0$), we can drop it to get an upper bound:

$$\frac{\kappa_L}{2}\|\boldsymbol{\Delta}\|_2^2 \leq \left(1 + \frac{1}{A_0}\right)\sum_{j \in \mathbb{S}_0}\lambda_j|\Delta_j|.$$

Taking the common regularization level $\lambda_{\text{chosen}} = \lambda = \rho = A_0 C_g\sqrt{\frac{\ln d_\alpha}{N}}$ with $A_0 > 1$ and note that $\sum_{j \in \mathbb{S}_0}\lambda_j|\Delta_j| = \lambda_{\text{chosen}}\|\boldsymbol{\Delta}_{\mathbb{S}_0}\|_1$, we obtain

$$\frac{\kappa_L}{2}\|\boldsymbol{\Delta}\|_2^2 \leq \left(1 + \frac{1}{A_0}\right)\lambda_{\text{chosen}}\|\boldsymbol{\Delta}_{\mathbb{S}_0}\|_1.$$

Using the Cauchy-Schwarz inequality $\|\mathbf{\Delta}_{\mathbb{S}_0}\|_1 \leq \sqrt{s_0}\|\mathbf{\Delta}_{\mathbb{S}_0}\|_2$, and since $\|\mathbf{\Delta}_{\mathbb{S}_0}\|_2 \leq \|\mathbf{\Delta}\|_2$:

$$\frac{\kappa_L}{2}\|\mathbf{\Delta}\|_2^2 \leq \lambda_{\text{chosen}}\left(1 + \frac{1}{A_0}\right)\sqrt{s_0}\|\mathbf{\Delta}\|_2.$$

If $\|\mathbf{\Delta}\|_2 \neq 0$, we can divide by $\|\mathbf{\Delta}\|_2$:

$$\|\mathbf{\Delta}\|_2 \leq \frac{2\lambda_{\text{chosen}}}{\kappa_L}\left(1 + \frac{1}{A_0}\right)\sqrt{s_0}.$$

Substituting $\lambda_{\text{chosen}} = A_0 C_g\sqrt{\frac{\ln d_\alpha}{N}}$:

$$\|\widehat{\boldsymbol{\alpha}} - \boldsymbol{\alpha}^*\|_2 \leq \frac{2A_0 C_g}{\kappa_L}\left(1 + \frac{1}{A_0}\right)\sqrt{s_0}\sqrt{\frac{\ln d_\alpha}{N}}$$

$$= \frac{2(A_0 + 1)C_g}{\kappa_L}\sqrt{\frac{s_0 \ln d_\alpha}{N}}.$$

This proves the $L_2$-rate with constant $C_{L2} = \frac{2(A_0+1)C_g}{\kappa_L}$.

For the $L_1$-rate, Lemma 4 yields the cone bound $\|\mathbf{\Delta}_{(\mathbb{S}_0)^c}\|_1 \leq C_{\text{cone}}\|\mathbf{\Delta}_{\mathbb{S}_0}\|_1$ with $C_{\text{cone}} = \frac{A_0+1}{A_0-1}$ (decomposability via [44]). Hence, assuming all $\lambda_j$ for penalized components are $\lambda_{\text{chosen}}$):

$$\begin{aligned}\|\mathbf{\Delta}\|_1 &= \|\mathbf{\Delta}_{\mathbb{S}_0}\|_1 + \|\mathbf{\Delta}_{(\mathbb{S}_0)^c}\|_1 \\ &\leq \|\mathbf{\Delta}_{\mathbb{S}_0}\|_1 + C_{\text{cone}}\|\mathbf{\Delta}_{\mathbb{S}_0}\|_1 = (1 + C_{\text{cone}})\|\mathbf{\Delta}_{\mathbb{S}_0}\|_1 \\ &\leq (1 + C_{\text{cone}})\sqrt{s_0}\|\mathbf{\Delta}_{\mathbb{S}_0}\|_2 \quad \text{(by Cauchy-Schwarz)} \\ &\leq (1 + C_{\text{cone}})\sqrt{s_0}\|\mathbf{\Delta}\|_2.\end{aligned}$$

Substituting $C_{\text{cone}} = \frac{A_0+1}{A_0-1}$:

$$1 + C_{\text{cone}} = 1 + \frac{A_0 + 1}{A_0 - 1} = \frac{A_0 - 1 + A_0 + 1}{A_0 - 1} = \frac{2A_0}{A_0 - 1}.$$

So,

$$\|\mathbf{\Delta}\|_1 \leq \left(\frac{2A_0}{A_0 - 1}\right)\sqrt{s_0}\|\mathbf{\Delta}\|_2.$$

Now substitute the bound for $\|\mathbf{\Delta}\|_2$:

$$\begin{aligned}\|\widehat{\boldsymbol{\alpha}} - \boldsymbol{\alpha}^*\|_1 &\leq \left(\frac{2A_0}{A_0 - 1}\right)\sqrt{s_0}\left(\frac{2(A_0 + 1)C_g}{\kappa_L}\sqrt{\frac{s_0 \ln d_\alpha}{N}}\right) \\ &= \frac{4A_0(A_0 + 1)C_g}{(A_0 - 1)\kappa_L}s_0\sqrt{\frac{\ln d_\alpha}{N}}. \\ &= C_{L1}(A_0, C_g, \kappa_L)s_0\sqrt{\frac{\ln d_\alpha}{N}}\end{aligned}$$

where $C_{L1}(A_0, C_g, \kappa_L) = \frac{4A_0(A_0+1)C_g}{(A_0-1)\kappa_L}$. This concludes the proof of parameter consistency in $L_1$ and $L_2$ norms. $\qquad\square$

**Remark 2.** *The radius $\delta_R$ governing the RSC inequality Equation (21) is critical and is typically of the same order as the target statistical error. If $\|\mathbf{\Delta}\|_2$ exceeds $\delta_R$, the Lipschitz remainder $L_I\|\mathbf{\Delta}\|_2^3$ can dominate, and/or the uniform Hessian concentration in Assumption 12 may fail on such a large neighborhood of $\boldsymbol{\alpha}^*$. In general M-estimation analyses (e.g., [44]), one often proves an RSC with a tolerance term,*

$$\ell_N(\boldsymbol{\alpha}^* + \mathbf{\Delta}) - \ell_N(\boldsymbol{\alpha}^*) - \langle\nabla\ell_N(\boldsymbol{\alpha}^*), \mathbf{\Delta}\rangle \geq \frac{\kappa_L}{2}\|\mathbf{\Delta}\|_2^2 - \tau_L\frac{\ln d_\alpha}{N}\|\mathbf{\Delta}\|_1^2,$$

*valid on a larger set. Under Assumption 12, we obtain sufficiently strong control to dispense with this tolerance and derive the cleaner quadratic curvature bound Equation (21). If Assumption 12 were weakened (e.g., only yielding a bound of the form $C_H\sqrt{\frac{\ln d_\alpha}{N}}\frac{\|\mathbf{\Delta}\|_1}{\sqrt{s_0}}\|\mathbf{\Delta}\|_2$ on $\mathbf{\Delta}^\top(\boldsymbol{H}_N - \boldsymbol{I})\mathbf{\Delta}$ over the cone), then a tolerance term proportional to $\|\mathbf{\Delta}\|_1^2$ would naturally appear in the RSC.*

**Lemma 4** (Cone Condition for $L_1$-Penalized M-Estimators). *Let $\widehat{\alpha}$ be any local minimizer of $Q(\alpha) = \ell_N(\alpha) + P(\alpha)$, where $\ell_N(\alpha)$ is a differentiable loss function and the penalty $P(\alpha)$ is a sum of component-wise $L_1$ penalties:*

$$P(\alpha) = \sum_{j=1}^{d_\alpha} \lambda_j |\alpha_j|.$$

*Let $\mathbb{S}_0$ be the true support of $\alpha^*$. Let $\Delta = \widehat{\alpha} - \alpha^*$. Suppose the regularization parameters are chosen such that for some constant $A_0 > 1$:*

$$\lambda_j \geq A_0 |[\nabla \ell_N(\alpha^*)]_j| \quad \text{for all } j \in (\mathbb{S}_0)^c. \tag{25}$$

*and $\lambda_j$ of similar order for $j \in \mathbb{S}_0$. For simplicity, we often set $\lambda_j = \lambda_{chosen}$ for all penalized components, where $\lambda_{chosen} \geq A_0 \|\nabla \ell_N(\alpha^*)\|_\infty$. If the RSC condition from Equation (21) holds for $\Delta$:*

$$\ell_N(\alpha^* + \Delta) - \ell_N(\alpha^*) - \langle \nabla \ell_N(\alpha^*), \Delta \rangle \geq \frac{\kappa_L}{2} \|\Delta\|_2^2,$$

*then, for $A_0 > 1$, the error vector $\Delta$ satisfies the cone condition:*

$$\sum_{j \in (\mathbb{S}_0)^c} \lambda_j |\Delta_j| \leq \frac{A_0 + 1}{A_0 - 1} \sum_{j \in \mathbb{S}_0} \lambda_j |\Delta_j|. \tag{26}$$

*If all $\lambda_j$ for penalized components are equal to $\lambda_{chosen}$, this simplifies to:*

$$\|\Delta_{(\mathbb{S}_0)^c}\|_{pen,1} \leq \frac{A_0 + 1}{A_0 - 1} \|\Delta_{\mathbb{S}_0}\|_{pen,1},$$

*where $\| \cdot \|_{pen,1}$ refers to the $L_1$ norm over the components that are actually penalized. If all components were penalized with the same $\lambda_0$, this would be $\|\Delta_{(\mathbb{S}_0)^c}\|_1 \leq \frac{A_0+1}{A_0-1} \|\Delta_{\mathbb{S}_0}\|_1$.*

**Remark 3.** *In our case, $\lambda_j = \lambda$ for mean components $\mu_{kd}$, and $\lambda_j = \rho$ for off-diagonal precision components $(\Psi_k)_{rs}$, and $\lambda_j = 0$ for parameters not penalized like proportions or diagonal precision elements if they are not penalized towards a specific value. Moreover, the Cone Condition ensures that the error vector is primarily concentrated on the true support. The key is that the regularization parameter for the "noise" variables (off-support) must be sufficiently larger than the corresponding component of the score vector, allowing the penalty to effectively shrink noise components.*

*Proof of Lemma 4.* Let $\mathcal{P} \subset \{1, \ldots, d_\alpha\}$ denote the index set of *penalized* coordinates and write $\|x\|_{\text{pen},1} = \sum_{j \in \mathcal{P}} |x_j|$. In what follows, sums and $L_1$-norms are taken over $\mathcal{P}$. Assume the tuning condition holds on *all penalized* coordinates:

$$\lambda_j \geq A_0 \left| [\nabla \ell_N(\alpha^*)]_j \right| \qquad \forall j \in \mathcal{P}, \quad A_0 > 1, \tag{27}$$

equivalently $\lambda_{\text{chosen}} \geq A_0 \|\nabla \ell_N(\alpha^*)\|_{\infty,\text{pen}}$ when a common level is used. [2]

Since $\widehat{\alpha}$ is a local minimizer of $Q(\alpha)$, it satisfies the basic optimality inequality

$$\ell_N(\widehat{\alpha}) - \ell_N(\alpha^*) \leq P(\alpha^*) - P(\widehat{\alpha}).$$

Invoking the restricted curvature bound (RSC) Equation (21) for $\Delta = \widehat{\alpha} - \alpha^*$,

$$\ell_N(\widehat{\alpha}) - \ell_N(\alpha^*) \geq \langle \nabla \ell_N(\alpha^*), \Delta \rangle + \frac{\kappa_L}{2} \|\Delta\|_2^2.$$

Note that, only the nonnegativity of the quadratic remainder is needed for the cone; the explicit $\kappa_L > 0$ is used later for $L_2$-rates. Therefore,

$$\langle \nabla \ell_N(\alpha^*), \Delta \rangle + \frac{\kappa_L}{2} \|\Delta\|_2^2 \leq P(\alpha^*) - P(\widehat{\alpha}). \tag{28}$$

For the penalty difference, write $P(\alpha) = \sum_{j \in \mathcal{P}} \lambda_j |\alpha_j|$ and let $\mathbb{S}_0 \subseteq \mathcal{P}$ denote the true support of the penalized coordinates. Using the standard $L_1$ support inequality with $a = \alpha^*$ and $b = \widehat{\alpha} = \alpha^* + \Delta$,

$$\|a\|_1 - \|b\|_1 = \|a_{\mathbb{S}_0}\|_1 - \|b_{\mathbb{S}_0}\|_1 - \|b_{(\mathbb{S}_0)^c}\|_1 \leq \|a_{\mathbb{S}_0} - b_{\mathbb{S}_0}\|_1 - \|b_{(\mathbb{S}_0)^c}\|_1,$$

---

[2]If some coordinates are unpenalized, they are excluded from $\mathcal{P}$ and do not enter the cone inequality.

and since $(\boldsymbol{\alpha}^*)_{(\mathbb{S}_0)^c} = \mathbf{0}$ on $\mathcal{P}$, we obtain

$$P(\boldsymbol{\alpha}^*) - P(\widehat{\boldsymbol{\alpha}}) \;\le\; \sum_{j \in \mathbb{S}_0} \lambda_j |\Delta_j| \;-\; \sum_{j \in (\mathbb{S}_0)^c} \lambda_j |\Delta_j|.$$

Substitute this into Equation (28) and bound the linear term by the triangle inequality over $\mathcal{P}$:

$$\langle \nabla \ell_N(\boldsymbol{\alpha}^*), \boldsymbol{\Delta} \rangle \;\le\; \sum_{j \in \mathcal{P}} \big| [\nabla \ell_N(\boldsymbol{\alpha}^*)]_j \big| \, |\Delta_j| \;\le\; \sum_{j \in \mathcal{P}} \frac{\lambda_j}{A_0} \, |\Delta_j|,$$

where the last step uses Equation (27). We obtain

$$\frac{\kappa_L}{2} \, \|\boldsymbol{\Delta}\|_2^2 \;\le\; \sum_{j \in \mathbb{S}_0} \left( \lambda_j + \frac{\lambda_j}{A_0} \right) |\Delta_j| \;-\; \sum_{j \in (\mathbb{S}_0)^c} \left( \lambda_j - \frac{\lambda_j}{A_0} \right) |\Delta_j|.$$

Equivalently,

$$\frac{\kappa_L}{2} \, \|\boldsymbol{\Delta}\|_2^2 \;\le\; \left( 1 + \tfrac{1}{A_0} \right) \sum_{j \in \mathbb{S}_0} \lambda_j |\Delta_j| \;-\; \left( 1 - \tfrac{1}{A_0} \right) \sum_{j \in (\mathbb{S}_0)^c} \lambda_j |\Delta_j|. \tag{29}$$

Since the left-hand side is nonnegative, it follows that

$$\left( 1 - \tfrac{1}{A_0} \right) \sum_{j \in (\mathbb{S}_0)^c} \lambda_j |\Delta_j| \;\le\; \left( 1 + \tfrac{1}{A_0} \right) \sum_{j \in \mathbb{S}_0} \lambda_j |\Delta_j|,$$

and for $A_0 > 1$ this yields the cone inequality

$$\sum_{j \in (\mathbb{S}_0)^c} \lambda_j |\Delta_j| \;\le\; \frac{A_0 + 1}{A_0 - 1} \sum_{j \in \mathbb{S}_0} \lambda_j |\Delta_j|.$$

If all penalized coordinates share a common level $\lambda_{\text{chosen}}$, this becomes $\|\boldsymbol{\Delta}_{(\mathbb{S}_0)^c}\|_{\text{pen},1} \le \frac{A_0+1}{A_0-1} \|\boldsymbol{\Delta}_{\mathbb{S}_0}\|_{\text{pen},1}$, and if every coordinate is penalized equally it reduces to $\|\boldsymbol{\Delta}_{(\mathbb{S}_0)^c}\|_1 \le \frac{A_0+1}{A_0-1} \|\boldsymbol{\Delta}_{\mathbb{S}_0}\|_1$. $\qquad\square$

With the cone condition in Lemma 4 established on the penalized coordinates, we now consolidate the standing assumptions for the ranking-consistency analysis (Lemma 5) and for Theorem 3. The goal is to avoid duplication, align the SRUW assumptions used earlier with the GMM penalized framework here, and make explicit exactly which conditions are invoked downstream.

**Remark 4** (Alignment with SRUW assumptions). *Assumptions 4-6 were introduced for the SRUW model in the MNARz setting. In the present penalized GMM analysis for Theorem 3:*

- *Model uniqueness (SRUW-4). We condition on a fixed mixture structure (number of components $K$ and dimension $D$) and require identifiability of the GMM density; this role is played by Assumption 7 below. We do not require the SRUW tuple $(K_0, m_0, r_0, l_0, \mathbb{V}_0)$ explicitly here because no model selection over SRUW structures is performed in Theorem 3.*

- *Compactness and interiority (SRUW-5-6). These correspond directly to Assumption 8 below (compact parameter subset $\Theta_{\mathbb{V}}^t$ and $\boldsymbol{\alpha}^*$ interior), together with the eigenvalue and boundedness constraints therein. Thus SRUW-5-6 are subsumed by Assumption 8.*

*Hence, for the purposes of Lemma 5 and Theorem 3, it suffices to work with Assumptions 7-12 below; SRUW-4-6 need not be re-stated.*

To proceed rigorously towards Lemma 5 and, ultimately, Theorem 3, we now restate the full set of standing assumptions-consolidating those aligned with the $\mathbb{SRUW}$ framework and those introduced earlier for penalized likelihood analysis-into a unified assumption block.

**Assumption 13** (Standing Assumptions for proving Theorem 3). *The following holds:*

1. *Identifiability & Smoothness (Assumption 7): the GMM density $f_{\text{clust}}(\boldsymbol{y}; \boldsymbol{\alpha})$ is identifiable and $\ell(\boldsymbol{y}; \boldsymbol{\alpha}) = -\ln f_{\text{clust}}(\boldsymbol{y}; \boldsymbol{\alpha})$ is three times continuously differentiable in an open ball $\mathbb{B}(\boldsymbol{\alpha}^*, r_0)$.*

2. **Compactness & Bounds** (Assumption 8): $\boldsymbol{\alpha}^*$ is an interior point of a compact $\boldsymbol{\Theta}'_{\mathbb{V}} \subset \mathbb{B}(\boldsymbol{\alpha}^*, r_0)$; mixture weights, means, and covariance/precision eigenvalues satisfy the stated uniform bounds (with $\pi_{\min}, \eta, \sigma^2_{\min}, \sigma^2_{\max}, \theta_{\min}, \theta_{\max}$).

3. **Data-generating mechanism** (Assumption 9): $\boldsymbol{y}_1, \ldots, \boldsymbol{y}_N$ i.i.d. from $f_{clust}(\cdot; \boldsymbol{\alpha}^*)$ with latent $z_n \sim Mult(\boldsymbol{\pi}^*)$ and $\boldsymbol{y}_n | z_n = k \sim \mathcal{N}(\boldsymbol{\mu}^*_k, \boldsymbol{\Sigma}^*_k)$.

4. **Posterior responsibilities** (Assumption 10): $t_k(\boldsymbol{y}; \boldsymbol{\alpha}) \in (0, 1]$ for all $\boldsymbol{\alpha} \in \boldsymbol{\Theta}'_{\mathbb{V}}$.

5. **Fisher RE on a cone** (Assumption 11): restricted eigenvalue condition for $\boldsymbol{I}(\boldsymbol{\alpha}^*)$ on $\mathcal{C}(c_0, \mathbb{S}_0)$ with constant $\kappa_I > 0$.

6. **Uniform Hessian concentration** (Assumption 12): for radius $\delta_R \asymp \sqrt{s_0 \ln d_\alpha / N}$ and $N \gtrsim s_0 \ln d_\alpha$,

$$\sup_{\tilde{\boldsymbol{\alpha}} \in \mathbb{B}(\boldsymbol{\alpha}^*, \delta_R) \cap \boldsymbol{\Theta}'_{\mathbb{V}}} \sup_{\boldsymbol{\Delta} \in \mathcal{C}(c_0, \mathbb{S}_0), \|\boldsymbol{\Delta}\|_2 = 1} \left| \boldsymbol{\Delta}^\top \left( \boldsymbol{H}_N(\tilde{\boldsymbol{\alpha}}) - \boldsymbol{I}(\tilde{\boldsymbol{\alpha}}) \right) \boldsymbol{\Delta} \right| \leq C_H \sqrt{\frac{s_0 \ln d_\alpha}{N}}$$

with probability at least $1 - c_1^H d_\alpha^{-c_2^H}$.

7. **Penalty on penalized coordinates.** On the penalized index set $\mathcal{P}$, choose $\lambda_j$ so that

$$\lambda_j \geq A_0 \left\| \nabla \ell_N(\boldsymbol{\alpha}^*) \right\|_{\infty, pen} \qquad \text{for some } A_0 > 1,$$

e.g. a common level $\lambda_{chosen} = A_0 C_g \sqrt{\frac{\ln d_\alpha}{N}}$ with $C_g$ from Lemma 2. Unpenalized coordinates are excluded from $\mathcal{P}$ and from all $\| \cdot \|_{pen,1}$ norms.

**Assumption 14** (Working high-probability event for ranking analysis). *Let* $\mathcal{E}_{grad} = \{ \|\nabla \ell_N(\boldsymbol{\alpha}^*)\|_{\infty, pen} \leq C_g \sqrt{\ln d_\alpha / N} \}$ *be the event in Lemma 2, and* $\mathcal{E}_{RSC}$ *the event on which Equation (21) holds with curvature* $\kappa_L > 0$ *over* $\{ \boldsymbol{\Delta} \in \mathcal{C}(c_0, \mathbb{S}_0) : \|\boldsymbol{\Delta}\|_2 \leq \delta_R \}$. *Under Assumption 13 and for N large enough, both events hold with probability at least* $1 - \delta_N$, *where* $\delta_N = C_1^{grad} d_\alpha^{-C_2^{grad}} + c_1^H d_\alpha^{-c_2^H}$. *In the proof of Lemma 5, we condition on* $\mathcal{E}_{grad} \cap \mathcal{E}_{RSC}$.

**Definition 4** (Quantities for Ranking Consistency). *Let* $\boldsymbol{\alpha}^*$ *be the true GMM parameters, and* $\widehat{\boldsymbol{\alpha}}(\lambda)$ *be the estimator for a given regularization level* $\lambda$ *(with* $\rho$ *either tied to* $\lambda$ *or fixed).*

- *Noise level.*
$$\lambda_{noise} = C_g \sqrt{\frac{\ln d_\alpha}{N}},$$
*the uniform bound on the score at* $\boldsymbol{\alpha}^*$ *from Lemma 2.*

- *Effective curvature for means.*
$$H_{kj}^{eff}(\boldsymbol{\alpha}^*) = \mathbb{E}_{\boldsymbol{\alpha}^*} \left[ t_k(\boldsymbol{y}; \boldsymbol{\alpha}^*) (\boldsymbol{\Psi}_k^*)_{jj} \right] = (\boldsymbol{\Psi}_k^*)_{jj} \mathbb{E}_{\boldsymbol{\alpha}^*} [t_k(\boldsymbol{y}; \boldsymbol{\alpha}^*)],$$
*where the expectation is with respect to*
$$\boldsymbol{y} \sim f_{clust}(\cdot; \boldsymbol{\alpha}^*).$$
*By Assumption 8,* $\mathbb{E}_{\boldsymbol{\alpha}^*}[t_k(\boldsymbol{y}; \boldsymbol{\alpha}^*)] = \pi_k^* \geq \pi_{\min}$ *and* $(\boldsymbol{\Psi}_k^*)_{jj} \geq \theta_{\min}$ *(since* $\boldsymbol{\Psi}_k^* \succ 0$ *and* $e_j^\top \boldsymbol{\Psi}_k^* e_j \geq \lambda_{\min}(\boldsymbol{\Psi}_k^*)$*), hence*
$$H_{kj}^{eff}(\boldsymbol{\alpha}^*) \geq \pi_{\min} \theta_{\min} > 0.$$

- *KKT remainder for mean coordinates.*
$$R_{kj}(\widehat{\boldsymbol{\alpha}}(\lambda), \boldsymbol{\alpha}^*, \lambda) = \partial_{\mu_{kj}} \ell_N(\widehat{\boldsymbol{\alpha}}(\lambda)) - \partial_{\mu_{kj}} \ell_N(\boldsymbol{\alpha}^*) - H_{kj}^{eff}(\boldsymbol{\alpha}^*)(\widehat{\mu}_{kj}(\lambda) - \mu_{kj}^*).$$
*We assume that, with high probability and for* $\lambda$ *in the regime* $\lambda \asymp \sqrt{(\ln d_\alpha)/N}$,
$$\left| R_{kj}(\widehat{\boldsymbol{\alpha}}(\lambda), \boldsymbol{\alpha}^*, \lambda) \right| \leq E_{KKT\_rem}(\lambda),$$
*where* $E_{KKT\_rem}(\lambda)$ *depends on* $\|\widehat{\boldsymbol{\alpha}}(\lambda) - \boldsymbol{\alpha}^*\|$ *(cf. Lemma 3) and on bounds for Hessian/third derivatives (from Assumptions 7-12). In particular, when* $\|\boldsymbol{\Delta}\|_2$ *is of order* $\sqrt{s_0 \ln d_\alpha / N}$, *we take*
$$E_{KKT\_rem}(\lambda) \leq C_{rem} \lambda_{noise}$$
*for some constant* $C_{rem} \geq 0$ *over the relevant range of* $\lambda$.

- *Noise-screening threshold.*

$$\Lambda_N^* \;=\; \big(1 + C_{rem} + \epsilon_N\big)\,\lambda_{noise},$$

with a small margin $\epsilon_N > 0$.

- *Signal-preservation threshold.* For a coordinate $j$ and some $k_0$ attaining (or meeting) a signal condition for $|\mu_{k_0 j}^*|$,

$$\Lambda_S^*(j;\lambda) \;=\; H_{k_0 j}^{eff}(\boldsymbol{\alpha}^*)\,|\mu_{k_0 j}^*| \;-\; \big(1 + \epsilon_S\big)\,\lambda_{noise} \;-\; \big(1 + \epsilon_S\big)\,E_{KKT\_rem}(\lambda),$$

where $\epsilon_S > 0$ is a fixed margin. Choosing $\lambda$ so that $\Lambda_S^*(j;\lambda) > 0$ ensures the signal at $(k_0, j)$ persists (i.e., $\widehat{\mu}_{k_0 j}(\lambda) \neq 0$) under the KKT inequalities.

**Assumption 15** (**True Relevant/Irrelevant Variables**). *Let $\mathbb{S}_{\boldsymbol{\mu}}^*$ denote the set of indices $j$ such that at least one component mean has a nonzero $j$-th entry, i.e., $j \in \mathbb{S}_{\boldsymbol{\mu}}^*$ iff $\exists\, k \in \{1, \dots, K\}$ with $\mu_{kj}^* \neq 0$. For $j \notin \mathbb{S}_{\boldsymbol{\mu}}^*$, we have $\mu_{kj}^* = 0$ for all $k$.*

**Assumption 16** (**Minimum Signal Strength**). *For each $j \in \mathbb{S}_{\boldsymbol{\mu}}^*$, there exists $k_0 \in \{1, \dots, K\}$ such that, for all $\lambda$ up to some detection level $\lambda_{detect\_upper}$,*

$$H_{k_0 j}^{\text{eff}}(\boldsymbol{\alpha}^*)\,|\mu_{k_0 j}^*| \;\geq\; \lambda \;+\; (1 + \epsilon_S)\lambda_{\text{noise}} \;+\; (1 + \epsilon_S)E_{\text{KKT\_rem}}(\lambda), \tag{30}$$

*with $\epsilon_S > 0$ fixed. In particular, Equation (30) implies $\Lambda_S^*(j;\lambda) > \lambda$. Moreover, we assume the separation*

$$\min_{j \in \mathbb{S}_{\boldsymbol{\mu}}^*} \lambda_j^\dagger \;>\; \Lambda_N^*, \qquad \text{where } \lambda_j^\dagger := \inf\{\lambda' > 0 : \; \lambda' = \Lambda_S^*(j;\lambda')\}, \tag{31}$$

*i.e., the smallest regularization at which the $j$-th signal would vanish strictly exceeds the noise-screening threshold $\Lambda_N^*$.*

**Assumption 17** (**Regularization Grid**). *The grid $\mathcal{G}_\lambda = \{\lambda^{(1)}, \dots, \lambda^{(M_G)}\}$ covers a neighborhood of the noise threshold $\Lambda_N^*$ and extends up to*

$$\min_{j \in \mathbb{S}_{\boldsymbol{\mu}}^*} \lambda_j^\dagger \quad \text{with } \lambda_j^\dagger \text{ as in Equation (31)}.$$

*Assume $\rho$ is either tied to $\lambda$ (e.g., $\rho \asymp \lambda$) or fixed so that its effect is absorbed by constants in the bounds.*

**Remark 5.** *The remainder $R_{kj}(\widehat{\boldsymbol{\alpha}}, \boldsymbol{\alpha}^*, \lambda)$ in Definition 4 captures the deviation of the mean-coordinate KKT equation from its linearized form. It aggregates: (i) off-diagonal Fisher blocks acting on other coordinates of $\boldsymbol{\Delta}$, (ii) empirical-population curvature fluctuations $(\boldsymbol{H}_N - \boldsymbol{I})$, and (iii) population curvature drift $\boldsymbol{I}(\tilde{\boldsymbol{\alpha}}) - \boldsymbol{I}(\boldsymbol{\alpha}^*)$ from evaluating at $\tilde{\boldsymbol{\alpha}}$. Under the standing assumptions (smoothness/compactness: Assumptions 7-8, Hessian concentration: Assumption 12) and the $L_2$-error bound from Lemma 3, one obtains the bound*

$$|R_{kj}(\widehat{\boldsymbol{\alpha}}(\lambda), \boldsymbol{\alpha}^*, \lambda)| \;\leq\; E_{\text{KKT\_rem}}(\lambda) \;\leq\; C_{\text{rem}}\lambda_{\text{noise}}$$

*on the high-probability event $\mathcal{E}_{grad} \cap \mathcal{E}_{RSC}$, provided the sparsity/sample-size regime ensures $C_H \sqrt{\frac{s_0 \ln d_\alpha}{N}} + L_I \|\widehat{\boldsymbol{\alpha}}(\lambda) - \boldsymbol{\alpha}^*\|_2 \lesssim 1$ (e.g., $s_0 \lesssim \sqrt{N/\ln d_\alpha}$). A crude but sufficient bound for individual coordinates is $\|\boldsymbol{\Delta}\|_\infty \leq \|\boldsymbol{\Delta}\|_2 = O\big(\sqrt{s_0 \ln d_\alpha / N}\big)$ by Lemma 3. Assumption 16 then ensures the effective signal $H_{k_0 j}^{\text{eff}}(\boldsymbol{\alpha}^*)|\mu_{k_0 j}^*|$ dominates both the regularization level and the stochastic/remainder terms, yielding persistence of true signals and suppression of noise along the regularization path.*

**Lemma 5** (Ranking Consistency for Mean Parameters). *Under Assumptions 13, 15, 16, and 17, and on the high-probability event $\mathcal{E}_{grad} \cap \mathcal{E}_{RSC}$ from Assumption 14, let the grid $\mathcal{G}_\lambda$ contain points distributed across $[0, \lambda_{grid\_max}]$ with*

$$\lambda_{grid\_max} \;>\; \min_{j \in \mathbb{S}_{\boldsymbol{\mu}}^*} \lambda_j^\dagger, \qquad \lambda_j^\dagger := \inf\{\lambda' > 0 : \; \lambda' = \Lambda_S^*(j;\lambda')\}.$$

*Then, with probability at least*

$$P_{rank} \;\geq\; 1 \;-\; M_G \cdot D \cdot \delta_N, \quad \text{where } \delta_N := C_1^{grad} d_\alpha^{-C_2^{grad}} + c_1^H d_\alpha^{-c_2^H},$$

*$M_G = |\mathcal{G}_\lambda|$, and $D$ denotes the number of penalized mean coordinates per variable (e.g., $D = K$), the following hold:*

1. **Relevant Variables.** *For each $j \in \mathbb{S}_{\boldsymbol{\mu}}^*$,*

$$\mathcal{O}_K(j) \geq N_{signal}(j) := \#\{\lambda \in \mathcal{G}_\lambda : \lambda < \Lambda_S^*(j; \lambda)\}.$$

*Let $\eta_R := \min_{j \in \mathbb{S}_{\boldsymbol{\mu}}^*} N_{signal}(j)$. By the separation Equation (31),*

$$\eta_R > \#\{\lambda \in \mathcal{G}_\lambda : \lambda \leq \Lambda_N^*\}.$$

2. **Irrelevant Variables.** *For each $j \notin \mathbb{S}_{\boldsymbol{\mu}}^*$,*

$$\mathcal{O}_K(j) \leq N_{noise} := \#\{\lambda \in \mathcal{G}_\lambda : \lambda \leq \Lambda_N^*\}.$$

*Consequently, $\eta_R > N_{noise}$, yielding a strict separation in ranking scores between relevant and irrelevant variables with probability at least $P_{rank}$.*

*Proof of Lemma 5.* Work on the high-probability event

$$\mathcal{E}_{\text{rank}} := \mathcal{E}_{\text{grad}} \cap \mathcal{E}_{\text{RSC}}$$

(on which the uniform gradient bound $|S_{kj}^*| \leq \lambda_{\text{noise}}$ and the KKT-remainder bound $|R_{kj}(\widehat{\boldsymbol{\alpha}}(\lambda), \boldsymbol{\alpha}^*, \lambda)| \leq E_{\text{KKT\_rem}}(\lambda)$ hold simultaneously for all penalized mean coordinates $(k, j)$ and all $\lambda \in \mathcal{G}_\lambda$). By Assumptions 13-17 and the union bound,

$$\mathbb{P}(\mathcal{E}_{\text{rank}}) \geq 1 - M_G \cdot D \cdot \delta_N, \quad \delta_N := C_1^{\text{grad}} d_\alpha^{-C_2^{\text{grad}}} + c_1^H d_\alpha^{-c_2^H}.$$

The ranking score for variable $j$ is

$$\mathcal{O}_K(j) = \sum_{\lambda \in \mathcal{G}_\lambda} \mathcal{I}(\exists k \in \{1, \ldots, K\} : \widehat{\mu}_{kj}(\lambda) \neq 0).$$

**KKT expansion.** For a penalized mean coordinate $(k, j)$, the KKT condition reads

$$0 = \nabla_{\mu_{kj}} \ell_N(\widehat{\boldsymbol{\alpha}}(\lambda)) + \lambda \widehat{\xi}_{kj}, \qquad \widehat{\xi}_{kj} \in \begin{cases} \{\text{sign}(\widehat{\mu}_{kj}(\lambda))\}, & \widehat{\mu}_{kj}(\lambda) \neq 0, \\ [-1, 1], & \widehat{\mu}_{kj}(\lambda) = 0. \end{cases}$$

A first-order expansion at $\boldsymbol{\alpha}^*$ yields

$$\nabla_{\mu_{kj}} \ell_N(\widehat{\boldsymbol{\alpha}}(\lambda)) = \underbrace{[\nabla_{\mu_{kj}} \ell_N(\boldsymbol{\alpha}^*)]}_{=: S_{kj}^*} + H_{kj}^{\text{eff}}(\boldsymbol{\alpha}^*)(\widehat{\mu}_{kj}(\lambda) - \mu_{kj}^*) + R_{kj}(\widehat{\boldsymbol{\alpha}}(\lambda), \boldsymbol{\alpha}^*, \lambda),$$

hence

$$\lambda \widehat{\xi}_{kj} = -S_{kj}^* - H_{kj}^{\text{eff}}(\boldsymbol{\alpha}^*)(\widehat{\mu}_{kj}(\lambda) - \mu_{kj}^*) - R_{kj}(\widehat{\boldsymbol{\alpha}}(\lambda), \boldsymbol{\alpha}^*, \lambda). \tag{32}$$

**Relevant variables** ($j \in \mathbb{S}_{\boldsymbol{\mu}}^*$)**.** Fix $j \in \mathbb{S}_{\boldsymbol{\mu}}^*$. By Assumption 16 there exists $k_0$ with $\mu_{k_0 j}^* \neq 0$ satisfying the signal condition. Assume, for contradiction, that at a given $\lambda \in \mathcal{G}_\lambda$ we have $\widehat{\mu}_{k_0 j}(\lambda) = 0$. Then $\widehat{\xi}_{k_0 j} \in [-1, 1]$ and $\widehat{\mu}_{k_0 j}(\lambda) - \mu_{k_0 j}^* = -\mu_{k_0 j}^*$. Applying Equation (32) and taking absolute values,

$$\lambda \geq \left| H_{k_0 j}^{\text{eff}}(\boldsymbol{\alpha}^*) \mu_{k_0 j}^* - S_{k_0 j}^* - R_{k_0 j} \right| \geq H_{k_0 j}^{\text{eff}}(\boldsymbol{\alpha}^*) |\mu_{k_0 j}^*| - |S_{k_0 j}^*| - |R_{k_0 j}|.$$

On $\mathcal{E}_{\text{rank}}$,

$$\lambda \geq H_{k_0 j}^{\text{eff}}(\boldsymbol{\alpha}^*) |\mu_{k_0 j}^*| - \lambda_{\text{noise}} - E_{\text{KKT\_rem}}(\lambda).$$

Therefore, whenever

$$\lambda < H_{k_0 j}^{\text{eff}}(\boldsymbol{\alpha}^*) |\mu_{k_0 j}^*| - \lambda_{\text{noise}} - E_{\text{KKT\_rem}}(\lambda),$$

we must have $\widehat{\mu}_{k_0 j}(\lambda) \neq 0$. Since $\Lambda_S^*(j; \lambda) = H_{k_0 j}^{\text{eff}}(\boldsymbol{\alpha}^*)|\mu_{k_0 j}^*| - (1 + \epsilon_S)\lambda_{\text{noise}} - (1 + \epsilon_S)E_{\text{KKT\_rem}}(\lambda)$ is a stricter threshold,

$$\lambda < \Lambda_S^*(j; \lambda) \implies \widehat{\mu}_{k_0 j}(\lambda) \neq 0.$$

Hence the count of grid points for which variable $j$ is (at least in one component) active satisfies

$$\mathcal{O}_K(j) \geq N_{\text{signal}}(j) := \#\{\lambda \in \mathcal{G}_\lambda : \lambda < \Lambda_S^*(j; \lambda)\}.$$

Taking the minimum over $j \in \mathbb{S}_{\boldsymbol{\mu}}^*$ gives $\eta_R := \min_{j \in \mathbb{S}_{\boldsymbol{\mu}}^*} N_{\text{signal}}(j)$.

**Irrelevant variables** ($j \notin \mathbb{S}_{\boldsymbol{\mu}}^*$). Here $\mu_{kj}^* = 0$ for all $k$. For a given $\lambda$, the zero solution $\widehat{\mu}_{kj}(\lambda) = 0$ is KKT-feasible iff

$$\left| \nabla_{\mu_{kj}} \ell_N(\widehat{\boldsymbol{\alpha}}(\lambda)) \right| \leq \lambda.$$

Using the expansion with $\mu_{kj}^* = 0$ and $\widehat{\mu}_{kj}(\lambda) = 0$,

$$\left| S_{kj}^* + R_{kj}(\widehat{\boldsymbol{\alpha}}(\lambda), \boldsymbol{\alpha}^*, \lambda) \right| \leq \lambda.$$

On $\mathcal{E}_{\text{rank}}$, this is guaranteed whenever

$$\lambda \geq \lambda_{\text{noise}} + E_{\text{KKT\_rem}}(\lambda).$$

By definition of the noise threshold $\Lambda_N^* = (1 + \epsilon_N)\lambda_{\text{noise}} + (1 + \epsilon_N) \sup_{\lambda' \leq \Lambda_N^*} E_{\text{KKT\_rem}}(\lambda')$ and the monotone/worst-case domination in its definition, any $\lambda > \Lambda_N^*$ satisfies $\lambda \geq \lambda_{\text{noise}} + E_{\text{KKT\_rem}}(\lambda)$. Therefore, for each $j \notin \mathbb{S}_{\boldsymbol{\mu}}^*$ and all $\lambda > \Lambda_N^*$, all coordinates $\widehat{\mu}_{kj}(\lambda)$ equal zero, so the indicator $\mathcal{I}(\exists k : \widehat{\mu}_{kj}(\lambda) \neq 0) = 0$. Consequently,

$$\mathcal{O}_K(j) \leq N_{\text{noise}} := \#\{\lambda \in \mathcal{G}_\lambda : \lambda \leq \Lambda_N^*\}.$$

**Separation and probability.** By the separation Assumption Equation (31),

$$\eta_R > N_{\text{noise}},$$

yielding a strict gap between the scores of relevant and irrelevant variables on $\mathcal{E}_{\text{rank}}$. Finally, by the union bound over at most $M_G$ grid points and $D$ penalized mean coordinates per variable, we obtain

$$\mathbb{P}\Big(\text{the above conclusions hold for all } j\Big) \geq 1 - M_G \cdot D \cdot \delta_N,$$

which is $P_{\text{rank}}$ in the statement. This completes the proof. $\qquad\square$

**Theorem 4** (Selection Consistency of the Two-Step SRUW Procedure)**.** *Assume all assumptions in Lemma 3 and Lemma 5, together with Theorem 2 for the final $(K, m, r, \ell)$ choice, hold. Let $s_0 = |\mathbb{S}_0|$ and $M_{NR} = p - s_0$ be the number of non-$\mathbb{S}_0$ variables. Moreover, suppose the following assumptions hold:*

(a) ***False negative for $\mathbb{S}_0$.*** *For any true relevant variable $j \in \mathbb{S}_0$ and any intermediate set $\widehat{\mathbb{S}}_{cur} \subset \mathbb{S}_0$ with $j \notin \widehat{\mathbb{S}}_{cur}$, the probability of incorrectly rejecting $j$ from $\mathbb{S}$ by the $BIC_{diff}$ criterion is uniformly bounded:*

$$\sup_{\widehat{\mathbb{S}}_{cur} \subset \mathbb{S}_0} \mathbb{P}\big(BIC_{diff}(j \mid \widehat{\mathbb{S}}_{cur}) \leq 0\big) \leq p_S(N),$$

*where $s_0 \cdot p_S(N) \leq \epsilon_{S,FN}(N)$ and $\epsilon_{S,FN}(N) = o_N(1)$.*

(b) ***False positive for non-$\mathbb{S}_0$.*** *For any true non-relevant variable $j \notin \mathbb{S}_0$ (i.e., $j \in \mathbb{U}_0 \cup \mathbb{W}_0$), given that $\mathbb{S}_0$ has been correctly identified (i.e., $\widehat{\mathbb{S}}_{cur} = \mathbb{S}_0$), the probability of $BIC_{diff}(j \mid \mathbb{S}_0) > 0$ is bounded:*

$$\mathbb{P}\big(BIC_{diff}(j \mid \mathbb{S}_0) > 0\big) \leq p_N(N),$$

*where $p_N(N) \to 0$ as $N \to \infty$. For BIC, typically $p_N(N) = \mathcal{O}(N^{-\gamma_B})$ for some $\gamma_B > 0$ (e.g., $\gamma_B \geq \Delta\nu_{\min}/2$ where $\Delta\nu_{\min} \geq 1$ is the minimum parameter-penalty gap).*

(c) ***Consistency of BIC-penalized regressions for $\widehat{\mathbb{R}}[j \mid \mathbb{S}_0]$.:***

- *For $j \in \mathbb{W}_0$, $\mathbb{P}\big(\widehat{\mathbb{R}}[j \mid \mathbb{S}_0] = \emptyset\big) \geq 1 - p_{reg}(N)$.*
- *For $j \in \mathbb{U}_0$, $\mathbb{P}\big(\widehat{\mathbb{R}}[j \mid \mathbb{S}_0] = \mathbb{R}_0(j) \neq \emptyset\big) \geq 1 - p_{reg}(N)$.*

*where $(w_0 + u_0)\, p_{reg}(N) = o_N(1)$.*

Let the stopping-rule parameter for selecting $\widehat{\mathbb{S}}$ (and $\widehat{\mathbb{W}}$) be $c$, understood so that once $\widehat{\mathbb{S}}_{cur} = \mathbb{S}_0$ holds, each $j \notin \mathbb{S}_0$ is tested at most $c$ times by the $BIC_{diff}(\cdot \mid \mathbb{S}_0)$ screening before termination.

For a desired tolerance $\epsilon_{S,FP} > 0$ on the probability of including any false positive in $\widehat{\mathbb{S}}$, it suffices to choose $c$ so that

$$M_{NR}\left(1 - (1 - p_N(N))^c\right) \leq \epsilon_{S,FP} \qquad \Longleftrightarrow \qquad c \leq \frac{\ln\left(1 - \epsilon_{S,FP}/M_{NR}\right)}{\ln\left(1 - p_N(N)\right)}. \qquad (33)$$

For small $p_N(N)$, this is well-approximated by

$$c \lesssim \frac{\epsilon_{S,FP}}{M_{NR}\, p_N(N)}.$$

In particular, if $p_N(N)$ decays polynomially in $N$ (e.g., $N^{-\gamma_B}$ with $\gamma_B \geq 1/2$) and $M_{NR}\, p_N(N) \to 0$, then any fixed $c_{fixed} \geq 1$ (e.g., $c_{fixed} = 1$ or 3) ensures

$$\mathbb{P}\left(any\ false\ positive\ in\ \widehat{\mathbb{S}}\right) \leq M_{NR}\left(1 - (1 - p_N(N))^{c_{fixed}}\right) \leq M_{NR}\, c_{fixed}\, p_N(N) \to 0.$$

Then, under (a)-(c) and with $c$ chosen according to Equation (33) (or with $c = c_{fixed}$ such that $M_{NR}\, c_{fixed}\, p_N(N) \to 0$), the two-step SRUW procedure recovers the true model structure $(K_0, m_0, r_0, \ell_0, \mathbb{V}_0)$ with probability $\mathbb{P}(Success) \to 1$ as $N \to \infty$.

*Proof.* We show each stage succeeds with high probability and then combine them.

**Conditioning on ranking accuracy.** By Lemma 5, with probability at least $1 - P_{\text{fail,rank\_total}}$, the ranking event $\mathcal{E}_{\text{rank}}$ holds: all $s_0$ relevant variables in $\mathbb{S}_0$ appear before any pure-noise $\mathbb{W}_0$ variables, and $\mathbb{U}_0$ appear after $\mathbb{S}_0$. We condition on $\mathcal{E}_{\text{rank}}$ for the remainder of the argument.

**No false negatives in $\widehat{\mathbb{S}}$.** Fix $j \in \mathbb{S}_0$ and any $\widehat{\mathbb{S}}_{cur} \subset \mathbb{S}_0$ not containing $j$. By Assumption (a),

$$\mathbb{P}\left(\text{BIC}_{\text{diff}}(j \mid \widehat{\mathbb{S}}_{cur}) \leq 0\right) \leq p_S(N).$$

By a union bound over the $s_0$ true variables,

$$\mathbb{P}\left(\mathbb{S}_0 \not\subseteq \widehat{\mathbb{S}} \mid \mathcal{E}_{\text{rank}}\right) \leq s_0\, p_S(N) =: \epsilon_{S,FN}(N) \to 0.$$

Let $\mathcal{E}_{\text{S-noFN}}$ be the event $\mathbb{S}_0 \subseteq \widehat{\mathbb{S}}$. Then $\mathbb{P}(\mathcal{E}_{\text{S-noFN}} \mid \mathcal{E}_{\text{rank}}) \geq 1 - \epsilon_{S,FN}(N)$.

**No false positives in $\widehat{\mathbb{S}}$.** On $\mathcal{E}_{\text{rank}} \cap \mathcal{E}_{\text{S-noFN}}$ we have $\widehat{\mathbb{S}}_{cur} = \mathbb{S}_0$. For any $j \notin \mathbb{S}_0$, Assumption (b) gives

$$\mathbb{P}\left(\text{BIC}_{\text{diff}}(j \mid \mathbb{S}_0) > 0\right) \leq p_N(N) =: q.$$

Let $\mathcal{E}_{\text{S-noFP}}$ be the event that no non-relevant variable is ever added to $\widehat{\mathbb{S}}$ after $\mathbb{S}_0$ is reached.

*(A) Distribution-free bound.* For any fixed set $\mathcal{J}$ of candidates examined of any size,

$$\mathbb{P}\left(\exists j \in \mathcal{J} : \text{BIC}_{\text{diff}}(j \mid \mathbb{S}_0) > 0\right) \leq |\mathcal{J}|\, q$$

by a union bound. In particular,

$$\mathbb{P}\left(\mathcal{E}_{\text{S-noFP}}^c \mid \mathcal{E}_{\text{rank}} \cap \mathcal{E}_{\text{S-noFN}}\right) \leq M_{NR}\, q,$$

which tends to 0 if $M_{NR}\, p_N(N) \to 0$. This bound requires no independence and is always valid.

*(B) Sharper bound under c-run termination.* Assume the screening step proceeds through non-relevant candidates until it observes $c$ consecutive rejections, and that the $c$ decisions in such a run are (asymptotically) independent or, more generally, satisfy

$$\mathbb{P}\Big( \bigcap_{t=1}^{c}\{\mathrm{BIC}_{\mathrm{diff}}(j_t \mid \mathbb{S}_0) \leq 0\}\Big) \;\geq\; (1-q)^c$$

for any $c$ distinct non-relevant candidates $(j_1,\ldots,j_c)$. Then the probability that a given $c$-block fails (i.e., contains at least one FP) is

$$\mathbb{P}(\text{block error}) \;\leq\; 1 - (1-q)^c.$$

Partition the $M_{NR}$ non-relevant indices into $\lfloor M_{NR}/c\rfloor$ disjoint blocks of size $c$ (discarding a remainder if needed). By a union bound over blocks,

$$\mathbb{P}\big(\mathcal{E}_{\text{S-noFP}}^c \mid \mathcal{E}_{\mathrm{rank}} \cap \mathcal{E}_{\text{S-noFN}}\big) \;\leq\; \frac{M_{NR}}{c}\Big(1-(1-q)^c\Big) \;\leq\; M_{NR}\Big(1-(1-q)^c\Big).$$

Therefore, enforcing

$$M_{NR}\Big(1-(1-p_N(N))^c\Big) \;\leq\; \epsilon_{S,FP} \quad\Longleftrightarrow\quad c \;\leq\; \frac{\ln\big(1-\epsilon_{S,FP}/M_{NR}\big)}{\ln\big(1-p_N(N)\big)}$$

yields $\mathbb{P}(\mathcal{E}_{\text{S-noFP}}^c \mid \mathcal{E}_{\mathrm{rank}} \cap \mathcal{E}_{\text{S-noFN}}) \leq \epsilon_{S,FP}$. For small $p_N(N)$, $1-(1-p_N(N))^c \sim c\,p_N(N)$, so $c \lesssim \epsilon_{S,FP}/(M_{NR}\,p_N(N))$.

Combining (A)-(B): whenever the independence/decoupling condition for runs holds, we may use the sharper $(1-(1-q)^c)$ design in the theorem statement; otherwise, the distribution-free guarantee $M_{NR}\,p_N(N)$ is valid (and implies the sharper one whenever $c$ is fixed and $M_{NR}p_N(N) \to 0$).

**Put together** $\widehat{\mathbb{S}} = \mathbb{S}_0$. Let $\mathcal{E}_S^* := \mathcal{E}_{\text{S-noFN}} \cap \mathcal{E}_{\text{S-noFP}}$. Then, with either (A) or (B),

$$\mathbb{P}\big(\widehat{\mathbb{S}} = \mathbb{S}_0 \,\big|\, \mathcal{E}_{\mathrm{rank}}\big) \;\geq\; 1 - \epsilon_{S,FN}(N) - \epsilon_{S,FP}(N) \;\to\; 1,$$

for $\epsilon_{S,FP}(N) = M_{NR}\,p_N(N)$ in (A), or $\epsilon_{S,FP}(N) = M_{NR}\,(1-(1-p_N(N))^c)$ in (B).

**Consistency of $\widehat{\mathbb{W}}, \widehat{\mathbb{U}}, \widehat{\mathbb{R}}$.** Given $\widehat{\mathbb{S}} = \mathbb{S}_0$, the reverse scan uses BIC-penalized regressions to decide $\widehat{\mathbb{W}}$ and $\widehat{\mathbb{R}}$. By Assumption (c),

$$\mathbb{P}\big(\widehat{\mathbb{W}} = \mathbb{W}_0 \,\big|\, \mathcal{E}_{\mathrm{rank}} \cap \mathcal{E}_S^*\big) \;\geq\; 1 - (w_0 + u_0)\,p_{\mathrm{reg}}(N) - \epsilon_{W,FP}(N) \;\to\; 1,$$

and

$$\mathbb{P}\big(\widehat{\mathbb{R}} = \mathbb{R}_0 \,\big|\, \mathcal{E}_{\mathrm{rank}} \cap \mathcal{E}_S^* \cap \{\widehat{\mathbb{W}} = \mathbb{W}_0\}\big) \;\geq\; 1 - u_0\,p_{\mathrm{reg}}(N) \;\to\; 1.$$

**Final SRUW choice via BIC.** By Theorem 2, the final selection of $(K, m, r, \ell)$ is consistent, i.e., $\mathbb{P}(P_{\text{final\_choice}}) \to 1$.

Multiplying the stage-wise success probabilities,

$$\mathbb{P}(\text{Overall Success}) \;\geq\; \mathbb{P}(\mathcal{E}_{\mathrm{rank}}) \cdot \mathbb{P}(\mathcal{E}_S^* \mid \mathcal{E}_{\mathrm{rank}}) \cdot \mathbb{P}(\widehat{\mathbb{W}} = \mathbb{W}_0 \mid \cdots)$$
$$\cdot\, \mathbb{P}(\widehat{\mathbb{R}} = \mathbb{R}_0 \mid \cdots) \cdot \mathbb{P}(P_{\text{final\_choice}}) \;\to\; 1.$$

The slowest decaying term among $P_{\text{fail,rank\_total}}$, $\epsilon_{S,FN}(N)$, $\epsilon_{S,FP}(N)$, $(w_0+u_0)p_{\mathrm{reg}}(N)$, $u_0 p_{\mathrm{reg}}(N)$ governs the overall rate. In particular, under the sharper design (B) with $c$ chosen as above,

$$\epsilon_{S,FP}(N) \;=\; M_{NR}\big(1-(1-p_N(N))^c\big) \;\leq\; \epsilon_{S,FP},$$

while the distribution-free design (A) yields $\epsilon_{S,FP}(N) = M_{NR}\, p_N(N) \to 0$ whenever $M_{NR}\, p_N(N) \to 0$. $\qquad\square$

**Theorem 5** (Equivalence of MNARz-SRUW and MAR on Augmented Data for SRUW). *Consider an observation* $(\boldsymbol{y}_n, \boldsymbol{c}_n)$ *where* $\boldsymbol{y}_n = (\boldsymbol{y}_n^S, \boldsymbol{y}_n^U, \boldsymbol{y}_n^W)$ *and* $\boldsymbol{c}_n$ *is its missingness pattern. Let the complete-data likelihood for* $\boldsymbol{y}_n$ *given cluster* $z_{nk} = 1$ *under the SRUW model* $(K, m, r, \ell, \mathbb{V})$ *be:*

$$f_{SRUW}(\boldsymbol{y}_n \mid z_{nk} = 1; \theta_k) = f_k(\boldsymbol{y}_n^S; \boldsymbol{\alpha}_k)\, f_{reg}(\boldsymbol{y}_n^U \mid \boldsymbol{y}_n^R; \boldsymbol{\theta}_{reg})\, f_{indep}(\boldsymbol{y}_n^W; \boldsymbol{\theta}_{indep}).$$

*Assume an MNARz-SRUW mechanism for missing data, where the probability of the missingness pattern* $\boldsymbol{c}_n$ *depends only on the cluster membership* $z_{nk}$:

$$\begin{aligned}
f(\boldsymbol{c}_n \mid \boldsymbol{y}_n, z_{nk} = 1; \psi_k) &= f(\boldsymbol{c}_n \mid z_{nk} = 1; \psi_k) \\
&= \prod_{d \in S \cup U \cup W} \rho_{kd}^{c_{nd}} \left(1 - \rho_{kd}\right)^{1 - c_{nd}},
\end{aligned} \tag{34}$$

*where* $\rho_{kd} \in (0,1)$ *are components of the missingness parameter* $\psi_k$ *(thus* $f(\boldsymbol{c}_n \mid z_{nk} = 1; \psi_k)$ *does not depend on* $\boldsymbol{y}_n$*). The observed data under this MNARz-SRUW model is* $(\boldsymbol{y}_n^o, \boldsymbol{c}_n)$*, and its likelihood is:*

$$L_{MNARz\text{-}SRUW}(\boldsymbol{y}_n^o, \boldsymbol{c}_n; \boldsymbol{\theta}, \boldsymbol{\psi}) = \int \sum_{k=1}^K \pi_k\, f_{SRUW}(\boldsymbol{y}_n \mid z_{nk} = 1; \theta_k)\, f(\boldsymbol{c}_n \mid z_{nk} = 1; \psi_k)\, d\boldsymbol{y}_n^m. \tag{35}$$

*Now consider the augmented observed vector* $\tilde{\boldsymbol{y}}_n^o = (\boldsymbol{y}_n^o, \boldsymbol{c}_n)$*. Assume* $\tilde{\boldsymbol{y}}_n^o$ *is i.i.d. from a mixture model in which the conditional law of the missing part* $\boldsymbol{y}_n^m$ *is MAR with respect to* $\tilde{\boldsymbol{y}}_n^o$ *(i.e., the missingness mechanism does not depend on* $\boldsymbol{y}_n^m$ *given* $(\boldsymbol{y}_n^o, \boldsymbol{c}_n)$*). Define the \*\*augmented observed-data\*\* likelihood as*

$$\tilde{f}_{MAR}(\tilde{\boldsymbol{y}}_n^o; \boldsymbol{\theta}, \boldsymbol{\psi}) = \sum_{k=1}^K \pi_k \left( \int f_{SRUW}(\boldsymbol{y}_n^o, \boldsymbol{y}_n^m \mid z_{nk} = 1; \theta_k)\, d\boldsymbol{y}_n^m \right) f(\boldsymbol{c}_n \mid z_{nk} = 1; \psi_k), \tag{36}$$

*where* $f_{SRUW}(\cdot \mid z_{nk} = 1; \theta_k)$ *is the same component density as above.*

*Then, for fixed parameters* $(\boldsymbol{\theta}, \boldsymbol{\psi})$*, the observed-data likelihood under MNARz-SRUW for* $(\boldsymbol{y}_n^o, \boldsymbol{c}_n)$ *is identical to the likelihood of the augmented observation* $\tilde{\boldsymbol{y}}_n^o = (\boldsymbol{y}_n^o, \boldsymbol{c}_n)$ *under the MAR interpretation:*

$$L_{MNARz\text{-}SRUW}(\boldsymbol{y}_n^o, \boldsymbol{c}_n; \boldsymbol{\theta}, \boldsymbol{\psi}) = \tilde{f}_{MAR}(\tilde{\boldsymbol{y}}_n^o; \boldsymbol{\theta}, \boldsymbol{\psi}).$$

*Proof of Theorem 5.* Starting from Equation (35), by the MNARz assumption $f(\boldsymbol{c}_n \mid \boldsymbol{y}_n, z_{nk} = 1; \psi_k) = f(\boldsymbol{c}_n \mid z_{nk} = 1; \psi_k)$ is independent of $\boldsymbol{y}_n^m$, hence

$$\begin{aligned}
L_{\text{MNARz-SRUW}}(\boldsymbol{y}_n^o, \boldsymbol{c}_n; \boldsymbol{\theta}, \boldsymbol{\psi}) &= \int \sum_{k=1}^K \pi_k\, f_{\text{SRUW}}(\boldsymbol{y}_n^o, \boldsymbol{y}_n^m \mid z_{nk} = 1; \theta_k)\, f(\boldsymbol{c}_n \mid z_{nk} = 1; \psi_k)\, d\boldsymbol{y}_n^m \\
&= \sum_{k=1}^K \pi_k\, f(\boldsymbol{c}_n \mid z_{nk} = 1; \psi_k) \left( \int f_{\text{SRUW}}(\boldsymbol{y}_n^o, \boldsymbol{y}_n^m \mid z_{nk} = 1; \theta_k)\, d\boldsymbol{y}_n^m \right) \\
&= \sum_{k=1}^K \pi_k\, f_k(\boldsymbol{y}_n^o \mid z_{nk} = 1; \theta_k)\, f(\boldsymbol{c}_n \mid z_{nk} = 1; \psi_k),
\end{aligned}$$

where we set $f_k(\boldsymbol{y}_n^o \mid z_{nk} = 1; \theta_k) := \int f_{\text{SRUW}}(\boldsymbol{y}_n^o, \boldsymbol{y}_n^m \mid z_{nk} = 1; \theta_k)\, d\boldsymbol{y}_n^m$. Comparing with Equation (36) yields

$$L_{\text{MNARz-SRUW}}(\boldsymbol{y}_n^o, \boldsymbol{c}_n; \boldsymbol{\theta}, \boldsymbol{\psi}) = \tilde{f}_{\text{MAR}}(\tilde{\boldsymbol{y}}_n^o; \boldsymbol{\theta}, \boldsymbol{\psi}),$$

as claimed. $\qquad\square$

By Theorem 5, the observed-data likelihood under MNARz-SRUW equals the augmented observed-data likelihood under MAR for every $(\boldsymbol{\theta}, \boldsymbol{\psi})$ and each observation; hence, the full-sample likelihoods coincide pointwise in $(\boldsymbol{\theta}, \boldsymbol{\psi})$. The following corollary shows the resulting estimator-level equivalence.

**Corollary 1** (Estimator equivalence under augmentation). *Under the conditions of Theorem 5, for any fixed $(\boldsymbol{\theta}, \boldsymbol{\psi})$ and for each observation,*

$$L_{\textit{MNARz-SRUW}}(\boldsymbol{y}_n^o, \boldsymbol{c}_n; \boldsymbol{\theta}, \boldsymbol{\psi}) \;=\; \tilde{f}_{\textit{MAR}}(\tilde{\boldsymbol{y}}_n^o; \boldsymbol{\theta}, \boldsymbol{\psi}).$$

*Hence the full-sample observed-data likelihoods coincide, and therefore the sets of maximum likelihood estimators for $(\boldsymbol{\theta}, \boldsymbol{\psi})$ under MNARz-SRUW based on $(\boldsymbol{y}^o, \boldsymbol{c})$ and under MAR on the augmented data $\tilde{\boldsymbol{y}}^o = (\boldsymbol{y}^o, \boldsymbol{c})$ are identical (up to label switching). Moreover, if the same priors on $(\boldsymbol{\theta}, \boldsymbol{\psi})$ are used in both formulations, the resulting Bayesian posteriors coincide.*

*Proof of Corollary 1.* By Theorem 5, for every observation $n$ and every fixed $(\boldsymbol{\theta}, \boldsymbol{\psi})$,

$$L_{\text{MNARz-SRUW}}(\boldsymbol{y}_n^o, \boldsymbol{c}_n; \boldsymbol{\theta}, \boldsymbol{\psi}) \;=\; \tilde{f}_{\text{MAR}}(\tilde{\boldsymbol{y}}_n^o; \boldsymbol{\theta}, \boldsymbol{\psi}), \qquad \tilde{\boldsymbol{y}}_n^o = (\boldsymbol{y}_n^o, \boldsymbol{c}_n).$$

Let $\mathcal{D} = \{(\boldsymbol{y}_n^o, \boldsymbol{c}_n)\}_{n=1}^N$ denote the sample, assumed i.i.d. under the mixture model as in the theorem. The full-sample observed-data likelihoods are then

$$L_{\text{MNARz-SRUW}}(\mathcal{D}; \boldsymbol{\theta}, \boldsymbol{\psi}) \;=\; \prod_{n=1}^N L_{\text{MNARz-SRUW}}(\boldsymbol{y}_n^o, \boldsymbol{c}_n; \boldsymbol{\theta}, \boldsymbol{\psi})$$

$$\tilde{L}_{\text{MAR}}(\mathcal{D}; \boldsymbol{\theta}, \boldsymbol{\psi}) \;=\; \prod_{n=1}^N \tilde{f}_{\text{MAR}}(\tilde{\boldsymbol{y}}_n^o; \boldsymbol{\theta}, \boldsymbol{\psi}).$$

By pointwise equality for each factor, we obtain the full-sample equality

$$L_{\text{MNARz-SRUW}}(\mathcal{D}; \boldsymbol{\theta}, \boldsymbol{\psi}) \;=\; \tilde{L}_{\text{MAR}}(\mathcal{D}; \boldsymbol{\theta}, \boldsymbol{\psi}) \quad \text{for all } (\boldsymbol{\theta}, \boldsymbol{\psi}).$$

Consequently, the sets of maximum likelihood estimators coincide:

$$\arg \max_{(\boldsymbol{\theta}, \boldsymbol{\psi})} L_{\text{MNARz-SRUW}}(\mathcal{D}; \boldsymbol{\theta}, \boldsymbol{\psi}) \;=\; \arg \max_{(\boldsymbol{\theta}, \boldsymbol{\psi})} \tilde{L}_{\text{MAR}}(\mathcal{D}; \boldsymbol{\theta}, \boldsymbol{\psi}),$$

up to the usual permutations of mixture component labels (label switching), since both likelihoods are invariant under relabelings.

For the Bayesian statement, let $\Pi$ be a common prior on $(\boldsymbol{\theta}, \boldsymbol{\psi})$ admitting a density $\pi(\boldsymbol{\theta}, \boldsymbol{\psi})$ with respect to a common dominating measure. The posteriors are

$$\pi_{\text{MNARz}}(\boldsymbol{\theta}, \boldsymbol{\psi} \mid \mathcal{D}) \;\propto\; \pi(\boldsymbol{\theta}, \boldsymbol{\psi}) \, L_{\text{MNARz-SRUW}}(\mathcal{D}; \boldsymbol{\theta}, \boldsymbol{\psi})$$

$$\pi_{\text{MAR}}(\boldsymbol{\theta}, \boldsymbol{\psi} \mid \mathcal{D}) \;\propto\; \pi(\boldsymbol{\theta}, \boldsymbol{\psi}) \, \tilde{L}_{\text{MAR}}(\mathcal{D}; \boldsymbol{\theta}, \boldsymbol{\psi}).$$

Since the likelihoods coincide pointwise in $(\boldsymbol{\theta}, \boldsymbol{\psi})$, the unnormalized posteriors agree, hence their normalizing constants (integrals over the same parameter space) are also equal. Therefore

$$\pi_{\text{MNARz}}(\boldsymbol{\theta}, \boldsymbol{\psi} \mid \mathcal{D}) \;=\; \pi_{\text{MAR}}(\boldsymbol{\theta}, \boldsymbol{\psi} \mid \mathcal{D}),$$

again, modulo label switching. $\qquad\square$

**Remark 6.** *In the MAR interpretation of the augmented data $\tilde{\boldsymbol{y}}_n^o$, the components $\boldsymbol{y}_n^o$ have density $f_k(\boldsymbol{y}_n^o \mid z_{nk} = 1; \theta_k)$ in cluster $k$, and the components $\boldsymbol{c}_n$ (which are fully observed within $\tilde{\boldsymbol{y}}_n^o$) have density $f(\boldsymbol{c}_n \mid z_{nk} = 1; \psi_k)$ in cluster $k$. The MAR assumption applies to $\boldsymbol{y}_n^m$, meaning its missingness mechanism, given $\boldsymbol{y}_n^o$ and $\boldsymbol{c}_n$, does not depend on $\boldsymbol{y}_n^m$ itself. The likelihood of $\tilde{\boldsymbol{y}}_n^o$ is formed by integrating out $\boldsymbol{y}_n^m$ from $f(\boldsymbol{y}_n, \boldsymbol{c}_n \mid z_{nk} = 1)$, which leads to the product $f_k(\boldsymbol{y}_n^o \mid z_{nk} = 1; \theta_k) \, f(\boldsymbol{c}_n \mid z_{nk} = 1; \psi_k)$ for each cluster $k$.*

## E Detailed EM Algorithms for the Two-Stage Procedure

This section gives complete EM derivations for both stages. In Stage A (ranking), we run a penalized GMM on the *imputed and standardized* data $\bar{Y} = \text{std}(\tilde{Y})$, where $\tilde{Y}$ is the single-imputed matrix defined in Algorithm 1 which is used for ranking only. In Stage B (role assignment), we fit the unpenalized SRUW model under MNARz by exploiting the equivalence to MAR on the augmented data $(Y, C)$. Throughout, $\Psi_k = \Sigma_k^{-1}$, the SRUW partition is $V = (\mathbb{S}, \mathbb{R}, \mathbb{U}, \mathbb{W})$, and $t_{nk}$ denotes EM responsibilities. We write $\ell(\cdot)$ for (penalized) log-likelihoods and $Q(\cdot\,; \cdot)$ for EM $Q$-functions.

**Stage A (ranking): Penalized EM for adaptive-regularized GMM**

**Penalized objective.** Given $\bar{\boldsymbol{y}}_1, \ldots, \bar{\boldsymbol{y}}_N \in \mathbb{R}^D$, we maximize the penalized observed-data log-likelihood

$$\ell_{\text{pen}}(\boldsymbol{\alpha}) = \sum_{n=1}^{N} \log\Big[ \sum_{k=1}^{K} \pi_k \, \phi(\bar{\boldsymbol{y}}_n \mid \boldsymbol{\mu}_k, \boldsymbol{\Sigma}_k) \Big] - \lambda \sum_{k=1}^{K} \|\boldsymbol{\mu}_k\|_1 - \rho \sum_{k=1}^{K} \sum_{i \neq j} \boldsymbol{P}_{k,ij} \, |\boldsymbol{\Psi}_{k,ij}|. \quad (37)$$

*Assumption:* For each EM run at a fixed $(\lambda, \rho)$, the weights $\boldsymbol{P}_k$ are computed from the warm start and then held fixed.[3]

At iteration $t$, with responsibilities $t_{nk}^{(t)}$, the penalized $Q$-function is

$$Q_{\text{pen}}(\boldsymbol{\alpha}; \boldsymbol{\alpha}^{(t-1)}) = \sum_{n=1}^{N} \sum_{k=1}^{K} t_{nk}^{(t)} \Big\{ \log \pi_k - \tfrac{1}{2} \log |\boldsymbol{\Sigma}_k| - \tfrac{1}{2} (\bar{\boldsymbol{y}}_n - \boldsymbol{\mu}_k)^\top \boldsymbol{\Psi}_k (\bar{\boldsymbol{y}}_n - \boldsymbol{\mu}_k) \Big\}$$
$$- \lambda \sum_{k=1}^{K} \|\boldsymbol{\mu}_k\|_1 - \rho \sum_{k=1}^{K} \sum_{i \neq j} \boldsymbol{P}_{k,ij} \, |\boldsymbol{\Psi}_{k,ij}|. \quad (38)$$

**E-step.**

$$t_{nk}^{(t)} = \frac{\pi_k^{(t-1)} \phi(\bar{\boldsymbol{y}}_n \mid \boldsymbol{\mu}_k^{(t-1)}, \boldsymbol{\Sigma}_k^{(t-1)})}{\sum_{\ell=1}^{K} \pi_\ell^{(t-1)} \phi(\bar{\boldsymbol{y}}_n \mid \boldsymbol{\mu}_\ell^{(t-1)}, \boldsymbol{\Sigma}_\ell^{(t-1)})}, \qquad n_k^{(t)} = \sum_{n=1}^{N} t_{nk}^{(t)}. \quad (39)$$

**M-step: mixing weights.** $\pi_k^{(t)} = n_k^{(t)}/N$.

**M-step: means $\boldsymbol{\mu}_k$ with $\ell_1$ penalty.** Given $\boldsymbol{\Psi}_k^{(t-1)}$, the subproblem in $\boldsymbol{\mu}_k$ is convex. Let

$$\bar{\boldsymbol{m}}_k^{(t)} := \frac{1}{n_k^{(t)}} \sum_{n=1}^{N} t_{nk}^{(t)} \bar{\boldsymbol{y}}_n.$$

Then the gradient of the smooth part is

$$\nabla_{\boldsymbol{\mu}_k} \Big[ \tfrac{1}{2} \sum_{n=1}^{N} t_{nk}^{(t)} (\bar{\boldsymbol{y}}_n - \boldsymbol{\mu}_k)^\top \boldsymbol{\Psi}_k^{(t-1)} (\bar{\boldsymbol{y}}_n - \boldsymbol{\mu}_k) \Big] = n_k^{(t)} \boldsymbol{\Psi}_k^{(t-1)} (\boldsymbol{\mu}_k - \bar{\boldsymbol{m}}_k^{(t)}).$$

The KKT optimality for coordinate $j$ is

$$n_k^{(t)} (\boldsymbol{\Psi}_k^{(t-1)})_{jj} \mu_{kj} + n_k^{(t)} \sum_{v \neq j} (\boldsymbol{\Psi}_k^{(t-1)})_{jv} \mu_{kv} - n_k^{(t)} (\boldsymbol{\Psi}_k^{(t-1)} \bar{\boldsymbol{m}}_k^{(t)})_j \in \lambda \, \partial |\mu_{kj}|.$$

A coordinate-descent update is

$$\mu_{kj} \leftarrow \frac{1}{n_k^{(t)} (\boldsymbol{\Psi}_k^{(t-1)})_{jj}} \mathcal{S}_\lambda\Big( n_k^{(t)} (\boldsymbol{\Psi}_k^{(t-1)} \bar{\boldsymbol{m}}_k^{(t)})_j - n_k^{(t)} \sum_{v \neq j} (\boldsymbol{\Psi}_k^{(t-1)})_{jv} \mu_{kv} \Big),$$

where $\mathcal{S}_\lambda(u) = \text{sign}(u) \, \max\{|u| - \lambda, 0\}$.

**M-step: precisions $\boldsymbol{\Psi}_k$ via weighted graphical lasso.** With $\boldsymbol{\mu}_k^{(t)}$ fixed, define the responsibility-weighted covariance

$$\boldsymbol{S}_k^{(t)} = \frac{1}{n_k^{(t)}} \sum_{n=1}^{N} t_{nk}^{(t)} (\bar{\boldsymbol{y}}_n - \boldsymbol{\mu}_k^{(t)})(\bar{\boldsymbol{y}}_n - \boldsymbol{\mu}_k^{(t)})^\top.$$

Then

$$\boldsymbol{\Psi}_k^{(t)} \in \arg\min_{\boldsymbol{\Psi} \succ 0} \Big\{ -\log \det \boldsymbol{\Psi} + \text{tr}(\boldsymbol{S}_k^{(t)} \boldsymbol{\Psi}) + \frac{2\rho}{n_k^{(t)}} \sum_{i \neq j} \boldsymbol{P}_{k,ij} \, |\boldsymbol{\Psi}_{ij}| \Big\},$$

with diagonals unpenalized; standard `glasso` solvers apply.

---

[3]If $\boldsymbol{P}_k$ is updated during EM, one must add a majorization step to preserve monotonicity.

**Stage B (role assignment): EM for SRUW under MNARz**

**Observed likelihood via augmentation.** Let $\mathcal{D}_{\mathrm{MAR}} \dot\cup \mathcal{D}_{\mathrm{MNAR}} = [D]$. Under MNARz,

$$\ell(\Theta; \boldsymbol{Y}, \boldsymbol{C}) = \sum_{n=1}^{N} \log\Big[ \sum_{k=1}^{K} \pi_k \, f_{k,\mathrm{MAR}}^o(\boldsymbol{y}_n^o; \boldsymbol{\alpha}_k, \boldsymbol{\xi}) \, f_c^{\mathrm{MNARz}}(\boldsymbol{c}_{n,\mathrm{MNAR}}; \boldsymbol{\psi}_k) \Big],$$

is a standard mixture on the augmented observation $(\boldsymbol{y}_n^o, \boldsymbol{c}_{n,\mathrm{MNAR}})$.

**Complete-data log-likelihood and $Q$-function.** With $\Theta = (\boldsymbol{\pi}, \{\boldsymbol{\alpha}_k\}, \boldsymbol{\xi}, \{\boldsymbol{\psi}_k\})$ and latent $\{z_{nk}\}$,

$$Q(\Theta; \Theta^{(t-1)}) = \sum_{n=1}^{N} \sum_{k=1}^{K} t_{nk}^{(t)} \Big\{ \log \pi_k + \mathbb{E}[\log f_k(\boldsymbol{Y}_n; \boldsymbol{\alpha}_k, \boldsymbol{\xi}) \mid \boldsymbol{y}_n^o, z_{nk}{=}1] + \log f_c^{\mathrm{MNARz}}(\boldsymbol{c}_{n,\mathrm{MNAR}}; \boldsymbol{\psi}_k) \Big\}.$$

**E-step.**

$$t_{nk}^{(t)} = \frac{\pi_k^{(t-1)} \, f_{k,\mathrm{MAR}}^o(\boldsymbol{y}_n^o; \boldsymbol{\alpha}_k^{(t-1)}, \boldsymbol{\xi}^{(t-1)}) \, f_c^{\mathrm{MNARz}}(\boldsymbol{c}_{n,\mathrm{MNAR}}; \boldsymbol{\psi}_k^{(t-1)})}{\sum_{\ell=1}^{K} \pi_\ell^{(t-1)} \, f_{\ell,\mathrm{MAR}}^o(\boldsymbol{y}_n^o; \boldsymbol{\alpha}_\ell^{(t-1)}, \boldsymbol{\xi}^{(t-1)}) \, f_c^{\mathrm{MNARz}}(\boldsymbol{c}_{n,\mathrm{MNAR}}; \boldsymbol{\psi}_\ell^{(t-1)})}.$$

**M-step: $\pi_k$ and MNARz parameters.**

$$\pi_k^{(t)} = \frac{1}{N} \sum_{n=1}^{N} t_{nk}^{(t)}, \qquad \hat\rho_k^{(t)} = \frac{\sum_{n=1}^{N} t_{nk}^{(t)} \sum_{d \in \mathcal{D}_{\mathrm{MNAR}}} c_{nd}}{\sum_{n=1}^{N} t_{nk}^{(t)} |\mathcal{D}_{\mathrm{MNAR}}|}, \quad \boldsymbol{\psi}_k^{(t)} = \log \frac{\hat\rho_k^{(t)}}{1 - \hat\rho_k^{(t)}}.$$

**M-step: SRUW data model updates.** Write

$$f_k(\boldsymbol{y}; \boldsymbol{\alpha}_k, \boldsymbol{\xi}) = f_{\mathrm{clust}}(\boldsymbol{y}^{\mathbb{S}}; \boldsymbol{\mu}_k^{\mathbb{S}}, \boldsymbol{\Sigma}_k^{\mathbb{S}}) \, f_{\mathrm{reg}}(\boldsymbol{y}^{\mathbb{U}} \mid \boldsymbol{y}^{\mathbb{R}}; \boldsymbol{a}, \boldsymbol{\beta}, \boldsymbol{\Omega}) \, f_{\mathrm{indep}}(\boldsymbol{y}^{\mathbb{W}}; \boldsymbol{\gamma}, \boldsymbol{\Gamma}).$$

Let $\mathbb{E}_k[\cdot \mid \boldsymbol{y}_n^o]$ denote Gaussian conditional expectations under component $k$. Then the sufficient statistics are the *mixture-weighted* moments

$$\mathbb{E}[\cdot \mid \boldsymbol{y}_n^o] = \sum_{k=1}^{K} t_{nk}^{(t)} \mathbb{E}_k[\cdot \mid \boldsymbol{y}_n^o, z_{nk}{=}1].$$

*Cluster block $\mathbb{S}$:* for each $k$, compute

$$\bar{\boldsymbol{y}}_k^{\mathbb{S},(t)} = \frac{1}{n_k^{(t)}} \sum_{n=1}^{N} t_{nk}^{(t)} \mathbb{E}_k[\boldsymbol{y}_n^{\mathbb{S}} \mid \boldsymbol{y}_n^o], \qquad \boldsymbol{S}_k^{\mathbb{S},(t)} = \frac{1}{n_k^{(t)}} \sum_{n=1}^{N} t_{nk}^{(t)} \mathbb{E}_k[\boldsymbol{y}_n^{\mathbb{S}} (\boldsymbol{y}_n^{\mathbb{S}})^\top \mid \boldsymbol{y}_n^o],$$

and set $\boldsymbol{\mu}_k^{\mathbb{S},(t)} = \bar{\boldsymbol{y}}_k^{\mathbb{S},(t)}$ and $\boldsymbol{\Sigma}_k^{\mathbb{S},(t)} = \boldsymbol{S}_k^{\mathbb{S},(t)} - \bar{\boldsymbol{y}}_k^{\mathbb{S},(t)} (\bar{\boldsymbol{y}}_k^{\mathbb{S},(t)})^\top$.

*Regression block $\mathbb{U} \mid \mathbb{R}$:* let $\boldsymbol{X}_n = [1 \ (\boldsymbol{y}_n^{\mathbb{R}})^\top]$. Form the *mixture-weighted* global moments

$$\boldsymbol{A} = \sum_{n=1}^{N} \sum_{k=1}^{K} t_{nk}^{(t)} \mathbb{E}_k[\boldsymbol{X}_n^\top \boldsymbol{X}_n \mid \boldsymbol{y}_n^o], \qquad \boldsymbol{B} = \sum_{n=1}^{N} \sum_{k=1}^{K} t_{nk}^{(t)} \mathbb{E}_k[\boldsymbol{X}_n^\top \boldsymbol{y}_n^{\mathbb{U}} \mid \boldsymbol{y}_n^o],$$

then $\begin{bmatrix} \boldsymbol{a}^{(t)} \\ \boldsymbol{\beta}^{(t)} \end{bmatrix} = \boldsymbol{A}^{-1} \boldsymbol{B}$, and

$$\boldsymbol{\Omega}^{(t)} = \frac{1}{N} \sum_{n=1}^{N} \sum_{k=1}^{K} t_{nk}^{(t)} \mathbb{E}_k\Big[ (\boldsymbol{y}_n^{\mathbb{U}} - \boldsymbol{a}^{(t)} - \boldsymbol{y}_n^{\mathbb{R}} \boldsymbol{\beta}^{(t)})(\boldsymbol{y}_n^{\mathbb{U}} - \boldsymbol{a}^{(t)} - \boldsymbol{y}_n^{\mathbb{R}} \boldsymbol{\beta}^{(t)})^\top \Big| \boldsymbol{y}_n^o \Big].$$

*Independent block $\mathbb{W}$:*

$$\boldsymbol{\gamma}^{(t)} = \frac{1}{N} \sum_{n=1}^{N} \sum_{k=1}^{K} t_{nk}^{(t)} \mathbb{E}_k[\boldsymbol{y}_n^{\mathbb{W}} \mid \boldsymbol{y}_n^o], \quad \boldsymbol{\Gamma}^{(t)} = \frac{1}{N} \sum_{n=1}^{N} \sum_{k=1}^{K} t_{nk}^{(t)} \mathbb{E}_k[\boldsymbol{y}_n^{\mathbb{W}} (\boldsymbol{y}_n^{\mathbb{W}})^\top \mid \boldsymbol{y}_n^o] - \boldsymbol{\gamma}^{(t)} (\boldsymbol{\gamma}^{(t)})^\top.$$

**Numerical stability.** In Stage A, we use warm starts and pathwise $(\lambda, \rho)$; diagonals unpenalized and enforce $\boldsymbol{\Psi}_k \succ 0$. In Stage B, if some $n_k^{(t)}$ is tiny, we add a small ridge to $\boldsymbol{\Sigma}_k^{\mathbb{S}}$ or merge/discard components per BIC.

# F  Additional Experiments

## F.1  Metric Details

We outline some metrics that we use to evaluate the clustering accuracy and imputation error, which are summarized below:

Table 3: Summary of evaluation metrics for clustering and imputation quality

| Name | Formula | Range |
|------|---------|-------|
| **Imputation error** | | |
| NRMSE | $\dfrac{\sqrt{\frac{1}{N}\sum_{i=1}^{N}(y_i - \hat{y}_i)^2}}{\sigma}$ | $[0,1]$ |
| WNRMSE | $\dfrac{\sum_{c=1}^{C}(\text{NRMSE}_c \times w_c)}{\sum_{c=1}^{C} w_c}$ | $[0,1]$ |
| **Clustering similarity** | | |
| ARI | $\dfrac{\text{RI} - \mathbb{E}[\text{RI}]}{\max(\text{RI}) - \mathbb{E}[\text{RI}]}$ | $[0,1]$ |
| **Composite** | | |
| CIIE | $\alpha \times (1 - \text{NRMSE}) + \boldsymbol{\beta} \times \text{Similarity Score}$ | $[0,1]$ |

## F.2  Assessing Integrated MNARz Handling

We conduct an additional experiment to showcase the performance of handling MNAR pattern within the variable selection framework of which the setups are similar to [62]. Datasets were simulated with $n = 100$ observations, $K = 3$ true clusters with proportions $\boldsymbol{\pi} = (0.5, 0.25, 0.25))$, and $D = 6, 9$ variables. True cluster memberships $\mathbf{Z}$ were drawn according to $\boldsymbol{\pi}$. The complete data $\mathbf{Y}$ was generated as $Y_{nd} = \sum_{k'=1}^{K} Z_{nk'}\delta_{k'd} + \epsilon_{nd}$, where $\epsilon_{nd} \sim \mathcal{N}(0,1)$, and $\boldsymbol{\delta}$ defined cluster-specific mean shifts with signal strength $\tau = 2.31$ with $\delta_{11} = \delta_{14} = \delta_{22} = \delta_{25} = \delta_{33} = \delta_{36} = \tau$, others zero). For a specific MNAR scenario, the class-specific intercept component $\psi_k^z$ was 0, and variable-specific slopes $\psi_j^y$ were $(1.45, 0.2, -3, 1.45, 0.2, -3)$, resulting in $P(M_{ij} = 1|Y_{ij}, Z_{ik} = 1) = \Phi(\psi_j^y Y_{ij})$.

In Figure 4 and 5, we plot the boxplots of ARI and NRMSE over 20 replications. We observe that MNAR-based approaches attain better results over methods not designed for MNAR patterns. These methods deliver competitive performance even when the true missingness mechanism is more complex (MNARy, MNARyz); thereby, supporting the conclusion in [62]. Furthermore, our framework demonstrates even slightly higher ARI and lower NRMSE in some cases compared to standard MNARz.

## F.3  Sensitivity on the choice of c

As discussed in [13] and further investigated through Theorem 4, the choice of the hyperparameter $c$ plays a crucial role in the stepwise construction of the relevant variable set $\widehat{\mathbb{S}}$, balancing the risk of stopping the selection process too early with the risk of incorrectly including irrelevant variables. Our theoretical work, particularly Theorem 4 and the formulation for selecting $c$ in Equation (33), suggests that $c$ should ideally be determined by considering the probability of a single incorrect inclusion ($p_N(N)$), the number of non-relevant variables ($M_{NR}$), and a desired tolerance for overall false positives in $\widehat{\mathbb{S}}$ ($\epsilon_{S,FP}$). The experimental results presented here provide practical insights into this interplay and generally support our theoretical conclusions.

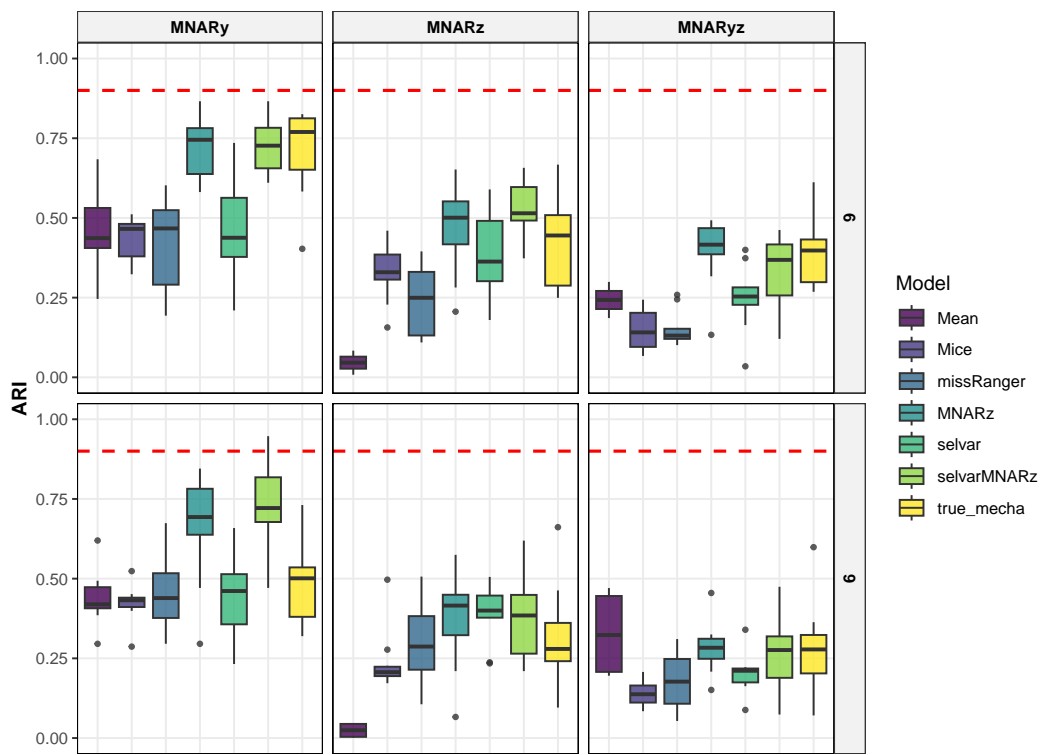

Figure 4: Boxplot of the ARI obtained over 20 replications of simulated data. The theoretical ARIs are represented by a red dashed line.

As illustrated in Figures 6 and 7, the two datasets (Dataset 1: $D = 7, s_0 = 3$; Dataset 2: $D = 14, s_0 = 2$) show different optimal ranges for $c$. This aligns with the theoretical expectation that $c$ is not a universal constant but rather interacts with dataset-specific factors, which are encapsulated by $p_N(N)$ and $M_{NR}$. For Dataset 1, the excellent performance with $c = 2$ (achieving 100% correct selection of relevant variables) points to a very low $p_N(N)$. This suggests that after the three true relevant variables are found, the remaining four non-$\mathbb{S}_0$ variables consistently produce $\text{BIC}_{\text{diff}} \leq 0$, making a small stopping value like $c = 2$ both effective and safe. The continued strong performance at $c = 7$ (90% correct) further indicates that $p_N(N)$ is small enough that even scanning all $M_{NR} = 4$ non-relevant variables rarely leads to a false inclusion. This situation is consistent with a theoretical scenario where $p_N(N) \ll 1/M_{NR}$, minimizing the risk of false positives regardless of $c$ within a reasonable range.

In contrast, Dataset 2, which has more non-$\mathbb{S}_0$ variables ($M_{NR} = 12$), demonstrates a clearer trade-off. Good performance is observed for $c = 2$ (80%) and $c = 3$ (90%), implying that $p_N(N)$ is still relatively small. However, the notable decline in performance when $c = 7$ (only 5% correct $\mathbb{S}_0$ selection) empirically validates a key theoretical concern: if $c$ is set too high and $p_N(N)$ is not sufficiently close to zero, the algorithm examines more non-$\mathbb{S}_0$ variables while awaiting $c$ consecutive negative $\text{BIC}_{\text{diff}}$ values. Each additional variable examined increases the cumulative chance of a Type I error (a non-$\mathbb{S}_0$ variable having $\text{BIC}_{\text{diff}} > 0$ by chance). If the likelihood of obtaining $c$ correct rejections in a row, $(1 - p_N(N))^c$, diminishes significantly as more variables are processed, false inclusions become more probable. The poor result for $c = 7$ in Dataset 2 suggests its $p_N(N)$ value makes it unlikely to achieve seven consecutive correct rejections before a false positive occurs among the $M_{NR} = 12$ candidates. This aligns with the theoretical relationship $c \approx \ln(M_{NR}/\epsilon_{S,FP})/p_N(N)$: for a given $\epsilon_{S,FP}$, a higher $p_N(N)$ or a larger $M_{NR}$ would generally favor a smaller $c$ to maintain that error tolerance, or a large fixed $c$ might lead to a poorer effective $\epsilon_{S,FP}$. The finding that $c = 3$ is effective for Dataset 2 is consistent with the heuristic used in prior work [13], suggesting it offers a practical compromise when $p_N(N)$ is small but non-negligible, and $M_{NR}$ is moderate. Furthermore, the relative stability of cluster number selection across different

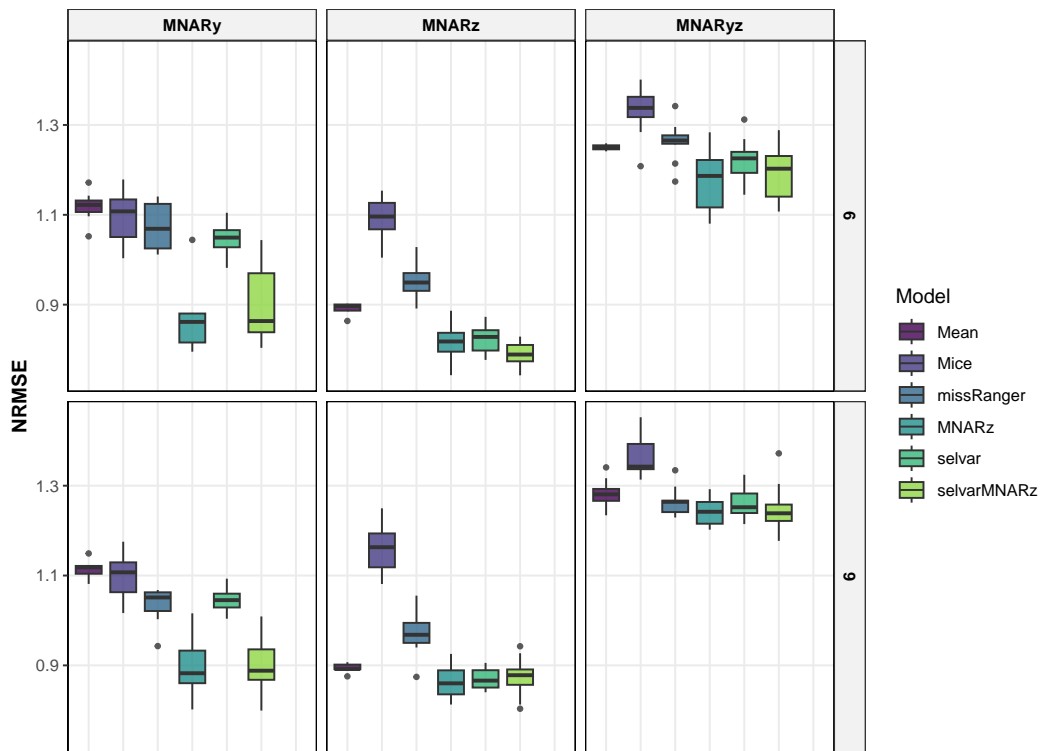

Figure 5: Boxplot of the NRMSE obtained over 20 replications of simulated data

values of $c$ suggests that $c$'s main influence is on the selection of variables for $\mathbb{S}$, as predicted by theory, with more indirect effects on the overall model choice, primarily if $\widehat{\mathbb{S}}$ is significantly misidentified.

### F.4 Computational Times and Scalability

**Complexity Analysis**

We count arithmetic operations up to absolute constants (big-Oh). Throughout this part we denote: $N \in \mathbb{N}$ (samples), $D \in \mathbb{N}$ (variables), $K \in \mathbb{N}$ (mixture components). Let $M_{\mathrm{EM}}$ be a uniform upper bound on the per-fit number of EM iterations until convergence (Assumption A2 below). In Stage A (ranking), the M-step contains a graphical-lasso solve per component with at most $M_{\mathrm{glasso}}$ outer iterations. Covariance inverses and log-determinants are computed by Cholesky factorizations.

**Assumptions.** We work under the following explicit conditions.

- **A1.** The ground-truth SRUW partition has $D_{\mathrm{eff}} := |\mathbb{S}_{\mathrm{true}}| + |\mathbb{U}_{\mathrm{true}}| \ll D$.
- **A2.** Each EM fit terminates in at most $M_{\mathrm{EM}}$ iterations. For population/regularized EM, this is justified by established convergence rates: either sublinear convergence $M_{\mathrm{EM}} = \mathcal{O}_p(1/\sqrt{N})$ [31] or geometric convergence $M_{\mathrm{EM}} = \mathcal{O}(\log(1/\epsilon))$ to achieve $\epsilon$-accuracy [76]. Our analysis uses the more conservative bounded iteration assumption for clarity.
- **A3.** With probability $1 - o(1)$, the Stage-A ranking lists all $D_{\mathrm{eff}}$ informative variables before any purely irrelevant variables (proved in Theorem 3 of the paper).
- **A4.** Let $C_{\mathrm{glasso}}(d)$ denote the arithmetic cost of one graphical-lasso solve of size $d \times d$. In general, $C_{\mathrm{glasso}}(d) = \Theta(M_{\mathrm{glasso}} d^3)$. Under connected-component decomposition with largest block size $s_{\max}$ (as in [73, 41]), $C_{\mathrm{glasso}}(d) = \Theta\left(M_{\mathrm{glasso}} \sum_c p_c^3\right) \leq \Theta\left(M_{\mathrm{glasso}} d\, s_{\max}^2\right)$.

**Per-iteration costs for a $d$-variate GMM.** One EM iteration for a $K$-component Gaussian mixture in dimension $d$ has:

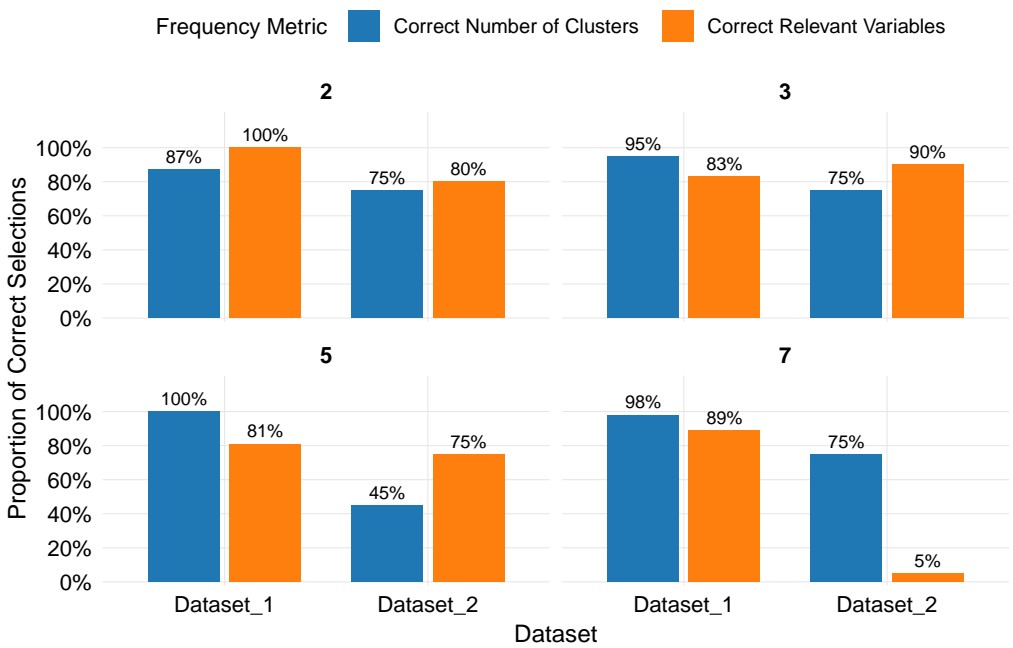

Figure 6: Proportions of choosing correct number clusters and relevant variables for simulated dataset in Section 5 under varying c. We run the experiment over 50 replications

- *E-step:* Responsibilities $t_{nk}$ require evaluating Gaussian log-densities for all $(n, k)$. With per-component $\boldsymbol{\Sigma}_k^{-1}$ and $\log \det \boldsymbol{\Sigma}_k$ fixed within the iteration, each evaluation uses a matrix-vector product and a quadratic form, costing $\Theta(d^2)$. Hence $\Theta(NKd^2)$.
- *M-step (means, weights):* Weighted sums are $\Theta(NKd)$.
- *M-step (covariances):* Weighted second moments yield $\Theta(NKd^2)$. The per-component matrix factorization/inversion is $\Theta(d^3)$, hence $\Theta(Kd^3)$.

Therefore one EM iteration in dimension $d$ costs

$$\Theta\big(NKd^2\big) + \Theta\big(Kd^3\big),$$

and one EM fit (up to convergence) costs

$$\mathcal{C}_{\text{EM}}(d) = \Theta\Big(M_{\text{EM}}\big(NKd^2 + Kd^3\big)\Big). \tag{40}$$

The classical SRUW selection starts from all $D$ variables and eliminates one at a time. At step $j$ ($j = D, D-1, \ldots, 2$), to remove one variable it evaluates $j$ candidates; each evaluation requires an EM fit in dimension $j - 1$. Using Equation (40), the total cost is

$$\sum_{j=2}^{D} j \cdot \mathcal{C}_{\text{EM}}(j-1) = \Theta\Big(M_{\text{EM}} \sum_{j=2}^{D} j\big(NK(j-1)^2 + K(j-1)^3\big)\Big).$$

Using the polynomial sums:

$$\sum_{j=1}^{D} j^3 = \frac{D^2(D+1)^2}{4} = \Theta(D^4)$$

and

$$\sum_{j=1}^{D} j^4 = \frac{D(D+1)(2D+1)(3D^2+3D-1)}{30} = \Theta(D^5)$$

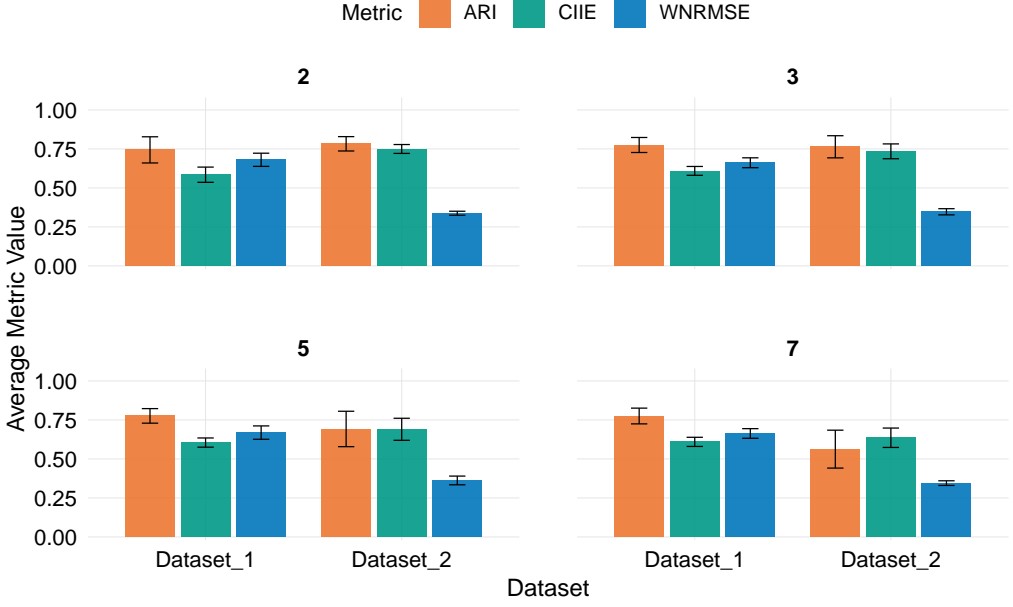

Figure 7: Clustering performance of simulated dataset in Section 5 under varying c. We run the experiment over 50 replications

, we obtain the tight bound

$$\mathcal{C}_{\text{stepwise}} = \Theta\big(M_{\text{EM}}\,(NKD^4 + KD^5)\big). \tag{41}$$

The $D^5$ term renders this approach impractical beyond moderate $D$.

**Stage A (Ranking).** For each $(\lambda, \rho)$ on a grid of size $M_{\text{grid}}$, we run a penalized EM. Per iteration: the E-step remains $\Theta(NKD^2)$; the M-step adds $K$ graphical-lasso solves. By Assumption A4,

$$\text{per iter cost} = \Theta\big(NKD^2\big) + \Theta\big(K\,C_{\text{glasso}}(D)\big).$$

Therefore the total Stage-A cost is

$$\mathcal{C}_{\text{rank}} = \Theta\Big(M_{\text{grid}}\,M_{\text{EM}}\,\big(NKD^2 + K\,C_{\text{glasso}}(D)\big)\Big). \tag{42}$$

Two explicit regimes follow immediately from A4:

$$\textbf{(Dense/general)} \quad C_{\text{glasso}}(D) = \Theta(M_{\text{glasso}}D^3)$$
$$\Rightarrow \mathcal{C}_{\text{rank}} = \Theta\Big(M_{\text{grid}}M_{\text{EM}}\big(NKD^2 + KM_{\text{glasso}}D^3\big)\Big). \tag{43}$$

$$\textbf{(Connected components of size } s_{\max}) \quad C_{\text{glasso}}(D) = \Theta\big(M_{\text{glasso}}Ds_{\max}^2\big)$$
$$\Rightarrow \mathcal{C}_{\text{rank}} = \Theta\Big(M_{\text{grid}}M_{\text{EM}}\big(NKD^2 + KM_{\text{glasso}}Ds_{\max}^2\big)\Big). \tag{44}$$

**Stage B (Role assignment).** A single forward/backward pass evaluates a constant number of SRUW-MNARz fits per newly considered variable. Under A3 the pass stops after $D_{\text{eff}}$ additions; hence the Stage-B complexity is

$$\mathcal{C}_{\text{role}} = \Theta\Big(M_{\text{EM}}\big(NKD_{\text{eff}}^2 + KD_{\text{eff}}^3\big)\Big). \tag{45}$$

Since $D_{\text{eff}} \ll D$ (A1), $\mathcal{C}_{\text{role}}$ is strictly lower order than $\mathcal{C}_{\text{rank}}$ and Stage A dominates. Combining Equation (43)-Equation (44) and Equation (45), the two-stage complexity is

$$\mathcal{C}_{\text{two-stage}} = \mathcal{C}_{\text{rank}} + \mathcal{C}_{\text{role}} = \Theta\Big(M_{\text{grid}}M_{\text{EM}}\big(NKD^2 + K\,C_{\text{glasso}}(D)\big)\Big) + o\big(\mathcal{C}_{\text{rank}}\big). \tag{46}$$

From Equation (41) and Equation (46), the speedup factor satisfies

$$\frac{\mathcal{C}_{\text{stepwise}}}{\mathcal{C}_{\text{two-stage}}} = \Omega\left(\frac{NKD^4 + KD^5}{M_{\text{grid}}\left(NKD^2 + K\,C_{\text{glasso}}(D)\right)}\right).$$

Two explicit lower bounds follow.

- If $N \geq M_{\text{glasso}}D$ (E-step dominates Stage A): using Equation (43),

$$\frac{\mathcal{C}_{\text{stepwise}}}{\mathcal{C}_{\text{two-stage}}} = \Omega\left(\frac{D^2}{M_{\text{grid}}}\right).$$

- If $N < M_{\text{glasso}}D$ (glasso dominates Stage A): using Equation (43),

$$\frac{\mathcal{C}_{\text{stepwise}}}{\mathcal{C}_{\text{two-stage}}} = \Omega\left(\frac{D^2}{M_{\text{grid}}\,M_{\text{glasso}}}\right).$$

Under connected-component sparsity with block size $s_{\max}$ Equation (44) the second case strengthens to

$$\frac{\mathcal{C}_{\text{stepwise}}}{\mathcal{C}_{\text{two-stage}}} = \Omega\left(\frac{D^2}{M_{\text{grid}}} \cdot \min\left\{1, \frac{N}{M_{\text{glasso}}\,s_{\max}^2}\right\}\right).$$

In all regimes the speedup is $\Omega\left(D^2/M_{\text{grid}}\right)$, and strictly larger when sparsity (small $s_{\max}$) is present.

**Empirical Validation**

We report wall-clock times (seconds) for representative scenarios, confirming the predicted polynomial speedups:

| Scenario | $N$ | $D$ | $K$ | SelvarMNARz (s) | Clustvarsel (s) | Speedup |
|---|---|---|---|---|---|---|
| Varying $D$ | 750 | 15 | 4 | 12.2 | 190 | $\sim 15\times$ |
| Varying $D$ | 750 | 21 | 4 | 14.4 | 640 | $\sim 44\times$ |
| Varying $D$ | 750 | 27 | 4 | 15.5 | 2054 | $\sim 132\times$ |
| Varying $N$ | 1000 | 20 | 4 | 19.0 | 838 | $\sim 44\times$ |
| Varying $K$ | 750 | 20 | 12 | 9.77 | 1242 | $\sim 127\times$ |

These measurements are consistent with the theory: the two-stage method scales like $D^2$ in the dense case (and better under sparsity), whereas backward stepwise scales as $D^4$-$D^5$.

**Dependency of Computational Times under Degree of Missingness**

We analyze the computational dependency of our framework on the proportion of missing data, revealing an advantageous property: runtime *decreases* with increasing missing rates due to the efficient handling of incomplete data patterns in the EM algorithm.

The key insight stems from the E-step computational complexity in Stage B, which operates directly on the observed data patterns. Let $X_n$ denote the number of observed entries in observation $\boldsymbol{y}_n$. The E-step cost for Gaussian mixture models scales as:

$$\mathcal{C}_{\text{E-step}} = \Theta\left(\sum_{n=1}^{N}\sum_{k=1}^{K} X_n^2\right) = \Theta\left(KN\mathbb{E}[X^2]\right)$$

For different missingness mechanisms:

- **MCAR at rate** $r$: $X \sim \text{Binom}(D, 1-r)$, yielding

$$\mathbb{E}[X^2] = \text{Var}(X) + \mathbb{E}[X]^2 = D(1-r)r + D^2(1-r)^2$$

This produces a quadratic decrease in E-step work as $r \uparrow 1$.

- **MAR/MNARz:** The same complexity bound applies, with $X_n$ representing the random count of observed coordinates per observation. Each per-record Gaussian density evaluation scales quadratically with observed entries, maintaining the $\Theta(KN\mathbb{E}[X^2])$ complexity.

Stage A employs efficient single imputation and remains largely insensitive to missing rates, while Stage B inherits the beneficial $\mathbb{E}[X^2]$ scaling. Consequently, total runtime decreases monotonically with increasing missingness rates.

We empirically verified this property by fixing $(N, D, K) = (1000, 14, 4)$ and systematically increasing the missing data rate under two mixed missingness scenarios. The results, summarized in Table 4 and Figure 8, confirm the theoretical predictions.

Table 4: Runtime (seconds) with increasing missing rate under mixed mechanisms.

| Missing Mechanism | 20% | 30% | 50% | 80% |
|---|---|---|---|---|
| Mixed (MAR+MNAR) | 54.1 | 46.2 | 32.9 | 23.9 |
| Mixed (MCAR+MAR+MNAR) | 34.9 | 23.8 | 25.5 | 22.8 |

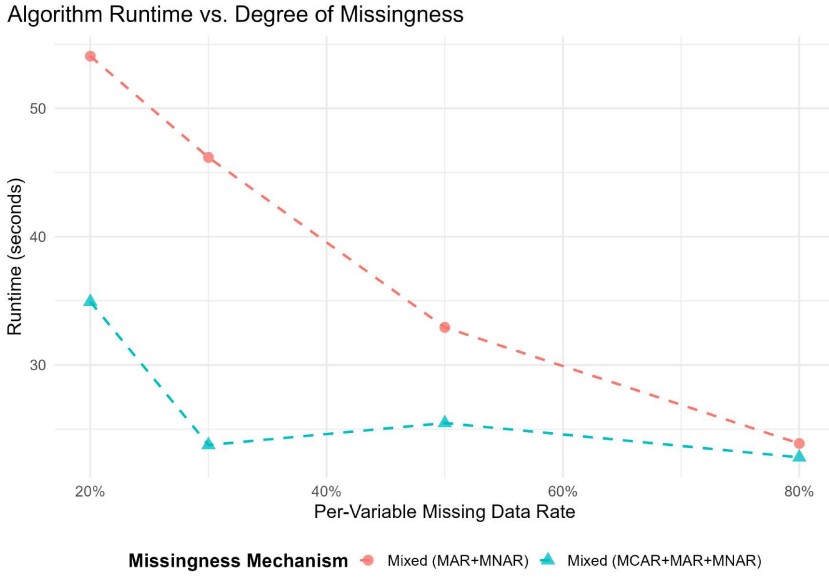

Figure 8: Runtime as the proportion of missing data increases.

Three key observations emerge:

1. Runtime consistently decreases with higher missing rates, as predicted by the $\mathbb{E}[X^2]$ scaling. The MNARz block processes smaller observed patterns during E-step calculations, reducing computational burden for conditional expectations and complete-data sufficient statistics.

2. The slight variation at 30%-50% missingness in the three-mechanism scenario likely stems from random allocation of missing positions. When missing data concentrates in clustering variables ($\mathbb{S}$), the number of EM iterations for convergence may vary, but the overall inverse relationship persists.

3. This property provides significant practical benefits. Users can expect faster processing on datasets with higher missing rates-a valuable characteristic for real-world applications where extensive missing data is common.

The combination of mathematical analysis and empirical results demonstrates that our framework not only handles high missing rates effectively but also becomes more computationally efficient as missing data increases, making it particularly suitable for challenging real-world datasets with substantial missingness.

### F.5 More on Simulated Dataset

**Performance under Model Misspecification.** Previous simulations focused on "pure" MAR and MNAR scenarios to clarify the benefits of our model when combined with the MNARz mechanism. However, real-world missingness mechanisms are often mixed. To assess robustness, we conducted additional simulations with a dataset of $D = 14$ variables, where the true clustering variable set is $\mathbb{S}^{\star} = \{1, 2\}$ (remaining variables play roles of $\mathbb{R}, \mathbb{U}, \mathbb{W}$ according to scenario 8 in section 5. We generated missing data using a mixture of mechanisms:

- **MCAR:** Some variables have missing values completely at random
- **MAR:** Missingness depends only on observed components $y_n^o$
- **MNARy:** Missingness depends directly on unobserved values $\boldsymbol{y}^m$ (actively violating MNARz assumption)

The MNARy mechanism (see [62] for precise formulation) specifically tests our model's robustness to misspecification, as it depends on the unobserved values rather than cluster assignments. We compare our method against several baselines, including a novel two-step approach designed to isolate the benefits of joint modeling:

1. **MNARz + SelvarMix**: Estimate GMM-MNARz model and impute missing data, then run SelvarMix [13] on the imputed dataset
2. **Multiple Imputation variants**: Using both missRanger and gcimputeR (cite here) with Mclust
3. **VarSelLCM**: A competing joint modeling approach

Table 5: Experiment with mixed MAR+MNARy (True set $\{1, 2\}$)

| Method | ARI | NRMSE | Relevant Variables Selected |
|---|---|---|---|
| SelvarMNARz (ours) | 0.808 | 0.176 | 1, 2 *(correct)* |
| VarSelLCM | 0.779 | – | 1-11 *(extra variables)* |
| missRanger + Mclust | 0.799 | 0.190 | – |
| gcimputeR + Mclust | 0.774 | 0.413 | – |
| MNARz + SelvarMix | 0.328 | 0.070 | 1 *(missing variable)* |

Table 6: Experiment with mixed MCAR+MAR+MNARy (True set $\{1, 2\}$)

| Method | ARI | NRMSE | Relevant Variables Selected |
|---|---|---|---|
| SelvarMNARz (ours) | 0.808 | 0.181 | 1, 2 *(correct)* |
| VarSelLCM | 0.772 | – | 1-11 *(extra variables)* |
| missRanger + Mclust | 0.783 | 0.198 | – |
| gcimputeR + Mclust | 0.755 | 0.401 | – |
| MNARz + SelvarMix | 0.328 | 0.087 | 1 *(missing variable)* |

The poor performance of the `MNARz + SelvarMix` baseline prompted a deeper investigation. To test whether properly handling imputation uncertainty could rescue the decoupled strategy, we also ran Multiple Imputation (MI) variants (missRanger-MI and MNARz-MI with random initializations) and pooled the results. Using the default `Rmixmod` backend for `SelvarMix` yielded the results below.

Table 7: Performance on Mixed (MAR+MNARy) Data (Rmixmod backend)

| Method | ARI | NRMSE | Relevant Variables |
|---|---|---|---|
| SelvarMNARz (Ours) | 0.778 | 0.162 | 1, 2 |
| Decoupled (SI MNARz) | 0.328 | 0.263 | 1 |
| Decoupled (MI missRanger) | 0.336 | 0.257 | 1 |
| Decoupled (MI MNARz) | 0.328 | 0.331 | 1 |

Table 8: Performance on Mixed (MCAR+MAR+MNARy) Data (Rmixmod backend)

| Method | ARI | NRMSE | Relevant Variables |
|---|---|---|---|
| SelvarMNARz (Ours) | 0.773 | 0.164 | 1, 2 |
| Decoupled (SI MNARz) | 0.328 | 0.387 | 1 |
| Decoupled (MI MNARz) | 0.336 | 0.322 | 1 |
| Decoupled (MI missRanger) | 0.328 | 0.248 | 1 |

From Table 7 and Table 8, we deduce two key takeaways: (i) Our joint model retains high ARI and correct selection under mixed mechanisms, including MNARy misspecification; (ii) MI does not repair the decoupled pipeline in this setting, with results mirroring single-imputation. This suggests two contributing factors:

1. **Uncertainty Propagation.** Decoupled pipelines treat completed data as observed, discarding posterior uncertainty in $y^m$. Our EM-based framework propagates this uncertainty through all parameter and role updates.

2. **Downstream Stability.** The variable-selection backend (SelvarMix with its Rmixmod engine) shows instability (e.g., sensitivity to local maxima); MI then averages multiple weak fits. The joint estimation procedure is empirically more stable in this context.

Finally, under these mixed mechanisms, the class-level MNARz parameters in our model often converge to similar values across clusters for variables whose missingness is effectively MAR or MCAR, while remaining discriminative where missingness is truly class-linked. This helps explain the model's robustness to misspecification.

**Effect of Initialization on EM Stability and Accuracy.** We compared a hierarchical clustering (HC) based initialization (Ward's linkage on Euclidean distances with cluster centers extracted from the dendrogram cut at $K$) against multiple random initializations (MIs). HC places initial centers in high–density regions, yielding more stable responsibilities at the first E–step and fewer poor local optima than purely random starts. As shown in Table 9, HC attains higher ARI on 7 of 8 scenarios, and trails slightly once, confirming its overall robustness and improved convergence behavior for the EM algorithm.

Table 9: Comparison of EM initialization methods under 8 data scenarios in Section 5 (higher ARI is better). MIs: Multiple random initializations.

| Scenario | MIs (ARI) | HC (ARI) |
|---|---|---|
| 1 | 0.283 | **0.317** |
| 2 | 0.479 | **0.533** |
| 3 | **0.551** | 0.536 |
| 4 | 0.441 | **0.654** |
| 5 | 0.760 | **0.771** |
| 6 | 0.711 | **0.778** |
| 7 | 0.716 | **0.780** |
| 8 | 0.785 | **0.786** |

HC initialization offers a more sensible and stable warm start than multiple random restarts, typically improving both EM convergence and final clustering accuracy (ARI), with negligible overhead relative to the overall EM cost.

### F.6 More on Transcriptome Dataset

**Background and Prior Analyses**

**Dataset.** We analyze the *Arabidopsis thaliana* transcriptome comprising 1267 genes measured across 27 experimental conditions aggregated from seven projects P1-P7. Genes were preselected for differential expression at least once in the hypocotyl growth switch time course (Project 6), making P6 biologically central. Following [37, 39], we retain all 1267 genes: 1149 are fully observed; 118

contain missing entries (107 with one, 10 with two, 1 with three). Overall, $9.3\%$ of genes have any missingness and the global missing rate is $0.38\%$.

**Prior findings.** SelvarClust [37] (complete cases) found that including irrelevant variables degrades homogeneity; variable selection produced more coherent clusters and recovered known co-expression groups (e.g., a cluster of 15 genes co-clustered with 4 well-studied markers). *SelvarClustMV* [39] (MAR setting) expanded the gene set by reprocessing previously excluded genes and treating missing data within an EM framework, concluding that P6 (hypocotyl switch) and P7 (isoxaben treatment) are clustering-relevant, together with P1-P4, whereas P5 (nematode infection) is not primarily grouping. Both studies support P2 (iron signaling) as a core axis for defining co-expression groups. Note that the 2012 analysis assumes MAR for the missing entries. In contrast, our framework explicitly models class-dependent missingness (MNARz), while retaining MAR/MCAR as limiting cases through parameterization.

### Additional Interpretation of Our Results on the Transcriptome

**Global outcome.** Fitting `SelvarMNARz` for $K \in 2, \ldots, 20$ with $c = 5$, spectral distance weights $\boldsymbol{P}_k$, and $\mathrm{p}_k LC$ structure, the selected model yields 18 clusters and a global role assignment with P1-P4 in $\mathbb{S}$ and P5-P7 in $\mathbb{U}$. This agrees with prior work on P5 (not clustering-relevant) but differs by reclassifying P6-P7 from $\mathbb{S}$ (in [37, 39]) to $\mathbb{U}$.

**Cluster-level diagnostics clarify the difference.** Let $R_k^2$ denote the coefficient of determination from regressing the $\mathbb{U}$-block (here, P5-P7) on the $\mathbb{S}$-block (P1-P4) within cluster $k$. Table 2 shows pronounced heterogeneity:

- Several clusters (e.g., 6, 7, 8, 10, 12, 18) have $R_k^2 > 0.60$, indicating that P5-P7 are largely explained by P1-P4 *within those clusters*. For these groups, assigning P5-P7 to $\mathbb{U}$ is appropriate.

- A large aggregate (Cluster 1) exhibits low $R^2$ alongside flat $\mathbb{S}$-profiles and detectable $\mathbb{U}$-activity, suggesting local decoupling of P5-P7 from P1-P4. This mirrors the biological intuition behind earlier inclusion of P6-P7 in $\mathbb{S}$.

Hence, both the earlier global $\mathbb{S}$-assignment (P6-P7 in $\mathbb{S}$) and our global $\mathbb{U}$-assignment (P6-P7 in $\mathbb{U}$) are locally valid—but *on different clusters*. The discrepancy is explained by heterogeneity: the relationship between early axes (P1-P4) and late/stress projects (P5-P7) is cluster-specific.

**Why does our global BIC prefer P5-P7 in $\mathbb{U}$?** Two factors are at play:

1. *Global model selection.* BIC aggregates fit-complexity tradeoffs across all clusters. Because many clusters exhibit high $R_k^2$ (P5-P7 explained by P1-P4), the global criterion prefers a parsimonious $\mathbb{S}$ (P1-P4) and assigns P5-P7 to $\mathbb{U}$.

2. *MNARz identifiability and shrinkage.* Under MNARz, class-specific missingness parameters $\rho_{kd}$ absorb class-linked absence patterns. When a project effectively behaves as MAR/MCAR for many clusters, the fitted $\rho_{kd}$ across $k$ becomes nearly homogeneous and the conditional dependence of P5-P7 on P1-P4 becomes tighter, further favoring $\mathbb{U}$ globally.

**Biological reading consistent with both views.** P2 (iron signaling) is reaffirmed as a core driver. P5-P7 behave as late/stress outputs strongly coupled to P1-P4 in many clusters (high $R_k^2$), but retain independent variation in a sizable subset (e.g., Cluster 1). This reconciles the prior decision to place P6-P7 in $\mathbb{S}$ (emphasizing hypocotyl-centric signals) with our global $\mathbb{U}$ assignment (emphasizing aggregate parsimony across all clusters).

**Practical implication.** Our diagnostics suggest a natural extension: *cluster-adaptive* role assignment or a hierarchical prior tying per-cluster roles, which would allow P6-P7 to enter $\mathbb{S}$ *only* where local evidence (low residual error) warrants it while retaining parsimony elsewhere. This aligns with the heterogeneity revealed by $R_k^2$ and preserves the strengths of both global viewpoints.

Our unified $\mathbb{S}/\mathbb{U}$ decision is globally efficient and statistically supported by MNARz-aware likelihood, while cluster-level diagnostics uncover biologically meaningful deviations. Together, they provide a coherent picture: P1-P4 are the principal axes; P5-P7 are predominantly redundant but locally informative in specific clusters, explaining the divergence from prior MAR-based analyses.

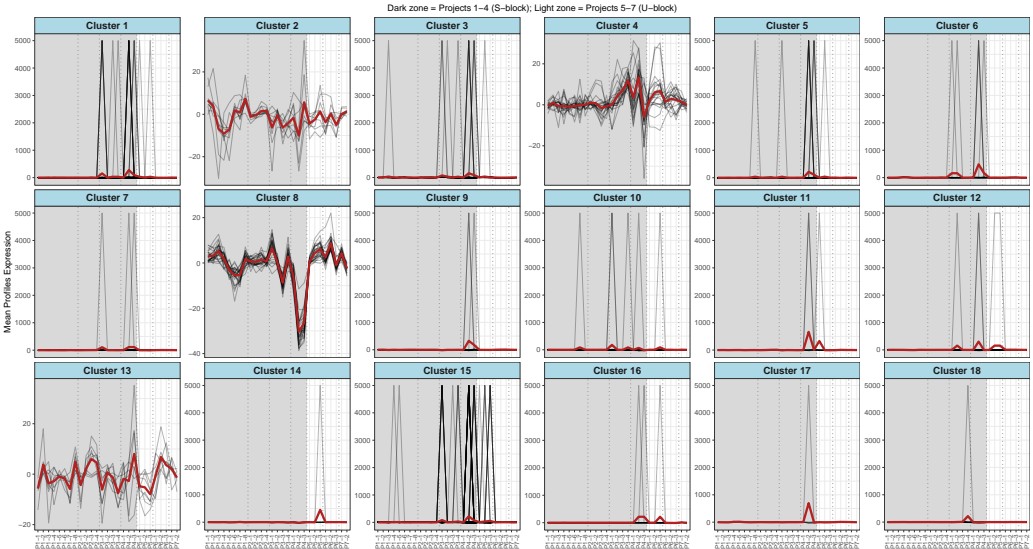

Figure 9: Mean expression profiles for each of the 18 clusters, displayed in separate panels. Light region indicates irrelevant P.

# G    Additional Details on Related Work in the Literature

## G.1    Model-based Clustering

Model-based clustering conceptualizes clustering as a statistical inference problem, where the data are assumed to be generated from a finite mixture of probability distributions, each corresponding to a latent cluster. Unlike heuristic-based methods, this paradigm enables principled inference, allowing for parameter estimation via likelihood-based techniques and objective model selection to determine the number of clusters.

A prototypical instance is the GMM, wherein each cluster is characterized by a multivariate Gaussian distribution. Parameter estimation is typically conducted using the EM algorithm [16], yielding soft assignments in which each observation is associated with posterior probabilities across clusters. GMMs offer more flexibility than simpler methods like $k$-means, as they can model clusters with varying shapes and overlapping regions via covariance structures.

Bayesian mixture models (MMs) extend this framework by treating both the parameters and the number of components as random variables with prior distributions. This fully Bayesian approach enables comprehensive uncertainty quantification over both model parameters and cluster allocations. For a fixed number of components, inference is often performed using Markov Chain Monte Carlo (MCMC) or variational methods. However, Bayesian MMs face the label switching problem due to the symmetry of the likelihood with respect to component labels, which renders the posterior non-identifiable without additional constraints. Strategies such as relabeling algorithms and identifiability constraints have been proposed to address this issue [65]. A notable advantage of the Bayesian approach is the incorporation of priors on model complexity, e.g., via Reversible Jump MCMC or birth-death processes, to infer the number of clusters. Despite their flexibility and robustness, fully Bayesian clustering methods can be computationally demanding, particularly when MCMC chains converge slowly or require extensive post-processing to resolve label ambiguity [29].

## G.2    Variable Selection for Model-based Clustering

Historically, variable selection was performed using best-subset or stepwise selection approaches, typically guided by information criteria such as AIC [1] or BIC [58]. While effective for moderate-dimensional settings, these methods become computationally prohibitive as the number of features $D$ increases.

The advent of penalized likelihood methods improved scalability and enabled sparse modeling. The LASSO [70], a seminal technique using an $L_1$ penalty, enables both variable selection and coefficient shrinkage. Predecessors include the nonnegative garrote [9] and ridge regression [28], the latter using an $L_2$ penalty, which does not induce sparsity. Efficient algorithms such as LARS [18] and coordinate descent have facilitated high-dimensional applications of these methods. However, the time complexity of LARS limits its stability when $D$ is very large.

Law et al. [32] proposed a wrapper approach that jointly performs clustering and variable selection through greedy subset evaluation, introducing the notion of feature saliency to assess variable importance in cluster discrimination. Andrews and McNicholas [3] developed the VSCC algorithm, combining filter and wrapper strategies: variables are ranked based on within-cluster variance from an initial clustering, then incrementally added subject to correlation thresholds to enhance separability. Their R package implementation, vscc, offers efficient noise filtering prior to model-based refinement.

Raftery and Dean [54] introduced a model selection framework for variable selection within GMMs, categorizing variables as clustering, candidate, or noise. Each candidate is evaluated using BIC to compare models where the variable does or does not influence clustering. Their framework accommodates statistical dependence between irrelevant and clustering variables via regression, avoiding overly simplistic independence assumptions. Scrucca and Raftery [59] enhanced this method, introducing computational heuristics in the clustvarsel R package to streamline EM evaluations.

Expanding this idea, Maugis et al. [38] proposed a three-role framework, relevant, irrelevant, and redundant variables, with redundancies modeled via linear regression on relevant variables. This design improves accuracy in scenarios with correlated predictors. Nevertheless, the stepwise search remains computationally demanding as dimensionality increases.

To address scalability, Celeux et al. [13] proposed a two-stage approach: first, variables are ranked by penalized likelihood using the method of Zhou et al. [75], which penalizes component means and precisions. Then, a linear scan through this ranking assigns roles. This heuristic drastically reduces computation, preserves model interpretability, and achieves strong empirical performance, though theoretical guarantees for recovering the correct $\mathbb{SRUW}$ partition remain open.

Extensions to categorical data include adaptations of the Raftery-Dean method to Latent Class Analysis (LCA) by Dean and Raftery [15], with further refinement by Fop et al. [20], resulting in improved class separation in clinical datasets. Bontemps and Toussile [8] considered mixtures of multinomial distributions, using slope-heuristic-adjusted penalization to improve model selection under small-sample conditions.

### G.3 Handling Missing Data

In practical datasets, missing values often occur and are categorized as MCAR, MAR, or MNAR. Under the MCAR assumption, missingness is unrelated to any data values; MAR allows dependence on observed data; MNAR, the most complex case, involves dependence on unobserved or latent variables.

Under MAR, Multiple Imputation (MI) [57] has emerged as a robust strategy to address uncertainty. Instead of single imputation, MI generates multiple completed datasets via draws from predictive distributions, followed by Rubin's combination rules to aggregate inference. MICE [71], a flexible implementation, fits univariate models conditionally and iteratively imputes missing values, supporting mixed data types and nonlinear relationships.

High-dimensional settings pose new challenges, where full joint modeling becomes unstable. To address this, Zhao and Long [74] incorporated Lasso and Bayesian Lasso into the MICE framework to enhance prediction accuracy through regularized imputation, effectively reducing overfitting and variable selection bias in high-$D$ regimes.

Machine learning techniques have also advanced imputation. MissForest [63] uses random forests to iteratively impute variables based on others, capturing nonlinearities in a nonparametric manner. Although not explicitly designed for clustering, the ensemble mechanism implicitly reflects local data structures akin to clustering. Deep learning methods, such as MIWAE [36], adopt autoencoder architectures to generate multiple imputations from learned latent spaces, enabling further extensions

like Fed-MIWAE [4] for privacy-sensitive contexts. These models assume a latent manifold structure and are best suited for large sample sizes, though interpretability remains a challenge.

Addressing MNAR data in clustering models remains difficult due to identifiability issues and the need for strong assumptions or auxiliary information. Two general strategies exist: (1) Selection models, specifying a joint model for data and missingness mechanisms, and (2) Pattern mixture models, defining distributions conditioned on missingness patterns and modeling their influence on cluster membership. In [62], Sportisse et al. investigated identifiability conditions in MNAR selection models and proposed methods to augment clustering models with missingness-informed constraints to improve identifiability.

