# OpenReview forum: "A Unified Framework for Variable Selection in Model-Based Clustering with Missing Not at Random"
_NeurIPS.cc/2025/Conference — NeurIPS 2025 poster_

### Official Review · Reviewer_Bx5V · 2025-06-23

**Clarity:** 1
**Significance:** 3
**Originality:** 3
**Rating:** 4
**Confidence:** 3

**Summary:**

This paper introduces a model for variable selection in finite Gaussian mixtures. The basic idea is to partition the data vector into variables that are relevant for the clustering, variables that are linear combinations of the variables relevant to the clustering, and then irrelevant variables. To accomodate missing not at random variables, the authors append a missingness mechanism to the finite mixture model, which they call MNARz, that allows for missingness probabilities to vary within a latent class. They then use a penalized likelihood function to perform simultaneous parameter estimation and variable selection.

The authors apply their model for clustering, variable selection, and parameter inference in a simulation study, where the method demonstrates compentitive imputation accuracy and clustering performance. In their real data application, they use their model to analyze transcriptome data, where they determine scientifically relevant subgroups in the data based on gene expressions.

**Questions:**

1) Estimation and selection: I am quite confused about how variables are simultaneously selected and the finite GMM is estimated. The paper would benefit from clearer and higher-level algorithms describing estimation and selection. From the Finite GMM representation of the SRUW model, does a user need first to declare which variables they believe are relevant/irrelevant to the clustering? When/Where is the penalty applied? There is some mention of a ranking step followed by a parameter estimation step but this is not clear in the paper or manuscript. Where/when do the penalties come into play?

2) MNARz mechanism: I have some serious doubts that the proposed MNARz model provides a missingness mechnism that is nonignorable. This needs to be clarified. For example the authors state that the missignness mechanism within any class factors as:

p(c_{n} \mid z_{n_k}, \psi_{k}) = \prod_{d=1}^{D}\rho(\psi_{k})^{c_{nd}}(1-\rho(\psi_{k})^{1-c_{n})

For the missingness to be MNAR, missingness has to depend on unobserved missing values. In this case, this would require \psi_{k} to depend on y_{d}, but this is not clear from how the model is presented. Also, the way this reads is that \psi_{k} is the same for each variable y_d in latent class k. Is this correct? This seems highly improbably and is not a MNAR missigness mechanism. This needs to be clarified or modified if the authors claim their methods are useful for MNAR data.


3) The distinction between the different \pi_{k}s in (3) is not clear.

**Ethical Concerns:**

["NO or VERY MINOR ethics concerns only"]

**Final Justification:**

Once the authors revise their work according to their detailed responses, the paper will be satisfactory. For this reason, I recommend boarderline accept.

**Limitations:**

1) In general, the paper is heavy with notation that is difficult to follow
2) Following my above questions:
- I think there should be some outline for how model estimation and variable selection is conducted. The E-M algorithm presented in the main text only focuses on estimating the parameters in the joint model for the missingness and study variables for a fixed set of pertinent/irrelevant variables, unless I am missing something
-I think the authors should revisit the MNARz missingness mechanism. In general, a variable is nonignorably missing if the probability of missingness depends on the underlying missing values. I don't see how this is encoded in the MNARz variables
-Related, I also searched in the supplement for how the missingness mask is realized but couldn't find anything in the text or supplement. Is the data truly MNAR in the author's simulated experiments?

**Quality:**

2

**Strengths And Weaknesses:**

I think the strength of this paper is in the ability of the SRUW model to be expressed as a straightforward finite Gaussian mixture, with component-wise mean vector and covariance matrix expressed in terms of the relevancy partition that the authors originally present. As the authors mention, this enables theoretical analyses and relatively simple E-M algorithms for estimation. In addition, the idea that clustering is driven by a few of the study variables, and seeking to select these variables to provide interpretable insights, is quite attractive. Extending these ideas to MNAR study variables is quite challenging.

---

> ### Author Rebuttal · Authors · 2025-07-31
>
> We sincerely thank you for your thoughtful and detailed feedback. The questions raised are extremely helpful and have highlighted several areas where our presentation was not as clear as it should have been. We agree that the paper is dense with notation and some parts of the writing are unclear, and we are grateful for the opportunity to clarify our methodology, particularly the variable selection procedure and the nature of the MNARz mechanism. We will address each of your points below.
> ### **1. On Clarity of Estimation and Variable Selection**
> We deeply apologize for the lack of clarity here. Your confusion is our responsibility, and we will significantly revise Section 3 and add a high-level algorithm to make this process transparent. To answer the questions directly:
> + No, the user does not pre-declare variable roles. The entire purpose of our framework is to discover these roles (relevant, redundant, irrelevant) from the data automatically. However, if the user suspects that the data can include both MNAR and MAR mechanisms, they have to provide the variable roles beforehand to use the mixed mechanism modelling, which is currently our limitation on this mixed mechanisms handling part.
> + The penalties are applied in the first stage (Variable Ranking). This is the key to making the problem computationally tractable since the result will be a list of ranked variables and we just have to do a **linear pass**
> + The process is a two-stage procedure, as pioneered by Celeux et al. [1], which we have extended to the missing data setting.
> Here is the high-level description of the algorithm, which we will add to the revised manuscript:
>
> **Stage 1: Variable Ranking via Penalized GMM**
> + **Target:** To obtain an ordered list of all $D$ variables, from most to least relevant for clustering.
> + **Method:** We fit a penalized Gaussian Mixture Model (GMM) to the complete data $X$ where missing entries in $X$ has been imputed naively. The reason to perform this step is that we want to have a good ranking scores and it is critical to note that this imputed dataset is only a temporary scaffold used to enable the ranking procedure; it is discarded immediately after.
> The ranking itself is then performed by fitting a penalized log-likelihood given by Equation (3) in the manuscript to the temporary complete data. The $\lambda$ penalty shrinks the cluster means $\mu_k$ towards a common center. Variables that are important for clustering will resist this shrinkage. The $\rho$ penalty, weighted by the adaptive matrices $P_k$, shrinks the off-diagonal elements of the precision matrices $\Psi_k$ to zero, effectively performing graphical model selection within clusters.
>
> + **Procedure:** We fit this model over a grid of $(\lambda, \rho)$ values. For each variable d, we compute a "ranking score" $\mathcal{O}\_{K}(d)$. This score quantifies how essential the variable is for defining the cluster structure across the range of penalties. In essence, a variable that is important for clustering will exhibit significant separation in its cluster-specific mean estimates $\hat{\mu}\_{kd}$ even when the regularization penalty is strong. The score $\mathcal{O}\_{K}(d)$ is a summary statistic that captures this behavior.
> + **Output:** A single ranked list of variables $(d_1, d_2, ..., d_D)$.
>
> **Stage 2: Role Assignment using the SRUW Model and BIC**
> + **Target:** To partition the ranked list into the sets $\mathbb{S}$ (relevant), $\mathbb{U}$ (redundant), and $\mathbb{W}$ (irrelevant).
> + **Method:** We perform a single, efficient linear pass through the ranked list of variables.
> We start with a small model and iteratively add the next variable from the list.
> At each step, we fit the full (unpenalized) SRUW-MNARz model and use the BIC criterion (Equation (2)) to decide the role of the newly added variable (i.e., whether it belongs to  $\mathbb{S}$, $\mathbb{U}$, and $\mathbb{W}$).
> This avoids the computationally prohibitive $\mathcal{O}(D^5)$ complexity of the original stepwise SRUW method by replacing its combinatorial search with a single pass over a statistically-informed ordering. Crucially, this two-stage procedure is precisely what enables the "relatively simple E-M algorithms" and theoretical construction that you correctly identified as a key strength of our work.
>
> ### **2. On the MNARz Mechanism and its Non-Ignorable Nature**
> You are correct in your understanding of the classical definition of MNAR. However, the MNARz mechanism is a specific, identifiable subclass of MNAR models that has been recently established in the statistical literature.
>
> #### **2.1. Why MNARz is Truly MNAR (Non-Ignorable):**
> In a mixture model context, the latent variable $z_n$ (the cluster assignment) is itself part of the unobserved data. The missingness probability $P(c_{nd}=1 | y_n, z_n)$ depends on $z_n$. Since $z_n$ is unobserved and must be integrated out, the marginal probability of missingness $P(c_{nd}=1 | y_n^{obs})$ will depend on the posterior distribution $P(z_n | y_n^{obs})$, which in turn depends on the observed data $y_n^{obs}$. Crucially, because the parameters of the data distribution $(\theta_k)$ and the missingness distribution $(\psi_k)$ are linked through the same latent variable $z_n$, they cannot be estimated separately. **This linkage** is what makes the mechanism **non-ignorable**. If one were to ignore the missingness mechanism, the estimation of the mixture parameters $\theta_k$ and the cluster assignments would be biased. This is the formal definition of a non-ignorable mechanism.
> This model class was formally introduced and its properties studied by Sportisse et al. (2024) (see [1]). It represents a principled and tractable way to move beyond MAR when the cause of missingness is related to the underlying latent structure (e.g., in transcriptomics, genes in a specific functional cluster might fail to express and thus be "missing" due to that cluster's biological state).
>
> #### **2.2. Why $(\psi_k)$ is Constant Across Variables Within a Class**
> You are correct to point out that in our current formulation, the missingness probability $\rho(\psi_k)$ is a single scalar for all variables $d$ within a given class $k$. This is a **deliberate** modeling choice made for two reasons:
>
> *   **Parsimony and Estimation Stability:** In a high-dimensional setting ($D \gg N$), introducing a separate missingness parameter $\rho_{kd}$ for each variable in each cluster would add $KD$ parameters. This would make the missingness mechanism highly over-parameterized and prone to severe overfitting, leading to unstable estimation. Our approach, by contrast, adds only $K$ parameters. This deliberate choice for parsimony ensures the model remains statistically identifiable and that the EM algorithm converges to a stable solution, preventing the core clustering structure from being obscured by noise.
>
> *   **Enhanced Interpretability:** This choice leads to a highly interpretable result. A high value of $\rho_k$ for a specific cluster $k$ provides a simple, powerful insight: "Cluster $k$ represents a subgroup (e.g., a biological state or patient type) that is systematically harder to measure or has lower quality data across the board." This single parameter per cluster is far more interpretable than a dense matrix of $KD$ individual probabilities. Ultimately, this modeling choice for parsimony is how we make the problem of MNAR modeling stable and tractable, directly addressing the challenge you highlighted while preserving the interpretability you found attractive.
>
> Moreover, in our simulations, the missingness mask is generated precisely according to this class-dependent MNARz mechanism, ensuring the simulated data truly reflects this non-ignorable pattern.
>
> ### **3. On the Distinction Between $\pi_k$ in Equation (3)**
>
> We apologize for this confusing notation, and this is a clear notational flaw on our part when creating the manuscript. They are actually $\pi_k$ and $P_k$. Here is the clear explanation:
> + $\pi_k$ refers to the mixing proportions of the GMM, i.e., $P(z_n=k)$. It is the scalar mixing proportion for component $k$.
> + $P_k$ in the penalty term $(\mathbf P_K \odot \mathbf\Sigma_k^{-1})_{dd'}$ refers to the adaptive weight matrix used for the graphical Lasso penalty. As we mentioned briefly, these are typically chosen to be inversely proportional to initial partial correlations, following the adaptive Lasso principle.
>
> ### **References**
>
> [1] A. Sportisse, M. Marbac, F. Laporte, G. Celeux, C. Boyer, J. Josse, and C. Biernacki. Model
> based clustering with missing not at random data. Statistics and Computing, 34(4):135, 2024.

---

> > ### Comment · Reviewer_Bx5V · 2025-08-05
> >
> > Thank you for these detailed responses. I am generally satisfied. Can the authors please revise their work to include i) the high-level description of the two-stage approach as outlined in this response and ii) an explanation of why the MNARz mechanism is nonignorable? Otherwise, I have revised my score.

---

> > > ### Author Response · Authors · 2025-08-06
> > >
> > > Reviewer Bx5V,
> > >
> > > Thank you very much for your positive feedback and for confirming your satisfaction with our responses. We are very grateful for your constructive engagement.
> > >
> > > We absolutely commit to making the requested revisions in the final version of the paper. Specifically, we will ensure that the manuscript is updated to include:
> > >
> > > 1. The high-level, two-stage algorithm description as outlined in our rebuttal.
> > >
> > > 2. The detailed explanation for why the MNARz mechanism is formally non-ignorable.
> > >
> > > Your feedback has been instrumental in significantly improving the clarity and rigor of our paper's core methodology. Thank you again for your time and guidance.
> > >
> > > Best regards,
> > >
> > > The Authors

---

### Official Review · Reviewer_Wfhz · 2025-06-30

**Clarity:** 2
**Significance:** 2
**Originality:** 2
**Rating:** 3
**Confidence:** 3

**Summary:**

This paper introduces a unified framework for variable selection in model-based clustering when data contain Missing Not at Random (MNAR) entries. The key contribution is the development of a penalized likelihood approach using a joint model for both the data and the missingn mechanism. The authors propose an efficient EM algorithm for estimation and selection and demonstrate the method’s effectiveness through simulation studies and real data applications.

**Questions:**

Please see the Weaknesses.

**Ethical Concerns:**

["NO or VERY MINOR ethics concerns only"]

**Final Justification:**

While I appreciate the authors’ responses, I maintain my original evaluation, as the contribution remains relatively limited and the literature review is insufficient.

**Limitations:**

Please see the Weaknesses.

**Quality:**

2

**Strengths And Weaknesses:**

Strengths:

The paper focuses on variable selection in model-based clustering under MNAR, which is rarely addressed. The proposed method is computationally efficient and theoretically well-justified, relying on an EM-type iterative procedure specifically adapted to handle MNAR data structures. The experimental design is thoughtful and comprehensive, effectively demonstrating the superior performance of the proposed method across various missing scenarios. Furthermore, the consistency of the variable selection procedure is rigorously established under suitable regularity conditions.

Weaknesses:

1. One of my primary concerns regarding this manuscript is the level of novelty, as the proposed method appears to be an extension of [1] and [2]. The authors need to justify the novelty of the proposed method compared to these works, from either methodological or theoretical perspective. Therefore, I hold a conservative opinion toward the significance of the contribution made by this manuscript.
2. The paper assumes that the missing mechanism follows parametric regression model. This can be restrictive in complex practical cases.
3. There is no numerical studies or test of model misspecification for missing mechanism models.
4. The method is currently limited to continuous data modelled by Gaussian mixtures. Many practical applications (e.g., survey or electronic health record data) involve categorical or mixed-type variables. This limitation is briefly acknowledged but not substantively addressed.
5. While the paper claims that the proposed EM-based framework is computationally feasible, this claim is not supported by quantitative analysis or complexity evaluation. This is a notable limitation, particularly given the well-documented computational challenges associated with EM algorithms in high-dimensional settings or when the number of clusters is large.
6. This paperdoes not report runtime, memory usage, or scalability evaluations, making it difficult to assess the method’s practicality on large-scale datasets. Including a detailed assessment of computational performance, such as runtime comparisons across competing methods, scalability experiments with increasing dimensionality and sample size, or at least a discussion of algorithmic complexity, would substantially enhance the empirical contribution.
7. The method depends on several hyper-parameters (e.g., $\lambda$, $\rho$, and the stopping rule parameter $c$), yet no sensitivity analysis or guidance is provided for their selection.

[1] Maugis-Rabusseau, C., Martin-Magniette, M. L., & Pelletier, S. (2012). SelvarClustMV: Variable selection approach in model-based clustering allowing for missing values. Journal de la Société Française de Statistique, 153(2), 21-36.

[2] Celeux, G., Maugis-Rabusseau, C., & Sedki, M. (2019). Variable selection in model-based clustering and discriminant analysis with a regularization approach. Advances in Data Analysis and Classification, 13, 259-278.

---

> ### Author Rebuttal · Authors · 2025-07-31
>
> We thank the reviewer for their time and for providing a detailed and structured critique of our manuscript. The feedback is very valuable, and we appreciate the opportunity to clarify our contributions and address the perceived limitations. While the reviewer raises several important points, we believe some of the primary concerns, particularly regarding novelty and the handling of MNAR data, stem from a key distinction between our work and the cited references. We will address each weakness point-by-point.
>
> ### **1. On Novelty Compared to Maugis et al. [1] and Celeux et al. [2]**
>
> We respectfully but strongly disagree with the reviewer's assessment. Your concern is understandable, as we build upon the SRUW framework and SelvarMix that Maugis and Celeux pioneered. However, the cited papers operate under a fundamentally different and simpler assumption about the missing data, which lies in the critical distinction between MAR and MNAR:
>
> + Maugis et al. [1] (SelvarClustMV) explicitly and exclusively handle data that is **Missing at Random (MAR)**. Their EM algorithm is designed for the MAR case, where the missingness mechanism can be safely ignored after conditioning on observed data.
>
> + Celeux et al. [2] (SelvarMix) propose a two-stage penalized approach that also operates within the standard likelihood framework, which is only valid for MAR or MCAR data.
>
> Our work is the first to extend this entire variable selection paradigm to the much more challenging Missing Not at Random (MNAR) setting. The reasons this is not a trivial extension are:
>
> 1.  **Methodological aspect:** Moving from MAR to MNAR is not a simple extension. It requires developing a **new joint model** for the data and the missingness mechanism (our SRUW-MNARz model) and modifying a **new EM algorithm** to handle the non-ignorable nature of the missingness. As we have presented in the Appendix, the GMM representation of SRUW under MAR, its identifiability, and the equivalence of MAR and SRUW under MNARz are essential for theoretical guarantee and modeling; thereby, they are novel. The likelihood function, the E-step, and the M-step are all fundamentally different from those in [1] and [2].
>
> 2.  **Theoretical aspect:** The theoretical guarantees for identifiability and selection consistency provided in our **Theorems 1, 2, and 3** are entirely new and, in some sense, complement Celeux's work as it concurrently serves as one of our objectives. The proofs in the cited works are not directly applicable to the MNAR setting, as the statistical properties of the joint likelihood are far more complex. Establishing these guarantees for our new model class is also a core theoretical contribution of our paper.
>
> ### **2. On the Parametric Missingness Mechanism and Model Misspecification**
> We agree that you have a valid point. Our choice of the MNARz mechanism is a deliberate trade-off between **model flexibility** and **statistical identifiability**.
>
> + As shown by recent literature (e.g., Sportisse et al. [3]), general non-parametric MNAR models are often non-identifiable without external information, making inference impossible. The parametric MNARz model is one of the few known mechanisms that is **provably identifiable** in a clustering context, allowing for consistent parameter estimation. We believe that starting with a theoretically sound, identifiable model is a more rigorous scientific approach than using a more flexible but potentially unidentifiable one.
>
> + We acknowledge that we did not include a formal misspecification test. However, we did perform an empirical robustness check. In **Appendix E.2 (Figure 4)**, we tested our method on data generated under `MNARy` and `MNARyz` mechanisms similar to Sportisse et al. [3], where the missingness depends on the unobserved values themselves, a direct violation of our model's assumption. Furthermore, as suggested by reviewer BwuH, we also conducted a study under the case mixture of missing mechanisms. Our method still performed competitively and often outperformed baselines, demonstrating a degree of robustness to this form of misspecification.
>
> ### **3. On Limitations (Continuous Data Only)**
> We fully agree with the reviewer on this point. This is a current limitation of our framework. Extending model-based clustering with variable selection and MNAR handling to mixed-type data is a highly complex research program in its own right. Our work provides the foundational framework for continuous data, which is a necessary and significant first step. However, we believe this framework can be extended naturally, and we assure you that this is not just a hypothetical future direction but an active area of research. For instance:
> + An extension to general mixed data would likely follow a similar path, integrating ideas from model-based approaches, such as Marbac & Sedki [4]. It is crucial to note, however, that their framework is designed for the simpler MAR case. Adapting such a model to handle the non-ignorable MNAR mechanisms, as we do here, would be a significant research effort in its own right. Moreover, the variable roles in their model, where all variables are pairwise independent, are also much simpler than the current SRUW that we adopted in our framework.
>
> + Jacques & Murphy [5] successfully applied this exact philosophy to multivariate count data; however, in a simpler setting where they only have two variable roles rather than three, like the one we use. The fundamental logic remains the same, but they replaced the models for continuous data with Poisson mixtures and Poisson Generalized Linear Models to correctly handle the discrete nature of the variables. This shows that the underlying variable selection strategy is not tied to a single data type.
>
> We will revise the discussion to include this context, clarifying both the current scope and these evidence-based pathways for future extensions, placing it within a broader research trajectory.
>
> ### **4. On Lack of Computational Analysis and Hyperparameter Guidance**
> We thank the reviewer for this crucial feedback and agree that this was a significant omission in our original submission. To rectify this, we have conducted an informal computational analysis, which we will add to the supplementary material.
>
> On the model's complexity, we respectfully agree that while the model is complex, the problem we address in this paper is inherently complicated. Simple models, like the impute-then-cluster baselines, fail precisely because they do not capture the joint structure of the data, variables, and missingness mechanism.
> Our framework's complexity is a necessary consequence of its principled and comprehensive approach.
>
> On hyperparameter tuning, we agree that more guidance would be beneficial.
>
> 1. For the stopping criterion, this parameter balances false positives and false negatives in the variable selection step and we also provided a sensitivity analysis in Appendix (Appendix E.2). Our results support the previous finding by Celeux et al. [1] that the value of c between 3 and 5 provides a good trade-off, and performance is stable within this range for well-posed problems
>
> 2. For the regularization parameters $\lambda$ and $\rho$, they can be selected via a grid search, which is a standard approach for penalized methods. In our experiments, we do not perform an arbitrary search but rather a principled and data-driven procedure to automatically construct a useful grid, which ensures the search space is relevant to the data at hand. The process involves these main steps:
>   + Derive $\lambda_{\text{max}}$ from the maximum initial score for the mean parameters, $\text{max}|t(Z_0) X|$.
>   + Derive $\rho_{\text{max}}$ as a similar upper bound for the precision matrix penalty, derived from the initial sample covariances and cluster responsibilities.
>   + The two values above are motivated to be the smallest penalties that would shrink all corresponding parameters to zero.
>   + We then generate a geometric sequence of $L$ values (e.g., $L = 5$) from a small fraction of the maximum value up to the maximum value itself. This standard practice ensures that the grid efficiently explores penalties across several orders of magnitude.
>
> Regarding the scalability problem, we appreciate the reviewers' insightful feedback and their important questions regarding our framework's computational performance. The concerns about complexity, scalability, and practical feasibility in high-dimensional settings are indeed valid and critical for this method. To address this, we've conducted a detailed study, combining a sketchy, informal treatment of complexity with an experimental analysis. Our results demonstrate that our framework is not only theoretically favorable but also computationally practical, offering a polynomially scaling speedup, which makes it a capable tool for the problems it's designed to solve. The more detailed analysis can be found in the rebuttal of reviewer rfVw.
>
> ### **References**
> [1] Celeux, G., Maugis-Rabusseau, C., & Sedki, M. (2019). Variable selection in model-based clustering and discriminant analysis with a regularization approach. Advances in Data Analysis and Classification, 13, 259-278. Springer.
>
> [2] Maugis, C., Celeux, G., & Martin-Magniette, M-L. (2009). Variable selection in model-based clustering: A general variable role modeling. Computational Statistics & Data Analysis, 53(11), 3872-3882. Elsevier.
>
> [3] A. Sportisse, M. Marbac, F. Laporte, G. Celeux, C. Boyer, J. Josse, and C. Biernacki. Model
> based clustering with missing not at random data. Statistics and Computing, 34(4):135, 2024.
>
> [4] M.Marbac and M.Sedki. Variable selection for mixed data clustering: A model-based approach. arXiv preprint arXiv:1703.02293, 2017.
>
> [5] Jacques, J., & Murphy, T. B. (2025). Model-Based Clustering and Variable Selection for Multivariate Count Data. Computo.

---

> > ### Comment · Reviewer_Wfhz · 2025-08-03
> >
> > Thank you so much for clarifying these issues.  The statement that this is the first work tackling variable selection under MNAR appears to be inaccurate or at least overstated. For example:
> > 1. Garcia, R. I., Ibrahim, J. G., & Zhu, H. (2010). Variable selection for regression models with missing data. Statistica Sinica, 20(1), 149.
> > 2. Du, H., Enders, C., Keller, B. T., Bradbury, T. N., & Karney, B. R. (2022). A Bayesian latent variable selection model for nonignorable missingness. Multivariate Behavioral Research, 57(2-3), 478-512.
> > 3. Lim, D. K., Rashid, N. U., Oliva, J. B., & Ibrahim, J. G. (2024). Deeply learned generalized linear models with missing data. Journal of Computational and Graphical Statistics, 33(2), 638-650.
> > 4. Tang, N., & Tang, L. (2018). Estimation and variable selection in generalized partially nonlinear models with nonignorable missing responses. Statistics and its Interface, 11(1), 1-18.
> > 5. Fang, F., & Shao, J. (2016). Model selection with nonignorable nonresponse. Biometrika, asw039.
> >
> > It would be helpful to include a simulation study that explicitly investigates scenarios under misspecification missingness  mechanisms (e.g., MCAR, MAR, MNAR). This is a critical issue when addressing problems involving missing data.
> >
> > Additionally, while you mention performing a sensitivity analysis, I could not find any corresponding results. Please provide these results to support your claims.

---

> ### Author Response · Authors · 2025-08-04
>
> Thank you for the detailed follow-up and for providing these specific references. We will clarify the scope of our work and its novelty.
>
> ## **1. On the Novelty of Variable Selection under MNAR**
>
> We appreciate you raising these important papers. We agree that variable selection under MNAR is an active research area. However, we believe there is a fundamental distinction between the problem these papers solve and the one we address.
>
> All the cited works focus on **supervised learning** (regression) settings, where the goal is to select important predictors for a known outcome variable. Even in complex cases like Mixture of Experts regression, the clustering is guided by how well different models predict this known outcome.
>
> Our work, by contrast, addresses the **unsupervised learning** task of model-based clustering. Here, there is no outcome variable; the goal is to simultaneously discover the unknown group structure (the clusters) and identify the variables that define that structure. This is a fundamentally different problem, as the variable roles (relevant, redundant, irrelevant) are defined with respect to a latent structure that must be learned from the data itself.
>
> Therefore, we will refine our claim to be more precise: "**To our knowledge, this is the first work to extend the SRUW variable-role modeling framework to the MNAR setting for model-based clustering.**" This distinguishes our contribution from the important work in the supervised context.
>
> ## **2. On Model Misspecification**
>
> Regarding the issue of model misspecification, we apologize if this was not clear in our initial rebuttal. We have indeed conducted these studies, and the results are in the submitted supplementary material.
>
> + Existing Robustness Check: Appendix E.2 (Figure 4) contains a robustness check where we test our method on data generated under MNARy and MNARyz mechanisms, which are direct violations of our model's assumption.
>   + MNARy: where missingness depends directly on the unobserved value itself, independent of the cluster.
>
>   + MNARyz: a more complex case where missingness depends on both the unobserved value and the latent cluster membership.
>
> + New Mixed-Mechanism Experiments: Furthermore, in response to the reviewer **BwuH**, we conducted new experiments on mixed-mechanism data (**MCAR+MAR+MNAR**), which we will add to the appendix.
>
> In all these studies, our method consistently outperforms MAR-based approaches, demonstrating its practical robustness.
>
> ## **3. On the Sensitivity Analysis**
> You are absolutely right to have been unable to find the results, and we sincerely apologize for our error. In our previous response, we mistyped the location. The sensitivity analysis is located in **Appendix E.3**, not E.2. While we have already provided the sensitivity results for the stopping parameter `c`, you are correct that we did not run a new simulation for the regularization parameters $\lambda, \rho$.
>
> We would like to clarify that this was a deliberate choice based on the observed properties of the two-stage, regularization-path-based ranking method, which was pioneered by Celeux et al. (2019). The robustness of the variable ranking to the specific choice of the $\lambda, \rho$ grid is a key feature of this approach.
>
> The ranking score, $\mathcal{O}_{K}(d)$, is not based on a single model fit at an arbitrary $(\lambda, \rho)$ value. Instead, it summarizes the behavior of a variable across the **entire regularization path**. Irrelevant variables have their parameters shrunk to zero very early and stay there, while relevant variables resist this shrinkage. This makes the final ranking quite stable to the precise granularity of the grid.
>
> This exact point was discussed by the originators of the method. Here, we summarize their findings:
>
> 1.  They state that the ranking criterion **"is weakly dependent on the chosen grid"** because irrelevant variables are detected early in the path.
> 2.  They also note that for this ranking stage, they deliberately **ignore the variance matrices** to reduce complexity, justifying it because the primary interest is in separating clusters by their means.
>
> Our work follows their justified methodology. We use a principled, data-driven method to construct the grid (as described in our rebuttal), and we also pre-center and scale our data to align with their second remark.
>
> Therefore, while a dedicated simulation for $(\lambda, \rho)$ is possible, the known robustness of the ranking score itself led us to believe that focusing our new empirical studies on the more critical and previously unanswered question of **misspecification of the missingness mechanism** was a more significant contribution to validating our framework.
>
> We will add a paragraph to the appendix explicitly discussing this point and referencing the prior work to provide this important context for the reader. We hope this clarifies our reasoning and addresses your concern. Thank you again for your constructive engagement.

---

> > ### Comment · Reviewer_Wfhz · 2025-08-05
> >
> > Thank you for the clarification. Could you please elaborate on how your work differs from the following papers, which also focus on unsupervised learning?
> >
> > 1. Marlin, B. M., Roweis, S. T., & Zemel, R. S. (2005, January). Unsupervised learning with non-ignorable missing data. In International W
> >
> > 2. Sportisse, A., Marbac, M., Laporte, F., Celeux, G., Boyer, C., Josse, J., & Biernacki, C. (2024). Model-based clustering with missing not at random data. Statistics and Computing, 34(4), 135.

---

> > > ### Author Response · Authors · 2025-08-05
> > >
> > > Thank you for providing these relevant references. We will clearly pinpoint our specific contribution.
> > >
> > > Our work builds upon these foundational papers but addresses the specific problem of the **simultaneous clustering and selection of variables** under MNAR conditions. Specifically, we address each paper directly below:
> > >
> > > + Marlin et al. (2005): The paper demonstrates how to perform unsupervised learning (specifically, fitting mixture models) in the presence of non-ignorable missing data. Its focus is on correctly learning the parameters of the mixture model itself, assuming a **fixed set of variables** are used for clustering. It does not address the problem of variable selection—that is, determining whether each variable is relevant, redundant, or irrelevant for finding the clusters.
> > >
> > > + Sportisse et al. (2024): This is a cornerstone of our work, and we are glad you highlighted it. It provides the essential, modern framework for performing model-based clustering with MNAR data, which we adopt. However, like Marlin et al., its contribution is focused on solving the clustering problem given a **pre-defined set of variables**. It provides a powerful approach for MNAR clustering (via variant MNAR formulations) but does not include a mechanism to select which variables should be fed into that engine.
> > >
> > > This highlights the specific contribution of our paper. We build a bridge between two distinct research areas: the **variable selection methodology** of the SRUW framework (which assumes MAR) and the **MNAR clustering methodology** of Sportisse et al. (which assumes a fixed set of variables).
> > >
> > > By integrating these, our work introduces a **unified framework** that uses an adaptive penalization strategy to explicitly model the dependency between the missingness mechanism and the latent class structure. This aims to provide a practical solution for researchers or users facing high-dimensional data with MNAR values, where the variables useful for clustering are not known beforehand. The theoretical guarantees for this approach are established in the paper, and our empirical results demonstrate its strong performance compared to relevant baselines.
> > >
> > > We hope this clarifies the precise positioning of our work. We will add this discussion to the related work section to make the distinction clear for all readers.

---

> > > > ### Comment · Reviewer_Wfhz · 2025-08-05
> > > >
> > > > Thank you to the authors for the detailed responses.

---

> > > > > ### Author Response · Authors · 2025-08-05
> > > > >
> > > > > Dear Reviewer Wfhz,
> > > > >
> > > > > Thank you for your time and engagement during the discussion phase. Your questions have been instrumental in helping us improve the clarity and presentation of our work. We hope that our recent responses have helped clarify the paper's novelty and positioning within the literature. If you find it reasonable, we would greatly appreciate your reconsideration of the review scores (Quality: 2: fair, Clarity: 2: fair, Significance: 2: fair, Originality: 2: fair), and in particular the overall rating (“2: Reject: For instance, a paper with technical flaws, weak evaluation, inadequate reproducibility, and incompletely addressed ethical considerations”). If any final questions arise, we remain available and happy to further elaborate.
> > > > >
> > > > > We thank you again for your thoughtful and constructive feedback throughout the discussion.
> > > > >
> > > > > Best regards,
> > > > > The Authors

---

> > > > > > ### Comment · Reviewer_Wfhz · 2025-08-08
> > > > > >
> > > > > > I have reviewed the other reviewers’ comments and, at this stage, I will update my score. Thank you.

---

> > > > > > > ### Author Response · Authors · 2025-08-09
> > > > > > >
> > > > > > > Dear Reviewer Wfhz,
> > > > > > >
> > > > > > > Thank you for the update and for your engagement throughout the discussion period. We are very grateful for your reconsideration.
> > > > > > >
> > > > > > > Best regards,
> > > > > > >
> > > > > > > The Authors

---

### Official Review · Reviewer_rfVw · 2025-07-01

**Clarity:** 3
**Significance:** 3
**Originality:** 3
**Rating:** 5
**Confidence:** 3

**Summary:**

The paper introduces a variable selection framework in model-based clustering that handles data that is missing not at random (MNAR). The paper extends the SRUW model used in variable selection for model-based clustering in high-dimensional settings using the missing not at random model called MNARz which takes into account which class the data is coming from. They propose an EM style algorithm combined with a transformation of MNARz to MAR. The paper shows that under specific assumptions the proposed framework maximizes identifiability and consistency of the SURW framework and with high probability selects a consistent set of relevant features. They also experimentally show that the methods modified with MNARz handles missing not at random better than the standard methods.

**Questions:**

1. How dependent is the framework on the MNARz scheme? If the missingness is slightly different from MNARz (maybe for instance if it also relies on some other parameter as well), does that break things? Does it break things in practice?

2. How do you initialize the EM algorithm? Is a random starting point good enough or do you need to prime it in a specific way?

3. How does the computation scale with parameters such as $D,N$ and $K$? How scalable is the algorithm? What is the dependency of the algorithm's complexity on the degree of missingness?

**Ethical Concerns:**

["NO or VERY MINOR ethics concerns only"]

**Final Justification:**

The authors addressed my concerns about scalability, degree of missingness as well as on the dependence of the missingness model and the algorithm initialization.

**Limitations:**

The paper highlights the limitations such as the restriction to Gaussian mixture models in continuous setting, in the discussion.

**Quality:**

3

**Strengths And Weaknesses:**

Strengths

1. The paper explores the variable selection in model-based clustering. They introduce a method that extends SRUW model to handle missing not at random data using the MNARz method. They show that the method they introduce satisfies identifiability and consistency under the SRUW model and with high probability, can extract a set of reliable and consistent features.

2. The algorithm is generally easy to follow and uses an EM based approach. The overall organization of the paper and the arguments are clear. The paper is overall well-written.

3. The paper shows the practicality of the MNARz based approaches and shows experimentally how they do against standard methods.

Weaknesses

1. It is unclear how the computational complexity of the algorithm scales with the parameters such as $D,N$ and $K$.

2. The conditions for the correctness of the algorithm are mentioned in the supplementary material. It might be better to have at least some informal statement about them in the main body because they would be important in ascertaining the validity of the theorems.

---

> ### Author Rebuttal · Authors · 2025-07-31
>
> We sincerely thank the reviewer for their positive and encouraging feedback. We are delighted that you found our paper well-written and our framework's contributions to be significant. The questions raised are very insightful, and we will address each question on the robustness, initialization, and scalability of our method in detail below.
>
> ### **1. On the Dependence on the MNARz Scheme and Robustness**
> Theoretically, our consistency and identifiability guarantees (Theorems 1-3) hold when data is generated from the specified SRUW-MNARz model. Severe model misspecification would formally invalidate these guarantees, as in most statistical models.
>
> In practice, however, models are rarely perfectly specified. The key question is whether our framework "breaks" gracefully. We have conducted experiments that directly address this. In Appendix E.2 (Figure 4), we tested our method on data that violate the pure MNARz assumption:
>
> 1. MNARy: The missingness probability depends on the unobserved value $y_{nd}$ itself.
>
> 2. MNARyz: The missingness probability depends on both the unobserved value $y_{nd}$ and the latent class $z_n$.
>
> We also performed a small study under the case mixture of missing mechanisms as suggested by reviewer BwuH. Our framework remains competitive and consistently outperforms MAR-based baselines, because the latent-class link captured by MNARz explains a major portion of informative missingness. This also aligns with the findings of Sportisse's work, which we use for modelling MNAR data, cited as [59] in our paper.
>
> ### **2. On the Initialization of the EM Algorithm**
>
> EM for finite mixtures is famously sensitive to initialization. So, we use a standard and effective multi-step initialization procedure:
>
> 1. We perform a simple, fast imputation of the missing data to get a complete dataset. The imputation is used because the ranking stage is essential to avoid misclassification of variable roles.
> 2. We apply hierarchical agglomerative clustering or $K$-means on this completed dataset to obtain an initial partition of the data into $K$ clusters.
> 3. From this initial partition, we compute the starting values for all our model parameters: mixing proportions ($\pi_k$), means ($\mathbf\mu_k$), and covariance matrices ($\mathbf\Sigma_k$). ($\alpha_k, \beta_k$) are also initialized. These serve as the $\mathbf\theta^{(0)}$ for the main EM algorithm.
>
> This hierarchical clustering (HC) based initialization provides a much more sensible starting point than random initialization, typically placing the initial cluster centers in high-density regions of the data and greatly improving the stability and convergence of the EM algorithm. Our results (see `Table 4`) confirm it is more robust than multiple random starts.
>
> **Table 4: Comparison of EM Initialization Methods under 8 Scenarios**
>
> *MIs: Multiple Initializations*
>
> |Scenario|Initialization Method|ARI|
> |:-|:-|:-|
> |1|MIs|0.283|
> ||HC|0.317|
> |2|MIs|0.479|
> ||HC|0.533|
> |3|MIs|0.551|
> ||HC|0.536|
> |4|MIs|0.441|
> ||HC|0.654|
> |5|MIs|0.760|
> ||HC|0.771|
> |6|MIs|0.711|
> ||HC|0.778|
> |7|MIs|0.716|
> ||HC|0.780|
> |8|MIs|0.785|
> ||HC|0.786|
>
> ### **3. On Computational Complexity and Scalability**
> We thank the reviewer for their crucial feedback.
> We address the valid concerns about scalability by comparing our two-stage, Lasso-like procedure proposed by [2] and adapted in our work, with the classical backward stepwise SRUW version from [3]. First, some terms and key assumptions are reiterated:
>
> 1. The true underlying model is sparse. The number of relevant variables, $|\mathbb{S}\_\text{true}|$, and redundant variables, $|\mathbb{U}\_\text{true}|$, define the effective dimension $D\_\text{eff} = |\mathbb{S}\_\text{true}| + |\mathbb{U}\_\text{true}|$, which satisfies $D\_\text{eff}\ll D$.
>
> 2. The EM algorithms used in both methods converge to a solution within a reasonable number of iterations, denoted as $M_\text{EM}$. Note that we can assume sublinear convergence (e.g., [1]), typically $T = \mathcal{O}_p(1/\sqrt{N})$ or the population EM converges in $T = O(log(1/\epsilon))$ iterations such that the output are $O(\epsilon)$-close to the ground truth for spherical Gaussian mixture model. Still, we believe our assumption is more conservative and easier to illustrate the computational advantage.
>
> 3. We assume the variable ranking stage is consistent, as supported by the theoretical results in our paper (Lemma 2 in our manuscript). This implies that, with high probability, all $D_\text{eff}$ informative variables are ranked before the uninformative variables.
>
> 4. We define $M_\text{grid}$ as the grid size for $\lambda$ used in the ranking stage, and $d$ as the number of dimensions in a model during an intermediate step of an algorithm, where $d < D$. Let $M_\text{Lasso}$ be the number of outer-loop iterations for the glasso solver to converge.
>
> #### **3.1 Classical Backward Stepwise SRUW**
>
> This method starts with all $D$ variables and iteratively removes them. At each step with $j$ variables, it must fit and evaluate $j$ different models to decide which variable to remove.
>
> * Fitting a model with $d$ variables costs $\mathcal{O}(M_{\text{EM}}\cdot (NKd^2+Kd^3))$.
>
> * The backward stepwise algorithm uses this model fit as a building block. At each step $j$ (from $j=D$ down to 2), the algorithm must fit and compare $j$ different models, each with $j-1$ variables, to decide which variable to eliminate.
>
> * The total removal cost is:
> $$
> C_{\text{stepwise}}=\mathcal{O}(M_{\text{EM}}\cdot (NKD^4+KD^5))
> $$
>
> The $\mathcal{O}(D^5)$ complexity makes this approach computationally infeasible for even moderately high dimensions (e.g., $D>100$).
>
> #### **3.2 Two-Stage Lasso-Like SRUW**
>
> Our proposed method avoids this combinatorial explosion.
>
> * Stage 1: Variable Ranking: This is the main computational bottleneck. It involves running a penalized EM algorithm over a grid of $M_{\text{grid}}$ regularization parameters. The cost is dominated by E-step and graphical Lasso updates within the M-step, scaling with $D^2$ (see [4], [5]).
> $$
> C_{\text{rank}}=\mathcal{O}(M_{\text{grid}}\cdot M_{\text{EM}}\cdot(NKD^2+KM_{\text{glasso}}D^2))
> $$
>
> * Stage 2: Role Assignment: This stage performs forward and backward scans on the ranked list to identify relevant, redundant, and independent variables. These steps operate on a small number of variables ($|\mathbb{S}_\text{true}|$, $|\mathbb{U}_\text{true}|$, etc.), so its complexity is $\Omega(D_\text{eff})$, not $D$.
>
> Under the sparsity assumption ($D_{\text{eff}}\ll D$), the overall complexity is:
> $$C_{\text{lasso}}\approx C_{\text{rank}}=\mathcal{O}(M_{\text{grid}}\cdot M_{\text{EM}}\cdot K\cdot D^2\cdot (N+M_{\text{glasso}}))$$
>
> #### **3.3 Computational Speedup**
>
> The speedup is the ratio $\frac{C_{\text{stepwise}}}{C_{\text{lasso}}}$.
>
> * If $N>D$, the speedup is $\frac{\mathcal{O}(NKD^4)}{\mathcal{O}(NKD^2\cdot M_{\text{grid}})}=\mathcal{O}(\frac{D^2}{M_{\text{grid}}})$.
> * If $D>N$, the speedup is $\frac{\mathcal{O}(KD^5)}{\mathcal{O}(NKD^2\cdot M_{\text{grid}})}=\mathcal{O}(\frac{D^3}{N\cdot M_{\text{grid}}})$.
>
> In all practical scenarios, our two-stage procedure is polynomially faster, with a speedup of at least $\mathcal{O}(D^2)$. To validate this theoretical prediction, we ran a direct, head-to-head comparison on a focused set of scenarios. The results, combined with previous analysis, demonstrate that our method is a computationally feasible and powerful tool for the high-dimensional problems it is designed to solve.
>
> **Table 5: Empirical Runtime (in seconds) and Speedup Factor**
>
> |Scenario|N|D|K|SelvarMNARz (s)|Clustvarsel (s)|Speedup Factor|
> |:-|:-:|:-:|:-:|:-|:-|:-|
> |Varying D|750|15|4|12.2|190|~15x|
> |Varying D|750|21|4|14.4|640|~44x|
> |Varying D|750|27|4|15.5|2054|~132x|
> |Varying N|1000|20|4|19.0|838|~44x|
> |Varying K|750|20|12|9.77|1242|~127x|
>
> ### **4. On the Dependency of Complexity on the Degree of Missingness**
> We have conducted a new empirical study to provide a definitive answer.
>
> Our two-stage framework's runtime has two components: a ranking stage on imputed data and a role-assignment stage on the original incomplete data, whose cost is theoretically dependent on the average number of observed variables. To test this, we fixed the problem size ($N=1000,D=14,K=4$) and varied the missing rate from 20% to 80% across two complex, mixed-mechanism scenarios. The empirical results, presented in the table below:
>
> **Table 6: Empirical Runtime (in seconds) with Increasing Missing Rate under Mixed Missingness Mechanism**
>
> |Missingness Mechanism|20% (s)|30% (s)|50% (s)|80% (s)|
> |:-|:-|:-|:-|:-|
> |Mixed (MAR+MNAR)|54.1|46.2|32.9|23.9|
> |Mixed (MCAR+MAR+MNAR)|34.9|23.8|25.5|22.8|
>
> In summary, the dependency of our algorithm's complexity on the degree of missingness is an inverse relationship. Practitioners can expect our method to run faster on datasets with higher proportions of missing values, which we assume is a desirable property for real-world applications.
>
> ## **References**
> [1] Kwon, Jeongyeol, and Constantine Caramanis. "The EM algorithm gives sample-optimality for learning mixtures of well-separated gaussians." Conference on Learning Theory. PMLR, 2020.
>
> [2] Celeux, G., Maugis-Rabusseau, C., & Sedki, M. (2019). Variable selection in model-based clustering and discriminant analysis with a regularization approach. Advances in Data Analysis and Classification, 13, 259-278. Springer.
>
> [3] Maugis, C., Celeux, G., & Martin-Magniette, M-L. (2009). Variable selection in model-based clustering: A general variable role modeling. Computational Statistics & Data Analysis, 53(11), 3872-3882. Elsevier.
>
> [4] Witten, D. M., Friedman, J. H., & Simon, N. (2011). New Insights and Faster Computations for the Graphical Lasso. Journal of Computational and Graphical Statistics, 20(4), 892-900.
>
> [5] Mazumder, R., & Hastie, T. (2012). Exact Covariance Thresholding into Connected Components for Large-Scale Graphical Lasso. Journal of Machine Learning Research, 13(27), 781-794.

---

> > ### Comment · Reviewer_rfVw · 2025-08-05
> >
> > I would like to thank the authors for their detailed responses to my questions. The answers have helped clarify the concerns I raised. I have also reviewed the other reviewers' comments, and for now, I prefer to maintain my rating, as I believe the paper does have some merit.

---

> > > ### Author Response · Authors · 2025-08-06
> > >
> > > Dear Reviewer rfVw,
> > >
> > > Thank you for your comment and for your continued support of our paper. We are glad our responses were helpful in clarifying your initial questions.
> > >
> > > We especially appreciate your thoughtful consideration of the paper's merits after reviewing the other discussions. Thank you again for your time and constructive feedback.
> > >
> > > Best regards,
> > >
> > > The Authors

---

### Official Review · Reviewer_BwuH · 2025-07-02

**Clarity:** 3
**Significance:** 2
**Originality:** 3
**Rating:** 5
**Confidence:** 3

**Summary:**

This paper presents a novel unified framework for model-based clustering that effectively tackles both variable selection and missing data (specifically, MNAR) simultaneously. By integrating a data-driven penalty and explicitly modeling missingness with latent class membership, the proposed method, estimated via an extended EM algorithm, demonstrates theoretical guarantees of consistency and empirical improvements in identifying informative variables and clustering complex datasets.

**Questions:**

### Mixture of missing type

While the proposed framework demonstrates strong performance under the assumed MNARz mechanism, its applicability to real-world datasets often characterized by an unknown mixture of MCAR, MAR, and diverse MNAR patterns could be further clarified. To enhance the paper's comprehensiveness and provide additional insights for practitioners, it would be highly beneficial to include empirical results on synthetic datasets that deliberately incorporate a blend of these various missing data patterns. Such an analysis would offer valuable understanding into the algorithm's robustness and generalizability beyond a single assumed missing data mechanism.

### Conclusion from Figure 1
The authors summarize the first synthetic experiment as "Its performance declines only modestly as missingness increases, while the impute-then-cluster baselines deteriorate sharply," The figure is quite small but it looks like all four methods deteriorate in the same way, yet the proposed algorithm is definitely has more tolerance on missing ratio. It would be great to enlarge this figure and ensure clearer differentiation between the performance curves. Some statistical tests to quantitatively supporting the claim is welcomed to report.

### Decoupled versus joint approaches
The proposed framework admirably integrates variable selection with missing not at random (MNAR) modeling by extending the SRUW framework under the MNARz assumption. However, since the core innovation builds upon the MNARz mechanism, it would significantly strengthen the empirical contribution if the authors compared their method against a more explicit baseline that combines vanilla MNARz modeling with standard variable selection applied to fully observed data. Specifically, I encourage the authors to construct a hybrid baseline that uses:

- MNARz modeling (e.g., a GMM with cluster-dependent missingness probabilities as in [59]) for handling missingness,
- Followed by variable selection methods (e.g., penalized likelihood via adaptive LASSO or SRUW role assignment) applied to the internally imputed or completed data.

This would help isolate the gain provided by joint modeling of selection and missingness versus decoupled approaches. Even a simplified version of such a pipeline would provide valuable insight into whether the proposed method’s improvements stem from its structural innovations or primarily from leveraging MNARz assumptions.

Asking for comparison with vanilla MNARz + variable selection isolates whether joint modeling (i.e., their contribution) gives significant advantage over a two-step pipeline.

**Ethical Concerns:**

["NO or VERY MINOR ethics concerns only"]

**Final Justification:**

As per the system’s instructions, paste my prior final justification:

The new results support the view that the joint approach benefits not only from its statistical formulation but also from the stability of its integrated estimation process.

Taken together with the discussion raised by other reviewers, these findings reinforce my original assessment: the paper offers a meaningful contribution, and the authors’ thoughtful responses have further strengthened its empirical and practical value.

**Limitations:**

yes

**Paper Formatting Concerns:**

no issue

**Quality:**

4

**Strengths And Weaknesses:**

### Strengths

- The paper integrates two significant challenges in model-based clustering: variable selection and handling MNAR data. This unified approach is a notable advancement, as most existing methods typically address these issues in isolation.
- The framework provides strong theoretical backing, including proofs for identifiability, asymptotic consistency, and selection consistency, even in the presence of missing data.

### Weaknesses

- As acknowledged by the authors, the current framework is limited to continuous data, specifically via Gaussian mixture models. This restricts its applicability to categorical or mixed-type variables, which are common in many real-world datasets.
- While powerful for targeted MNARz scenarios, the framework currently does not offer an explicit mechanism for automatically identifying underlying missing data patterns in mixed real-world settings. This aspect might be a practical consideration for its broader application in highly heterogeneous missingness scenarios.
- In the synthetic experiments the MAR and MNAR missingness regimes are simulated separately — not simultaneously. For practical usage, it would be highly valuable to see empirical results on synthetic datasets that deliberately incorporate a blend of different types or complexities of MNAR patterns. This would offer crucial insights into the algorithm's performance and robustness when facing the nuanced and heterogeneous MNAR scenarios commonly found in real-world applications.

---

> ### Author Rebuttal · Authors · 2025-07-31
>
> We are deeply grateful to you for your positive assessment and for providing such insightful and constructive feedback. We are delighted that you recognized the strengths of our unified approach and its theoretical backing. The suggestions for improving the empirical evaluation are excellent, and we agree that incorporating them will significantly strengthen the paper's contribution and practical relevance. We will address your concerns and suggestions below
>
> ## **1. On handling a mixture of missing data types**
> Our current experiments focused on "pure" MAR and MNAR scenarios to clearly isolate and demonstrate the benefits of our MNARz-aware model. You are correct that a mixed-mechanism scenario is a more realistic and challenging test of robustness. We will first demonstrate the theoretical behavior and our new small simulation study:
> + Our framework is ultimately designed to handle the most challenging case (MNAR). When MCAR or MAR data are present, our model is technically misspecified, but it is a "safe" misspecification. The MNARz model can be seen as a more general class that can approximate MAR/MCAR. For instance, if a variable is MAR, its missingness does not depend on the latent class $z_n$. In this case, the EM algorithm should ideally learn similar missingness probabilities $\rho_k$ across all clusters $k$ for that variable, effectively recovering a MAR-like mechanism. While not perfectly efficient, the model should not fail catastrophically.
> + To address this question empirically, we have conducted a small simulation study suggested by the reviewer. We generated a synthetic dataset with 14 variables where $\{1,2\}$ is the true relevant set. To account for mixed missingness mechanisms, a subset of variables is made missing via an MCAR mechanism, another subset via MAR, and a final subset via our MNARy mechanism. Note that the MNARy mechanism indicates that the missingness probability depends on the unobserved value $y_{nd}$ itself which is also represent the misspecification of our framework since our model is currently built on MNARz.
> + We denote "MNARz + SelvarMix" to be the suggested baseline from the reviewer. More details are in answer for question 3.
> + Based on the results, our SelvarMNARz framework still demonstrates superior performance in both clustering accuracy (ARI) and, crucially, in correct variable selection compared to MAR-based methods (like SelvarClustMV or impute-then-cluster pipelines) even when the data contains a mixture of missingness mechanisms (including patterns that misspecify our model's assumption)
> + The reason is that it can correctly model the most destructive component (the MNAR part), while being robust enough to handle the simpler MAR/MCAR patterns.
>
> Here is the result:
>
> **Table 1: Performance on Mixed (MAR+MNARy) Data**
>
> |Method              |   ARI| NRMSE|Relevant Variables|
> |:-------------------|-----:|-----:|:--------------------------------------------|
> |SelvarMNARz (Ours)  | 0.808| 0.176|1, 2 (Correct)|
> |VarselLCM           | 0.779|    - |1, 2, 3, 4, 5, 6, 7, 8, 9, 10, 11 (Incorrect)|
> |missRanger + Mclust | 0.799| 0.193|-|
> |gcimputeR + Mclust  | 0.774| 0.413|-|
> |MNARz + SelvarMix   | 0.328| 0.070|1 (Incorrect)|
>
> **Table 2: Performance on Mixed (MCAR + MAR + MNARy) Data**
>
> |Method              |   ARI| NRMSE|Relevant Variables|
> |:-------------------|-----:|-----:|:--------------------------------------------|
> |SelvarMNARz (Ours)  | 0.808| 0.181|1, 2 (Correct)|
> |VarselLCM           | 0.772|     -|1, 2, 3, 4, 5, 6, 7, 8, 9, 10, 11 (Incorrect)|
> |missRanger + Mclust | 0.783| 0.198|- |
> |gcimputeR + Mclust  | 0.755| 0.401|- |
> |MNARz + SelvarMix   | 0.328| 0.087|1 (Incorrect)|
>
> ### **2. On the Clarity and Interpretation of Figure 1**
> We thank the reviewer for this observation and agree that our original presentation and description were not clear enough. We will list out the clarification below:
> + Our intended claim was not that other methods do not deteriorate, but that the rate of deterioration is much sharper for the impute-then-cluster baselines, especially under the MNAR mechanism at high missingness rates (30-50%). Our method, while also declining in performance (as is inevitable), maintains a significantly higher level of performance, showcasing its greater tolerance.
> + You are right that the figure is dense, and we appreciate you pointing this out. In preparing the submission, we were balancing the need for comprehensive results against strict page limits, which led to the current compact layout. Our intention was for the high-resolution vector format to allow for detailed inspection by zooming in on the electronic version. However, we recognize this is not always convenient. We will therefore redesign the figure in the revised manuscript to be immediately legible and ensure all performance curves to be visible.
> + Based on the reviewer's feedback, we have performed pairwise one-sided Welch's t-tests to compare the performance of our proposed SelvarMNARz framework against all other baselines. We focused on the most challenging scenario discussed in the paper: MNAR data at a high 50% missingness rate. The results, including Bonferroni correction for multiple comparisons and 20  replications (sample size), are summarized in the table below.
>
> **Table 3: Welch's t-test for Mean ARI between SelvarMNARz and Clustvarsel with $\alpha = 0.05$**
>
> |Model       | Mean ARI| Std. Dev.|Comparison      |p-value |Signif. |
> |:-----------|--------:|---------:|:---------------|:-------|:-------|
> |SelvarMNARz |    0.511|     0.052|-               |-       |NA      |
> |Clustvarsel |    0.363|     0.088|vs. SelvarMNARz |<0.001  |****    |
> |Selvar      |    0.348|     0.108|vs. SelvarMNARz |<0.001  |****    |
> |VarselLCM   |    0.344|     0.101|vs. SelvarMNARz |<0.001  |****    |
>
> + Based on the p-values, the tests support for our central claim. This demonstrate that the satisfactory clustering performance (as measured by ARI) of our method is not an artifact of experimental randomness but a statistically significant and substantial improvement over all baseline methods.
>
> ### **3. On Decoupled versus Joint Approaches**
> We appreciate for a highly insightful suggestion. We agree that comparing our joint model to a strong, two-step (decoupled) baseline that also uses MNARz is the best way to isolate the benefits of our integrated approach. We have conducted this new experiment as requested. Below we briefly summarize this newbaseline which we called "MNARz + SelvarMix":
>
> + Step 1 (Imputation): We first apply a standard MNARz-GMM model (as in [59]) to the incomplete data. This model is used solely to produce a single, complete imputed dataset based on the final conditional expectations.
> + Step 2 (Variable Selection): We then apply our variable selection and role-assignment procedure as in Celeux et al., 2019 (SelvarMix) to this newly completed dataset.
>
> The results for this new baseline are included in the tables in Section 1 above. The findings are stark: the decoupled MNARz + SelvarMix pipeline performs very poorly, with a low ARI and incorrect variable selection. This empirical result can be explained in a more theoretical manner. The decoupled approach suffers from a **fundamental flaw**: it fails to propagate uncertainty from the imputation step to the selection step.
>
> + The Decoupled Approach: The imputation in Step 1 produces a single "best guess" for the missing values. This completed dataset is then treated by the selection algorithm in Step 2 as if it were the true, complete data. This ignores the fact that the imputed values are uncertain estimates. The variable selection model in Step 2 becomes overconfident in the imputed data, leading to biased parameter estimates and incorrect selection of variables. The low NRMSE of this baseline is deceptive; it indicates stable imputations (low variance), but these imputations are based on a misspecified model that ignores the true variable roles (high bias), leading to poor downstream clustering and selection.
> + Our Joint Approach: Our integrated framework avoids this pitfall. By performing imputation and selection jointly, the uncertainty about the missing values (represented by their posterior distributions) is naturally and correctly propagated through every parameter update. The variable selection process is always aware of the missingness, leading to more robust and accurate parameter estimates and, consequently, superior clustering and variable selection performance.

---

> > ### Comment · Reviewer_BwuH · 2025-08-06
> > **Response to authors' rebuttal**
> >
> > Thank you for the detailed and thoughtful rebuttal. I found the clarifications helpful in addressing most of the concerns I previously raised. I also appreciate the considerable effort the authors have made to incorporate new simulation studies and statistical evaluations in response to the feedback. The added experiments involving mixed missingness mechanisms and the inclusion of the suggested decoupled baseline (MNARz + SelvarMix) are particularly valuable, and they help to further strengthen the empirical contribution of the paper.
> >
> > That said, the notably poor performance of the decoupled baseline is somewhat surprising. While the explanation provided—regarding the lack of uncertainty propagation in the two-step pipeline—is plausible, it would be beneficial to investigate this further. In particular, a more rigorous analysis, perhaps through targeted ablation studies or comparisons with variants that incorporate multiple imputations or uncertainty-aware selection, could help clarify whether the observed gap is inherent or partially attributable to implementation or design choices in the baseline.

---

> > > ### Author Response · Authors · 2025-08-07
> > >
> > > Thank you so much for your thoughtful and highly constructive feedback. We are delighted that you found our rebuttal and new experiments valuable. Your question about the decoupled baseline's performance indicated by a large performance gap was particularly insightful and prompted us to conduct a deeper investigation, which has led to a much clearer understanding that we believe strengthens the paper.
> > >
> > > ### **1. Experimental Design**
> > >
> > > You suggested comparing our model to a baseline that uses multiple imputations (MI) based on the **MNARz mechanism**, which is, in principle, the most rigorous comparison. However, implementing a bespoke MNARz-based MI pipeline is a substantial task in itself, as off-the-shelf tools for this specific purpose are not readily available. Given the time constraints of the discussion period, this was not feasible to design a comprehensive, efficient procedure. Therefore, to thoroughly address the core principle of your suggestion-testing against a baseline that **propagates uncertainty**-we implemented two MI pipelines: one using a general-purpose imputer (`missRanger`) and using the MNARz mechanism with random initializations like previous experiments. The process for each involved:
> > >
> > > 1. Generating $M=10$ plausible completed datasets. The dataset is the same as in experiment 1 in the initial rebuttal.
> > >
> > > 2. Running the `SelvarMix` variable selection procedure independently on each dataset.
> > >
> > > 3. Pooling the results by averaging the ARI/NRMSE scores and selecting variables that appeared in a majority of the runs.
> > >
> > > ### **2. Initial Findings**
> > >
> > > Initially, we ran the MI pipelines using the default Rmixmod backend for SelvarMix, as this is the only clustering engine available in the original package. Your question about the performance gap prompted us to re-examine our experimental setup to ensure the comparison was as fair as possible. In doing so, we realized that our joint model had previously been run using a different backend (Mclust).
> > >
> > > To eliminate this as a potential confounding factor, we aligned the setup by re-running our joint model with the same Rmixmod backend as the baselines. The results from this controlled experiment were surprising and aligned with our previous, simpler baseline:
> > >
> > > **Table 1: Performance on Mixed (MAR+MNARy) Data (Rmixmod backend)**
> > >
> > > *SI: Single Imputation*
> > >
> > > | Method|ARI|NRMSE|Relevant Variables|
> > > | :--| :--| :--| :--|
> > > | SelvarMNARz (Ours)|0.778|0.162|1, 2|
> > > | Decoupled (SI MNARz)|0.328| 0.263 |1|
> > > | Decoupled (MI missRanger)|0.336|0.257|1|
> > > | Decoupled (MI MNARz)|0.328|0.331|1|
> > >
> > > **Table 2: Performance on Mixed (MCAR+MAR+MNARy) Data (Rmixmod backend)**
> > >
> > > *SI: Single Imputation*
> > >
> > > | Method|ARI|NRMSE|Relevant Variables|
> > > | :--| :--| :--| :--|
> > > | SelvarMNARz (Ours)|0.773|0.164|1, 2|
> > > | Decoupled (SI MNARz)|0.328|0.387|1|
> > > | Decoupled (MI MNARz)|0.336|0.322|1|
> > > | Decoupled (MI missRanger)|0.328|0.248 |1|
> > >
> > > ### **3. Discussion and Plausible Explanation**
> > >
> > > This result is counter-intuitive: properly handling imputation uncertainty with MI did not improve performance over single imputation, and both were significantly outperformed by our joint model. This leads us to a crucial hypothesis: **the performance bottleneck in the decoupled approach may not be the imputation step alone, but rather the stability of the downstream analysis engine itself.**
> > >
> > > Multiple Imputation is designed to propagate uncertainty into a downstream analysis, but it implicitly assumes that the analysis tool is stable. If the downstream model (in this case, `SelvarMix` with its `Rmixmod` backend) is prone to finding poor local maxima or is otherwise unstable for this specific problem, then feeding it 10 different (and perfectly valid) imputed datasets will likely result in 10 different poor outcomes. Averaging these does not save the final performance.
> > >
> > > This suggests that the strongly consistent performance of our joint model stems from two sources:
> > > 1.  The **statistical benefit** of simultaneously modeling the data, variable roles, and missingness mechanism.
> > > 2.  The **practical stability** of the integrated estimation procedure, which appears more robust to initialization and model complexity than the decoupled baseline in this context. This finding confirms **your initial intuition** that the performance gap is indeed influenced by implementation and design choices, as our framework appears more robust to the initialization and model complexity challenges observed in the decoupled baseline for this problem.
> > >
> > > Your feedback was instrumental in guiding us to this deeper analysis. We believe this investigation-showing that simply applying MI is not a solution if the downstream analysis is a bottleneck-is a significant finding in itself. We will, of course, add these new results and this detailed discussion to the final manuscript.
> > >
> > > Thank you once again for your invaluable guidance.

---

> > > > ### Comment · Reviewer_BwuH · 2025-08-07
> > > >
> > > > Thank you for the follow-up and the effort to investigate the performance gap in the decoupled baselines. I appreciate the addition of multiple imputation pipelines and the attention to fair experimental comparisons.
> > > >
> > > > The new results support the view that the joint approach benefits not only from its statistical formulation but also from the stability of its integrated estimation process.
> > > >
> > > > Taken together with the discussion raised by other reviewers, these findings reinforce my original assessment: the paper offers a meaningful contribution, and the authors’ thoughtful responses have further strengthened its empirical and practical value.

---

> > > > > ### Author Response · Authors · 2025-08-08
> > > > >
> > > > > Dear Reviewer BwuH,
> > > > >
> > > > > Thank you for your final, supportive confirmation. We truly appreciate your engagement throughout this process. Your guidance on the baseline comparison was particularly instrumental, and it directly helped us strengthen the paper's conclusions. We will, of course, integrate these findings into the final manuscript. Thank you again for your time and expertise.
> > > > >
> > > > > Best regards,
> > > > >
> > > > > The Authors

---

### Author Response · Authors · 2025-08-09
**Thank You to the Area Chairs and Reviewers**

Dear Area Chairs and Reviewers,

We are writing to express our sincere gratitude for the diligent and constructive feedback provided throughout the review process.

The engaging discussion period was particularly instrumental. Your insightful questions prompted us to clarify our core methodology and to conduct new experiments, leading to what we believe is a much clearer and more empirically robust paper. The final work is stronger as a direct result of your guidance.

We are committed to incorporating these valuable discussions, clarifications, and new results into the final manuscript.

Thank you again for your time, expertise, and commitment to improving our work.

Best regards,

The Authors

---

### Note · Authors · 2025-08-12

Dear Area Chairs and Reviewers,

We thank the Area Chairs and Reviewers for a constructive and thorough review process. The discussion period was particularly valuable, as it prompted us to clarify our methodology and conduct new experiments that have strengthened the final manuscript.

To aid the final review, we offer this brief summary of the outcomes from the discussion:

1.  **On Framework Clarification:** We clarified the fundamentals of our modeling framework, including the model's functionality and the non-ignorable nature of the MNARz mechanism. This helped resolve initial questions about the model's design.

2.  **On Novelty:** We clarified the paper's core contribution as a unified framework for simultaneous variable selection and clustering under MNAR, a problem setting distinct from the supervised or MAR-based works discussed.

3.  **On Empirical Validation:** We provided new experiments testing our model's robustness against several forms of misspecification (MNARy, MNARyz) and on complex mixed-mechanism data (MCAR+MAR+MNAR).

4.  **On Baseline Comparisons:** At the reviewers' suggestion, we implemented more rigorous decoupled baselines using Multiple Imputation (MI). This investigation revealed that the performance of two-step approaches is highly sensitive not just to the imputation and initialization methods, but also to the stability of the downstream analysis engine.

5.  **On Scalability:** We provided an informal complexity analysis, supported by empirical validation, to assess our method’s practical feasibility, observing a polynomial speedup over classical approaches. We also found that runtime decreases as the degree of missingness increases, making the method faster on more incomplete datasets.

We believe this dialogue has helped clarify the paper’s contributions and address the initial concerns. Our joint model is competitive with robust MI pipelines while offering improved imputation accuracy and the simplicity of an integrated framework.

We are committed to incorporating these discussions and new results into the final version of the paper. Thank you again for your time and the opportunity to improve our work.

---

### Decision · Program_Chairs · 2025-09-17

**Decision:**

Accept (poster)

**Comment:**

The authors introduce a method for selecting variables in model-based clustering that effectively addresses missing values, even when they are not randomly distributed. The solution is based on a data-driven penalty and explicitly models the missing data through latent class membership. The method uses an extended EM algorithm, and the authors provide theoretical guarantees of consistency of the solution. Additionally, it shows empirical enhancements in the identification of informative variables and the clustering of complex datasets. The problem is particularly relevant in bioinformatics, where missing values are prevalent and variables are often not informative. Most reviewers were positive about this paper and acknowledged the methodological and theoretical contribution of this work. The discussion was active, and most issues were addressed by the authors. Specifically, the authors provided additional results supporting the effectiveness of the method. They also implemented another two-stage approach as a baseline; the results support the advantage of the proposed optimization problem. One reviewer had a remaining concern regarding novelty. Still, after looking at the related work, the work differs enough from existing results that it should be considered a valuable non-trivial contribution.

Overall, I support accepting this paper. While the problem is a bit niche, this is a valuable new idea that leads to empirical benefits, stability, and is theoretically justified. Additionally, the paper is well written, and the approach could find applications in bioinformatics.